

# Instanton density operator in lattice QCD from higher category theory

**Jing-Yuan Chen**

Institute for Advanced Study, Tsinghua University, Beijing, 100084, China

## Abstract

A natural definition for instanton density operator in lattice QCD has long been desired. We show this problem is, and has to be, solved by higher category theory. The problem is solved by refining at a conceptual level the Yang-Mills theory on lattice, in order to recover the homotopy information in the continuum, which would have been lost if we put the theory on lattice in the traditional way. The refinement needed is a generalization—through the lens of higher category theory—of the familiar process of Villainization that captures winding in lattice XY model and Dirac quantization in lattice Maxwell theory. The apparent difference is that Villainization is in the end described by principal bundles, hence familiar, but more general topological operators can only be captured on the lattice by more flexible structures beyond the usual group theory and fibre bundles, making the language of categories natural and necessary. The key structure we need for our particular problem is called multiplicative bundle gerbe, based upon which we can construct suitable structures to naturally define the 2d Wess-Zumino-Witten term, 3d skyrmion density operator and 4d hedgehog defect for lattice $S^3$ (pion vacua) non-linear sigma model, and the 3d Chern-Simons term, 4d instanton density operator and 5d Yang monopole defect for lattice $SU(N)$ Yang-Mills theory; moreover, the structures behind the non-linear sigma model and the Yang-Mills theory are related via an implicit Yang-Baxter equation. In a broader perspective, higher category theory enables us to rethink more systematically the relation between continuum quantum field theory and lattice quantum field theory. We sketch a proposal towards a general machinery that constructs the suitably refined lattice degrees of freedom for a given non-linear sigma model or gauge theory in the continuum, realizing the desired topological operators on the lattice.

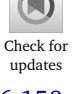
doi:10.21468/SciPostPhys.19.6.158

# 1 Introduction

Quantum chromodynamics (QCD), which describes the strong interaction between quarks and gluons, is a theory that has a simple and elegant form but from which extremely rich dynamics emerges. The dynamics is so non-trivial that most substantial computations of interest are out of the reach of usual analytical means. Wilson pioneered the development of lattice QCD [1], which puts QCD on a spacetime lattice of Euclidean signature, so that, at the fundamental level, the quantum path integral of the theory receives a non-perturbative, UV complete definition, while at the practical level, many problems of interest can henceforth be computed numerically [2, 3]. In this sense, in many practical scenarios lattice QCD is the essential embodiment of QCD.

One of the most important aspects in the richness of QCD is the existence of *instanton* [4], a topological configuration of the Yang-Mills gauge field, whose presence leads to significant consequences in the observed properties of QCD [5–7]. Yet a curious problem then arises. While the instanton configurations are well-defined in the continuum, and moreover it is intuitive that in lattice QCD these configurations must have been somehow effectively captured in the fluctuations of the lattice Yang-Mills path integral, *there is no lattice operator that can be defined in an unambiguous, mathematically natural manner to explicitly represent the instanton*. Yet such an operator is desired, if we want to compute the correlations of instantons among themselves or with other operators, or to study further formal, non-perturbative problems. This problem has been well-known for over four decades [8]. It has a simple origin, which we will review below along with its current workaround solutions [9].

The primary goal of this work is to solve this problem. We find we must understand more deeply what it really means to "put a continuum path integral onto the lattice". We are nat-

urally brought to the use of *higher category theory*, which returns us a conceptually refined definition of lattice Yang-Mills path integral which represents the continuum Yang-Mills theory, especially its topological aspects, better than the traditional definition does. Based upon this lesson, our more general goal—though not fully achieved within the present work—is to establish a machinery that does the following: Given a continuum quantum field theory of interest—think of a non-linear sigma model or gauge theory whose field takes continuous values and has topological configurations—construct the suitable field contents on the lattice so that the topological aspects of the continuum theory are adequately captured.

Our goal of the present work is to introduce the new concepts and principles. An immediate numerical implementation is beyond the scope of the present work. However there would be no fundamental obstacle, and indeed, a more explicit technical description will be presented in a subsequent work [10]. We do anticipate that, using our newly introduced concepts, actual numerical computations that involve explicit instanton operators can be implemented and carried out in the near future.

We stress that being able to define topological operators on the lattice is not only useful for numerical purposes, but also important for analytical studies as well as fundamental understandings. For early examples, being able to define the vortex operator in $S^1$ non-linear sigma model on the lattice led to the discovery of the Berezinskii-Kosterlitz-Thouless transition in 2d [11–13] and allowed an explicit lattice derivation for the 2d boson-vortex duality [14] (with T-duality [15] being its special case); while being able to define the monopole operator in lattice $U(1)$ gauge theory allowed explicit lattice derivations for the 3d boson-vortex duality and the 4d electro-magnetic duality [16–18]. As we will see, these previous examples played a crucial role in motivating our present work. Later, lattice construction has also found an important position in the developments of topological quantum field theory, from both the high energy [19,20] and the condensed matter perspective [21–23]. The thoughts from topological quantum field theory have also deeply influenced our present work, even though the theories we consider, including QCD and others, are not purely topological and contain interesting dynamics at various energy scales. Therefore, in addition to the potential application to the numerics of lattice QCD, theoretical appeal is in itself a major motivation of this work, in hope to facilitate future analytical studies, and to deepen the understanding of the theories themselves by placing the problem in a broader context.

## 1.1 Problem and vague ideas

Let us first introduce the origin of the problem, and sketch some intuitive but vague ideas towards a solution.

We are interested in $SU(N)$ Yang-Mills gauge field in the continuum in 4d. The instanton density and the total instanton number (on an oriented closed 4d manifold $\mathcal{M}$, or an oriented infinite 4d manifold $\mathcal{M}$ with decaying field strength towards infinity) are given by

$$\mathcal{I} := \frac{1}{2}\mathbf{tr}\left[\frac{F}{2\pi} \wedge \frac{F}{2\pi}\right], \qquad I := \int_{\mathcal{M}} \mathcal{I} \in \mathbb{Z}. \tag{1}$$

The instanton number $I$ is the second Chern number of the $SU(N)$ principal bundle of the gauge field over $\mathcal{M}$, and can be non-zero when the principal bundle is topologically non-trivial. In the quantum path integral of a gauge theory, all possible principal bundles are to be summed over.

We want to realize such topological configurations in lattice gauge theory. In the traditional lattice gauge theory [1], a lattice gauge field is to assign to each (oriented) lattice link $l$ an element from the gauge group $G$, so the total configuration space is $\prod_{\text{links } l} G_l$. (We emphasize an important conceptual point: Gauge redundancy does not require any extra treatment on

the lattice, because it is merely $\prod_{\text{vertices } v} G_v$, i.e. an element from $G$ at each vertex, which is a locally finite size space for finite dimensional, compact $G$, and only leads to a product of *local constant factors*—hence unimportant—in the partition function [1]. At the level of observables, the Elitzur's theorem [24] means we do not need to demand any observable to be gauge invariant, since the gauge non-invariant part will essentially automatically vanish anyways.) Thus, in our case, to assign an instanton number to a lattice gauge configuration is to have a function

$$\prod_{\text{links } l} SU(N)_l \to \mathbb{Z}. \tag{2}$$

But the configuration space on the left-hand-side is connected. Thus, if we want to map the configurations to different values of instanton numbers, regardless of how we do so in details, we must encounter discontinuities in the assignment, which is unnatural.

From this simple argument it is easy to see the same problem occurs in more general cases, whenever we want to define the lattice counterpart of "topological configurations" for continuous-valued fields in the continuum. Such cases mainly include non-linear sigma models, whose traditional lattice realizations map each vertex to a point on a "vacua" target manifold, and gauge theories, whose traditional lattice realizations map each link to an element in a Lie group.

In the practice of lattice QCD, the current solutions (see e.g. [9] for a review) are to allow discontinuous assignments, as long as the discontinuities are designed to only occur at field configurations of small weights in the Euclidean lattice path integral. There are several ways to do so. An early way is to forbid those lattice field configurations which appear "highly non-smooth", thus cutting the connected configuration space into disconnected pieces containing "smooth enough" configurations only, and then assign an instanton number to each piece by a procedure of interpolation to the continuum [8]. Another way, close in spirit but much more efficient in practice, is to design a procedure to flow those apparently "highly non-smooth" lattice configurations to more smooth ones, so that the interpolation to a continuum field configuration becomes obvious [25, 26]; discontinuities occur at where the flow bifurcates. Another direction of development is to define suitable Dirac operators on the lattice, and use a lattice version of the Atiyah-Singer index theorem to define the instanton number as the computed index [27–31], which may jump when the lattice Dirac operator becomes non-local as the gauge field varies.

These methods to define instanton number on the lattice have all been studied deeply. The flow based methods and the Dirac operator based methods are both practically used for computing the topological susceptibility, $\langle I^2 \rangle / V$, the variance of the instanton number per spacetime volume. On the other hand, these definitions have important unsatisfactory aspects. On the practical side, if we want to compute correlations that involve local instanton densities at given spacetime positions, as opposed to the total instanton number, it seems the current methods are not sufficient to give an adequate local lattice definition (perhaps except for the first kind of method, which is nonetheless not really used in practical computations for various reasons). On the fundamental side, the problem is even more apparent—discontinuities indicate that these definitions are not sufficiently mathematically natural, and therefore it is hard to anticipate the aforementioned deepened understanding of the theory itself or the facilitation towards future analytical studies; moreover, the Dirac operator based methods have the additional problem of requiring an extra structure on the spacetime, the spin structure, i.e. fermion boundary conditions, which should not have been needed for defining the $SU(N)$ instanton configurations.

What can be done to solve the problem, then? There are two ideas to explore:

1. If the lattice theory has discrete degrees of freedom to begin with, then we can use their values to define discrete topological numbers without encountering discontinuity.

Moreover, the definition should have some local expression so that the local density of the topological number is also defined.

2. Suppose the lattice degrees of freedom are still continuous-valued. But instead of assigning a discrete topological number to a field configuration, we assign a probability profile of the topological number:

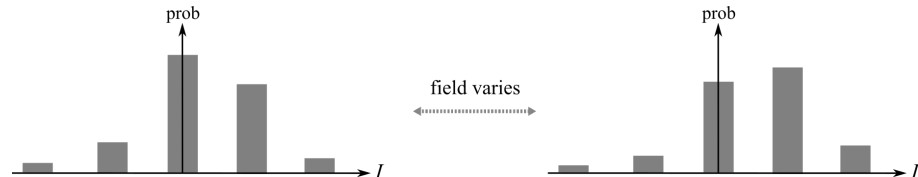

Figure 1: Probability profiles and cross over.

As we continuously vary the field on the lattice, the probability profile changes continuously. Nevertheless, the most probable topological number can jump when the two largest probabilities cross over at some "highly non-smooth" configurations, and this is intuitively how this idea is related to the previous methods that allow discontinuities. Moreover, the assignment of the probability profile to a given field should have some local expression, so that we can also define the probability profile of the local density of the topological number.

At the level of classical action, these ideas seem *ad hoc* and diverted from the original continuum theory, in which there seems to be no discrete-valued local fields and $\mathcal{I}$ depends on $F$ deterministically. However, we are in a quantum theory. The apparent degrees of freedom we use to present the path integral are nothing fundamental, as they are to be integrated out anyways. So these concerns raised around the classical action might not be relevant. On the very contrary, quantum mechanically there are good arguments in support of both ideas:

1. The very problem itself, that we should somehow get discrete topological numbers on the lattice, suggests that it is a good idea to find a presentation of our theory that involve discrete-valued degrees of freedom on the lattice. In fact, even in the continuum, the summation over different principal bundles is a discrete degree of freedom, though seemingly not manifested locally in the classical action.

2. If we intuitively think of the field on the lattice as some kind of "sampling" of the field in the continuum, then something deterministic in the continuum becoming probabilistic on the lattice is natural, because from a "sampling" we should not expect a deterministic inference of the "full original data", but a probabilistic inference. And this especially rings in the context of Euclidean path integral.

Most interestingly, these two ideas are not mutually exclusive, nor orthogonal, but complementary. Let us start from the first idea, i.e. we want to find such a presentation for our theory of interest on the lattice, that not only involves the "traditional" continuous-valued fields, but also some "new" fields, some of which are discrete-valued. In the Euclidean path integral, the "traditional" fields and the "new" fields are coupled, i.e. integrated over with a joint weight which depends on the fields smoothly. For a given configuration of all these fields, a topological operator density has an explicit local expression, such that the associated total topological number is only determined by the discrete-valued fields, hence there is no discontinuity. On the other hand, for a given configuration of the "traditional" fields only, we can integrate out the "new" fields, and since those "new" fields are weighted probabilistically conditioned on

the given "traditional" fields, so will be the topological operator density and hence the total topological number.

The question, then, becomes how to naturally find such "new" fields and joint weights, given a continuum theory of interest. This is where the lessons from the previous examples [11–14, 16–18] and the power of category theory come in. The purpose is to build a natural correspondence between the lattice and the continuum. The idea can be summarized as

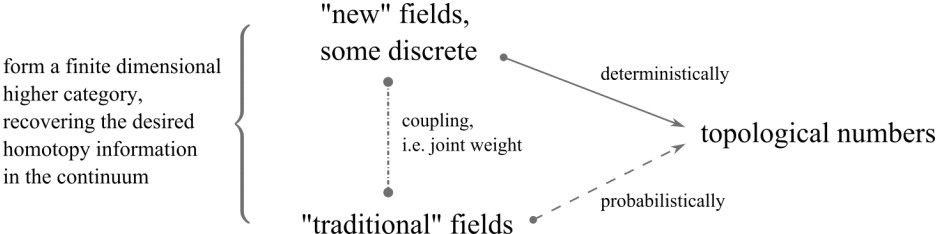

Figure 2: Summarization of main idea.

Here the "homotopy information" means how a field changes gradually from one place to another in the continuum; this is an infinite dimensional piece of information, but since a lot of details are unimportant, the topological part of the information can be effectively reduced to finite dimensional by category theory (these kinds of mathematical problems were indeed one major motivation why category theory was invented and developed in the first place), to be used as the lattice degrees of freedom. The "traditional" lattice fields are in no sense more "fundamental" than the "new" lattice fields, only that they are the lowest order topological approximation to the continuum in a suitable sense.

## 1.2 Why category theory

The idea sketched above has been realized before, though only in limited examples, and not organized into such a general perspective. It first appeared in what is now known as the Villain model [11, 12, 32], which we will review in details in Section 2. Briefly speaking, this is a lattice construction for $S^1$ non-linear sigma model, but such that, in addition to the "traditional" angular variable $e^{i\theta_v}$ on the vertices, there is also an integer variable $m_l$ on the links. They have a joint weight in the Euclidean path integral so that, summing out the integers $m_l$, we will retrieve a theory that resembles the traditional lattice $S^1$ non-linear sigma model (XY model) in terms of $e^{i\theta_v}$ only. Because of this, some might view the Villain model as merely an approximation to the seemingly "more actual" XY model. However, the real gist of the Villain model is that it allows the topological observables of winding number and vortex to be explicitly defined in terms of the integer variable $m_l$ [11–14, 16]. There is something more profound in the Villain model than simply "approximating" the XY model.

What are the lessons to be extracted from the Villain model? There are two directions of thinking, and both will lead to higher category theory if we dig deep enough. In fact, the two directions of thinking splice back again in the language of higher category theory, whence bring us a natural solution for our goal.

The more geometrical direction of thinking is to first understand the "continuum meaning" of the integer variable $m_l$ on the links in the Villain model. Think of the lattice as being embedded in the continuum. Then, starting from the angular variable $e^{i\theta_v}$ at vertex $v$, moving in the continuum along the path traced out by the link $l$ towards a neighboring vertex $v'$, the integer $m_l$ can be thought of as parametrizing the winding of $e^{i\theta(x)}$ around the $S^1$ before

reaching $e^{i\theta_{\nu'}}$:

$$\text{lattice } \theta_{\nu'} - \theta_\nu + 2\pi m_l \quad \longleftrightarrow \quad \text{continuum } \int_{x \in l} d\theta(x). \tag{3}$$

In this sense, the Villian model *topologically refines* the XY model: it captures more information of the continuum theory than the traditional XY model does; in particular, it recovers the homotopy information, so that winding and vortex can be explicitly defined.

This suggests that more generally, when the desired continuum theory has continuous-valued fields, the traditionally defined lattice theory misses the homotopy information from the continuum; but we can refine the lattice theory by suitably including more lattice fields in order to capture the essential homotopy information of interest from the continuum. This is admittedly vague, but category theory is what it takes to make this program substantial. Category theory is the mathematical language that deals with relations, relations between relations, essential contents, and so on, in a manner that is highly general, flexible but at the same time rigorous. It is therefore the natural language to help us rethink what it really means to "essentially capture" the continuum theory onto the lattice.

Let us now turn to the other, more algebraical direction of thinking, as we speak of the "essential information of interest", which is the topological information in our context. Soon after the invention of Villain model, it was understood that the Villainization process of introducing the integer variable is, mathematically, to implement the universal cover $\mathbb{Z} \to \mathbb{R} \to S^1$ over $S^1$, so that the fundamental group $\pi_1(S^1) \cong \mathbb{Z}$—the topological characterization of winding and vortex—is explicitly captured into the newly introduced $\mathbb{Z}$ variables, $\pi_1(S^1) \xrightarrow{\sim} \pi_0(\mathbb{Z}) \cong \mathbb{Z}$. With this understanding, the Villainization process has soon been generalized to lattice gauge theories, with the target space $S^1$ above replaced by Lie groups such as $U(1)$ or others with non-trivial $\pi_1$ [17,18,33], so that the monopole operators can be explicitly defined and worked with. We will review these ideas and these known constructions in details in Section 2.

For $SU(N)$ Yang-Mills theory, Villainization would not help, as $SU(N)$ already has trivial $\pi_1$ and is its own universal cover; meanwhile the instanton configurations in 4d comes from $\pi_3(SU(N)) \cong \mathbb{Z}$. It turns out that there is a mathematical notion called *3-connected cover*, which is to $\pi_3$ just like the universal cover (1-connected cover) is to $\pi_1$. This seems to be what we might need. However, in basically all cases of interest, the 3-connected covers are infinite dimensional spaces, and are hence contradictory to the very purposes of defining lattice theories, especially the purpose of performing numerical computations.

Category theory comes to rescue. In the recent years, Villainization has been reformulated as realizing the universal cover into a category [34–36]. With this perspective in mind, instead of realizing the 3-connected cover as a single infinite dimensional space, one has the new option of realizing it as a higher category [37], which involves multiple "layers" of spaces relating to one another via suitable maps, and moreover each layer can be chosen to be a finite dimensional space. This higher category realization of the 3-connected cover, of which the key part is known as a *multiplicative bundle gerbe* [38], is what we need to put on the lattice in order to capture the $\pi_3$ of the field in the continuum theory, and describing how this works is indeed the primary purpose of this paper.

Most interestingly, this is also where the geometrical and the algebraical directions of thinking splice back together. In the geometrical direction of thinking, we are led to consider the paths, surfaces and so on in the target space. It turns out that, these geometrical objects precisely form a choice of the higher categorical realization of the 3-connected cover [39, 40]—albeit that, in this particular choice, infinite dimensional spaces are involved. But in the categorical sense, or say the algebraic sense, this choice of higher categorical realization is not unique, and there are realizations that are essentially equivalent, but with each layer being finite dimensional [37] and hence suitable for lattice theory. Therefore, the language of higher

category theory indeed unifies the different directions of inspirations that can be drawn from the Villain model, and thereby solves our problem.

In a broader scope, our work directs towards a framework that turns the problem of "how to 'discretize' a continuum quantum field theory (QFT) onto the lattice while retaining the topological operators for the continuous-valued fields" into a well-posed mathematical problem. The general framework is only a sketched one at this stage (though our current limited development is already sufficient for our primary goal), as we will discuss in Section 6, and we believe it can be made more complete in the future. For non-linear sigma models, our proposal can be summarized into a diagram:

$$
\mathcal{M} \to \mathcal{T} \quad \Longrightarrow \quad
\begin{array}{cccc}
\mathcal{L}_d & \Delta_d\mathcal{M} & \Delta_d\mathcal{T} & \mathbf{ET}_d \\
\downarrow\cdots\downarrow & \downarrow\cdots\downarrow & \downarrow\cdots\downarrow & \downarrow\cdots\downarrow \\
\cdots & \cdots & \cdots & \cdots \\
\downarrow\downarrow\downarrow\downarrow & \downarrow\downarrow\downarrow\downarrow & \downarrow\downarrow\downarrow\downarrow & \downarrow\downarrow\downarrow\downarrow \\
\mathcal{L}_2 & \Delta_2\mathcal{M} & \Delta_2\mathcal{T} & \mathbf{ET}_2 \\
\downarrow\downarrow\downarrow & \downarrow\downarrow\downarrow & \downarrow\downarrow\downarrow & \downarrow\downarrow\downarrow \\
\mathcal{L}_1 & \Delta_1\mathcal{M} & \Delta_1\mathcal{T} & \mathbf{ET}_1 \\
\downarrow\downarrow & \downarrow\downarrow & \downarrow\downarrow & \downarrow\downarrow \\
\mathcal{L}_0 & \mathcal{M} & \mathcal{T} & \mathcal{T}
\end{array}
\quad . \tag{4}
$$

The left of the "⇒" describes a field in the continuum—simply a smooth function from the spacetime manifold $\mathcal{M}$ to some target manifold $\mathcal{T}$. The right is what we need for the lattice: Briefly speaking, the second and third columns (simplicial higher categories) are the continuum spacetime and the target space, where $\Delta_n X$ means the space of (singular) $n$-simplices in $X$ and is in general an infinite dimensional space. The first column is the lattice, with the subscript labelling the dimension of the cells; this column is discrete, but nonetheless captures the essential information of the second column in the intuitive way—the lattice just fills up the continuum. (If the lattice is cubical instead of simplicial, the simplicial $\Delta_n$ will be replaced by the cubical version.) The mathematical problem that becomes well-posed is to find the last column: we want a finite dimensional structure (in general a simplicial higher category) that is nonetheless topologically equivalent—up to whatever topological information that we care about—to the third column which involves infinite dimensional spaces. The horizontal arrows between columns are suitably defined maps (higher anafunctors). The map from the first column to the last column represents a field on the lattice. Remarkably, this process of considering the spaces of (singular) simplices in the continuum and then looking for categorical equivalence resonates with the historical development of the subject of higher homotopy theory itself [41–44]. The proposal for gauge theories is similar,

$$
\begin{array}{cc}
\mathcal{PM} & G \\
\downarrow\downarrow & \downarrow\downarrow \\
\mathcal{M} & *
\end{array}
\quad \Longrightarrow \quad
\begin{array}{cccc}
\mathcal{L}_d & \Delta_d\mathcal{M} & \Delta_d|BG| & \mathbf{BEG}_d \\
\downarrow\cdots\downarrow & \downarrow\cdots\downarrow & \downarrow\cdots\downarrow & \downarrow\cdots\downarrow \\
\cdots & \cdots & \cdots & \cdots \\
\downarrow\downarrow\downarrow\downarrow & \downarrow\downarrow\downarrow\downarrow & \downarrow\downarrow\downarrow\downarrow & \downarrow\downarrow\downarrow\downarrow \\
\mathcal{L}_2 & \Delta_2\mathcal{M} & \Delta_2|BG| & \mathbf{BEG}_2 \\
\downarrow\downarrow\downarrow & \downarrow\downarrow\downarrow & \downarrow\downarrow\downarrow & \downarrow\downarrow\downarrow \\
\mathcal{L}_1 & \Delta_1\mathcal{M} & \Delta_1|BG| & G \\
\downarrow\downarrow & \downarrow\downarrow & \downarrow\downarrow & \downarrow\downarrow \\
\mathcal{L}_0 & \mathcal{M} & |BG| & *
\end{array}
\quad , \tag{5}
$$

where the left means Wilson lines assign (as an anafunctor) $G$-values to spacetime paths, and on the right, $|BG|$ is the classifying space of $G$, and the last column, i.e. the structure to be found, can be thought of as related to that in the non-linear sigma model case via the categorical process of delooping. These diagrams will be explained in Section 6.

Prior to the present work, in the recent years higher category theory is already becoming important in theoretical physics, especially in the context of generalized global symmetries and classifications of phases of matter (e.g. [34–36, 45–47]); higher gauge theories have also been proposed to describe exotic field theories [48–50]; moreover, some of these studies indeed have an emphasis on lattice theories [35, 45, 47, 49]. In the present work, however, the way higher category theory appears has some notable differences with the previous works:

- Physically, the present work is not a study of the low energy, universal properties of phases, but a study (to facilitate numerics in prospect) of the dynamics of particular theories at generic energy scales. Moreover, the theories we study are by no means "exotic". They are familiar QFTs—pion effective theory and Yang-Mills theory in QCD—that describe fundamental particle physics, even though there is no obvious involvement of higher categories in their familiar continuum presentations.

- Lattice theories with discrete higher categories are way much better studied than those with continuous ones. The present work deals with continuous ones, and as we have seen, the very reason that higher categories appear is to rescue continuity. The key mathematical feature needed for handling continuous higher categories is the use of simplicially modeled weak categories and anafunctors. This point seems not to have been well appreciated in the theoretical physics context before.

- The categories involved in the present work are not inherently equipped with a linear structure, unlike those used in the classification of low energy phases. Here the quantum mechanical linearity simply results from the fact that in the end we are building a well-defined path integral.

Note however, that if we apply the categorical formalism in this work to discrete groups, we will straightforwardly recover the previously developed group cohomology based lattice models [19, 23]. Therefore, we believe the present work is hinting a more general framework that encompasses the study of topological aspects in both the UV physics and the IR physics.

The previous literature which could somehow hint our present work is [38], which introduced the higher categories we need, i.e. multiplicative bundle gerbes, in the context of Wess-Zumino-Witten terms and Chern-Simons terms in the continuum. The surprise, however, is that the seemingly overkilling mathematical formality there in the continuum becomes natural and necessary in the practical use of lattice QFT. And of course the crucial advantage of the lattice over the continuum is that the path integral measure is explicitly locally well-defined. Moreover, the systematic topological relation found in our present work between QFT in the continuum and on the lattice will allow us to work on more general problems, with more general mathematical structures, in the future.

In a recent work [51] that appeared as this manuscript was being finalized, bundle gerbe techniques have been employed to compute the Wess-Zumino-Witten integral in lattice non-linear sigma model. However, the degrees of freedom there are the traditional fields on vertices, hence the result still has the discontinuity problem. By contrast, the main point of the present work is that the degrees of freedom themselves form a bundle gerbe structure.

Some other recent works [52, 53], which appeared during the course of preparation of this manuscript, used bundle gerbe (without multiplicative structure) on the lattice for a very different physical context. The goal there is to study the higher Berry phase [54, 55] on 1d spatial lattice using matrix product states, and the bundle gerbe realizes an element in $H^3(X;\mathbb{Z}) \xleftarrow{\sim} H^2(\Omega_* X, \mathbb{Z})$, where $X$ is the parameter space (which *a priori* has nothing multiplicative) at each point on the 1d spatial system. By contrast, in our present work, the multiplicative bundle gerbe realizes an element (the generator) of $H^4(|BG|;\mathbb{Z})$, which can transgress $H^4(|BG|;\mathbb{Z}) \rightarrow H^3(|G|,\mathbb{Z})$ if we forget about the multiplicative structure on $G$. While their

physical context and hence the categorical structure are different from our present work, the purpose to introduce finite dimensional higher categorical structures on the lattice is the same: to keep the lattice problem locally finite dimensional meanwhile capturing the essential homotopy information from the continuum. This coincidence shows that such categorical way of thinking might be becoming broadly useful in tackling traditionally difficult problems in different branches of theoretical physics.

This work is organized as follows. In Section 2, we review in details the known examples of lattice theories with well-defined topological operators for continuous-valued fields; they include the Villain model and its variants, and the spinon decomposition. In Section 3, we explain the fundamental difficulty to go beyond the known examples if we stick with the familiar toolbox of group theory and/or fibre bundles. In Section 4, we introduce our main constructions for 1) lattice pion effective theory with skyrmion operator and 2) lattice Yang-Mills theory with instanton operator, respectively, using an intuitive explanation rather than the systematic language of category theory. In Section 5, we first cast the previously known examples in the language of strict higher categories, and then explain how the picture can be generalized to more flexible higher categories that would lead to our main constructions. In Section 6, we sketch our more general proposal towards systematically connecting continuum QFT and lattice QFT. Finally, Section 7 contains our further, scattered thoughts.

## 2 Known examples

We begin by reviewing the known examples of lattice QFTs in which the topological operators of continuous-valued fields are naturally defined after making suitable refinements on the lattice. These known examples belong to two kinds: (generalized) Villainization, and spinon decomposition. We will extract the common rationale behind these constructions, putting them into an organized picture. While the examples themselves are familiar, in our review we will make special emphasis on some conceptual points which are not commonly discussed but will become important. This will help us understand why no more example can be (and indeed, has been) found along this rationale, and henceforth think about how to step back and then reach beyond.

### 2.1 Villainized $S^1$ non-linear sigma model: Winding and vortex

The first example is $S^1$ non-linear sigma model (nl$\sigma$m). In 1d, there is the topological configuration of winding; in 2d and above, there is the topological defect of vortex, where a winding occurs around the vortex core. They are characterized by $\pi_1(S^1) \cong \mathbb{Z}$.

On the lattice, the traditional theory, known as the XY model, has an $S^1$ variable $e^{i\theta_v}$ at each vertex $v$. On the link $l$ between $v$ and $v'$, the path integral is weighted by some positive increasing function $W_{XY}(e^{id\theta_l} + c.c)$, where $d\theta_l := \theta_{v'} - \theta_v$, so that configurations with better aligned $e^{i\theta}$ have higher weights. The partition function then reads

$$Z_{XY} = \left[ \prod_{v'} \int_{-\pi}^{\pi} \frac{d\theta_{v'}}{2\pi} \right] \prod_l W_{XY}(e^{id\theta_l} + c.c). \tag{6}$$

A usual choice for $W_{XY}$ is $W_{XY}(x) = \exp[(x-2)/2T]$, where $T$ can be interpreted as the temperature in statistical mechanics context, and $R = T^{-1/2}$ can be interpreted as the $S^1$ radius in QFT context.[1] However, minor quantitative changes in the detailed choice for the

---

[1]Here we assumed the lattice is uniform, since we have implicitly set each lattice length to be 1. Otherwise the weight on each link should depend on the length of the link in order for the physics to appear uniform. This consideration is understood in all the discussions below.

weight should not matter for long distance observables, in the sense of renormalization. The theory has a 0-form $U(1)$ global symmetry $e^{i\theta_v} \to e^{i\theta_v} e^{i\alpha_v}$ with $\alpha_v$ satisfying $e^{id\alpha_l} = 1$.[2,3] (We do not say "$e^{i\alpha}$ is a constant" because if the spacetime has multiple pieces disconnected from each other, $e^{i\alpha}$ can take different values between the pieces, and this is sometimes referred to as "locally constant".)

For the general reason explained in Section 1.1, topological operators—windings and vortices—cannot be defined naturally in the XY model. For instance, consider a 1d lattice which forms a loop, with the $e^{i\theta}$ configuration indicated by the arrows; here we pictured two configurations:

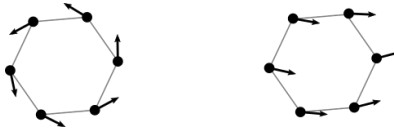

Figure 3: XY configurations on a loop that appear to have different winding numbers.

We feel the configuration on the left should have a winding number $w = 1$. However, by turning each arrow individually (note, this cannot be done in the continuum), this configuration can be continuously deformed to the one on the right, which, we feel, should have $w = 0$. So a deterministic assignment of winding number would certainly run into discontinuities. To avoid this, we can, instead, say the two configurations, respectively, have high probabilities with $w = 1$ and $w = 0$, and the probabilities for different $w$ crossover during the deformation process.

The Villain model is the natural refinement of the XY model that makes this concrete. Originally, on each link $l$, the variable under consideration is $e^{id\theta_l} \in U(1)$. In the Villain model, the link variable is extended to $\gamma_l \in \mathbb{R}$, with the constraint that $e^{i\gamma_l} = e^{id\theta_l}$. We will interpret $\gamma_l$ below. If we choose a $2\pi$ range for $\theta$, say $\theta \in (-\pi, \pi]$, then we can write $\gamma_l = d\theta_l + 2\pi m_l = \theta_{v'} - \theta_v + 2\pi m_l$, where $m_l \in \mathbb{Z}$; but the value of $m_l$ itself is not physically meaningful, because if we change the $2\pi$ range for $\theta$, the value of $m_l$ will change accordingly to keep $\gamma_l$ unchanged. Since the $m$ part is not fixed by $e^{i\theta}$, it is an independent degree of freedom (d.o.f.) to be summed over in the path integral. The XY model is supposed to be the Villain model with $m_l$ summed over, i.e.

$$W_{XY}(e^{id\theta_l} + c.c) \approx \sum_{m_l \in \mathbb{Z}} W_1(\gamma_l), \tag{7}$$

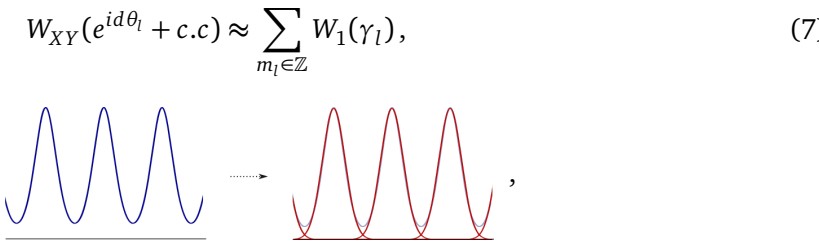

Figure 4: $W_{XY}$ as a function of $d\theta_l$ decomposed as a sum of $W_1$ over $m_l \in \mathbb{Z}$.

as a periodic function of $d\theta_l$, where $W_1$ is some positive even function decreasing with $|\gamma_l|$ (for each value of $m_l$, $W_1(\gamma_l)$ is pictured above as one red peak in $d\theta_l$). Here the "$\approx$" is

---

[2]Our use of $S^1$ versus $U(1)$ is based on whether it is thought of as a space, or as a Lie group with a special point being the identity.

[3]The global symmetry here is actually $U(1) \rtimes \mathbb{Z}_2 \cong O(2)$, where the $\mathbb{Z}_2$ part takes $e^{i\theta_v} \to e^{-i\theta_v}$. This $\mathbb{Z}_2$ part will not play a crucial role in our discussion below; we can explicitly break it and our key points below will not be altered.

because, as we said before, the weight can change slightly without changing the physics, in the sense of renormalization; the usual choice $W_{XY}(e^{id\theta_l} + c.c) = \exp[(\cos d\theta_l - 1)/T]$ is often approximated by a sum of Gaussians, where $W_1(\gamma_l) = \exp[-\gamma_l^2/2T]$, with $m_l$ controlling the center of the Gaussian.[4] The partition function of the Villain model is therefore [11, 12, 32]

$$Z = \left[\prod_{v'}\int_{-\pi}^{\pi}\frac{d\theta_{v'}}{2\pi}\right]\left[\prod_{l'}\sum_{m_{l'}\in\mathbb{Z}}\right]\prod_l W_1(\gamma_l). \tag{8}$$

Now we need to understand the following questions:

– What is the rationale behind the extension from $e^{id\theta_l} \in U(1)$ to $\gamma_l \in \mathbb{R}$?

– How does Villainization enable us to define windings and vortices?

– In what sense things are continuous/smooth in the Villain model?

to appreciate that the Villain model is useful and natural.

   Geometrically, it is intuitive to understand the meaning of $\gamma_l \in \mathbb{R}$ in relation to the continuum $S^1$ nl$\sigma$m. Think of the lattice as being embedded in the continuum. Then $e^{i\theta_v}$ at different vertices $v$ are like samplings from $e^{i\theta(x)}$ with $x$ generic points in the continuum. The lattice link $l$ connecting $v$ and $v'$ is a path in the continuum. Along this path $l$, the field $e^{i\theta(x\in l)}$ interpolates and traces out a path in $S^1$ going from $e^{i\theta_v}$ to $e^{i\theta_{v'}}$.

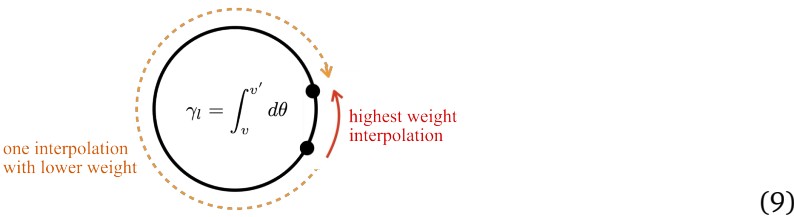

$$\tag{9}$$

Figure 5: Meaning of $\gamma_l$.

Then $\gamma_l \in \mathbb{R}$, satisfying $e^{i\gamma_l} = e^{id\theta_l}$, is nothing but the (signed) length of this path in $S^1$, $\gamma_l = \int_{x\in l} d\theta(x)$, with $m_l \in \mathbb{Z}$ describing the different winding choices for the interpolating path. It is then intuitive why the weight $W_1(\gamma_l)$ is chosen to be decreasing with $|\gamma_l|$.

   With this understanding of $\gamma$, the natural definition for winding number in 1d is obviously

$$w := \oint_{1d}\frac{\gamma}{2\pi} := \sum_l \frac{\gamma_l}{2\pi} = \sum_l m_l \in \mathbb{Z}. \tag{10}$$

It is easy to confirm our intuition before. Given a $e^{i\theta}$ configuration, while the $2\pi\mathbb{Z}$ part of $\gamma_l$ is not determined by $e^{i\gamma_l} = e^{id\theta_l}$, the weight $W$ will prefer the choice that makes $\gamma_l$ closest to 0. In this example of configuration

---

[4]Sometimes this Gaussian approximation is said to be the motivation to perform Villainization. We emphasize that it is not. While bringing in many conveniences for further analytical studies (as we will see soon), the Gaussian approximation is not an important point at the fundamental level. The important point of Villainization is to make it possible to define topological operators [11, 12].

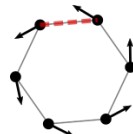

Figure 6: $w = 1$ being the most probable value given the $e^{i\theta_l}$ configuration.

it amounts to the most probable choice being each $\gamma_l \approx 2\pi/(\text{number of links})$, and thus $w = 1$. In terms of $m_l$, since we have chosen $\theta \in (-\pi, \pi]$, we find $d\theta_l \gtrsim 0$ on most links, except for the indicated link, where $\theta_{v'} \gtrsim -\pi, \theta_v \lesssim \pi$ so that $d\theta_l \gtrsim -2\pi$, and therefore the most probable configuration for $m$ is to have $m_l = 1$ at the indicated link and $m_l = 0$ elsewhere. Thus $w = 1$ is the most probable winding number given these $e^{i\theta_v}$ configurations, but other winding numbers are also possible. If we have chosen some other range $(a, a+2\pi]$ for $\theta$, then the most probable $m_l$ configuration would change, but the physically meaningful $\gamma_l$ and $w$ do not depend on this choice.

In 2d and above, we can define the topological defect of vortex. The vorticity at a plaquette $p$ is nothing but the winding number around $p$, so it is defined as

$$v_p := \frac{d\gamma_p}{2\pi} = dm_p \in \mathbb{Z}, \tag{11}$$

where $d\gamma_p$ is the lattice curl around the plaquette (it can be a square, or 2d cell of other shapes). Clearly it satisfies

$$\oint_{2d} v := \sum_p v_p = 0, \qquad dv_c = 0, \tag{12}$$

which means on a closed oriented 2d surface the total vorticity must be 0,[5] and in 3d or above (here $c$ labels a 3d lattice cube, or 3d lattice cell of other shapes) the vortex forms a $(d-2)$-dimensional defect without boundary if viewed on the dual lattice.

Now that vortices are naturally defined on the lattice for $d \geq 2$, we can independently control their fugacity:

$$Z = \left[\prod_{v'} \int_{-\pi}^{\pi} \frac{d\theta_{v'}}{2\pi}\right] \left[\prod_{l'} \sum_{m_{l'} \in \mathbb{Z}}\right] \prod_l W_1(\gamma_l) \prod_p W_2(v_p), \tag{13}$$

where the subscripts on $W$ denote the dimension of the lattice cells on which the weight is defined. A usual convenient choice is $W_2 = \exp[-Uv_p^2/2]$, where $U$ suppresses the vortices.

Being able to unambiguously define the vortices and control their fugacity is tremendously important for understanding the role played by vortices in the renormalization behavior near the BKT transition in 2d [11–14, 56] and spontaneous symmetry breaking (SSB) transition in 3d and above. Let us elaborate on this point. Once $W_2$ is non-trivial, i.e. non-constant, the "recovery of XY model" (7) can no longer happen exactly no matter what we choose for $W_{XY}$ and $W_1$, because now the $m_l$ summation, through the $m_l$ dependence of $W_2(v_p)$ will generate effective couplings of $e^{id\theta_l}$ between different neighboring links.[6] Naively, one might

---

[5]On non-orientable ones such as a Klein bottle, it is easy to see the total vorticity is only well-defined mod 2, and thus can take any even number, and which specific one depends on some choice in the definition.

[6]One may note this effect is analogous to the idea of Symanzik improvement [57–61], but being generated rather than being put in by hand. We will discuss the possible relation in the last section of the paper. This is one way to think about why the introduction of the vortex fugacity weight helps with renormalization.

worry that this may ruin the physics of the XY model. But recall we emphasized that in the renormalization sense, it is unimportant to recover the XY model to the exact details. Quite the opposite, the introduction of the vortex fugacity weight in the model *helps* control the renormalization behavior. The physical intuition is that, as we coarse-grain the lattice, the effects of such fugacity is going to be effectively generated anyways, so having such a weight explicitly in the model helps us keep track of the associated effects, e.g. whether vortices becomes more or less important at larger length scales. In 2d, [14, 56] carefully analyzed the renormalization running of both $W_1$ and $W_2$ to understand the BKT transition; since $W_2$ should indeed have non-trivial running, keeping it as a weight that can indeed run is, therefore, obviously better than fixing $W_2 = 1$. [7] The same physical intuition is understood in the more general models to be introduced in this paper.

If we want to completely forbid the vortices, we can use an $S^1$ Lagrange multiplier [15]

$$W_2^{forbid}(v_p) := \int_{-\pi}^{\pi} \frac{d\widetilde{\theta}_p}{2\pi} \, e^{i\widetilde{\theta}_p v_p} = \delta_{v_p, 0}, \tag{14}$$

and this will hence prohibit the disordered phase.[8] This is something the traditional XY model cannot achieve without Villainization.[9] The Lagrange multiplier $S^1$ field $e^{i\widetilde{\theta}_p}$ can be thought of as living on $(d-2)$-dimensional cells on the dual lattice, and it has a $(d-2)$-form $U(1)$ global symmetry $e^{i\widetilde{\theta}_p} \to e^{i\widetilde{\theta}_p} e^{i\widetilde{\alpha}_p}$ with $e^{i\widetilde{\alpha}_p}$ satisfying $e^{id^\star\widetilde{\alpha}_l} = 1$, where $d^\star$ is like $d$ but performed on the dual lattice.[10,11] This vortex-forbidding symmetry is the conservation of winding number, because a vortex in spacetime is a change of winding number in space. Using the Villain model on the lattice, we can easily see the celebrated mixed anomaly between the original 0-form $U(1)$ and this dual $(d-2)$-form $U(1)$ symmetry: We can try to introduce a background $U(1)$ gauge field for the original symmetry, and find the only way to make it appear consistently in $W_2^{forbid}$ is to let it explicitly break the dual $U(1)$.[12] (In 1d, while $W_2^{forbid}$ cannot be defined, one can define a topological theta term $e^{i\widetilde{\Theta}\sum_l \gamma_l/2\pi} = e^{i\widetilde{\Theta}w}$ in the path integral. One can discuss

---

[7]In 1d, there is an even stronger result that the Villain model with Gaussian $W_1$ is the "perfect action" under renormalization [62].

[8]Unfortunately, an $S^1$ nl$\sigma$m with vortices forbidden is often wrongfully said to be "non-compact" in the literature, but the theory really is still a compact $S^1$ theory, because: 1) the legitimate local boson number (or angular momentum) creation/annihilation operator is still integer quantized, $e^{in\theta_v}, n \in \mathbb{Z}$, and 2) there can be non-trivial windings around non-contractible loops. Mathematically, $m$ being closed does not mean it is exact. By contrast, an actually non-compact $\mathbb{R}$ theory does not require $n \in \mathbb{Z}$, and moreover there is no winding number. However, traditionally this topological distinction was not well-appreciated, so that an $S^1$ nl$\sigma$m with vortices forbidden has been called "non-compact", leading to confusions.

[9]In the XY model, the vortex fugacity cannot be controlled directly since vorticity is not well-defined, however one can anticipate to suppress vorticity by suppressing large values of $d\theta_l$ mod $2\pi$ in the choice of $W_{XY}$, only that such control is indirect, not as explicitly meaningful as the $W_2$ fugacity in the Villain model. On the other hand, obviously $W_2^{forbid}$ can only be defined in the Villain model but not in the XY model.

[10]One may think of $v_p$ as a 2-cochain and $\widetilde{\theta}_p$ as a 2-chain (hence a $(d-2)$-cochain on the dual lattice), and $d^\star$ acting on cochains on the dual lattice is the same as the boundary $\partial$ acting on chains on the original lattice up to some conventional $\pm$ signs.

[11]This symmetry can be seen via the lattice version of integration by parts $\sum_p \widetilde{\theta}_p d\gamma_p = -\sum_p d^\star\widetilde{\theta}_l \gamma_l + $ (boundary terms). The boundary terms might or might not be 0 depending on the boundary condition, and hence the said symmetry might or might not be respected on the boundary.

[12]The introduction of $U(1)$ background gauge field is to replace $\gamma_l \to \gamma_l - A_l$ in $W_1$, where the background $A_l$ is $U(1)$ in the sense that any local $2\pi\mathbb{Z}$ shift $A_l \to A_l + 2\pi N_l$ can be absorbed by the dynamical field $m_l \to m_l + N_l$. However, this changes the value of $v_p := d\gamma_p/2\pi = dm_p$ by $dN_p$ in $W_2$. To remedy this, in $W_2$ we might replace $v_p$ by $(d\gamma_p - dA_p)/2\pi$, which is no longer $\mathbb{Z}$-valued. For a generic $W_2$, there is no particular problem, but for $W_2 = W_2^{forbid}$, the $\widetilde{\theta}_p$ and hence $\widetilde{\alpha}_p$ will cease to be $U(1)$-valued but $\mathbb{R}$. Alternatively, we can replace $v_p$ by $d\gamma_p/2\pi + S_p$ in $W_2$, where $S_p \in \mathbb{Z}$ is the Dirac string part for $A_l$ such that the background flux $F_p := dA_p + 2\pi S_p$ (see Section 2.2) remains invariant under the $N_l$ shift; $d\gamma_p/2\pi + S_p$ also remains invariant. But $S_p$ can at most be required to be closed on the lattice (closedness is a requirement that can be imposed locally, while exactness is a non-local requirement; in a complementary view, if $S_p$ is required to be exact, it is equivalent to $A_l$ being $\mathbb{R}$ rather

the notion of a dual "$(-1)$-form global symmetry" of $\widetilde{\Theta}$ and its mixed anomaly with the original $U(1)$ symmetry [63].)

An analytical convenience for choosing $W_1$ and $W_2$ to be Gaussian is the following.[13] In 2d, by performing Hubbard-Stratonovich transformations for both $W_1$ and $W_2$ and then summing out $m$, and viewing the result on the dual lattice, one can derive the exact *boson-vortex duality* between the lattice and the dual lattice [14] (here the terms are those on the exponent):

$$-\frac{1}{2T}\sum_l(d\theta+2\pi m)_l^2-\frac{U}{2}\sum_p dm_p^2$$

$$\Downarrow \text{ Hubbard-Stratonovich fields } \widetilde{\gamma}_l/2\pi\in\mathbb{R} \text{ and } \widetilde{\theta}_p+2\pi\widetilde{\kappa}_p\in\mathbb{R}$$

$$-\frac{T}{2}\sum_l\frac{\widetilde{\gamma}_l^2}{(2\pi)^2}+i\sum_l\frac{\widetilde{\gamma}_l}{2\pi}(d\theta+2\pi m)-\frac{1}{2U}\sum_p(\widetilde{\theta}+2\pi\widetilde{\kappa})^2+i\sum_p\widetilde{\theta}dm_p \quad (15)$$

$$\Downarrow \text{ sum out } m_l, \text{ enforcing } \widetilde{\gamma}_l-d^\star\widetilde{\theta}_l=:2\pi\widetilde{m}_l\in 2\pi\mathbb{Z}$$

$$-\frac{T}{2(2\pi)^2}\sum_l(d^\star\widetilde{\theta}+2\pi\widetilde{m})_l^2-\frac{1}{2U}\sum(\widetilde{\theta}+2\pi\widetilde{\kappa})_p^2+i\sum_v\theta_v d^\star\widetilde{m}_v\,.$$

Note that the $\mathbb{R}/2\pi\mathbb{Z}$ part of the Hubbard-Stratonovich field for $W_2$ is nothing but the $\widetilde{\theta}$ in $W_2^{forbid}$. The $1/2U$ term explicitly breaks the dual $U(1)$ global symmetry of $\widetilde{\theta}$. As $U\to\infty$, the Hubbard-Stratonovich transformed $W_2$ reduces to $W_2^{forbid}$ as expected, and the dual $U(1)$ symmetry emerges. In this limit, the boson-vortex duality becomes a self-duality (with $2\pi/\widetilde{T}=T/2\pi$) [15], which is the lattice version of the T-duality in string theory. In $d\geq 3$, the derivation for boson-vortex duality is exactly the same, and one can easily see that in $d=3$ the resulting dual theory is a $U(1)$ gauge theory (see Section 2.2) coupled to a $U(1)$ nl$\sigma$m Higgs field [16], while in more general dimensions it is a $(d-2)$-form $U(1)$ theory (see Section 2.3) coupled to a $(d-3)$-form $U(1)$ field; when $U\to 0$ the $(d-3)$-form field cease to exist and a $(d-2)$-form dual $U(1)$ global symmetry emerges.

All these discussions show that Villainization is a topological refinement to better connect the lattice theory to the continuum, and is tremendously useful in the analytical studies of important non-perturbative physics.

To prepare for our later discussions, however, we need some further understandings of the Villain model. We begin by reinterpreting Villainzation as gauging a $\mathbb{Z}$ global symmetry. While such kind of group theoretic interpretation will no longer be possible in the more general cases that we aim at (and this is a crucial point—we will only have category theoretic interpretation in general), it is helpful for bringing up some important ideas that we want to discuss.

Suppose we begin with an $\mathbb{R}$-valued theory, where $\theta_v\in\mathbb{R}$ instead of $S^1$. Each link has weight $W_1(d\theta_l)$, so the theory has a 0-form $\mathbb{R}$ global symmetry $\theta_v\to\theta_v+\alpha_v$, $\alpha_v\in\mathbb{R}$, $d\alpha_l=0$. We want to reduce this $\mathbb{R}$ global symmetry to $U(1)$, and we can do so by gauging the $2\pi\mathbb{Z}$ subgroup of the global symmetry. Denoting the $2\pi\mathbb{Z}$-valued dynamical gauge field by $2\pi m_l$, the gauging process is to replace $d\theta_l$ by $d\theta_l+2\pi m_l$ in $W_1$ and sum over $m_l$, and the gauge invariance is $\theta_v\to\theta_v+2\pi k_v$, $m_l\to m_l-dk_l$ for any $k_v\in\mathbb{Z}$; moreover, the gauge flux $dm_p$ can have its own dynamical weight, some $W_2(dm_p)$ on each plaquette $p$. Now, we basically

than $U(1)$), it is unlike $d\gamma_p/2\pi=dm_p$ which is exact by definition. Now that $S_p$ might be non-exact in a Dirac quantized flux situation, when $W_2=W_2^{forbid}$, it will explicitly break the dual $U(1)$ symmetry parametrized by $\widetilde{\alpha}$, demonstrating the said mixed anomaly.

[13]We emphasize that while the dualities below are exactly derived at the lattice level by choosing the weights to be Gaussian, if the weights are modified by not too much, the physics of the dualities should still hold in the IR. Hence this is a convenience, rather than something fundamental.

obtain the Villain model, except here $\theta_v \in \mathbb{R}$. But there is the $2\pi\mathbb{Z}$ gauge invariance $k_v$ that we can exploit, to gauge fix each $\theta_v$ to $(-\pi, \pi]$. Thus, we obtain the Villain model by gauging the $2\pi\mathbb{Z}$ subgroup from an $\mathbb{R}$ theory and then fixing the $2\pi\mathbb{Z}$ gauge on $\theta_v$.

A first observation from this reinterpretation is that the Villain model relies on the fact that $S^1 = \mathbb{R}/2\pi\mathbb{Z}$. More exactly, it relies on finding the universal cover of $S^1$, which is $\mathbb{R}$:

$$
\begin{array}{c}
2\pi\mathbb{Z} \to \mathbb{R} \\
\downarrow \\
S^1
\end{array}
\qquad . \tag{16}
$$

While such a two row notation is standard for a fibre bundle in mathematics, in our context there is an extra meaning to have two rows—different rows are fields that live on lattice cells of different dimensions: the lowest row contains fields that live at the 0-dimensional vertices, $e^{i\theta_v} \in S^1$, while the row above are fields that live at the 1-dimensional links, $\gamma_l = d\theta_l + 2\pi m_l \in \mathbb{R}$ subjected to $e^{i\gamma_l} = e^{id\theta_l}$, and $m_l \in \mathbb{Z}$ that helps parametrize $\gamma_l$ as $d\theta_l + 2\pi m_l$.

Why is finding the universal cover such a useful thing to do? This leads to the algebraic motivation behind Villainization, in complementary to the geometric motivation (9). It is because Villainization leads to an isomorphism from $\pi_1(S^1)$ to $\pi_0(\mathbb{Z})$, through the universal cover $\mathbb{R}$ which is a non-trivial $\mathbb{Z}$ bundle over $S^1$. Generally, for a fibre bundle $F \to E \to B$, their homotopy groups satisfy the long exact sequence[14]

$$
\cdots \to \pi_n(F) \to \pi_n(E) \to \pi_n(B) \to \pi_{n-1}(F) \to \pi_{n-1}(E) \to \pi_{n-1}(B) \to \cdots . \tag{17}
$$

The reason to find the universal cover $E$ of $B$ is so that $\pi_1(E)$ is trivial and $\pi_0(E) = \pi_0(B)$ (which is trivial as well if $B$ is connected), hence the long exact sequence leads to an isomorphism $\pi_1(B) \xrightarrow{\sim} \pi_0(F)$. In our case, $\pi_1(S^1)$ is what characterizes the winding of $e^{i\theta}$ in the continuum, which becomes ambiguous on the lattice due to the said discontinuity problem; by capturing this information into $\pi_0(\mathbb{Z})$, which is counted by $m$, the discontinuity problem is resolved because $\mathbb{Z}$ is discrete to begin with. Using the same idea, we can use the Villainization process to capture general $\pi_1$ topological information, see Section 2.3.

The $\mathbb{Z}$ gauge theory perspective also brings us to the front of an important conceptual question: In what sense things are continuous/smooth in the Villain model?

First of all, this is a question because apparently $\gamma_l = \theta_{v'} - \theta_v + 2\pi m_l$ is no longer continuous in the original $S^1$ variables $e^{i\theta_v}$, and therefore if we think of $e^{i\theta_v} \in S^1$ and $m_l \in \mathbb{Z}$ as some kind of "fundamental local d.o.f.", the path integral weight $W_1(\gamma_l)$ appears discontinuous in $e^{i\theta_v}$.

This question arises because $e^{i\theta_v} \in S^1$ and $m_l \in \mathbb{Z}$ are not a good set of variables to simultaneously think about. We can either simultaneously think about $\theta_v \in (-\pi, \pi]$ and $m_l \in \mathbb{Z}$, or simultaneously think about $e^{i\theta_v} \in S^1$ and $\gamma_l \in \mathbb{R}$ subjected to the constraint $e^{i\gamma_l} = e^{id\theta_l}$. The path integral weight is smooth either way (being smooth in the second way implies that in the first way).

The $\mathbb{Z}$ gauge theory perspective helps us understand this important conceptual point. In gauge theory, it is common to either describe the d.o.f. by gauge fixing, or by looking at gauge invariant combinations:

- Apparently, $\theta_v \in (-\pi, \pi]$ and $m_l \in \mathbb{Z}$ are the $\mathbb{Z}$ gauge fixed d.o.f.. The path integral weight is smooth in $\theta_v \in (-\pi, \pi]$, but the desired continuity from $\theta_v \gtrsim -\pi$ to $\theta_v \lesssim \pi$ in only recovered by absorbing the $2\pi\mathbb{Z}$ shift into the neighboring $m_l$'s, or in other words, the path integral, smooth in $\theta_v \in (-\pi, \pi]$, becomes smooth in $e^{i\theta_v}$ only after summing over all the $m_l$'s, see (7), but not before the sum.

---

[14]Which means the image of each arrow is the kernel of the next arrow.

- On the other hand, $e^{i\theta_v} \in S^1$ and $\gamma_l \in \mathbb{R}$ are $\mathbb{Z}$ gauge invariant. Physical observables must be built out of them. In terms of these $\mathbb{Z}$ gauge invariant variables, the path integral weight is smooth as expected.[15] The price paid is, the independent $\mathbb{Z}$ gauge invariant variables are not locally factorized, due to the link constraint $e^{i\gamma_l} = e^{id\theta_l}$, and this is a common feature of gauge theory.[16]

For instance, consider a lattice consisting of a single plaquette only, as shown below. The locally factorized $\mathbb{Z}$ gauge fixed d.o.f., forming the space $(-\pi, \pi]^4 \times \mathbb{Z}^4$, are shown on the left, while an independent set of $\mathbb{Z}$ gauge invariant fields can be chosen as on the right, forming the actual configuration space $(S^1)^4 \times \mathbb{R}^4 \big|_{\text{link constraints } e^{i\gamma} = e^{id\theta}} \cong S^1 \times \mathbb{R}^3 \times \mathbb{Z}$:

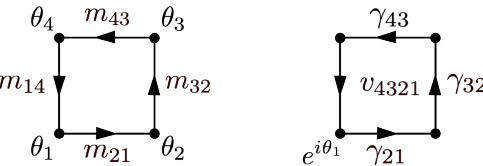

Figure 7: Apparent vs invariant degrees of freedom on a lattice consisting of a single plaquette.

The $\mathbb{Z}$ gauge fixed space $(-\pi, \pi]^4 \times \mathbb{Z}^4$ is glued along suitable boundaries into the actual configuration space $(S^1)^4 \times \mathbb{R}^4 \big|_{\text{link constraints } e^{i\gamma} = e^{id\theta}} \cong S^1 \times \mathbb{R}^3 \times \mathbb{Z}$ (rather than into the naive $(S^1)^4 \times \mathbb{Z}^4$)—this is when we express the $\mathbb{Z}$ gauge invariant variables $e^{i\theta_v}$ and $\gamma_l$ in term of the $\mathbb{Z}$ gauge fixed $\theta_v$ and $m_l$. The path integral weight is smooth over the actual configuration space. Moreover, the actual configuration space covers (which means it can be mapped back to) the $(S^1)^4$ of the XY model. On a more general lattice, the configuration space has a topology of $(S^1)^{N_0} \times \mathbb{R}^{N_1} \big|_{\text{each } e^{i\gamma_l} = e^{id\theta_l}} \cong (S^1)^{B_0} \times \mathbb{R}^{N_0 - B_0} \times \mathbb{Z}^{N_1 - (N_0 - B_0)}$, rather than the naive $(S^1)^{N_0} \times \mathbb{Z}^{N_1}$ or the $\mathbb{Z}$ gauge fixed $(-\pi, \pi]^{N_0} \times \mathbb{Z}^{N_1}$, where $N_0, N_1$ are the numbers of vertices and links, and $B_0$ is the zeroth Betti number, i.e. the number of pieces of the lattice disconnected from one another. The $(S^1)^{B_0}$ factor is where the $U(1)$ global symmetry acts on, the $\mathbb{Z}^{N_1 - (N_0 - B_0)}$ factor counts all possible winding and vortex configurations, and the $\mathbb{R}^{N_0 - B_0}$ factor is the space of independent $\gamma$'s given the winding and vorticity. The dependence on the topological number $B_0$ is a reflection that the configuration space is not a local factorization.

In summary, the apparent $\theta_v \in (-\pi, \pi]$ and $m_l \in \mathbb{Z}$ variables allow us to write the path integral measure in an explicitly locally factorized form, and they can be further glued into the actual configuration space which is not locally factorized; the path integral weight is not only smooth over the space of the apparent variables, but must also be smooth over the actual configuration space after the gluing. Alternatively, we can begin with the physical $e^{i\theta_v} \in S^1$ and $\gamma_l \in \mathbb{R}$ which are manifestly local and the weight and observables must be smooth in them; but then we must impose the constraint $e^{i\gamma_l} = e^{id\theta_l}$ on each link, making the constrained actual configuration space not locally factorized. It seems a little verbose here to describe this trade-off between smoothness and local factorizability, though fortunately the $\mathbb{Z}$ gauge theory perspective helps us understand this point, thanks to our familiarity with gauge theories. Later we will show the Villainization process can be recast in the language of the Lie groupoid $S^1 \times \mathbb{R} \rightrightarrows S^1$. There, this continuity and locality issue becomes naturally understood in terms

---

[15]The factors $W_1, W_2$ are smooth in $\gamma_l$, and do not otherwise depend on $e^{i\theta_v}$ due to the 0-form $U(1)$ global symmetry. If the $U(1)$ global symmetry is explicitly broken, there can be some vertex weight $W_0(e^{i\theta_v})$, which is still $\mathbb{Z}$ gauge invariant.

[16]For general gauge theories in the Hamiltonian formulation, it is familiar that the gauge invariant Hilbert space is not locally factorized. Although we are in a path integral rather than a Hamiltonian formulation, this aspect is similar.

of functors from the lattice to this Lie groupoid. When we tackle our main problems of $S^3$ nl$\sigma$m and $SU(N)$ Yang-Mills, the familiar gauge group approach becomes mathematically inadaquate, but these two alternative pictures of "apparently locally factorized d.o.f. glueing into a not locally factorized actual configuration space" and "apparently local physical variables being constrained down to a not locally factorized actual configuration space" remain valid, and is naturally understood from the functor perspective.

## 2.2 Villainized $U(1)$ gauge theory: Dirac quantization, monopole, Chern-Simons and instanton

Soon after the Villainization method appeared in the $S^1$ nl$\sigma$m context, it has been applied to $U(1)$ gauge theory as well [16–18]. In the recent years the Villainized $U(1)$ gauge theory (along with further generalizations) has attracted revived attention in the purview of (ordinary and higher form) symmetries and anomalies in topological terms [64] and topological orders [65], as well as the more exotic fractons [66]. The idea is extremely simple—just put those d.o.f. we have for Villainized $S^1$ nl$\sigma$m onto lattice cells of one higher dimension. This leads to natural lattice descriptions for Dirac quantization in 2d, monopole in 3d or higher, abelian Chern-Simons (CS) term in 3d and abelian instanton in 4d, and so on.

In the traditional $U(1)$ lattice gauge theory [1], on each link there is a $U(1)$ variable $e^{ia_l}$, which can be thought of as a Wilson line across that link. The flux around a plaquette is also $U(1)$-valued, $e^{ida_p}$, which can be thought of as a Wilson loop around the plaquette; the path integral of the traditional $U(1)$ lattice gauge theory is weighted by a positive increasing function $W(e^{ida_p} + c.c.)$ on each plaquette. If the gauge theory is coupled to matter, such as in lattice QED[17] or abelian Higgs model, $e^{ia_l}$ appears in the hopping of the matter particles. For examples, when coupled to fermion $\psi$ of charge $q_\psi \in \mathbb{Z}$, the hopping is $\bar{\psi}_{v'}e^{iq_\psi a_l}\psi_v$; when coupled to an XY model boson $e^{i\theta_v}$ of charge $q_\theta \in \mathbb{Z}$, the hopping is $e^{-id\theta_l + iq_\theta a_l}$. The charge must be integer due to the $U(1)$ nature of $e^{ia_l}$. The $U(1)$ gauge transformation is $e^{ia_l} \to e^{ia_l}e^{id\alpha_l}$, $\psi \to e^{iq_\psi \alpha_v}\psi$, $e^{i\theta_v} \to e^{iq_\theta \alpha_v}e^{i\theta_v}$ for arbitrary $e^{i\alpha_v} \in U(1)$.

The path integral of the gauge field is to integrate over $e^{ia_l} \in U(1)$ for all links $l$. As emphasized in the introduction, gauge redundancy is unimportant and does not require any treatment on the lattice. In the partition function, gauge redundancy is merely a $U(1)$ at each vertex, which is a locally finite size space and hence only leads to a product of *local* constant factors in the partition function [1]. And observables are not demanded to be gauge invariant, since any gauge non-invariant part will automatically vanish anyways, by Elitzur's theorem [24]. Therefore, gauge fixing or any other treatment about the gauge redundancy is not needed. This is a remarkable point, because in many cases in the continuum, gauge fixing involves solving (usually differential) equations over the spacetime manifold, generally leading to global issues, but these issues are artifacts from the choice of gauge fixing condition, rather than anything intrinsic to the gauge invariance itself. Any physical effect, local or global, must manifest on the lattice without any extra treatment about the gauge.

A pure $U(1)$ gauge theory has a 1-form $U(1)$ global symmetry $e^{ia_l} \to e^{ia_l}e^{i\beta_l}$, where $e^{i\beta_l}$ satisfies $e^{id\beta_p} = 1$, which does not change $e^{ida_p}$ and hence the path integral weight.[18] This is *not* a $U(1)$ gauge transformation in general, because when the spacetime has non-contractible

---

[17]It is understood that QED is not "renormalizable" in the sense that if we reduce the lattice unit length in the UV while changing the path integral weight in order to maintain the IR physics, then we expect, in analogy to the Landau pole in continuum, the path integral weight will run into some singularity at some finite unit length, i.e. the unit length cannot be made arbitrarily small, unless new physics is introduced in the UV. But at any finite unit length before that happens, the lattice model is still well-defined and we can still discuss its IR physics.

[18]By "1-form global" here, we do not mean $\beta$ is "constant". It means the $e^{i\beta}$ holonomy for any two loops (generally non-contractible) that can be deformed to each other must be the same. This is like, by "0-form global", it means $e^{i\alpha}$ for any two points can be connected by a path to each other must be the same, but not necessarily so for those that cannot.

loops, the closedness condition $e^{id\beta_p} = 1$ does not imply exactness, i.e. there might be no choice of $e^{i\alpha_v}$ such that $e^{i\beta_l} = e^{id\alpha_l}$. Thus, when the $U(1)$ gauge field is coupled to matter, while the $U(1)$ gauge invariance must still be there, the 1-form $U(1)$ global symmetry is explicitly broken.

Similar to the winding and vortex configurations in XY model, configurations which look like having non-trivial Dirac quantized fluxes or non-trivial monopoles do appear in fluctuations in the traditional $U(1)$ lattice gauge theory, but there is no natural way to actually define these topological operators. Being able to define and hence forbid (or at least highly suppress) the monopole operator is particularly important for application to Maxwell theory in reality, in which monopoles have not been observed; monopole proliferation will lead to the confinement phase [1, 16, 56, 67] rather than the realistic Coulomb phase, i.e. the 1-form $U(1)$ SSB phase.[19]

We have to Villainize the traditional theory to have natural definitions for the topological operators. That is, on each plaquette we now have the real-valued flux $f_p \in \mathbb{R}$ satisfying the constraint $e^{if_p} = e^{ida_p}$; if we fix the range $a_l \in (-\pi, \pi]$, then we can write $f_p = da_p + 2\pi s_p$, where $s_p \in \mathbb{Z}$ is to be thought of as the Dirac string variable (if viewed on the dual lattice) and summed over in the path integral. If we think of the plaquette as being embedded in the continuum, the lattice gauge flux $f_p \in \mathbb{R}$ can be thought of as the integral of the continuum field strength over the plaquette, $f_p = \int_{x \in p} da(x)$. Over a closed oriented 2d surface, we find the Dirac quantization condition

$$\oint_{2d} \frac{f}{2\pi} := \sum_p \frac{f_p}{2\pi} = \sum_p s_p \in \mathbb{Z} \tag{18}$$

(just like the winding number in the $S^1$ nl$\sigma$m). On each lattice cube $c$ (or 3d cell of other shapes), we can define the monopole number

$$m_c := \frac{df_c}{2\pi} = ds_c \in \mathbb{Z}, \tag{19}$$

(just like the vorticity in the $S^1$ nl$\sigma$m) which satisfies

$$\oint_{3d} m := \sum_c m_c = 0, \qquad dm_h = 0, \tag{20}$$

where $h$ denotes a hypercube (or 4d cell of other shapes). So monopoles are $(d-3)$-dimensional defects without boundary, if viewed on the dual lattice. The Villainized $U(1)$ gauge theory reads

$$Z = \left[ \prod_{l'} \int_{-\pi}^{\pi} \frac{da_{l'}}{2\pi} \right] \left[ \prod_{p'} \sum_{s_{p'} \in \mathbb{Z}} \right] \prod_p W_2(f_p) \prod_c W_3(m_c). \tag{21}$$

(If there are charged matter fields, Villainization of the gauge field makes no change to its couplings to those matter fields.) The usual Gaussian choices for the weights are $W_2(f_p) = \exp[-f_p^2/2e^2]$ (with $e^2$ the usual Maxwell coupling), $W_3(m_c) = \exp[-Um_c^2/2]$. Again, if we want to completely forbid the monopoles and hence prohibit the confinement

---

[19]In the Coulomb vs the confinement phase, the Wilson loops' exponential suppression is proportional to the perimeter vs the (minimal) bounded area, generalizing the long vs short ranged correlation for order parameters in 0-form symmetry SSB. When coupled to matter field, both phases have perimeter law, but a closer inspection shows in the Coulomb phase the perimeter law can be realized as a zero law [68].

phase—as it should for the Maxwell theory in reality—we can use the Lagrange multiplier [64, 65][20]

$$W_3^{forbid}(m_c) := \int_{-\pi}^{\pi} \frac{d\widetilde{a}_c}{2\pi} e^{i\widetilde{a}_c m_c} = \delta_{m_c,0} \, , \tag{22}$$

where $\widetilde{a}_c$ can be thought of as living on $(d-3)$-dimensional cells on the dual lattice, and has a dual $(d-3)$-form $U(1)$ global symmetry $e^{i\widetilde{a}_c} \to e^{i\widetilde{a}_c} e^{i\widetilde{\beta}_c}$ satisfying $e^{id^*\widetilde{\beta}_p} = 1$. Again, the original 1-form $U(1)$ global symmetry (exists only in a pure gauge theory) has a mixed anomaly with this dual $(d-3)$-form $U(1)$. (In $d=2$, while $W_3^{forbid}$ cannot be defined, one can define the topological theta term $e^{i\widetilde{\Theta} \sum_p f_p/2\pi}$, and discuss the "$(-1)$-form global symmetry" of $\widetilde{\Theta}$ and its mixed anomaly with the 1-form $U(1)$ [63].) And again, dualities can be derived just like in the $S^1$ nl$\sigma$m case; a remarkable case is the electromagnetic duality in 4d [16], which is self dual with $2\pi/\widetilde{e}^2 = e^2/2\pi$ when both charged matter particles and monopoles are forbidden (or both present).

The Villainized $U(1)$ gauge theory can be thought of as gauging a 1-form $\mathbb{Z}$ global symmetry from an $\mathbb{R}$ gauge theory, and then gauge fixing the 1-form $\mathbb{Z}$ by fixing the range of $a_l \in (-\pi, \pi]$. This uses the universal cover central extension

$$\begin{array}{c} 2\pi\mathbb{Z} \to \mathbb{R} \\ \downarrow \\ U(1) \end{array} \, , \tag{23}$$

which is similar to the structure in $S^1$ nl$\sigma$m, except everything is in one higher dimension, and thus the space $S^1$ becomes the group $U(1)$ because consecutive link variables can be naturally composed. We would like to reiterate the conceptual point made at the end of Section 2.1. The configuration space for Villainized $U(1)$ pure gauge theory is $U(1)^{B_1} \times \mathbb{R}^{N_1-B_1} \times \mathbb{Z}^{N_2-(N_1-B_1)}$ rather than the naive $U(1)^{N_1} \times \mathbb{Z}^{N_2}$ or the 1-form $\mathbb{Z}$ gauge fixed $(-\pi, \pi]^{N_1} \times \mathbb{Z}^{N_2}$, where $N_2, N_1$ are the numbers of plaquettes and links, and $B_1$ is the first Betti number. The $U(1)^{B_1}$ factor is the space on which the 1-form $U(1)$ global symmetry acts, while the $\mathbb{Z}^{N_2-(N_1-B_1)}$ factor counts all possible quantized flux and monopole configurations. The appearance of the topological number $B_1$ shows the configuration space is not locally factorized, but this is *not* due to the $U(1)$ gauge invariance (since the space for $U(1)$ gauge redundancy is just $U(1)^{N_0}$ which is local); again this comes from Villainization. Later, we will recast the Villainized $U(1)$ gauge theory in the language of the Lie 2-group $U(1) \times \mathbb{R} \rightrightarrows U(1) \rightrightarrows *$ [34, 35], which is the delooping of the Lie groupoid used for $S^1$ nl$\sigma$m.

All the above are straightforward generalizations from the $S^1$ nl$\sigma$m, by putting everything in one higher dimension. There are also aspects which do not have familiar counterparts in nl$\sigma$m. They are the abelian CS term and abelian instanton.

When monopole is forbidden, the Dirac quantized real-valued flux $f$ is a representative element for the first Chern class $c_1$ in the image of $H^2(|BU(1)|; \mathbb{Z}) \to H^2(|BU(1)|; \mathbb{R})$. Taking the cup product with itself will give an element in the image of $H^4(|BU(1)|; \mathbb{Z}) \to H^4(|BU(1)|; \mathbb{R})$, which is the abelian instanton number.[21]

---

[20]Similar to the situation in footnote 8, a $U(1)$ gauge theory with monopoles forbidden has often been called "non-compact" in the literature, which is confusing, because it is in fact still a compact $U(1)$ gauge theory rather than a non-compact $\mathbb{R}$ gauge theory. The topological distinctions are whether the Wilson loop operators have to have quantized charges, and whether it is possible to have non-zero Dirac quantized fluxes over non-contractible 2d surfaces.

[21]The classifying space of a group $G$ is usually denoted as $BG$, but in this work we will reserve the notation $BG$ for the category obtained by delooping $G$ (see Section 5.1), while the classifying space will be denoted as $|BG|$, the geometric realization of the category $BG$ (see Section 5.4).

More explicitly, on a hypercube $h$ (or 4d cell of other shapes), we can use cup product to define the abelian instanton density over a hypercube [64, 65][22]

$$\mathcal{I}_h := \left( \frac{f}{2\pi} \cup \frac{f}{2\pi} \right)_h . \tag{24}$$

The cup product satisfies the Leibniz rule, so clearly in 5d and above, the instanton non-conservation defect, $d\mathcal{I}$, is proportional to the monopole defect $df/2\pi$. Moreover, in the below, suppose monopoles are forbidden with $W_3^{forbid}$, so that $ds = df/2\pi = 0$ after integrating out the Lagrange multiplier, then we have

$$\mathcal{I}_h = \frac{d\mathcal{C}_h}{2\pi} + (s \cup s)_h , \qquad \mathcal{C}_c := \frac{1}{2\pi}(a \cup da + a \cup 2\pi s + 2\pi s \cup a)_c . \tag{25}$$

Here $\mathcal{C}_c$ is the CS density which will be discussed soon.[23] This equation implies that the total abelian instanton number over a closed oriented 4d spacetime is quantized as expected:

$$I := \oint_{4d} \mathcal{I} = \sum_h \mathcal{I}_h = \sum_h (s \cup s)_h \in \mathbb{Z} . \tag{26}$$

Topological theta term in 4d can hence be defined.[24]

If the 4d spacetime is a spin manifold, it is well-known [19] that the quantization is even stronger: $\sum_h (s \cup s)_h \in 2\mathbb{Z}$ for any $s_p$ satisfying $ds_c = 0$. Therefore, in fermion-related contexts, there is another convention that calls $\mathcal{I}/2$ rather than $\mathcal{I}$ the abelian instanton density, and $I/2$ rather than $I$ the total abelian instanton number.

The CS density $\mathcal{C}_c$ is only well-defined as $e^{i\mathcal{C}_c} \in U(1)$, because under the 1-form $2\pi\mathbb{Z}$ shift $a_l \to a_l + 2\pi n_l$ (which effectively restores the $2\pi$ periodicity of $a_l$) and $s_p \to s_p - dn_p$ that keeps the physical flux $f_p$ invariant, $\mathcal{C}_c$ might shift by $2\pi\mathbb{Z}$. Now, on oriented 3d spacetime (or 3d submanifold embedded in higher dimensional spacetime), one may include another factor in the path integral, the CS phase (recall we supposed monopoles are forbidden):

$$W_{CS}^k := e^{ik \sum_c \mathcal{C}_c} , \tag{27}$$

with any CS level $k \in \mathbb{Z}$. Under $U(1)$ gauge transformation, the CS weight changes by a boundary factor, therefore if the 3d spacetime has a boundary, Dirichlet boundary condition is needed to avoid boundary gauge transformation. It is easy to check, using the expression (25), that a non-trivial CS phase breaks the 1-form $U(1)$ global symmetry of the $U(1)$ gauge field to a $\mathbb{Z}_{2k}$ subgroup,[25] and moreover this 1-form $\mathbb{Z}_{2k}$ global symmetry is anomalous.[26]

---

[22]On a hypercube, one choice of the cup product is the following. Suppose the hypercube has corner vertices given by coordinates $x, y, z, \tau \in \{0, 1\}$. There are a total of six pairs of plaquettes $p$ and $p_f$ on the hypercube, such that the center of $p_f$ is shifted from the center of $p$ by $\hat{x}/2 + \hat{y}/2 + \hat{z}/2 + \hat{\tau}/2$; for example one such pair is the $xy$-plaquette $p$ centered at $x = y = 1/2, z = \tau = 0$ and the $z\tau$-plaquette $p_f$ centered at $x = y = 1, z = \tau = 1/2$. Multiply $f_p f_{p_f}$ for each such pair, and then add up the contributions from all six pairs, we get $(f \cup f)_h$. One can show the cup product satisfies the Leibniz rule under lattice exterior derivative. The choice of cup product is not unique, but any choice is required to satisfy the Leibniz rule. On more general 4d lattice, the choice of cup product is given by a branching structure.

[23]On a cubic lattice, a choice of cup product is defined using the shift $\hat{x}/2 + \hat{y}/2 + \hat{z}/2$, and this choice is compatible with the 4d choice made above when the 3d is embedded in 4d as the $xyz$ hyperplane.

[24]One can also let the theta become local and dynamical, but then for consistency we will need to introduce the Villainization integer field for this theta (on the dual lattice), which will couple to the CS density. This is the lattice axion theory.

[25]To get the factor of 2, we need the property $\beta \cup s = s \cup \beta + d(\cdots)$ when $\beta$ and $s$ are both closed. This $(\cdots)$ is denoted by the cup-1 product $\beta \cup_1 s$, and this is how the notion of higher cup product is motivated.

[26]Which means if a 2-form $\mathbb{Z}_{2k}$ background is introduced, the CS phase will not be gauge invariant under the 1-form $\mathbb{Z}_{2k}$ gauge transformation. It is well-known and easy to check on the lattice [65, 69] that gauging a $\mathbb{Z}_n$

If the 3d spacetime is endowed with a spin structure, then level $k \in \mathbb{Z}/2$ is also possible [19]. The $e^{i\pi}$ ambiguity in $e^{ik\sum_c C_c}$ for half-integer $k$ can be absorbed by an extra fermionic path integral $z_\chi[s] = \pm 1$ that depends on $s_p$ mod 2 as well as a choice of the spin structure [70], so that the well-defined combination valid for any $k \in \mathbb{Z}/2$ is

$$W_{CS}^k := e^{ik\sum_c C_c}(z_\chi[s])^{2k}. \tag{28}$$

The explicit construction of $z_\chi[s]$ can be found in [70] for simplicial complex and in [65] for cubic lattice along with an intuitive Berry phase interpretation. Because of this, in fermion-related contexts, there is another convention that calls $\mathrm{k} := 2k \in \mathbb{Z}$ rather than $k \in \mathbb{Z}/2$ the abelian CS level.

The 3d $U(1)$ Chern-Simons-Maxwell theory on lattice reads [71, 72]

$$Z_{kCS} = \left[\prod_{l'} \int_{-\pi}^{\pi} \frac{da_{l'}}{2\pi}\right] \left[\prod_{p'} \sum_{s_{p'} \in \mathbb{Z}}\right] W_{CS}^k \prod_p W_2(f_p) \prod_c W_3^{forbid}(m_c). \tag{29}$$

It is important to note that the theory becomes ill-defined when the Maxwell weight $W_2$ becomes trivial, $W_2 = 1$, i.e. when one attempts to define a "purely topological CS theory" on the lattice. This problem was originally analyzed in $\mathbb{R}$ gauge theory [73], and stays the same in Villainized $U(1)$ gauge theory.[27] In fact, a "purely topological CS theory" is expected to be impossible, because the gapless chiral boundary mode must be non-topological. So it is natural to include a non-topological Maxwell term [71, 72]. Even in the continuum, a Maxwell term with tiny $1/e^2$ is secretly understood in the regularization of the eta-invariant [75]; when we are on the lattice, the necessity to include a Maxwell term just gets better exposed.

While the CS-Maxwell theory is non-topological, it is a free theory if the Maxwell weight $W_2$ is chosen to be Gaussian as usual. In this case, the CS-Maxwell theory can be solved. It reproduces all the interesting properties from a continuum $U(1)$ CS theory [76], but in an explicit, UV complete fashion. These include: the Wilson loop flux attachment, with the framing interpolating from point-split framing [76] (determined by the cup product) at small $1/e^2$ to geometrical framing [77] (determined by the metric) at large $1/e^2$ [78]; the ground state degeneracy; the chiral boundary mode and, most non-trivially, the associated gravitational anomaly understood in a microscopic exposition. We will present these details in a separate work [71].

---

subgroup of this $\mathbb{Z}_{2k}$ (which means introducing a 2-form $\mathbb{Z}_n$ background field and then promoting this background to dynamical—this will essentially make the $a \cup s + s \cup a$ terms in $\mathcal{C}$ rescale by $1/n$) is equivalent to (after rescaling $a$ by $1/n$—which leads to some unimportant local constant in the path integral measure) dividing the CS level by $n^2$. So only those $\mathbb{Z}_n$ subgroups of this $\mathbb{Z}_{2k}$ where $n^2$ divides $k$ will be non-anomalous. (And if $n^2$ divides $2k$ but not $k$, the theory can be made non-anomalous by introducing fermions; see below.)

[27]First consider the $\mathbb{R}$-valued CS term on the lattice, $\propto \oint_{3d} a \cup da$, $a_l \in \mathbb{R}$. We vary $a$ to find the equation of motion, which means we want $\oint_{3d}(\delta a \cup da|_{\text{EoM}} + da|_{\text{EoM}} \cup \delta a) = 0$ for any $\delta a$. But the two terms are in general unequal, unlike the wedge product in the continuum. Therefore we cannot conclude $da|_{\text{EoM}} = 0$, which means there are undesired zero modes that can be added to any given $da$ configuration while leaving the action invariant. Moreover, unlike the gauge redundancy which occurs at each vertex locally, these extra zero modes have non-local profiles. Thus, they make the Gaussian path integral ill defined.

The problem is the same in Villainized $U(1)$ gauge theory with monopoles forbidden, because locally (though not globally) this theory looks the same as $\mathbb{R}$ gauge theory and hence inherits the same problem: Given any configuration of the gauge flux $f_p \in \mathbb{R}$, it is easy to see any shift $\Delta f$ (due to shifts of $a$ and $s$) that satisfies $d\Delta f_c = 0$ and $\oint_{3d}(\delta a \cup \Delta f + \Delta f \cup \delta a) = 0$ for any $\delta a_l \in \mathbb{R}$ will leave the CS term invariant. The partition function thus diverges, with one infinite factor from each of such non-local zero modes, and the number of such non-local zero modes depends non-locally on all the details of the lattice and the cup product.

Imposing non-local constraints can directly forbid these zero modes [69, 74]. However, in general we want a QFT to be local, and this can be achieved by having a non-topological but local Maxwell term that removes these non-local zero modes [71–73].

## 2.3 More general villainizations, including $\mathbb{Z}_2$ vortex in $\mathbb{R}P^2$ non-linear sigma model, and $\mathbb{Z}_N$ monopole in $PSU(N)$ gauge theory

Villainization has many more applications. The most obvious is to work with multiple $U(1)$, which is useful in studying topological order and Hall conductivity [65,79,80]. Another obvious direction is to work with $q$-form $U(1)$ gauge fields, where $q = 0, 1$ reduce to the previous cases; by the same steps as before, we can derive boson-vortex-type dualities between $q$-form $U(1)$ gauge theory and $(d-q-2)$-form $U(1)$ gauge theory, and demonstrate the associated mixed anomaly between the $q$-form $U(1)$ and $(d-q-2)$-form dual $U(1)$ global symmetries. Interestingly, sometimes Villainization is even useful for dealing with discrete abelian gauge groups for more subtle purposes (compared to our main purpose of avoiding discontinuities): An important case is $2\mathbb{Z} \to \mathbb{Z} \to \mathbb{Z}_2$ for defining spin-c connection in footnotes 33 (also footnote 29); also, $n\mathbb{Z} \to \mathbb{Z} \to \mathbb{Z}_n$ helps manifest the Coulomb phase in $\mathbb{Z}_n$ gauge theory [81]; $n\mathbb{Z}_n \to \mathbb{Z}_{n^2} \to \mathbb{Z}_n$ facilitates a nice lattice implementation of Dijkgraaf-Witten twisted abelian topological order [80,82]; and there are further applications in more exotic models [66].

It is common to develop the impression that Villainization is to deal with $U(1)$, or at most including other abelian groups built out of (or being a subgroup of) $U(1)$. This is not the case. Through our algebraic motivation discussed below (16), it should be clear that the real purpose of Villainization is to capture $\pi_1$, and has nothing to do with whether the symmetry or gauge group is abelian or not. This leads to many more applications.

We begin with nl$\sigma$m, i.e. 0-form theory. Suppose the nl$\sigma$m target space $\mathcal{T}$ has a nontrivial $\pi_1(\mathcal{T}) \cong \Gamma$, with $\Gamma$ some discrete group, not necessarily abelian. To capture the $\Gamma$ winding/vorticity, we can Villainize the traditional $\mathcal{T}$ lattice nl$\sigma$m by the universal cover $\widetilde{\mathcal{T}}$

$$
\begin{array}{c}
\Gamma \to \widetilde{\mathcal{T}} \\
\downarrow \\
\mathcal{T}
\end{array} \tag{30}
$$

Note that $\widetilde{\mathcal{T}}$ does not have to be a group, only $\Gamma$ does. Let us take the $\mathbb{R}P^2$ nl$\sigma$m as an example, which describes the physics of nematicity in systems like liquid crystals. The target space has $\pi_1(\mathbb{R}P^2) \cong \mathbb{Z}_2$, and hence there is $\mathbb{Z}_2$ winding in 1d and $\mathbb{Z}_2$ vorticity in higher dimensions; $\mathbb{R}P^2$ also has higher $\pi_n$'s, but for now we ignore their physical effects and only focus on the $\pi_1$ effects. The structure

$$
\begin{array}{c}
\mathbb{Z}_2 \to S^2 \\
\downarrow \\
\mathbb{R}P^2 \,,
\end{array} \tag{31}
$$

can be implemented on the lattice as an $S^2$ nl$\sigma$m with a $\mathbb{Z}_2$ global symmetry gauged. Thus, the Villainized partition function reads [83]

$$
Z = \left[ \prod_{v''} \int_{S^2} \frac{d^2\hat{n}_{v''}}{4\pi} \right] \left[ \prod_{l'} \sum_{\sigma_{l'}=\pm 1} \right] \prod_{l=\langle v'v \rangle} W_1(\hat{n}_{v'} \cdot \sigma_l \hat{n}_v) \prod_p W_2(D\sigma_p), \tag{32}
$$

where $W_1, W_2$ are some positive increasing functions, and $D\sigma_p := \prod_{l \in \partial p} \sigma_l$ describes the $\mathbb{Z}_2$ vortex. (We can also introduce $W_2^{forbid}$ that uses a $\mathbb{Z}_2$ Lagrange multiplier field to forbid the $\mathbb{Z}_2$ vortex. In 1d, while there is no $W_2$, we can have a topological $\mathbb{Z}_2$ theta term for the $\mathbb{Z}_2$ winding number.) The $\mathbb{Z}_2$ gauge invariance here is $\hat{n}_v \to s_v \hat{n}_v$, $\sigma_{l=\langle v'v \rangle} \to s_{v'} \sigma_l s_v^{-1}$. Note that we have not fixed the $\mathbb{Z}_2$ gauge here, which is fine because it is merely a local, finite factor of 2 on each vertex; this is in contrast to the $\mathbb{Z}$ gauge invariance before, which is of infinite size and hence must be fixed. (If we do want to fix the $\mathbb{Z}_2$ gauge, we can, for instance, require every $\hat{n}_v$

to live on the upper hemisphere which is sufficient to specify a nematic variable living on $\mathbb{R}P^2$.)
With this model, we can understand the point raised before, that in what sense a link variable
takes value in $\widetilde{\mathcal{T}}$ which is not a group in general. Consider a nematic order parameter pointing
along $\pm\hat{n}_v$ at vertex $v$, and focus on, say, its $+\hat{n}_v$ end. Moving along the link $l$, this end will
gradually move and reach some other direction in $S^2$, denoted by $\sigma_l\hat{n}_{v'} \in S^2$; correspondingly,
the $-\hat{n}_v$ end will move and reach $-\sigma_l\hat{n}_{v'}$.

Next we move to gauge theory, i.e. 1-form theory. The mathematical structure for Vil-
lainization is the central extension of a group $G$, not necessarily abelian, to its universal cov-
ering group:

$$\begin{array}{c} \Gamma \to \widetilde{G} \\ \downarrow \\ G \end{array} \quad . \tag{33}$$

Here $\Gamma$ has to be abelian because the $\Gamma$-valued field lives on plaquettes, and the composition of
adjacent plaquettes has no specified order, unlike the links. An important example is $PSU(N)$
lattice gauge theory, which contains $\mathbb{Z}_N$ monopoles [33]. Recall $PSU(N) := SU(N)/Z(SU(N))$
where the center of $SU(N)$ is $Z(SU(N)) = e^{i2\pi\mathbb{Z}_N/N}\mathbf{1}_{N\times N} \cong \mathbb{Z}_N$. Note that $PSU(N)$ has higher
$\pi_n$'s inherited from $SU(N)$ (the next non-trivial one being $\pi_3$ inherited from $SU(N)$, which is
the main problem we will tackle in the work), but here we only focus on the $\pi_1$ physics, which
arises from the mod-out of the center. The structure

$$\begin{array}{c} \mathbb{Z}_N \to SU(N) \\ \downarrow \\ PSU(N), \end{array} \tag{34}$$

can be implemented on the lattice as an $SU(N)$ gauge theory with the 1-form $Z(SU(N)) \cong \mathbb{Z}_N$
global symmetry gauged.

We first briefly review the traditional $SU(N)$ lattice gauge theory defined by Wilson [1–3].
The dynamical d.o.f. is $g_l \in SU(N)$ at each link $l$, thought of as a Wilson line along the link.
The path integral is weighted by a plaquette weight which is a positive, increasing function
of $(\mathbf{tr}Dg_p + c.c.)$, where the $SU(N)$ flux $Dg_p$ is the Wilson loop around a plaquette, i.e. the
ordered product of the $g_l$'s around $p$ starting from some chosen vertex:

$g_{43}$      $g_{14}$   $g_{32}$     $Dg_p := g_{14}g_{43}g_{32}g_{21}$       $g_{21}$

$$Dg_p := g_{14}g_{43}g_{32}g_{21} \tag{35}$$

Figure 8: Definition of $Dg_p$.

(for abelian group, $De^{ia}{}_p = e^{ida_p}$). Gauge transformation $g_l \to h_{v'}g_lh_v^{-1}$ (where $l = \langle v'v\rangle$)
changes $Dg_p$ by a conjugation, hence the weight remains invariant.[28] Similarly, choosing
another starting vertex only changes $Dg_p$ by a conjugation, which does not change the weight;
the starting vertex can even be located away from the plaquette, as long as we conjugate the
flux by a suitable Wilson line. The flux $Dg_p$ satisfies $DDg_c = 1$ (lattice version of Bianchi
identity) on any cube $c$, where the definition and why it equals 1 is illustrated by the picture

---

[28]Therefore, the weight does not have to depend on $Dg_p$ through the trace, but through any function of the
eigenvalues.

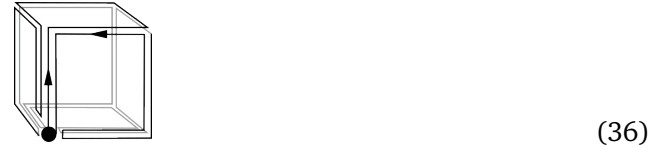

$$(36)$$

Figure 9: Definition of $DDg_c$, and why it equals $\mathbf{1}$ is self-evident.

A conceptual point, similar to that regarding the $\widetilde{\mathcal{T}}$-valued link variable before, is that now we have a $SU(N)$-valued plaquette variable $Dg_p$, which might seem problematic, because plaquette variables should be abelian as mentioned above. The solution is, this is not problematic because $Dg_p$ is not an *independent* plaquette variable, it is defined via link variables starting from a chosen vertex, and composition between plaquettes can be defined accordingly using the conjugation of Wilson lines built out of link variables (for example see the pictorial definition of $DDg$).

The flux $Dg_p$ respects the 1-form global symmetry $g_l \to g_l z_l$ for $z_l \in Z(SU(N)) \cong \mathbb{Z}_N$ satisfying $Dz_p = 1$. This is what we will gauge, in order to obtain the Villainized $PSU(N)$ gauge theory [33]:

$$Z = \left[ \prod_{l'} \int_{g_{l'} \in SU(N)} \right] \left[ \prod_{p'} \sum_{\sigma_{p'} \in Z(SU(N))} \right] \prod_p W_2(\mathbf{tr}(\sigma_p Dg_p) + c.c.) \prod_c W_3(D\sigma_c). \quad (37)$$

Here $W_2$ and $W_3$ are some positive, increasing functions, and $D(\sigma Dg)_c = D\sigma_c := \prod_{p \in \partial_c} \sigma_p$, with the orientations of $p$ here chosen to be consistent with that of $\partial c$, describes the $\mathbb{Z}_N$ monopole. (We can also use $W_3^{forbid}$ by introducing a $\mathbb{Z}_N$ Lagrange multiplier to forbid the $\mathbb{Z}_N$ monopoles.) We can also define $\mathbb{Z}_N$ total flux (similar to the $\mathbb{Z}$ total flux for $U(1)$ gauge field) and the associated $\mathbb{Z}_N$ topological theta term over a 2d surface [35]. For $N = 2$ the $\mathbb{Z}_N$ total flux represents the second Stiefel-Whitney class of the $PSU(2) \cong SO(3)$ gauge field.[29]

Moving further up to higher form gauge theories, since the $q \geq 2$ form independent variables can only be abelian for reasons explained above, only abelian examples exist, which have already been discussed at the beginning of this subsection.

This concludes what Villainization in its general form can do. It captures the $\pi_1$ of nl$\sigma$m target spaces or gauge groups by taking universal covers. An important step in furthering the understanding of Villainization appeared in [34,35], which turned out to be an important inspiration for our present work. The physical context there is to study the possible low energy phases of Yang-Mills theory. Under this context, the Villainized $PSU(N)$ gauge theory is interpreted in terms of the Lie 2-group $PSU(N) \ltimes SU(N) \rightrightarrows PSU(N) \rightrightarrows *$. Importantly, [34,35] shows the low energy phases of Yang-Mills theory admit more possibilities, which are described by more general Lie 2-groups gauge theories [49], $G \ltimes H \rightrightarrows G \rightrightarrows *$, in which $H$ might not fully cover $G$, leading to the exact sequence $* \to \ker(\tilde{t}) \to H \xrightarrow{\tilde{t}} G \to \mathrm{coker}(\tilde{t}) \to *$. We will review such structure in Sections 5.1.

---

[29] In addition to the $\mathbb{Z}_N$ total flux in 2d, in $d \geq 3$, even if we have used $W_3^{forbid}$ to forbid the $\mathbb{Z}_N$ monopoles, there remains a new piece of interesting topological configuration. We can Villainize $\sigma_p =: e^{i2\pi s_p/N}$ by introducing an $Nh_c \in N\mathbb{Z}$, forming $N\mathbb{Z} \to \mathbb{Z} \to \mathbb{Z}_N$, where the $\mathbb{Z}$ variable $ds_c - Nh_c$ is invariant under $s_p \to s_p + Nn_p, h_c \to h_c + dn_c$. Then $W_3^{forbid}$ is enforcing that there exists some $h_c$ such that $ds_c - Nh_c = 0 \in \mathbb{Z}$. While this implies $h_c$ is closed, it might not be exact in $\mathbb{Z}$ because in general $s_p/N \notin \mathbb{Z}$. Therefore we can have a closed, non-exact $h_c$ topological configuration; for $N = 2$ it represents the third integral Stiefel-Whitney class of the $PSU(2) \cong SO(3)$ gauge field, which will be important in footnote 33.

## 2.4 Spinon-decomposed $S^2$ non-linear sigma model: Berry phase, skyrmion and hedgehog

The Villain model and all its variants capture the $\pi_1$ of continuous-valued fields. There is another type of known examples, the $\mathbb{C}P^1$ representation, also known as the spinon decomposition, of $S^2$ nl$\sigma$m, which captures $\pi_2(S^2) \cong \mathbb{Z}$; this can be generalized to capture $\pi_2$ of more general target spaces such as $\mathbb{C}P^N$. Beyond these examples, there is no more known example that captures higher $\pi_n$'s of continuous-valued fields on the lattice, and we will explain why in Section 3. In this subsection we review how the $\mathbb{C}P^1$ representation works. It will bring up more discussions about the geometrical intuition as well as some technical points, which will be important for our main constructions in Section 4 and beyond.

A traditional $S^2$ nl$\sigma$m on lattice has a unit vector $\hat{n}_v \in S^2$ at each vertex $v$, and the link weight is a positive increasing function $W(\hat{n}_{v'} \cdot \hat{n}_v)$. There are $\pi_2$ topological configurations that cannot be naturally defined in the traditional lattice model: the skyrmion in 2d, and the hedgehog defect in 3d or above (that can be seen as the non-conservation of skyrmion number in the 2d space over time), which are characterized by $\pi_2(S^2) \cong \mathbb{Z}$. In fact, even around a 1d loop there is an important piece of physics that cannot be naturally defined, the Berry phase around the loop, whose $2\pi$ periodicity is due to the same topological information $\pi_2(S^2) \cong \mathbb{Z}$.[30]

Viewing $S^2$ as $\mathbb{C}P^1 := \mathbb{C}^2/\mathbb{C}_* \cong SU(2)/U(1)$ solves this problem [84–87]. This $\mathbb{C}P^1$ representation was originally developed and much more well-known in the continuum context, but on the lattice it becomes more crucial for capturing topology. Algebraically, having seen (16) and its variants, the idea now is obvious: to cover $S^2$ by $U(1) \to SU(2) \to S^2$, and then Villainize the $U(1)$:

$$
\begin{array}{c}
2\pi\mathbb{Z} \to \mathbb{R} \\
\downarrow \\
U(1) \to SU(2) \\
\downarrow \\
S^2
\end{array}
\qquad . \tag{38}
$$

This sequence of fibre bundles leads to the sequence of isomorphisms $\pi_2(S^2) \xrightarrow{\sim} \pi_1(U(1)) \xrightarrow{\sim} \pi_0(\mathbb{Z})$. The Berry phase is captured at the $U(1)$ stage, while the skyrmion and hedgehog are captured at the last stage.

The implementation goes as follows. Across a link $l = \langle v'v \rangle$, we introduce an $SU(2)$ variable $\mathcal{V}_l \in SU(2)$ that rotates $\hat{n}_v$ to $\hat{n}_{v'}$, i.e. subjected to the constraint $R_{\mathcal{V}_l}\hat{n}_v = \hat{n}_{v'}$, where $R_{\mathcal{V}_l}$ is the rotation matrix by casting $\mathcal{V}_l$ in the spin-1 representation; the constraint can be equivalently expressed as $\mathcal{V}_l(\hat{n}_v \cdot \vec{\sigma})\mathcal{V}_l^{-1} = \hat{n}_{v'} \cdot \vec{\sigma}$. This is like the constraint $e^{i\gamma_l}e^{i\theta_v} = e^{i\theta_{v'}}$ in the Villain model. This constraint does not uniquely fix $\mathcal{V}_l$ but leaves a $U(1)$ d.o.f., because after a given rotation we can still make an extra rotation around $\hat{n}_{v'}$ without changing $\hat{n}_{v'}$. In the spin-1/2 representation, the constraint implies $\mathcal{V}_l u_{\hat{n}_v} = e^{-ia_l} u_{\hat{n}_{v'}}$, where $\hat{n}_v = u_{\hat{n}_v}^\dagger \vec{\sigma} u_{\hat{n}_v}$ and $\hat{n}_v \cdot \vec{\sigma} = 2u_{\hat{n}_v} u_{\hat{n}_v}^\dagger - \mathbf{1}$ (this $u_{\hat{n}_v}$ is called *spinon*, which is why this $\mathbb{C}P^1$ representation is also called spinon decomposition), and $e^{ia_l}$ is the said $U(1)$ d.o.f., with $2a_l$ being the extra rotation angle around $\hat{n}_{v'}$. This dynamical $e^{ia_l}$ part is then viewed as a $U(1)$ gauge field, which we will

---

[30]In the previous $S^1$ nl$\sigma$m, in 0d there is also a phase, the $e^{i\theta}$ itself, which is well-defined without Villainization. In $U(1)$ gauge theory, in 1d there is also a phase, the Wilson loop $\prod_l e^{ia_l}$, which is again well-defined without Villainization. So the $S^2$ nl$\sigma$m is the first example where some physical phase requires suitable refinement of the traditional lattice theory to be well-defined.

Villainize as we did in Section 2.2. The partition function reads

$$Z = \left[ \prod_{v'} \int_{S^2} \frac{d^2\hat{n}_{v'}}{4\pi} \right] \left[ \prod_{l'} \int_{-\pi}^{\pi} \frac{da_{l'}}{2\pi} \right] \left[ \prod_{p'} \sum_{s_{p'} \in \mathbb{Z}} \right] \prod_l W_1(\mathbf{tr}\mathcal{V}_l + c.c.) \prod_p W_2(f_p) \prod_c W_3(m_c), \quad (39)$$

where $W_1$ is positive and increasing, $W_2, W_3$ are positive and decreasing with the absolute value of the arguments (or we can use $W_3^{forbid}$ in (22)). (This is for $d \geq 3$. For $d = 2$ we just ignore the $W_3$ part. For $d = 1$ we ignore the $s_p$ field and the $W_2$ and $W_3$ parts, and we can have an extra Berry phase factor, see (42) later.) The skyrmion configuration and hedgehog defect of the $S^2$ d.o.f. are then defined as the Dirac quantized flux and monopole of the $U(1)$ gauge theory. In particle physics, this is the familiar situation of an $SU(2)$ gauge field being Higgsed by an $S^2$ vacua down to a residual $U(1)$ gauge field [88,89]; the constraint $R_{\mathcal{V}_l}\hat{n}_v = \hat{n}_{v'}$ means the massive gauge bosons are set to be infinitely massive.

More explicitly, for $\hat{n}_v$ given by the spherical coordinates $(\theta_v, \phi_v)$, it is common to make a standard choice of $SU(2)$ matrix $\mathcal{U}_{\hat{n}_v}$ whose spin-1 representation would rotate $\hat{z}$ to $\hat{n}_v$:

$$\mathcal{U}_{\hat{n}_v} = e^{-i\sigma^z \phi_v/2} e^{-i\sigma^y \theta_v/2} e^{i\sigma^z \phi_v/2} = \begin{bmatrix} \cos(\theta_v/2) & -e^{-i\phi_v}\sin(\theta_v/2) \\ e^{i\phi_v}\sin(\theta_v/2) & \cos(\theta_v/2) \end{bmatrix} = \left[ u_{\hat{n}_v} \quad -i\sigma^y u_{\hat{n}_v}^* \right]. \quad (40)$$

Then $\mathcal{V}_l$ can be parametrized as $\mathcal{V}_l = \mathcal{U}_{\hat{n}_{v'}} e^{-i\sigma^z a_l} \mathcal{U}_{\hat{n}_v}^{-1}$, where $e^{-ia_l} \in U(1)$ is a new dynamical variable not fixed by the constraint $R_{\mathcal{V}_l}\hat{n}_v = \hat{n}_{v'}$; indeed, it manifests in $\mathcal{V}_l u_{\hat{n}_v} = e^{-ia_l} u_{\hat{n}_{v'}}$. This is like fixing $\theta_v \in (-\pi, \pi]$ in the Villain mode and writing $\gamma_l = \theta_{v'} + 2\pi m_l - \theta_v$. If we change the standard choice of $\mathcal{U}_{\hat{n}_v}$ by a $U(1)$ gauge transformation $\mathcal{U}_{\hat{n}_v} e^{i\psi_v \sigma^z}$, accompanying it by $a_l \to a_l + d\psi_l$ leaves $\mathcal{V}_l$ unchanged. The apparent singularity in our gauge choice (40) at $\theta_v = \pi$ (since $e^{i\phi_v}$ becomes ambiguous there) is like the apparent but not physically harmful discontinuity between $\theta_v = \pm\pi$ in the Villain model—$\mathcal{V}_l$ has nothing singular, just like $\gamma_l$ has nothing discontinuous in the Villain model.[31]

We claim that $e^{ia_l}$ is naturally interpreted as the Berry connection across the link, so that the Berry phase $e^{i\Phi}$ around a loop is defined by

$$e^{i\Phi} := e^{i\oint_{1d} a} = \prod_l e^{ia_l}, \qquad \text{or equivalently} \quad (41)$$

$$e^{-i\vec{\sigma}\cdot\hat{n}_v \Phi} := \prod_l \mathcal{V}_l \qquad \text{(path ordered, starting and ending at any } v \text{ on the loop),}$$

and the $U(1)$ Berry curvature is the Berry phase around a single plaquette, $e^{if_p} := e^{ida_p}$, or equivalently $e^{-i\vec{\sigma}\cdot\hat{n}_v f_p} = D\mathcal{V}_p$. For now we will accept this Berry connection interpretation and talk about its consequences; the main task of the later part of this subsection, (43) and below, is to understand this key claim.

With this Berry connection interpretation, a 1d theory weighted by the Berry phase reads

$$Z_{qBerry} = \left[ \prod_{v'} \int_{S^2} \frac{d^2\hat{n}_{v'}}{4\pi} \right] \left[ \prod_{l'} \int_{-\pi}^{\pi} \frac{da_{l'}}{2\pi} \right] e^{iq\Phi} \prod_l W_1(\mathbf{tr}\mathcal{V}_l + c.c.), \quad (42)$$

---

[31]If we still want to remove this unharmful singularity, we can simply leave the $U(1)$ gauge unfixed, as it only contributes a finite factor at each vertex. Thus, instead of sampling $\hat{n}_v \in S^2$ at each vertex, we will sample $\mathcal{U}_{\hat{n}_v} \in SU(2)$, which corresponds to multiplying an arbitrary $e^{i\sigma^z \psi_v/2}$ to the right of our standard choice (40). This is the common practice in the numerical implementation of [84,85]. On the other hand, in most implementations, the subsequent Villainization step is not adapted, despite the existing proposals [86,87].

for any $q \in \mathbb{Z}$. Viewing this 1d system as the worldloop of a spin, this is actually the simplest non-trivial case of putting a *coadjoint orbit theory* [76] onto the lattice. We will discuss more about coadjoint orbit theories on lattice in subsequent works. For odd $q$, the $SO(3)$ global symmetry of the spin becomes anomalous unless extended [90] to $SU(2)$,[32] and the interpretation is familiar: the total Berry phase over the sphere being $2\pi q$ means the spin is $q/2$.

The skyrmion in 2d is also easily understood. In the continuum, the skyrmion configuration is when the configuration of $\hat{n}$ over an oriented closed 2d surface wraps around the target space $S^2 \ni \hat{n}$; the Berry curvature can be regarded as $2\pi$ times the skyrmion density, so that the Dirac quantized flux of Berry curvature over the 2d space is $2\pi$ times the total skyrmion number. On the lattice, the $U(1)$ Berry curvature over a plaquette is Villainized as $f_p := da_p + 2\pi s_p \in \mathbb{R}$ (recall Section 2.2), which is $2\pi$ times the skyrmion density over the plaquette, on which the $W_2$ weight in (39) depends. The total skyrmion number $\sum_p f_p/2\pi = \sum_p s_p \in \mathbb{Z}$ is manifestly an integer. A topological theta term in 2d can thus be defined if desired.

In 3d or higher, the hedgehog defect is a skyrmion around a single cube, and hence counted by the Berry curvature monopole $m_c = df_c/2\pi = ds_c$. If we use $W_3^{forbid}$ in (22) to forbid the hedgehogs, there will be a dual $(d-3)$-form $U(1)$ global symmetry, and we can explicitly see on the lattice that it has the celebrated mixed anomaly with the 0-form $SO(3)$ global symmetry of the $S^2$.[33]

Now we shall elaborate on the key claim made above (41), that $e^{ia_l}$ should be understood as the lattice Berry connection. Consider the spinon decomposed link weight $W_1$ (ignoring the $W_2, W_3$ dependence on $a_l$ for now). It should be related to the weight $W$ in traditional lattice

---

[32]To see this, we introduce a background gauge field $V_l \in SU(2)$, which appears in $W_1$ as $\mathbf{tr}\mathcal{V}_l \to \mathbf{tr}(\mathcal{V}_l V_l^\dagger)$. If the background is $SO(3) \cong PSU(2)$ instead of $SU(2)$, then $V_l$ and $-V_l$ must be equivalent, and this is realized by absorbing an $e^{i\pi}$ shift into $e^{ia_l}$, which leaves $Z_{qBerry}$ invariant only for even $q$.

[33]This anomaly can be seen by introducing an $SO(3)$ background gauge field and finding that any consistent modification to the definition of the hedgehog breaks the dual $U(1)$. Alternatively, it can be seen by introducing a dual $U(1)$ background (Villainized) and finding that along its background Dirac string there is a $q = 1$ Berry phase integral (41), extending [90] the $SO(3)$ global symmetry to $SU(2)$ according to footnote 32. Below we focus on the first route.

The $SO(3)$ background gauge field appears in $W_1$ as $\mathbf{tr}\mathcal{V}_l \to \mathbf{tr}(\mathcal{V}_l V_l^\dagger)$, where the background field $V_l \in SU(2)$. But since the background should really be $SO(3) \cong PSU(2)$ rather than $SU(2)$, somehow $V_l$ and $-V_l$ must be equivalent. In $W_1$ we can absorb this sign ambiguity into $e^{ia_l}$. But then in $W_2$, the flux $f_p$ is ambiguous by $\pi$. The solution is to introduce a 2-form $\mathbb{Z}_2$ background $S_p \in \mathbb{Z}$ mod 2 that absorbs this ambiguity, so that the modified flux $f_p - \pi S_p$ is unambiguous—note that, in turn, the $2\mathbb{Z}$ ambiguity in $S_p$ needs to be absorbed by $s_p$. This $f_p - \pi S_p$ will be the argument of $W_2$. What has happened is that the 2-form $\mathbb{Z}_2$ background $S_p$ effectively reduces the 1-form $SU(2)$ background $V_l$ to $PSU(2) \cong SO(3)$, just like in (34) for dynamical gauge fields.

Interestingly, the skyrmion number $\oint_{2d}(f - \pi S)/2\pi$ becomes half-quantized. This is true even if we have demanded $S_p$ to be $\mathbb{Z}_2$ closed, i.e. $dS_c = 0$ mod 2, because the $\oint_{2d}$ can be around a non-contractible 2d surface. In fact, $\oint_{2d} S/2 := \sum_p S_p/2$ mod 1 characterizes the second Stiefel-Whitney class of the $SO(3)$ background. The flux $f_p - \pi S_p$ is no longer a $U(1)$ gauge flux, but a spin-c gauge flux associated with the $SO(3)$ background. Therefore in 2d we can have a non-trivial $\mathbb{Z}_2$ topological theta term coupled to this half-quantized spin-c flux, realizing the celebrated Haldane quantum spin chain phase [90, 91].

In $W_3$, $df_c/2\pi = ds_c$ is no longer a good hedgehog, because as we said, the $2\mathbb{Z}$ ambiguity in $S_p$ must be absorbed by $2s_p$. The unambiguous hedgehog defect should become $m_c := d(f - \pi S)_c/2\pi = ds_c - dS_c/2$; this is still an integer if we have demanded $dS_c = 0$ mod 2. In fact, it is better to describe the condition $dS_c = 0$ mod 2 along the lines of footnote 29, i.e. to introduce a $2H_c \in 2\mathbb{Z}$ background to form $2\mathbb{Z} \to \mathbb{Z} \to \mathbb{Z}_2$, such that the combination $dS_c - 2H_c \in \mathbb{Z}$ is unambiguous and is enforced to be 0 everywhere. Then the good hedgehog is $m_c := ds_c - H_c$. While $H_c = dS_c/2$ shows $H_c$ is closed, it might not be exact in $\mathbb{Z}$ since in general $S_p/2 \notin \mathbb{Z}$. This is nothing but a representative of the third integral Stiefel-Whitney class of the $SO(3)$ background. (Note that demanding the third integral Stiefel-Whitney class to vanish is a non-local condition and hence unphysical, in contrast to the previous closedness condition $dS_c = 0$ mod 2 which is local.) When the third integral Stiefel-Whitney class is indeed non-trivial, if we still use $W_3^{forbid}$ for $W_3$, the dual $U(1)$ global symmetry is explicitly broken, leading to a vanishing partition function, because there is no solution of $s_p$ that can make $m_c = ds_c - H_c$ vanish everywhere. This is the familiar fact that a spin-c gauge field cannot be free from monopole if the associated third integral Stiefel-Whitney class is non-trivial.

$S^2$ nl$\sigma$m as

$$W(\hat{n}_{v'} \cdot \hat{n}_v) \approx \int_{-\pi}^{\pi} \frac{da_l}{2\pi} W_1(\mathbf{tr}\mathcal{V}_l + c.c.),\tag{43}$$

in analogy to (7). It is technically useful to express $\mathbf{tr}\mathcal{V}_l$ in terms of the spinons $u_{\hat{n}_v}$:

$$\mathbf{tr}\mathcal{V}_l = \mathbf{tr}\mathcal{V}_l^\dagger = e^{ia_l} u_{\hat{n}_{v'}}^\dagger u_{\hat{n}_v} + c.c.\tag{44}$$

which turns out to be interpretable as the hopping of spinons. The dominating contribution to $W_1$ comes from the saddle $e^{ia_l} \approx u_{\hat{n}_v}^\dagger u_{\hat{n}_{v'}}/|u_{\hat{n}_v}^\dagger u_{\hat{n}_{v'}}|$. When $\hat{n}_v$ and $\hat{n}_{v'}$ are close to each other, we have $a \approx -iu^\dagger du$ at the saddle, recovering the familiar expression of Berry connection in the continuum. On the other hand, $|u_{\hat{n}_v}^\dagger u_{\hat{n}_{v'}}|$ control the fluctuation of $e^{ia_l}$ away from the saddle, and in particular, when $\hat{n}_v$ and $\hat{n}_{v'}$ are nearly opposite to each other, $u_{\hat{n}_{v'}}^\dagger u_{\hat{n}_v} \to 0$ and $W_1$ becomes insensitive to $e^{ia_l}$.

It is important to develop a geometrical understanding of the lattice Berry connection, in analogy to (9). Again think of the link $l$ as a path embedded in the continuum. If we have a continuum field configuration, $\hat{n}(x \in l)$ will trace out a path in the target space $S^2$. Of course, the paths in $S^2$ running from $\hat{n}_v$ to $\hat{n}_{v'}$ can take all kinds of shapes and form an infinite dimensional space, but we should "truncate away the unimportant details of how a generic path wiggles", and only keep an important $U(1)$ piece of information—from the continuum theory, we know that what really matters for defining the skyrmions and hedgehogs is the Berry curvature, and this is the piece of information we will keep:

- We view two continuum paths from $\hat{n}_v$ to $\hat{n}_{v'}$ as equivalent as long as the $U(1)$ Berry phase (half of the solid angle) bounded between them is 0, for instance

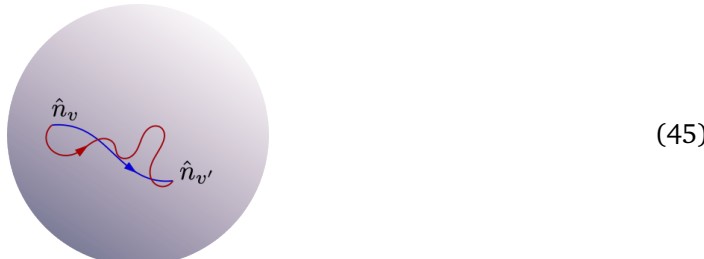

$$\tag{45}$$

Figure 10: Two paths that are considered equivalent for bounding 0 Berry phase.

More generally, when we consider the difference between two continuum paths from $\hat{n}_v$ to $\hat{n}_{v'}$, we only care about the $U(1)$ Berry phase (half of the solid angle) bounded between them. Therefore, the space of the equivalence classes of paths from $\hat{n}_v$ to $\hat{n}_{v'}$ forms $U(1)$, and this is what the different values of $e^{ia_l}$ represents.

Topologically, the space of all paths interpolating from $\hat{n}_v$ to $\hat{n}_{v'}$, though infinite dimensional, can be easily seen to have a $\pi_1 \cong \mathbb{Z}$, if we think of a path as a rubber band and wrap it around the sphere. After taking the equivalence relation as above, $\pi_1(U(1)) \cong \mathbb{Z}$ retains this piece of topological information. If only the starting point $\hat{n}_v$ of the paths is specified while the ending point $\hat{n}_{v'}$ is arbitrary, then the space of equivalence classes of paths forms $SU(2) \ni \mathcal{V}_l$, a non-trivial $U(1)$ bundle over $S^2 \ni \hat{n}_{v'}$. (We will have a slightly more formal discussion later at (48).)

Having explained the intended geometrical meaning of $e^{ia_l}$ and $\mathcal{V}_l$, let us now see that the saddle and fluctuations (44) in $W_1$ makes physical sense. $W_1(\mathcal{V}_l)$ on the lattice can be thought of as representing the (exponentiated) free energy of the equivalence class of continuum paths that $\mathcal{V}_l \in SU(2)$ collectively represents, and we expect that, given $\hat{n}_v$ and $\hat{n}_{v'}$, the weight should be maximized when the equivalence class contains the shortest geodesic between the given points. Here we draw a black curve representing the shortest geodesic connecting $\hat{n}_v$ and $\hat{n}_{v'}$ (the great circle on which they lie is indicated):

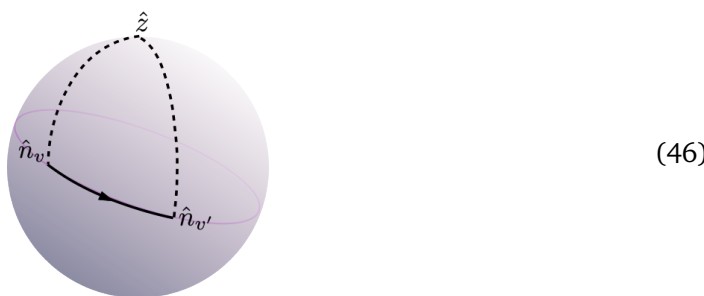

(46)

Figure 11: Saddle of lattice Berry connection.

The dashed curves are the shortest geodesics connecting each point to $\hat{z}$. It turns out that the phase $u_{\hat{n}_{v'}}^\dagger u_{\hat{n}_v}/|u_{\hat{n}_{v'}}^\dagger u_{\hat{n}_v}|$, which sets the saddle point for the lattice Berry connection $e^{ia_l}$, is equal to the continuum Berry phase, i.e. half of the solid angle, bounded by the triangular region.[34,35] The gauge dependence of the Berry connection is associated with the choice of the dashed curves, and here we are under the gauge choice (40). When we compute the Berry phase around a loop on the lattice, the dependence on the choice of dashed curves cancel out anyways, so what the saddle point $u_{\hat{n}_{v'}}^\dagger u_{\hat{n}_v}/|u_{\hat{n}_{v'}}^\dagger u_{\hat{n}_v}|$ really encodes geometrically is the shortest geodesic connecting $\hat{n}_v$ and $\hat{n}_{v'}$, in agreement with our physical intuition.

On the other hand, in (44), the fluctuation of $e^{ia_l}$ around the saddle point is constrained by $|u_{\hat{n}_v}^\dagger u_{\hat{n}_{v'}}| = \sqrt{(\hat{n}_v \cdot \hat{n}_{v'} + 1)/2}$, i.e. the larger the angular separation, the more fluctuation there will likely be, and when $\hat{n}_v, \hat{n}_{v'}$ become opposite, the fluctuation becomes arbitrary. This also agrees with our physical intuition.

Through (46), we can notice that the saddle point $u_{\hat{n}_{v'}}^\dagger u_{\hat{n}_v}/|u_{\hat{n}_{v'}}^\dagger u_{\hat{n}_v}|$ of the $e^{ia_l}$ fluctuation in $W_1$ sees two kinds of singularities, one is an artifact, while the other is meaningful and suitably handled:

---

[34]This equality is established by two facts: 1) If the separation of $\hat{n}_1$ and $\hat{n}_2$ is infinitesimal, then the infinitesimal phase $\arg\left(u_{\hat{n}_2}^\dagger u_{\hat{n}_1}\right)$ is by definition the continuum Berry connection (whose expression is, familiarly, $a = -iu^\dagger du = (1 - \cos\theta)d\phi/2$ under the gauge (40)) as we said below (44). 2) If $\hat{m}$ is any point on the same great circle as $\hat{n}_1$ and $\hat{n}_2$, then we have the additivity of the phases $\arg\left(u_{\hat{n}_2}^\dagger u_{\hat{n}_1}\right) = \arg\left(u_{\hat{n}_2}^\dagger u_{\hat{m}}\right) + \arg\left(u_{\hat{m}}^\dagger u_{\hat{n}_1}\right)$; this second fact is proven by parametrizing $\hat{m} = \alpha\hat{n}_1 + \beta\hat{n}_2$ with real numbers $\alpha, \beta$ (since they live on the same plane) and then evaluating $\left(u_{\hat{n}_2}^\dagger u_{\hat{m}}\right)\left(u_{\hat{m}}^\dagger u_{\hat{n}_1}\right)$ using (40) and this parametrization of $\hat{m}$—a straightforward calculation shows $\left(u_{\hat{n}_2}^\dagger u_{\hat{m}}\right)\left(u_{\hat{m}}^\dagger u_{\hat{n}_1}\right) = \left(u_{\hat{n}_2}^\dagger u_{\hat{n}_1}\right)(1 + \alpha + \beta)/2$. Now that we have the facts 1) and 2), we can cut the black geodesic in (46) into many infinitesimal segments so that, by 2), the phase $\arg\left(u_{\hat{n}_{v'}}^\dagger u_{\hat{n}_v}\right)$ is the sum of such phases from all the segments, and by 1), the phase of each segment is same as the continuum Berry connection, and therefore the total phase $\arg\left(u_{\hat{n}_2}^\dagger u_{\hat{n}_1}\right)$ indeed agrees with the continuum Berry phase, i.e. half of the solid angle bounded.

[35]To have another intuitive demonstration, if we denote the angular separation $\lambda = \arccos(\hat{n}_v \cdot \hat{n}_{v'})$, then $|u_{\hat{n}_v}^\dagger u_{\hat{n}_{v'}}| = \cos(\lambda/2)$. Thus, when $e^{ia_l}$ is at the saddle point $u_{\hat{n}_{v'}}^\dagger u_{\hat{n}_v}/|u_{\hat{n}_{v'}}^\dagger u_{\hat{n}_v}|$ so that $\mathrm{tr}\mathcal{V}_l$ in (44), and hence $W_1$, is maximized, we have $\mathrm{tr}\mathcal{V}_l = 2|u_{\hat{n}_v}^\dagger u_{\hat{n}_{v'}}| = e^{i\lambda/2} + e^{-i\lambda/2}$, which means the amount of rotation made by $R_{\mathcal{V}_l} \in SO(3)$ is indeed the angular separation $\lambda$, i.e. the black solid geodesic in (46) is the path along which $\hat{n}$ will be brought along if the amount of rotation gradually increase from 0 to $\lambda$ in $R_{\mathcal{V}_l}$.

- The first kind of singularity occurs when either $\hat{n}_v$ or $\hat{n}_{v'}$ is in the vicinity of $-\hat{z}$. In this case, the associated dashed curve and hence the solid angle changes rapidly. But this is an artifact due to our standard gauge choice (40) for $\mathcal{U}_{\hat{n}}$; in the gauge invariant Berry phase, the dependence on the dashed paths will cancel out anyways. Such kind of artifact is unavoidable if we use $U(1)$ gauge fixed $\mathcal{U}_{\hat{n}_v}$ and $e^{ia_l}$[36] to parametrize $\mathcal{V}_l$, because $SU(2)$ is a non-trivial $U(1)$ bundle over $S^2$. This is like, in the Villain model, the most probable choice of $m_l$ will jump when either $\theta_v$ or $\theta_{v'}$ moves across $\pm\pi$ if the $\mathbb{Z}$ gauge is fixed to $\theta_v \in (-\pi, \pi]$, but $\gamma_l$ does not jump. We would also like to remark that the Berry phase (41) can indeed be defined from the gauge invariant $\mathcal{V}_l$ directly, without referring to the $\mathcal{U}_{\hat{n}_v}$ and $e^{ia_l}$ parametrization.

- The second kind of singularity occurs when $\hat{n}_v$ and $\hat{n}_{v'}$ are nearly opposite. In this case, the black solid geodesic between them changes rapidly; as the two points become exactly opposite, there is no unique choice of the shortest geodesic, hence no unique choice for the most probable $e^{ia_l}$. Such singularity occurring in the most probable choice of $e^{ia_l}$ does not mean any physical observable or any weight becomes singular. Rather, it simply means all equivalence class of paths become equally probable, as we should expect when $\hat{n}_v$ and $\hat{n}_{v'}$ become opposite. Indeed, $\mathbf{tr}\mathcal{V}_l$ and hence the weight $W_1$ becomes insensitive to $e^{ia_l}$ as $|u_{\hat{n}_{v'}}^\dagger u_{\hat{n}_v}| \to 0$. This is like, in the Villain model, when $e^{i\theta_v}$ and $e^{i\theta_{v'}}$ become opposite, $\gamma_l$ taking $\pm\pi$ become equally probable.

Such suitable understanding of the apparent singularities in the Berry connection saddle is crucial for establishing the important topological fact that the link Berry connection d.o.f. lives on a non-trivial $U(1)$ bundle, i.e. $SU(2)$, over the space $S^2$ of the vertex d.o.f..[37]

It is straightforward to generalize this construction to $\mathbb{C}P^N$ nl$\sigma$m. The spinon $u_v$ will become an $(N+1)$-component complex unit vector taking value in $S^{2(N+1)-1}$, and in a $S^{2(N+1)-1}$ nl$\sigma$m the link weight is a function of $u_{v'}^\dagger u_v + c.c.$. Gauging the $U(1)$ phase global symmetry leads to the spinon-decomposed $\mathbb{C}P^N$ nl$\sigma$m with $u_{v'}^\dagger, e^{ia_l} u_v + c.c.$, and the $U(1)$ gauge field $e^{ia_l}$ is then Villainized. Generalizations can also be made to capture the $\pi_2$ of spaces beyond $\mathbb{C}P^N$.

A more interesting direction of generalization is to consider $\mathbb{R}P^2 \cong S^2/\mathbb{Z}_2$. In Section 2.3 we have captured its $\pi_1$, and now we can capture its $\pi_2$ inherited from the $S^2$. The spinon decomposition becomes $\mathbb{Z}_2 \ltimes U(1) \to SU(2) \to \mathbb{R}P^2$, where the gauge group is no longer abelian. Upon further Villainizing the $U(1)$ part, the $\pi_1 \cong \mathbb{Z}_2$ will act on the $\pi_2 \cong \mathbb{Z}$ by flipping the sign, leading to a non-trivial 2-group structure [92]. We will discuss such interplay between different $\pi_n$'s in subsequent works.

Finally, to prepare for the next section, we would like to cast the geometrical understandings (9) and (45) in more formal terms, relating them to the algebraic understandings (16) and (38) respectively.

Let $\mathcal{P}_*X$ and $\Omega_*X$ be the pointed path space and pointed loop space of $X$ respectively, i.e. the spaces of (parameterized) paths and loops starting from a given point. The space of all paths emanating from a given starting point is by definition $\mathcal{P}_*X$, while the space of all paths

---

[36]In numerical practices it is common not to fix the $U(1)$ gauge, but then these $U(1)$ gauge fluctuations which does not lead to any physics will cost numerical resources.

[37]More mathematically, given $\hat{n}_v$, specifying the $U(1)$ gauge of when $e^{ia_l} = 1$ is finding a section of $S^2 \ni \hat{n}_{v'}$ in $SU(2) \ni \mathcal{V}_l$—for the gauge choice (40), the section is specified by the equivalence class of paths that contains the path obtained by joining the two dashed curves in (46). Since $SU(2)$ is a non-trivial $U(1)$ bundle over $S^2$, there can be no global section, and the singularities developed encode the topology of the bundle; this is the familiar Dirac string story, placed along $-\hat{z}$ in this gauge—indeed, when either of $\hat{n}_v, \hat{n}_{v'}$ approaches $-\hat{z}$, the associated dashed curve becomes ambiguous. This explains the first kind of singularities. To explain the second kind, given $\hat{n}_v$, finding the saddle point for $e^{ia_l}$ for each possible $\hat{n}_{v'}$ is finding another section of $S^2 \ni \hat{n}_{v'}$ in $SU(2) \ni \mathcal{V}_l$, and this section is specified by the equivalence class of paths that contains the shortest geodesic in (46). Again the singularities developed encode the topology of the bundle, and for this section it occurs at $\hat{n}_{v'} = -\hat{n}_v$.

given both the starting and ending points can be identified with $\Omega_* X$, where the loop is formed by returning from the ending point to the starting point via some standard choice of path. Then obviously, for the $S^1$ Villain model,

$$\mathcal{P}_* S^1 \xrightarrow{length} \mathbb{R} \ni \gamma_l, \qquad \Omega_* S^1 \xrightarrow{length} 2\pi\mathbb{Z} \ni 2\pi m_l, \tag{47}$$

by taking the (signed) length of the image of the path in (9). Hence the Villainization fibre bundle $2\pi\mathbb{Z} \to \mathbb{R} \to S^1$ can be naturally recognized as $\Omega_* S^1 \to \mathcal{P}_* S^1 \to S^1$ after taking the (signed) length.

For $S^2$, we not only need to think about the continuum paths traced out by the links, but also the continuum surfaces—paths in the space of paths—swept out by the plaquettes. The intuitive discussion above suggests that the key information on a continuum surface swept out by a plaquette is (half of) its solid angle, being the integral of the continuum Berry curvature over the surface. This means the continuum field is reduced to the lattice field via

$$
\begin{array}{ccc}
\Omega_*^2 S^2 \to \mathcal{P}_* \Omega_* S^2 & \xrightarrow{\int_{2d} \text{Berry}} & 2\pi\mathbb{Z} \to \mathbb{R} \\
\downarrow & & \downarrow \\
\Omega_* S^2 \to \mathcal{P}_* S^2 & & U(1) \to SU(2) \\
\downarrow & & \downarrow \\
S^2 & & S^2.
\end{array}
\tag{48}
$$

The space of (topologically trivial) 2d surfaces emanating from a given path between two given points is homeomorphic to $\mathcal{P}_* \Omega_* S^2$; an element of it, geometrically a 2d disk on $S^2$, is thought of as being swept out by the continuum fields on a plaquette embedded in the continuum. Integrating such a surface with the continuum Berry curvature results in a value in $\mathbb{R}$; in particular, $\Omega_*^2 S^2$, the space of closed 2d spheres on $S^2$, indeed maps to $2\pi\mathbb{Z}$—the $\pi_2$ winding number which is equal to the quantized total Berry phase. This explains the map at the top row, which are fields on the plaquettes. Induced from that, a loop on $S^2$, i.e. an element of $\Omega_* S^2$, formed by an arbitrary path and a standard choice of "returning path" connecting two given points, is then mapped to the $U(1)$ Berry phase bounded by these two paths (the $2\pi\mathbb{Z}$ part is not determined because the bounding surface is not specified), and this is interpreted as the Berry connection; the dependence on the standard choice of "returning path", such as the dashed curves in (46), corresponds to the gauge dependence of the Berry connection. Consequently, an arbitrary path emanating from a given starting point is mapped to an equivalence class of paths according to (45), and they form $SU(2)$:

$$SU(2) \cong \mathcal{P}_* S^2 \times U(1)/Berry, \tag{49}$$

where the latter space means two elements of $\mathcal{P}_* S^2 \times U(1)$ are considered equivalent if the two paths in $\mathcal{P}_* S^2$ share the same starting and ending points, and moreover they bound a Berry phase that is equal to the difference in the two $U(1)$ phases.

In our main problem later, we will no long have the familiar language of Lie groups and fibre bundles, but such a picture of "truncating away the unimportant details" from the infinite dimensional space of continuum fields is what we need in order to understand how to think about and work with the more flexible yet unfamiliar categorical structures.

## 3 Difficulty beyond the known examples

The examples reviewed in Section 2 have all been worked out before. Yet what we uncovered through our review is the *relation* between the examples. They are not scattered; rather, they

are organized by the same rationale, capable of capturing the $\pi_1$ and $\pi_2$ of target spaces (or gauge groups), and moreover the rationale makes connection to the continuum.

With this, we can now understand why similar efforts trying to capture $\pi_{n\geq 3}$ (such as skyrmion in pion nl$\sigma$m and instanton in Yang-Mills) onto the lattice have not been successful. It is not because of bad luck. It will become clear in this section that it is mathematically impossible to achieve this goal within the familiar languages of Lie groups and fibre bundles. More flexible mathematical structures become necessary. These structures are not so easy to come up with by regular attempts. Or, even if someone comes up with them by a good strike physical intuition, the structures might seem "not mathematically nice enough" to be taken seriously. However, it turns out that, more systematic mathematical considerations will actually naturally lead to these structures, which will end up being physically intuitive.

Let us now think about $S^3$ nl$\sigma$m, which can describe the pion vacua. The skyrmion configuration is now over the 3d space, characterized by $\pi_3(S^3) \cong \mathbb{Z}$, and represents the baryons over the pion vacua [93]. The hedgehog defect in 4d represents the non-conservation of baryons, which we might want to be able to forbid on the lattice. Over a 2d space, we can also define a $U(1)$ phase, the Wess-Zumino-Witten (WZW) term, just like the Berry phase in 1d in $S^2$ nl$\sigma$m. Of course, $S^3$ also has higher $\pi_n$'s (e.g. the 4d WZW term is due to $\pi_5$), but in this work we will only focus on the physics due to $\pi_3$, the lowest non-trivial $\pi_n$.

In the continuum, for a field $g(x) \in SU(N)$ (with $|SU(2)| \cong S^3$, here $|G|$ means the manifold of a Lie group $G$), the WZW curvature, a 3-form analogue of the Berry curvature, is defined as $\mathbf{tr}[(g^{-1}dg)^3]/6(2\pi)^2$, which integrates to an integer—the skyrmion number—over a closed 3d manifold. The WZW curvature can be written as the exterior derivative of the WZW curving, a 2-form analogue of the Berry connection, which will not be globally well-defined if the skyrmion number is non-zero. Integrating the WZW curving over a closed 2d manifold yields the WZW term. We will review some technical details at the beginning of Section 4.

We show below that it is mathematically impossible to naturally define these $\pi_3$ related topological operators in $S^3$ nl$\sigma$m on the lattice, if we use the usual Lie group or fibre bundle approaches. Of course, our original motivating problem is $SU(N)$ lattice Yang-Mills theory, not $|SU(N)|$ lattice nl$\sigma$m (with $|SU(2)| \cong S^3$ the pion effective theory). The relation between the two is like the $U(1)$ gauge theory in Section 2.2 versus the $S^1$ nl$\sigma$m in Section 2.1. They are, roughly speaking, related by "putting everything in one higher dimension". Thus, if we have demonstrated the said impossibility for $S^3$ lattice nl$\sigma$m, the same must also be true for $SU(N)$ lattice Yang-Mills.

Before our full analysis of the problem, let us first discuss the role played by global symmetry. We are bringing this up because in the $S^1$ nl$\sigma$m, the Villainization involved elevating $S^1$ to $\mathbb{R}$, the universal cover of the $U(1)$ global symmetry, and in $S^2$ nl$\sigma$m, the spinon decomposition involved elevating $S^2$ to $SU(2)$, the universal cover of the $SO(3)$ global symmetry. This might generate a misleading impression that looking at (the universal cover of) the global symmetry is the key. But this is not true. First of all, conceptually, the existence of topological configurations is not tied with whether a global symmetry is respected. Moreover, for $|SU(N)|$ nl$\sigma$m (with $|SU(2)| \cong S^3$), denoting a field by $g$, the continuous part of the global symmetry is $(SU(N)_L \times SU(N)_R)/Z(SU(N)) \cong PSU(N)_C \times SU(N)'_R$, manifested as

$$g \to h_L g h_R^{-1} = h_C g h_C^{-1} h_R'^{-1}, \tag{50}$$

$$(h_L, h_R) \sim (h_L z, h_R z), \text{ i.e. } (h_C, h_R') \sim (h_C z, h_R'), \text{ for any } g \in SU(N), z \in Z(SU(N)),$$

and the universal cover of it is $SU(N)_L \times SU(N)_R$. But now we see that $SU(N) \to SU(N) \times SU(N) \to SU(N)$ is a trivial bundle, unlike in the examples of $S^1$ and $S^2$ before ((16) and (38)). It does not serve the desired purpose of "transmitting the desired $\pi_3$ to the $\pi_2$ in the layer above" via (17).

Now we are ready to see the problem in full. Based on the rationale of how we captured $\pi_1$ and $\pi_2$ before, it seems in order to capture $\pi_3 \cong \mathbb{Z}$ we naively need some sequence of fibre bundles of the form

$$
\begin{array}{c}
2\pi\mathbb{Z} \to \mathbb{R} \\
\downarrow \\
U(1) \to ??? \\
\downarrow \\
?? \to ? \\
\downarrow \\
S^3 \quad .
\end{array}
\tag{51}
$$

Topologically what we want is $\pi_3(S^3) \xrightarrow{\sim} \pi_2(\text{"??"}) \xrightarrow{\sim} \pi_1(U(1)) \xrightarrow{\sim} \pi_0(\mathbb{Z})$. Moreover, we can even have an interpretation of what the top layers represent: The $2\pi\mathbb{Z}$ on the cubes sum over to the skyrmion number, the $\mathbb{R}$ on a cube represent the WZW curvature on lattice, and the $U(1)$ on a plaquette the WZW curving on the lattice; these are all desired. It seems all we need is to fill out the question marks. But this is impossible. Look at the "??" slot. Topologically we need $\pi_2(\text{"??"}) \cong \mathbb{Z}$, but this "??" is a link variable, so we traditionally want it to be a group-valued variable, so that the variable can be composed when we compose consecutive links. The contradiction is, finite dimensional Lie groups always have trivial $\pi_2$, so this rationale fails.

What if we relax the requirement that the "??" slot should be a group, and hope that we somehow can still make sense of it as a link variable? The familiar examples of finite dimensional fibre bundles in physics are mostly principal or associated bundles, i.e. the transition functions between the fibres are described by Lie group actions, so we still encounter the same failure. In fact, after knowing our final solution in Section 5.5 and looking back, it can be shown [39] at full generality that any finite dimensional fibre cannot serve the purpose of transmitting the topological information from the layer below to above, $\pi_3(S^3) \xrightarrow{\sim} \pi_2(\text{"??"}) \xrightarrow{\sim} \pi_1(U(1))$.

Obviously the same failure occurs if we want to use this rationale to capture on the lattice any non-trivial $\pi_{n\geq 3}$ of general spaces.

Now, if we still want to solve our problem, we are left with two possibilities:

1. To work with infinite dimensional spaces.

2. To work with more flexible, finite dimensional structures beyond groups and fibre bundles.

Our very reason to be interested in lattice theories is the finite dimensionality of the local d.o.f. in the path integral, so of course our final solution will take the second route. However, it is important make connection to the first route, because the first route just points to the continuum theory itself.

Indeed, if we think of the lattice as being embedded in the continuum, the continuum field over the vertices, links, plaquettes and cubes organize into a fibre bundle sequence structure similar to that on the left panel of (48):

$$
\begin{array}{c}
\Omega_*^3 S^3 \to \mathcal{P}_* \Omega_*^2 S^3 \\
\downarrow \\
\Omega_*^2 S^3 \to \mathcal{P}_* \Omega_* S^3 \\
\downarrow \\
\Omega_* S^3 \to \mathcal{P}_* S^3 \\
\downarrow \\
S^3 ,
\end{array}
\tag{52}
$$

where every layer except for the bottom is infinite dimensional, as is expected for a continuum theory.[38] At the top layer, similar to (48), we indeed can map a 3d volume in $S^3$ (the image of the continuum field over the region of a lattice cube) to $\mathbb{R}$ by integrating over the continuum WZW curvature, leading to

$$
\begin{array}{ccc}
\Omega_*^3 S^3 \to \mathcal{P}_* \Omega_*^2 S^3 & \xrightarrow{\ \int_{3d} \mathrm{WZW}\ } & 2\pi\mathbb{Z} \to \mathbb{R} \\
\downarrow & & \downarrow \\
\Omega_*^2 S^3 & & U(1),
\end{array}
\tag{53}
$$

where the right-hand-side reproduces the desired structure in (51). The problem is, unlike in (48), this is not sufficient to reduce the $\mathcal{P}_*\Omega_* S^3$ slot to anything finite dimensional, because this slot is expected to become a $U(1)$ bundle over $\Omega_* S^3$, but the base $\Omega_* S^3$ is still infinite dimensional. So more has to be done to truncate away the unimportant details there in order to obtain something finite dimensional. More exactly, after the previous integral with continuum WZW, the remaining fibre bundle sequence structure in the lower layers is

$$
\begin{array}{c}
U(1) \to \frac{\mathcal{P}_*\Omega_* S^3 \times U(1)}{WZW} \\
\downarrow \\
\Omega_* S^3 \to \mathcal{P}_* S^3 \\
\downarrow \\
S^3,
\end{array}
\tag{54}
$$

where, similar to (49), $\mathcal{P}_*\Omega_* S^3 \times U(1)/WZW$ means two elements in $\mathcal{P}_*\Omega_* S^3 \times U(1)$ are considered equivalent, if the two surfaces in $\mathcal{P}_*\Omega_* S^3$ share the same boundary, and moreover they together bound a volume whose WZW phase is equal to the difference between the two $U(1)$ phases [39,40]. Our task is to recast this structure into a perspective that is more general than groups and fibre bundles—the perspective of category theory, and find a topologically equivalent but finite dimensional representative.

There is another idea, less geometrical and more algebraical, on what kind of infinite dimensional spaces we may want to use. In (16), $\mathbb{R}$ is the universal (i.e. 1-connected) cover of $S^1$, and in (38), $SU(2)$ is the 2-connected cover of $S^2$.[39] Then in (51) we might want the "?" slot to be the 3-connected cover over $S^3$. (This idea has also appeared in [92] recently.) But 3-connected covers are in general infinite dimensional, hence not directly useful for building lattice models. Then the task would be to find finite dimensional structure that effectively plays the role of a 3-connected cover.

Naturally, the geometrical/continuum idea and the algebraic idea come to confluence. In fact, the structure (54) already plays the effective role of a 3-connected cover [39,40] in the category theory sense,[40] albeit still involving infinite dimensional spaces. So no matter which idea we take, we are led to the task of finding a finite dimensional equivalence of this structure. Thus, the task has now become a well-posed mathematical problem—and whose answer turns out to be already known [37] in terms of multiplicative bundle gerbe [38]. The task of finding more general topological operators for more general continuous-valued lattice fields can be turned into well-posed mathematical problems in the same manner, and such relevance to physics provides a good motivation to study these more general mathematical problems.

---

[38]Along two consecutive links, the two paths in $S^3$ compose by concatenation in the obvious way—some reparametrization of the new path is needed but that does not affect anything to be discussed below. For more systematic treatment, see Section 5.

[39]The $m$-connected cover $X^{(m)}$ of a space $X$ means a covering space $X^{(m)} \to X$ whose $\pi_n(X^{(m)}) \xrightarrow{\sim} \pi_n(X)$ for $n > m$ and $\pi_n(X^{(m)})$ trivial for $n \le m$. $m$-connected covers of $X$ form the Whitehead tower over $X$.

[40]$\mathcal{P}_* S^3$ in (54) is the $\infty$-connected cover of $S^3$ because pointed path spaces are contractible. Then at the top level we mod out WZW, hence any topological information higher than $\pi_2$ of $\Omega_* S^3$ (i.e. those higher than $\pi_3$ of $S^3$) is being neglected. Thus we are essentially having a 3-connected cover of $S^3$.

# 4 Main construction

In this section we will introduce the construction that allows us to define the 2d WZW term (not the 4d one) and 3d skyrmion in $S^3$ lattice nl$\sigma$m, as well as the 3d CS term and 4d instanton in $SU(N)$ lattice Yang-Mills—which all originate from $\pi_3 \cong \mathbb{Z}$. The derivation process and the resulting structure lies in higher category theory, as said in the previous section. However, to explicitly present the resulting structure, no knowledge of category theory is required—in the end, structures are described by a set of rules; the familiar Lie groups are also described by a set of rules, except the "rules of the game" we need now are more flexible than those for a group, and anyways, these rules are all that is needed for a computer to carry out Monte-Carlo numerics. Therefore, in this section, we will first state these rules and explain the physical intuition behind, while the derivation and the systematic understanding in terms of higher category theory will be deferred to Sections 5 and 6.

We have explained in Section 3 that any fibre bundle covering $S^3$ or more generally $|SU(N)|$ cannot fulfill our goal. So let us now motivate what kind of covering, if not fibre bundle, we might need. Continuum theory provides a good hint. Consider an $S^3$ or more generally $|SU(N)|$ nl$\sigma$m in the continuum, parametrized by $g(x) \in SU(N)$. How do we show the continuum integral of the WZW curvature $\oint_{3d} \mathbf{tr}[(-ig^{-1}dg)^3]/6(2\pi)^2$ is an integer, which can be interpreted as the skyrmion number? We can first diagonalize $g = \mathcal{U}e^{i\lambda}\mathcal{U}^{-1}$, and find

$$
\begin{aligned}
\frac{1}{6}\mathbf{tr}[(g^{-1}dg)^3] &= d\left(\mathbf{tr}[\lambda(\mathcal{U}^{-1}d\mathcal{U})^2] - \frac{1}{2}\mathbf{tr}[e^{i\lambda}(\mathcal{U}^{-1}d\mathcal{U})e^{-i\lambda}(\mathcal{U}^{-1}d\mathcal{U})]\right) \\
&= d\left(\mathbf{tr}[d\lambda(\mathcal{U}^{-1}d\mathcal{U})] - \frac{1}{2}\mathbf{tr}[e^{i\lambda}(\mathcal{U}^{-1}d\mathcal{U})e^{-i\lambda}(\mathcal{U}^{-1}d\mathcal{U})]\right).
\end{aligned}
\tag{55}
$$

The parenthesis is the WZW curving (whose integral over a closed 2d surface gives the WZW term), and the two lines correspond to two different gauge choices. Note that neither $\lambda$ nor $\mathcal{U}$ is uniquely defined, since $g$ is invariant under $\lambda \to \lambda + 2\pi\kappa$ for any $\mathbb{Z}$-valued diagonal matrix $\kappa$, and under $\mathcal{U} \to \mathcal{U}\mathcal{V}$ for any $\mathcal{V}$ that commutes with $e^{i\lambda}$ (so $\mathcal{V}$ must be diagonal unless $g$ has eigenvalue degeneracy).[41] The WZW curving is in general not everywhere continuous, just like the Berry connection. If we cut the closed 3d space into many patches labeled by $\alpha, \beta \ldots$ that intersect along 2d common boundaries (this is known as a polyhedron decomposition of the space), across the 2d boundary between two patches $\alpha$ and $\beta$, the transformations above are allowed, constituting the transition functions $\kappa_{\alpha\beta}$ and $\mathcal{V}_{\alpha\beta}$ for the WZW curving. Substituting into the two gauge choices of the WZW curving above, we have respectively

$$
\begin{aligned}
\oint_{3d} \frac{i}{6(2\pi)^2}\mathbf{tr}[(g^{-1}dg)^3] &= \sum_{\text{patches } \alpha<\beta} \int_{2d \text{ between } \alpha,\beta} \mathbf{tr}\left[\kappa_{\alpha\beta}\frac{i(\mathcal{U}_\beta^{-1}d\mathcal{U}_\beta)^2}{2\pi}\right] \\
&= \sum_{\text{patches } \alpha<\beta} \int_{2d \text{ between } \alpha,\beta} \mathbf{tr}\left[\frac{d\lambda_\alpha}{2\pi}\frac{i\mathcal{V}_{\alpha\beta}^{-1}d\mathcal{V}_{\alpha\beta}}{2\pi}\right].
\end{aligned}
\tag{56}
$$

From either expression we can see the result is an integer: In the first gauge choice, recall $\kappa$ is a diagonal integer matrix, so after projecting $i(\mathcal{U}^{-1}d\mathcal{U})^2$ to the diagonal elements by $\kappa$, the integrand is some linear sum of 2d Berry curvatures with integer coefficients, hence integrating to an integer; in the second gauge choice, each diagonal component of $d\lambda/2\pi$ picks up some winding number (recall $\lambda$ will well-defined mod $2\pi$) upon integration, and so does each diagonal component of $i\mathcal{V}^{-1}d\mathcal{V}/2\pi$, hence also leading to an integer. More

---

[41]$g$ is also invariant under $e^{i\lambda} \to \sigma^{-1}e^{i\lambda}\sigma$, $\mathcal{U} \to \mathcal{U}\sigma$ where $\sigma \in S_N$ permutes the eigenvalues (the Weyl group). This will not come up in the calculation here.

explicitly, further using Stokes' theorem, either form above reduces to

$$\oint_{3d} \frac{i}{6(2\pi)^2} \mathbf{tr}[(g^{-1}dg)^3] = \sum_{\text{patches } \alpha<\beta<\gamma} \int_{1d \text{ between } \alpha,\beta,\gamma} \mathbf{tr}\left[\kappa_{\alpha\beta} \frac{i\mathcal{V}_{\beta\gamma}^{-1}d\mathcal{V}_{\beta\gamma}}{2\pi}\right] \tag{57}$$

$$= \sum_{\text{patches } \alpha<\beta<\gamma<\delta} \mathbf{tr}\left[\kappa_{\alpha\beta}\, n_{\beta\gamma\delta}\right]_{0d \text{ between } \alpha,\beta,\gamma,\delta} \in \mathbb{Z}, \tag{58}$$

where $n_{\beta\gamma\delta} := i(\ln \mathcal{V}_{\beta\gamma}^{\text{diag}} - \ln \mathcal{V}_{\beta\delta}^{\text{diag}} + \mathcal{V}_{\gamma\delta}^{\text{diag}})/2\pi$ is an integer diagonal matrix once we fix the logarithm branch cut convention.[42,43] In the same manner, we can also show that for a continuum Yang-Mills theory, the integral $\oint_{4d} \mathbf{tr} f^2/2(2\pi)^2$ gives an integer, the instanton number. The integrand is the exterior derivative of the CS 3-form, and at the 3d patch boundaries the CS 3-forms differ by the WZW curvature (plus some extra term, see e.g. [8]), and then the computation essentially reduces to that in the above.

Through this computation in the continuum, we can spot the appearance of some covering that is *not* a fibre bundle. The diagonalization of $g$ that we performed in order to find a useful explicit presentation of the WZW curving corresponds to the Weyl map

$$T \times SU(N)/T \to SU(N), \tag{59}$$

where $T \cong (S^1)^{N-1}$ is the maximal torus parameterized by $e^{i\lambda}$, and $SU(N)/T$ is parameterized by $\mathcal{U}$ with the diagonal $\mathcal{V}$ action mod out. But the Weyl map is not a fibre bundle over $SU(N)$, because when two eigenvalues in $e^{i\lambda}$ happen to be degenerate, the space of $\mathcal{V}$ that commutes with $e^{i\lambda}$ is enlarged to include non-diagonal matrices, but only the space of diagonal ones is being mod out. Thus the Weyl map violates the local triviality condition for a fibre bundle. When we express the WZW curving (55) by $\lambda$ and $\mathcal{U}$, we are further extending the covering into $\mathbb{R}^{N-1} \times SU(N) \to T \times SU(N)/T \to SU(N)$; since the diagonalization Weyl map is already not a fibre bundle, nor is it after this further extension.

While diagonalizing $SU(N)$ does not give rise to a fibre bundle, diagonalization is familiar enough to make sense of and work with. This is indeed how we will construct our non-fibre-bundle finite dimensional structure to solve our problem on the lattice. We will first present the construction for $S^3$ lattice nl$\sigma$m, which can be generalized to $|SU(N)|$ nl$\sigma$m. Next, similar to how we went from $S^1$ nl$\sigma$m in Section 2.1 to $U(1)$ gauge theory in Section 2.2, roughly speaking "putting everything in one higher dimension" will lead to the construction for $SU(N)$ lattice Yang-Mills. How to actually carry out this step is not as obvious as in the $U(1)$ case, and interestingly, if we carry out this step in the "literal" way, a troublesome issue that requires solving some generalized version of Yang-Baxter equation will come up.[44] Fortunately, if we

---

[42]In this derivation we have been consecutively using Stokes' theorem to reduce quantities onto the intersections between more and more patches. Mathematically, such a structure is known as a Deligne-Beilinson double cochain in the context of Deligne-Beilinson double cohomology (see e.g. [38]), where one direction of the cohomology is the (de Rham) $d$, and the other direction is the (Čech) transition between patches. For $U(1)$ gauge theory such a description is presented in details in e.g. [65]. Here, instead of $U(1)$ gauge connection 1-form, in $|SU(N)|$ nl$\sigma$m we have $U(1)$-valued WZW curving 2-form, and later in $SU(N)$ Yang-Mills theory we have $U(1)$-valued non-abelian CS 3-form, but the idea is similar.

One might also note the resemblance between Deligne-Beilinson double cohomology and BRST double cohomology (in particular, the structure we have described resembles the BRST descent equations). Their correspondence is via the notion of ananatural isomorphism to be introduced in Section 5.2.

[43]Just like the Berry connection is widely used in studying the topological and geometrical effects in the momentum space / Brillouin zone (as opposed to the real space), the WZW curving is also useful—though less well-known—in the same context, and is especially necessary when the system is interacting [94].

[44]It might seem surprising that some kind of Yang-Baxter equation is involved in Yang-Mills theory. In Sections 5.3 we will explain why the appearance of Yang-Baxter equation is completely natural when we "deloop" from a nl$\sigma$m to a gauge theory.

carry out this step in a way that better appeals to the relation with the continuum theory [10]—which will involve some techniques similar to the traditional work [8] but under a shifted mindset—then the generalized Yang-Baxter equation issue will be automatically resolved. In fact, our construction recovers [8] if we assume the gauge field strength is weak (which [8] requires) and take the saddle point approximation.

## 4.1 $S^3$ non-linear sigma model: Wess-Zumino-Witten, skyrmion and hedgehog

In the traditional $S^3$ nl$\sigma$m, the dynamical $S^3$ d.o.f. at each vertex is parametrized by $g_v \in SU(2) \cong S^3$. Across each link there is a link weight $W(\mathbf{tr}Dg_l + c.c.)$ where $Dg_{l=\langle v'v\rangle} := g_{v'}g_v^{-1} \in SU(2)$, and $W$ is a positive, increasing function. Note that $\mathbf{tr}Dg_l$ is indeed invariant under the $SO(4) \cong SU(2)_L \times SU(2)_R/\mathbb{Z}_2$ global symmetry (50) with $(Dh_L)_l = \mathbf{1} = (Dh_R)_l$.

Now we want to topologically refine the traditional theory, so that we can naturally define the topological operators such as WZW term and skyrmion. The result will be (70), which we reproduce here:

$$
Z = \left[\prod_{v'}\int_{SU(2)}dg_{v'}\right]\left[\prod_{l'}\sum_{m_{l'}=\pm}\int\frac{d^2\hat{n}_{l'}}{4\pi}\right]\left[\prod_{p'}\int_{-\pi}^{\pi}\frac{d\mathcal{W}_{p'}}{2\pi}\right]\left[\prod_{c'}\sum_{s_{c'}\in\mathbb{Z}}\right]
$$
$$
\times\prod_l W_1(\lambda_l,m_l)\prod_p W_2(e^{i\mathcal{W}_p}\mu^*_{g_{v\in\partial p},m_{l\in\partial p},\hat{n}_{l\in\partial p}}+c.c.)\prod_c W_3(\mathcal{S}_c)\prod_h W_4(d\mathcal{S}_h).
$$

We will now step by step introduce the d.o.f. and the desired properties of the path integral weights, i.e. "the rules of the game", using geometrical intuitions, leaving the formal mathematics to Sections 5 and 6.[45]

Let us first explain the link d.o.f. $m_l$ and $\hat{n}_l$, and the link weight $W_1$. The hint from continuum, which we discussed at the beginning of this section, suggests that we should perform diagonalization in order to find some useful cover over $SU(2)$. In the continuum, e.g. in getting (55), we diagonalized $g(x)$ itself. On the lattice, it turns out more natural to diagonalize $Dg_l \in SU(2)$ instead of $g_v \in S^3$. The fact that $g_v \in S^3$ is not naturally a group element while $Dg_l \in SU(2)$ is naturally a group element already suggests it is better to diagonalize $Dg_l$.[46] Furthermore, it is desired that whatever we do should manifest the $SO(4) \cong SU(2)_L \times SU(2)_R/\mathbb{Z}_2$ global symmetry (50). Under this transformation, $Dg_l \to h_L Dg_l h_L^{-1}$, the eigenvalues remain unchanged. This also suggests it is good to consider the diagonalization of $Dg_l$ rather than the diagonalization of $g_v$ (note, diagonalizing $g_v$ and $g_{v'}$ does not lead to a diagonalization of $Dg_l = g_{v'}g_v^{-1}$).

On each lattice link $l$, we should construct a suitable cover over $SU(2) \ni Dg_l$ (just like how $\mathbb{R} \ni \gamma_l$ covers $U(1) \ni e^{id\theta_l}$ in Section 2.1). According to Section 3, the cover, finite dimensional as we want, is necessarily a non-fibre bundle cover, which we shall now introduce. First, let us consider covering $SU(2)$ by two open patches, $SU(2)\backslash\{-\mathbf{1}\}$ and $SU(2)\backslash\{+\mathbf{1}\}$, although this is slightly different from what we will use in the end. The disjoint union of the two patches, $SU(2)\backslash\{-\mathbf{1}\} \sqcup SU(2)\backslash\{+\mathbf{1}\}$, which covers $SU(2)$, is indeed not a fibre bundle over

---

[45]In mathematical terms, the d.o.f. that we will describe is based on a bundle gerbe over $SU(N)$, following [95] (but with some necessary technical modifications, see footnote 138), which is then turned into a multiplicative bundle gerbe [38] using some geometrical intuition that gives a concrete implementation of the procedure in [96] (yet again with some crucial technical modifications, see footnote 133). See Section 5.5 for the formal discussions.

[46]In comparison, in a very recent work [51], the lattice d.o.f. are still the traditional vertex variables $g_v$ only, however bundle gerbe techniques have been employed to compute the lattice skyrmion number for "smooth enough" lattice configurations. (By constrast, in our work, we introduced new d.o.f., which, together with the traditional $g_v$, form a bundle gerbe type structure, and we do not require the lattice configuration to be "smooth".) There, the diagonalization is indeed performed on $g_v$ rather $Dg_l$.

$SU(2)$ since $\pm\mathbf{1}$ are special points. To understand why we choose patches in such way, let us diagonalize $Dg_l =: \mathcal{U}_l e^{i\lambda_l \sigma^z} \mathcal{U}_l^{-1}$. (Note the difference with the notations in (55): there we are diagonalizing $g$ while here $Dg_l$, moreover there $\lambda$ is a diagonal matrix while here a number, the coefficient of $\sigma^z$.) The first patch contains those $Dg_l$ elements whose $\lambda_l \in [0, \pi)$, while the second patch contains those $Dg_l$ elements whose $\lambda_l \in (0, \pi]$.[47] The key points are:

- The patches are defined using only the eigenvalues of $Dg_l$, ensuring the patches to remain invariant under the $SO(4) \cong SU(2)_L \times SU(2)_R / \mathbb{Z}_2$ transformation (50) which transforms $Dg_l$ by conjugation. In other words, if a patch contains some group element, the patch must contain the entire conjugacy class of that group element.

- The special points $\pm\mathbf{1}$ are where the eigenvalues of $Dg_l$ become degenerate. These are indeed special loci in the diagonalization, because the ambiguity $\mathcal{U}_l \to \mathcal{U}_l \mathcal{V}_l$ enhances from $U(1)$ to $SU(2)$ at these loci. It is anticipated that these special loci require special treatments in what we will do later.

In our actual construction, the link d.o.f. will take value in a non-fibre-bundle cover over $SU(2) \ni Dg_l$ given by

$$Y := \left(SU(2)\backslash\{-\mathbf{1}\}\right) \sqcup \left(SU(2)\backslash\{+\mathbf{1}\} \times S^2\right), \tag{60}$$

and what this extra $S^2$ does on top of the second patch will be explained soon by (62). Let us denote an element $y_l \in Y$ by $y_l = (Dg_l, m_l, \hat{n}_l)$, where $m_l = +$ means $y_l$ belongs to the $SU(2)\backslash\{-\mathbf{1}\}$ patch (so $m_l = +$ implies $Dg_l \neq -\mathbf{1}$), $m_l = -$ means $y_l$ belongs to the $SU(2)\backslash\{+\mathbf{1}\}$ patch (so $m_l = -$ implies $Dg_l \neq +\mathbf{1}$), and $\hat{n}_l \in S^2$ is only going to be meaningful when $m_l = -$ (i.e. when $m_l = +$, $\hat{n}_l$ will not appear anywhere in the theory and can be ignored).

Note that while $m_l$ is a two-valued label, it by no means forms a $\mathbb{Z}_2$ group, as there is no sensible group composition; nor is there a $\mathbb{Z}_2$ symmetry acting on $m_l$. And of course, the whole space $Y$ itself is not a group and cannot compose, either. We will see why this is not a problem.

In the lattice path integral, we will replace the traditional link weight $W(\mathbf{tr}Dg_l + c.c.)$ (note $\mathbf{tr}Dg_l + c.c. = 4\cos\lambda_l$) by some link weight $W_1(\lambda_l, m_l)$ over $Y$, with $m_l = \pm$ summed over (pretending there are no other weights that depend on $m_l$ for now):

$$W(\mathbf{tr}Dg_l + c.c.) \approx \sum_{m_l = \pm} W_1(\lambda_l, m_l). \tag{61}$$

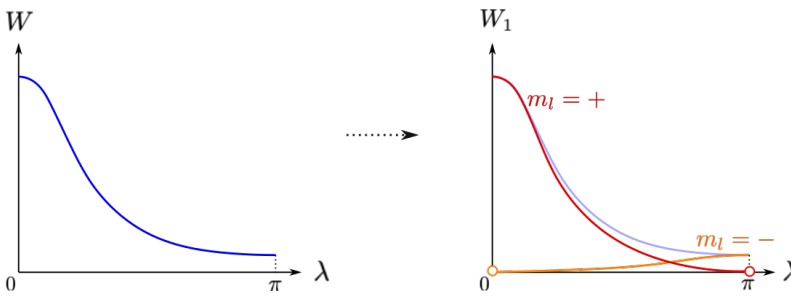

Figure 12: $W$ as a function of $\lambda_l$ decomposed as a sum of $W_1$ over $m_l = \pm$.

(At this point there is no dependence on $\hat{n}_l$, so $\int d^2\hat{n}_l / 4\pi$ yields a trivial factor 1.) This is similar in idea to (7) and (43), but now there is a new aspect that should be emphasized:

---

[47] $\lambda \in (-\pi, 0)$ is equivalent to $\lambda \in (0, \pi)$ upon exchanging the two eigenvalues.

Each patch of $Y$ does not cover the entire $SU(2) \ni Dg_l$, and we require $W_1$ to smoothly vanish towards the boundary of the image of each patch of $Y$, indicated by the hollow circles above, so to ensure the smoothness in $\lambda_l$ after summing over $m_l$.

We shall develop some physical intuition for what $y_l = (Dg_l, m_l, \hat{n}_l)$ is intended to mean, and why the link weight is decomposed as (61). Given our review of the known examples in Section 2, it is again useful to think of the lattice link as a path embedded in the continuum. Then along it the continuum field $g(x)$ traces out a path in $S^3$ interpolating from $g_v$ to $g_{v'}$. Of course the $g(x)$ interpolaton can take complicated shapes, but the infinite dimensional details of how the path wiggles are unimportant; the useful homotopy information is to be kept in $y_l$. Clearly the $Dg_l = g_{v'} g_v^{-1}$ part of $y_l$ indicates the relative position of the starting and the ending point. Now, as long as $Dg_l \neq -\mathbf{1}$, there is a unique shortest geodesic from $g_v$ to $g_{v'}$, given by $\{\mathcal{U}_l e^{i\lambda' \sigma^z} \mathcal{U}_l^{-1} g_v | 0 \leq \lambda' \leq \lambda_l\}$. The idea is that $m_l = +$ represents the contributions from all those continuum paths that are "close enough" to the geodesic. On the other hand $m_l = -$ represents the contributions from all other continuum paths. Schematically:

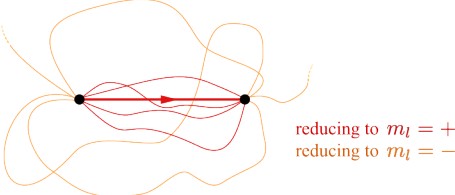

Figure 13: Schematic meaning of $m_l$.

How to define "close enough" in detail is unimportant, but when $Dg_l \to -\mathbf{1}$, fewer and fewer paths are considered "close enough" till none is (indeed, when $Dg_l = -\mathbf{1}$ there is no unique shortest geodesic), and when $Dg_l \to +\mathbf{1}$, more and more paths are considered "close enough" till all paths are. This explains the qualitative behavior of $W_1(\lambda_l, m_l)$ illustrated in (61).

It is helpful for both intuitive and practical purposes to pick a representative path for a given $y_l \in Y$; in particular we will use the representative to construct the $\mu$ function in the plaquette weight (65) later.

- Clearly, for the $m_l = +$ patch, the most natural choice of the representative path for $y_l = (Dg_l, +)$ is the shortest geodesic from $g_v$ to $g_{v'} = Dg_l\, g_v$, given by $\{\mathcal{U}_l e^{i\lambda' \sigma^z} \mathcal{U}_l^{-1} g_v | 0 \leq \lambda' \leq \lambda_l\}$, which is unique thanks to the $m_l = +$, i.e. $Dg_l \neq -\mathbf{1}$ condition.

- On the other hand, a good choice of the representative path for the $m_l = -$ patch is less obvious, and that is why we will need the $\hat{n}_l \in S^2$ d.o.f.: For $y_l = (Dg_l, -, \hat{n}_l)$, the choice of representative path is to first go from $g_v$ to $-g_v$ along some random direction $\hat{n}_l$ via $\{e^{i\lambda'' \hat{n}_l \cdot \vec{\sigma}} g_v | 0 \leq \lambda'' \leq \pi\}$, and then go from $-g_v$ to $g_{v'}$ via the shortest geodesic $\{\mathcal{U}_l e^{i\lambda' \sigma^z} \mathcal{U}_l^{-1} g_v | \pi \geq \lambda' \geq \lambda_l\}$.[48]

We illustrate the representative paths (one with $m_l = +$, one with $m_l = -$ and some choice of $\hat{n}_l$) by picturing $SU(2) \ni Dg_l$ as a 3d ball centered at $\mathbf{1}$ and with radial coordinate $\lambda$, so that whole the surface at $\lambda = \pi$ is identified to a single point $-\mathbf{1}$:

---

[48] Upon reversing the orientation of the link (i.e. exchanging $g_v$ and $g_{v'}$), the representative path for $m_l = +$ only reverses the running direction but the trajectory of the path remains the same. On the other hand, the trajectory of the representative path for $m_l = -$ changes. This is not a major issue: we can either fix some ordering of the vertices and hence the orientations of the links, or introduce an extra two-valued random variable to decide which orientation of the link is to be used when choosing the representative path. Here we will take the first approach, where on a (hyper)cubic lattice our fixed choice of the orientation of each link is our choice of the $+\hat{x}, +\hat{y}, \dots$ directions.

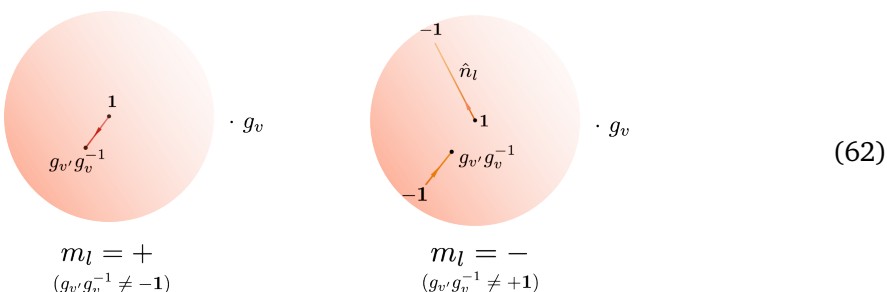

$$(62)$$

Figure 14: Representative path for $y_l$.

From this interpretation of $\hat{n}_l$, we can see that under the $SO(4)$ global symmetry transformation (50), not only does $\mathcal{U}_l \to h_L \mathcal{U}_l$, but also $\hat{n}_l \to R_{h_L} \hat{n}_l$, i.e. $\hat{n}_l \cdot \vec{\sigma} \to h_L(\hat{n}_l \cdot \vec{\sigma})h_L^{-1}$, so that the representative path transforms covariantly.

After introducing the link variable $y_l \in Y$ and the link weight $W_1$, we now move on to the introduction of the plaquette variable $e^{i\mathcal{W}_p} \in U(1)$ and the plaquette weight $W_2$. In the known examples in Section 2, the link variables always form a group, whose group composition is useful on the plaquette. Now we want to emphasize it is not necessary for the link variables to be composable—indeed, for our construction now $Y$ is not composable (even if we have chosen some representative paths, the space of these paths is not closed under concatenation). We want to show the plaquette variable can still be well-defined as long as we have specified the link variables around, without being able to compose them.

From the discussions in Section 3, it is clear that the new d.o.f. on the plaquette should be $U(1)$-valued, effectively representing the WZW curving over the plaquette. In the continuum, the WZW curving (55) is in general not continuously defined globally (just like the Berry connection). Correspondingly, on the lattice, the WZW curving $U(1)$ d.o.f. on the plaquette forms a non-trivial $U(1)$ bundle over the space of the vertex and link variables around the plaquette (just like in Section 2.4, the Berry connection $U(1)$ d.o.f. on the link forms a non-trivial $U(1)$ bundle over the space of the vertex d.o.f. at the ends of the link).

Let us first get a brief sense how there can be any non-trivial $U(1)$ bundle. Consider a plaquette with vertex and link variables labelled as

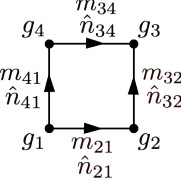

Figure 15: Degrees of freedom around a plaquette.

The base space of the bundle is parametrized by

$$\left\{(g_1, y_{21}, y_{32}, y_{34}, y_{41}) \in S^3 \times Y^4 \,|\, \Pi(y_{41})^{-1}\Pi(y_{34})^{-1}\Pi(y_{32})\Pi(y_{21}) = \mathbf{1}\right\}, \qquad (63)$$

where $\Pi$ is the projection from $Y \ni y_l$ to $SU(2) \ni Dg_l$. This space consists of multiple connected components, each labeled by one combination of the $m_l$'s. Let us first consider, say, the connected component labeled by $m_{21} = m_{32} = m_{34} = m_{41} = +$ (note this component does not involve any $n_l$). This connected component is then parametrized by

$$\left\{(g_1, g_2, g_3, g_4) \in (S^3)^4 \,|\, Dg_{21} \neq -\mathbf{1}, Dg_{32} \neq -\mathbf{1}, Dg_{43} \neq -\mathbf{1}, Dg_{14} \neq -\mathbf{1}\right\}, \qquad (64)$$

which has $\pi_2 \cong \mathbb{Z}$,[49] and therefore can indeed host non-trivial $U(1)$ bundle. For a different combination of the $m_l$'s, there will be a different constraint on the $Dg_l$'s, but the space of the allowed $g_v$'s still has non-trivial $\pi_2$, and moreover some $S^2 \ni \hat{n}_l$ will also come into play. Thus, each connected component of (63) can host a non-trivial $U(1)$ bundle.

It seems any non-trivial $U(1)$ bundle over such bizarre base space (63) will be extremely difficult to parametrize, let alone to prescribe a reasonable weight $W_2$ over them. Fortunately, the intuitive relation to the continuum makes the task much more manageable than it might seem. It is useful to borrow the ideas from the discussions of spinon decomposition below (44). We denote the WZW curving d.o.f. by $e^{i\mathcal{W}_p} \in U(1)$, and let the plaquette weight take the form

$$W_2 \left( e^{i\mathcal{W}_p} \mu^*_{g_{v\in\partial p}, m_{l\in\partial p}, \hat{n}_{l\in\partial p}} + c.c. \right), \tag{65}$$

where $W_2$ is positive and increasing in its argument. Here the function $\mu^*$ plays the role of $u^\dagger_{\hat{n}_{v'}} u_{\hat{n}_v}$ in (44). Let us briefly recall what happens in (44):

- The $U(1)$-valued dynamical variable to be introduced there, i.e. the Berry connection $e^{ia_l}$, represents half of the solid angle (i.e. the Berry phase) bounded by a generic curve between $\hat{n}_v$ and $\hat{n}_{v'}$ together with the two dashed curves in (46). The generic curve is the dynamical variable that we want to describe (with the detailed difference between two curves ignored as long as they bound zero Berry phase, see (45)), and the two dashed curves are fixed gauge choices.

  The maximum of (44) occurs when the generic curve coincides with (or bounds zero Berry phase with, due to (45)) the black geodesic curve in (46). This determines the phase $u^\dagger_{\hat{n}_v} u_{\hat{n}_{v'}} / |u^\dagger_{\hat{n}_v} u_{\hat{n}_{v'}}|$, the saddle point of $e^{ia_l}$.

- When either of the dashed curves becomes less and less well-defined as one of the end points approaches $-\hat{z}$, the gauge choice of the phase $u^\dagger_{\hat{n}_v} u_{\hat{n}_{v'}} / |u^\dagger_{\hat{n}_v} u_{\hat{n}_{v'}}|$ becomes singular, but this will not affect any physical observable and is hence unharmful. (Alternatively, we can consider all possible gauge choices in the path integral, hence resolving the already-unharmful gauge choice singularity.)

- When the black geodesic becomes less and less well-defined as the end points $\hat{n}_v$ and $\hat{n}_{v'}$ become antipodal, the magnitude $|u^\dagger_{\hat{n}_v} u_{\hat{n}_{v'}}|$ approaches 0, so that the weight becomes insensitive to $e^{ia_l}$. Thus, crucially, there is no physical singularity, as desired.

Now, point by point, we will construct the $\mu$ function in the same spirit, except the $U(1)$-valued dynamical variable to be introduced, the WZW curving $e^{i\mathcal{W}_p}$, now lives on the plaquette instead. By thinking of the plaquette as being embedded in the continuum, it is easy to picture the following desired properties for $\mu$:

- Since $W_2$ in (65) is a positive, increasing function, the saddle point of $e^{i\mathcal{W}_p}$ is given by $\mu/|\mu|$. We construct $\mu$ so that its phase $\mu/|\mu|$ is given by the continuum WZW curvature integrated over such a pyramid: The four base corners are at the $g_v$'s and the tip is at $\mathbf{1}$; the neighboring base corners are connected to each other by the aforementioned representative paths (62) for the given $m_l$'s and $\hat{n}_l$'s, while the tip is connected to each base

---

[49] First, $g_1$ is chosen freely from $S^3$; then $g_2$ is chosen from $S^3 \setminus \{-g_1\} \cong D_3$ and $g_3$ is chosen from $S^3 \setminus \{-g_2\} \cong D_3$; finally, and most non-trivially, $g_4$ is to be chosen from $S^3 \setminus \{-g_1, -g_3\}$. Generically $g_3 \neq g_1$, and in such generic cases the space $S^3 \setminus \{-g_1, -g_3\} \ni g_4$ is homotopic to $S^2$, which has $\pi_2 \cong \mathbb{Z}$. The presence of the $g_3 = g_1$ spot will not alter the $\pi_2$. To see this, note the space of $(g_3, g_4)$ under the additional assumption that $g_3 \neq g_1$ is homotopic to $S^2 \times S^2$; now we include the spot $g_3 = g_1$, at which the space of $g_4$ becomes homotopic to a point. Thus, the total space of $(g_3, g_4)$ is homotopic to such a space: start with $S^2 \times S^2$ (which represents $g_3 \neq g_1$), drag/collapse $S^2 \times \{\text{north pole}\} \subset S^2 \times S^2$ to a single point (which represents $g_3 = g_1$).

corner by the shortest geodesic; the four triangles on the side and the quadrangle at the base are then wrapped with some standard choice of interpolating surfaces (discussed below), forming a pyramid. Which side is called the "interior" or "exterior" of the pyramid does not matter, since the two choices only differ by a $2\pi$ phase. An illustration of the pyramid in $S^3 \ni g_v$, assuming each $m_l = +$, looks like

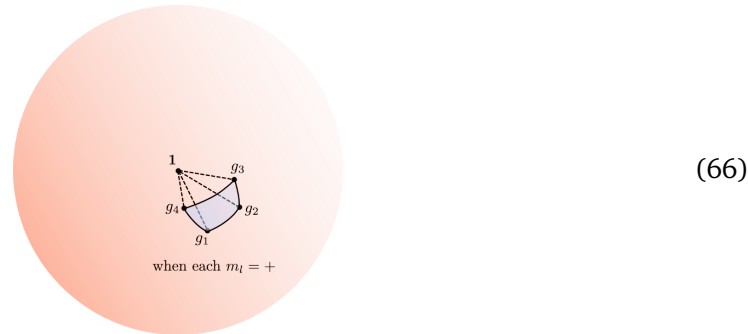

(66)

Figure 16: Saddle of lattice WZW curving.

Similar to the case of Berry connection in (46), here only the base quadrangle surface is physical, while the tip and the four triangles on the side are just some gauge choice; a gauge change can be absorbed by a redefinition of $e^{i\mathcal{W}_p}$. When six plaquettes piece up to a cube, we can compute the lattice WZW curvature over the cube, $e^{id\mathcal{W}_c}$, in which the dependence on the gauge choices (the tip and the triangles on the sides) cancel out.

Fluctuations of $e^{i\mathcal{W}_p}$ away from $\mu/|\mu|$ can be thought of as capturing the fluctuation of the interpolating surface at the base quadrangle away from the standard choice (recall the equivalence relation explained below (54)), just like the fluctuation of the Berry connection in the case of $S^2$ nl$\sigma$m (recall the discussion below (45)).

How to choose the standard interpolating surfaces in detail is not so important as long as the choice is "reasonable", approaching some notion of minimal surface when the $g_v$'s are close to each other and all $m_l = +$. For concreteness we will introduce two reasonable choices for constructing the standard choice of interpolating surface in the below. For now we explain the general idea of how topology is taken into account. It is easy to see that any choice of interpolating surface cannot be made continuously everywhere over the space of variables $g_{v \in \partial p}, m_{l \in \partial p}, \hat{n}_{l \in \partial p}$ in $\mu$, and singularities will be developed. Topology is taken into account by treating the singularities appropriately:

- When the choice of the interpolating surface for any of the side triangles of the pyramid becomes singular, the phase $\mu/|\mu|$ also becomes singular. But this does not matter because the side triangles are gauge choices anyways, just like the singularity in the case of Berry connection (when either $\hat{n}_v$ and $\hat{n}_{v'}$ approaches $-\hat{z}$ in (44)). Such singularity is unavoidable, indeed because we want the WZW curving to take value in a non-trivial $U(1)$ bundle over the space of the vertex and link variables.

- On the other hand, when the choice of the interpolating surface for the base quadrangle becomes singular, we require $|\mu| \to 0$, so that $W_2$ becomes insensitive to the value of the WZW curving $e^{i\mathcal{W}_p}$, and this agrees with our intuition, just like in the case of Berry connection (when $\hat{n}_v = -\hat{n}_{v'}$ in (44)). More generally, we want $|\mu| = 1$ when all $m_l = +$ and all $g_v$ equal, and $|\mu|$ decreases as the base quadrangle loop becomes larger and larger, until $|\mu| = 0$ when the choice of interpolating surface becomes singular.

As claimed, our construction of $\mu$ is, indeed, a generalization (in the sense of geometric interpretations) of what we did in spinon decomposition of $S^2$ nl$\sigma$m. And as mentioned before, for concreteness we will discuss two reasonable choices for the standard interpolating surfaces. The choices are of course non-unique; the descriptions in italic font below are some optionals. How to make a choice that works the best for numerical purpos can only be determined through future numerical investigations.

- Choice 1: *In our subsequent work [10], an interpolation procedure is introduced for gauge theory (which we will discuss in the next subsection) based on the technical aspects of [8]. One can use the similar idea to construct the interpolation surfaces in nl$\sigma$m. That is, we think of each plaquette as a surface parametrized by $(\sigma, \tau) \in [0,1]^2$ being embedded in the continuum, and we think of the base quadrangle (the standard choice of interpolating surface) in (66) as a field $g(\sigma, \tau) : [0,1]^2 \to S^3$, such that along each of its edges (i.e. when at least one of $\sigma, \tau$ takes value 0 or 1) $g(\sigma, \tau)$ is defined according to (62) (with some choice of parametrization—say, $\lambda$ increases with constant rate along the parametrization of the edge). When each $m_l = +$, we can choose the standard interpolating surface in $S^3$ by demanding $g(0, \tau)$ interpolates to $g(1, \tau)$ along the shortest geodesic as $\sigma$ increases for each fixed $\tau$. When some $m_l = -$, the interpolation as $\sigma$ increases becomes more complicated, and can be chosen in ways similar to the cube interpolations in [10] (the description there is for gauge theory, so the plaquette interpolation here is analogous to the cube interpolation there).*

- Choice 2: *We can also do the interpolation without relying on parametrizing the plaquette. We further cut the base quadrangle into two triangles by connecting $g_2$ and $g_4$ via the shortest geodesic (we will see that the special case of $g_2 = -g_4$ will naturally have $|\mu| = 0$), so that we have six triangles, four on the sides of the pyramid, and two on the base.*

  *First consider a triangular loop such that all three edges are given by the shortest geodesics, that is, when all $m_l$ involved in this triangle takes $+$. Just like two points in $S^2$ determine a great circle as long as the two points are not opposite, three points in $S^3$ (the three vertices of the triangle) determine a great sphere as long as the three points are not on a same geodesic circle.[50] The great sphere is cut into two pieces by the edges of the triangular loop, and one piece is always smaller than the other given that the three points are not on a geodesic circle, and we pick the smaller piece to be the interpolating surface.*

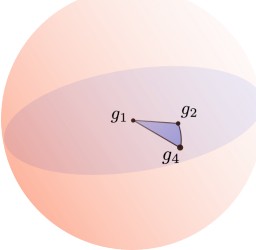

Figure 17: Interpolating surface.

---

[50]To see this, denote the three points by $p_1, p_2, p_3 \in SU(2)$. The below would be most easily pictured by setting $p_1 = \mathbf{1}$ though we will keep it general. Two points $p_1, p_2$ determine a geodesic circle $\ell_{21}$ (which is generated by diagonalizing $Dp_{21}$, letting the eigenvalue take any value between 0 and $2\pi$, and then multiplying the matrix back on $p_1$). Similarly $p_1, p_3$ determine a geodesic circle $\ell_{31}$, which is distinct from $\ell_{21}$ assuming the three points are not on a same geodesic circle. Now $\ell_{21}$ can be rotated to $\ell_{31}$ by an $SO(2)$ rotation living in the $SO(3) \subset SO(4)$ that keeps $p_1$ unchanged. Letting this $SO(2)$ rotation take angles from 0 to $2\pi$ generates the desired great sphere.

*Here we drew one of the triangles at the base of the pyramid, and we placed $g_1$ at the origin to make it easier to illustrate what a great sphere means.*

*In the special cases where the three points lie on a same geodesic circle, either the triangular loop is degenerate (i.e. one point lies on the shortest geodesic between the other two points) or the triangular loop itself is the geodesic circle. Obviously, for the former kind, we will take the interpolating surface to be trivial. On the other hand, for the latter kind, the choice of the interpolating surface will become singular, and this is the topological issues we discussed before—if the triangle loop is a side triangle of the pyramid, it is fine that the interpolating surface becomes singular since it is merely a gauge choice; while if the triangular loop is on the base, we will let $|\mu| = 0$.*

*Next consider a triangular loop such that one edge is flipped from $m_l = +$ to $m_l = -$. It seems it is a consistent, though perhaps crude, approximation to just set $|\mu| = 0$ whenever any $m_{l \in p} = -$. In that case, the description of the interpolating surface below will not be needed, and the $\hat{n}_l \in S^2$ variable can be ignored, so that the theory will be simplified. Whether this crude approximation is good enough to describe the physics of the $nl\sigma m$ is subjected to numerical investigation. For now we suppose we do not simply set $|\mu| = 0$ when some $m_{l \in p} = -$.*

*Since the representative path for $m_l = -$ is in general not a geodesic but two segments of geodesics, such "triangular loop" really looks like a quadrangular loop. The choice of the interpolating surface is illustrated as*

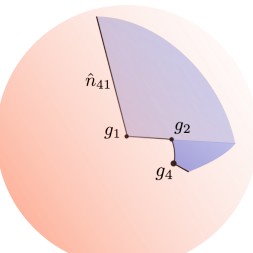

Figure 18: Interpolating surface.

*which is the union of two interpolating surfaces: one for the triangular formed by connecting $g_1, g_2, e^{i(\pi - 0^+)\hat{n}_{14} \cdot \vec{\sigma}} g_1$ with the shortest geodesics, and another for the triangular loop formed by connecting $g_2, g_4, -g_1$ with the shortest geodesics. The idea is that, when $m_{14} = -$ and $g_4 \to -g_1$, the interpolating surface would approach that of when $m_{14} = +$ and $g_4 \to e^{i(\pi - 0^+)\hat{n}_{14} \cdot \vec{\sigma}} g_1 \to -g_1$ from the $\hat{n}_{14}$ direction.*

*The treatments when the choice of interpolating surface becomes singular is the same as before.*

*When more $m_l = -$, the idea is the same.*

This completes our description of the crucial topological and dynamical properties of the plaquette weight $W_2$ that probabilistically weighs the WZW curving d.o.f. $e^{i\mathcal{W}_p}$.

Now that the WZW curving gained its physical meaning through the suitably constructed weight (65), we can readily use it in 2d spacetime to define the WZW phase at level $k \in \mathbb{Z}$:

$$W_{WZW}^k := e^{ik \oint_{2d} \mathcal{W}} := e^{ik \sum_p \mathcal{W}_p} . \tag{67}$$

Notably, a $k \neq 0$ WZW phase makes the $SO(4)$ global symmetry anomalous, although it does not directly break the symmetry. That is, if a non-trivial $SO(4)$ background gauge field is

introduced, the definition of the WZW phase will become ambiguous. To describe this anomaly we need to discuss how a topologically refined nl$\sigma$m is coupled to a topologically refined non-abelian (background) gauge field, and we will leave the detailed discussion to future works.[51]

Beyond 2d, the last step, of course, is to Villainize the lattice WZW curvature $e^{id\mathcal{W}_c} \in U(1)$ to the skyrmion density

$$\mathcal{S}_c := d\mathcal{W}_c/2\pi + s_c \in \mathbb{R}, \tag{68}$$

by introducing an $s_c \in \mathbb{Z}$ dynamical variable on each cube. We have a cube weight $W_3(\mathcal{S}_c)$ that is positive and decreases with $|\mathcal{S}_c|$. The total skyrmion number over a 3d surface is then defined as $\oint_{3d} \mathcal{S} = \sum_c s_c \in \mathbb{Z}$. A topological theta term can hence be defined. In 4d or above, we can define the hedgehog like defect $d\mathcal{S}_h = ds_h$ (where $h$ labels hypercubes), which represents the non-conservation of baryon number in the context of pion vacua effective theory in 4d spacetime. Again we can introduce a fugacity weight $W_4(d\mathcal{S}_h)$ for these defects, or forbid them using[52]

$$W_4^{forbid}(d\mathcal{S}) = \int_{-\pi}^{\pi} \frac{d\widetilde{\phi}_h}{2\pi}\, e^{i\widetilde{\phi}_h d\mathcal{S}_h}\,. \tag{69}$$

If we indeed use $W_4^{forbid}$, then there is the $(d-4)$-form dual $U(1)$ global symmetry $e^{i\widetilde{\phi}_h} \to e^{i\widetilde{\phi}_h} e^{i\widetilde{\alpha}_h}$, $e^{d^*\widetilde{\alpha}_c} = 1$, which in $d = 4$ is interpreted as the baryon conservation $U(1)$. Again there is a mixed anomaly between the original $SO(4)$ global symmetry and this dual $U(1)$ global symmetry. Just like the anomaly mentioned below (67), we will leave the detailed discussion of this anomaly to future works.[53]

Piecing up the discussions above, we have our first main result of this paper: The lattice $S^3$ nl$\sigma$m refined to include skyrmion reads, for $d \geq 4$,

$$Z = \left[\prod_{v'} \int_{SU(2)} dg_{v'}\right] \left[\prod_{l'} \sum_{m_{l'}=\pm} \int \frac{d^2\hat{n}_{l'}}{4\pi}\right] \left[\prod_{p'} \int_{-\pi}^{\pi} \frac{d\mathcal{W}_{p'}}{2\pi}\right] \left[\prod_{c'} \sum_{s_{c'} \in \mathbb{Z}}\right]$$

$$\prod_l W_1(\lambda_l, m_l) \prod_p W_2(e^{i\mathcal{W}_p} \mu^*_{g_{v\in\partial p}, m_{l\in\partial p}, \hat{n}_{l\in\partial p}} + c.c.) \prod_c W_3(\mathcal{S}_c) \prod_h W_4(d\mathcal{S}_h). \tag{70}$$

---

[51]Briefly speaking, the main task is to generalize the definition of the $\mu$ function to situations where the global symmetry background is non-trivial, and this is done using some technique to be introduced in Section 4.2, in relation to non-abelian CS phase factor. After doing so, we will find that under local gauge transformation of the background gauge field, the phase of this generalized $\mu$ function will transform. In $W_2$, we can absorb this phase transformation of $\mu$ into $e^{i\mathcal{W}_p}$, but then the WZW phase factor would not remain invariant unless $k = 0$.

[52]Very recently, [97] also discussed defining and forbidding defects in lattice nl$\sigma$m beyond the previously known examples (Villain and spinon decomposition), by discretizing the target space (e.g. the $S^3$ here). Here what we showed is that the same can be done without discretizing the target space—the vertex d.o.f. still takes continuous value in $S^3$ itself rather than some discrete points on $S^3$, and the $SO(4)$ global symmetry is still manifest. See footnote 151 for more discussions.

[53]One picture to describe the mixed anomaly is that the instanton of the $SO(4)$ background gauge field is charged under the dual $U(1)$. As we sketched in footnote 51, under gauge transformation of the $SO(4)$ background, the local WZW curving variable $e^{i\mathcal{W}_p}$ must transform accordingly to keep $W_2$ invariant. Then the remaining situation essentially becomes that of the gauge transformation of a Villainized 2-form $U(1)$ gauge field, constituting of $e^{i\mathcal{W}_p} \in U(1)$ and $s_c \in \mathbb{Z}$. A Villainized 3-form $U(1)$ background will be introduced as the refinement of the $SO(4)$ global symmetry background (similar to Section 4.2, but here the fields are not dynamical). This consists of $e^{iC_c} \in U(1)$, interpreted as the CS d.o.f. of the lattice $SO(4)$ background gauge field, and $I_h \in \mathbb{Z}$, such that $dC_h/2\pi + I_h$ is the background instanton density. In $W_3$, $d\mathcal{W}_c/2\pi + s_c \to d\mathcal{W}_c/2\pi + s_c - C_c/2\pi$ where $C_c$ absorbs the aforementioned 2-form $U(1)$ gauge transformation of $d\mathcal{W}_c$, and in $W_4$, $ds_h \to ds_h - dC_h/2\pi - I_h$ which is no long exact, hence violating the dual $U(1)$ global symmetry.

An alternative picture is, if we introduce a background gauge field for the dual $U(1)$ global symmetry, it is easy to see the Dirac string part of this $U(1)$ background will couple to $e^{i\mathcal{W}_p}$, generating a WZW phase whose level is the Dirac string charge. (This is similar to the second perspective mentioned at the beginning of footnote 33.) Then by footnote 51, this makes the $SO(4)$ global symmetry anomalous.

The d.o.f. together form a mathematical structure that counts the $\pi_3$ of the lattice $S^3$ nl$\sigma$m, to be explained with (140).[54] For $d = 3$, there is no $W_4$, but we can additionally consider a topological theta term

$$Z_\Theta = \left[\prod_{v'} \int_{SU(2)} dg_{v'}\right] \left[\prod_{l'} \sum_{m_{l'}=\pm} \int \frac{d^2\hat{n}_{l'}}{4\pi}\right] \left[\prod_{p'} \int_{-\pi}^{\pi} \frac{d\mathcal{W}_{p'}}{2\pi}\right] \left[\prod_{c'} \sum_{s_{c'} \in \mathbb{Z}}\right]$$
$$e^{i\Theta \sum_c \mathcal{S}_c} \prod_l W_1(\lambda_l, m_l) \prod_p W_2(e^{i\mathcal{W}_p}\mu^*_{g_{v\in\partial p}, m_{l\in\partial p}, \hat{n}_{l\in\partial p}} + c.c.) \prod_c W_3(\mathcal{S}_c), \qquad (71)$$

for any $\Theta \in U(1)$. For $d = 2$, there is no $s_c$ and $W_3$, but we can additionally consider a WZW term

$$Z_{kWZW} = \left[\prod_{v'} \int_{SU(2)} dg_{v'}\right] \left[\prod_{l'} \sum_{m_{l'}=\pm} \int \frac{d^2\hat{n}_{l'}}{4\pi}\right] \left[\prod_{p'} \int_{-\pi}^{\pi} \frac{d\mathcal{W}_{p'}}{2\pi}\right]$$
$$e^{ik \sum_p \mathcal{W}_p} \prod_l W_1(\lambda_l, m_l) \prod_p W_2(e^{i\mathcal{W}_p}\mu^*_{g_{v\in\partial p}, m_{l\in\partial p}, \hat{n}_{l\in\partial p}} + c.c.), \qquad (72)$$

for any $k \in \mathbb{Z}$.

We have described the crucial properties that the weight factors (and most particularly the $\mu$ function in $W_2$) should have, but how to optimize the weight factors in detail for best numerical performance is subjected to numerical investigation, and is indeed beyond the scope of the present work. Since some d.o.f. can no longer be group elements, the $W_2$ weight no longer has a simple analytic description in terms of the trace of some group element or so, and might need to be stored as a somewhat complicated function.[55] In practical implementation, if the phase $\mu/|\mu|$ slightly deviates from the value we described, there should be no crucial problem. Moreover, as a crude approximation, it is even consistent to set $\mu = 0$ when any of the $m_{l\in\partial p}$ involved is $-$; if this indeed works well numerically, then the implementation will be greatly simplified (and the $\hat{n}_l \in S^2$ can be entirely ignored).

It worths to reiterate the relation between the refined lattice nl$\sigma$m and the traditional lattice nl$\sigma$m. If we ignore the plaquette and cube d.o.f. and weights in (70), we can recover the traditional model via (61). But once the plaquette and cube d.o.f. and (reasonably chosen) weights are taken into account, there can be no exact recovery. This situation is similar to the inclusion of vortex fugacity weight in the Villain model, (13). As we emphasized there, instead of being a problem, this is expected (and analytically established in the Villain case [14]) to be a feature that *helps* control the renormalization, because in the renormalization process the effect of such d.o.f. and weights will be generated anyways, so having them in the model is expected to help keep track of the renormalization flow. Numerical computations will be needed to determine the renormalization behavior of the plaquette and cube weights.

Finally, we briefly explain how to generalize to nl$\sigma$m with target space $\mathcal{T} = |SU(N)|$ beyond $N = 2$. Again we diagonalize $Dg_l = \mathcal{U}_l e^{i\lambda_l^a \tau_a} \mathcal{U}_l^{-1}$, where $\tau_a$ ($a = 1, \cdots, N-1$) is a set of generators for the root lattice, and the space of eigenvalues is parametrized by $\lambda_l^a$ in the Weyl alcove, which for $SU(N)$ is an $(N-1)$-dimensional simplex (this generalizes $\lambda_l \in [0, \pi]$ for

---

[54]We emphasize again that in this paper we are only concerned with the topological physics due to $\pi_3 \cong \mathbb{Z}$. In the physical $d = 4$ spacetime, the actual $S^3 \cong |SU(2)|$ pion vacua effective theory contains a non-trivial 4d WZW term due to $\pi_5(S^3) \cong \mathbb{Z}_2$, and there are also topological effects due to $\pi_4(S^3) \cong \mathbb{Z}_2$; for $SU(N > 2)$, the pion-kaon vacua effective theory contains a non-trivial 4d WZW term due to $\pi_5(|SU(N > 2)|) \cong \mathbb{Z}$ [98, 99]. Hopefully our general framework to be sketched in Section 6 will lead to natural lattice definitions of these terms in future works.

[55]Some automated optimization program might be useful, for instance $W_2$ might be implemented as the output of some machine learning task.

$N=2$). Each of the co-dimension 1 faces of the simplex—there are $N$ of them—corresponds to a pair of adjacent eigenvalues becoming degenerate (this generalizes $\lambda_l = 0$ and $\lambda_l = \pi$ for $N = 2$). We cover $SU(N)$ by $N$ patches, each removing one pair of eigenvalue degeneracy, i.e. each removing one face of the Weyl alcove. Each patch can be labeled by the corner of the Weyl alcove that is opposite to the face being removed, which for $SU(N)$ turns out to be an element of the $\mathbb{Z}_N$ center (this generalizes the labels $m_l = \pm = \pm\mathbf{1}$ for the patches $\lambda_l \in [0, \pi)$ and $\lambda_l \in (0, \pi]$ respectively for $N = 2$), though this does not mean the $m_l$ labels are going to be able to compose like a $\mathbb{Z}_N$ group. For the $m_l = \mathbf{1}$ patch, the representative path is given by connecting a straight line in the Weyl alcove from the origin to the point that represents $\lambda_l^a$, and then conjugating this path by $\mathcal{U}_l$ before multiplying by $g_v$ on the right. For any other $m_l$ patch, the representative path has two segments, one segment is given by connecting a straight line in the Weyl alcove from the corner labeled by $m_l$ to the point that represents $\lambda_l^a$, and then conjugating this path by $\mathcal{U}_l$ before multiplying by $g_v$ on the right; the other segment is given by the edge of the Weyl alcove connecting the origin to the corner that labels $m_l$, and then conjugating this edge by an arbitrary element in $U(N)/H$ where, if we denote $m_l = e^{i2\pi\alpha/N}$, $H = U(\alpha) \times U(N - \alpha)$ is the subgroup that commutes with this edge (this generalizes the $\hat{n}_l \in U(2)/U(1)^2 \cong S^2$ for $N = 2$), before multiplying by $g_v$ on the right. For example, for $SU(3)$ the eigenvalues $\lambda_l$ interpolate in three ways in the Weyl alcove (the conjugation part cannot be read-off from here):

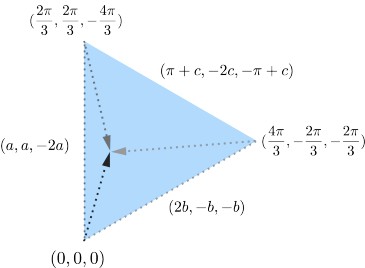

Figure 19: Three different interpolations of $\lambda_l$ in the Weyl alcove of $SU(3)$.

with deeper grey indicating higher weight in $W_1$ in this example. Thus we have described the non-fibre-bundle cover $Y$ over $SU(N)$ and the representative paths. The rest is essentially the same as $N = 2$. We weigh the lattice WZW curving $e^{i\mathcal{W}_p}$ by constructing a suitable function $\mu$, which involves suitably choosing interpolating 2d surfaces and integrating the continuum WZW curvature over the resulting pyramid—the closeness of the continuum WZW curvature is crucial here because that makes the integral independent of the choice of the interpolating 3d volume out of the $N(N-1)/2$-dimensional space of $SU(N)$. Finally we Villainize the lattice WZW curvature.

## 4.2 $SU(N)$ lattice gauge theory: Chern-Simons, instanton and Yang monopole

From the experience with Villainized $S^1$ lattice nl$\sigma$m and Villainized $U(1)$ lattice gauge theory introduced in Sections 2.1 and 2.2, it is intuitive to expect that, now that we have topologically refined the $|SU(N)|$ lattice nl$\sigma$m, the topologically refined $SU(N)$ lattice gauge theory can be obtained by "putting the d.o.f. on cells of one higher dimension". What this really means is the following: Traditionally, the lattice instanton is defined by interpolating the lattice gauge field to a continuum gauge field [8], and the problem is the interpolation choice will run into singularities or discontinuities as we vary the lattice gauge field (and the treatment in [8] is to disallow strongly fluctuating gauge fields); now what we have learned from the topological refinement of $|SU(N)|$ nl$\sigma$m is how to consider different possibilities of the interpolation of

the $|SU(N)|$ matter field at each level of lattice cell, and we are going to apply the idea to the different possibilities of interpolating the $SU(N)$ gauge field at each level of lattice cell, which is one dimension higher compared to the counterpart in nl$\sigma$m.

Again we will focus on $SU(2)$ in the below, since the generalization to $SU(N)$ using the Weyl alcove is straightforward.

Traditionally, we have a lattice gauge connection $g_l \in SU(2)$ on each lattice link, and $Dg_p \in SU(2)$ is the gauge flux around the plaquette $p$. We first describe how to refine the gauge flux on the plaquette. Recall in the case of nl$\sigma$m, on the link we refined $Dg_l \in SU(2)$ to $y_l \in Y$, and the patches (60) were chosen to be invariant under conjugation to manifest the $SO(4)$ global symmetry. Now, in gauge theory, on the plaquette we also refine $Dg_p \in SU(2)$ to $y_p \in Y$, and the patches being invariant under conjugation is desired because lattice gauge flux transforms by conjugation under gauge transformation and under changing the choice of the starting point. The plaquette weight $W_2$ for gauge theory has the same qualitative properties as the link weight (61) for nl$\sigma$m.

In nl$\sigma$m, an element $y_l = (Dg_l, m_l, \hat{n}_l)$ has been pictured in (62) as choosing an interpolating path from $g_v$ to $g_{v'}$, that will be used in designing the plaquette weight (65). Now in gauge theory, an element $y_p = (Dg_p, m_p, \hat{n}_p)$ can be interpreted as choosing a way of interpolating the gauge field over the plaquette, that will be useful later in designing the cube weight. When $m_p = +$ (which requires $Dg_p \neq -\mathbf{1}$), the interpolation is the following. Consider the holonomy around a portion of the plaquette (starting and ending at the lower left corner), indicated by the shaded area:

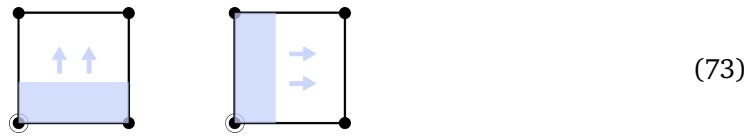

$$(73)$$

Figure 20: Portions of the plaquette with increasing sizes.

As the portion increases its size in either direction (indicated by the arrows), the holonomy around it interpolates along the shortest geodesic from $\mathbf{1}$ to $Dg_p$.[56] This essentially agrees with how the gauge field on the link is interpolated into the plaquette in [8]. When $m_p = -$ and some $\hat{n}_p$ is given, as the size of the portion increases, the holonomy interpolates in the alternative way as explained by (62) in the case of nl$\sigma$m.

On the cube, we introduce a $U(1)$ dynamical field $e^{i\mathcal{C}_c}$ [100] which is interpreted as the lattice version of the CS 3-form. Similar to the WZW curving d.o.f. in the case of nl$\sigma$m, here the CS d.o.f. forms a non-trivial $U(1)$ bundle over the space of $g_l$'s and $y_p$'s on the links and plaquettes around the cube, and is weighed by some

$$W_3(e^{i\mathcal{C}_c} \nu^*_{g_{l \in \partial c}, m_{p \in \partial c}, \hat{n}_{p \in \partial c}} + c.c.) \tag{74}$$

in analogy to (65). Just like the $\mu$ function in (65), here the $\nu$ function has the following properties: Its phase $\nu/|\nu|$ is given by interpolating the gauge fields on the plaquettes around the cube into the inside of the cube via some standardized procedure (as mentioned above, when all $m_p = +$, the plaquette interpolation is the same as that in [8], then we can also use the cube interpolation in [8]; when some $m_p = -$, some other interpolation into the cube will be used,

---

[56]Note this description of the plaquette interpolation is independent of the gauge choice except at the starting point of the loop (as indicated at the lower left corner of the plaquette), and gauge transformation at the starting point acts by conjugation on the holonomy around the shaded area of any size.

similar to the nl$\sigma$m case)[57] and then taking the continuum CS integral over the cube. Since the continuum CS term is gauge dependent, the phase $\nu/|\nu|$ will also be gauge dependent, but under gauge transformation it only changes by a lattice exterior derivative. Fluctuations of $e^{i\mathcal{C}_c}$ away from $\nu/|\nu|$ is interpreted as effectively capturing the fluctuations of the gauge field inside the cube away from the standard interpolation. Singularity in the phase $\nu/|\nu|$ due to singularity in the gauge dependence of the continuum CS term does not matter since such singularity will always drop out in physical observables, while singularity in the choice of the standard interpolation of gauge field into the cube should occur at where $|\nu|$ decreases to 0. (The idea here has been partly developed in [100]. In that paper, there is also a dynamical lattice CS $U(1)$ field weighted with some saddle. However, [100] does not include the dynamical variables on the plaquettes introduce above and those on the hypercubes to be introduced below. Moreover, in [100] the counterpart of $|\nu|$ is a constant, rather than a function which can, crucially, vanish under suitable situations. Hence the problem of discontinuity persists.)

The above are the key requirements for the $\nu$ function in the cube weight. For concreteness, we will present one particular way to construct the standard interpolations and the $\nu$ function in a separate work [10]—because the procedure is highly technical and takes some length to describe. Some highly technical aspects are borrowed from [8], but used in a conceptually different way; moreover, at the technical level, in [10] we also improved some expressions from [8] so that the construction can be described in terms of Wilson loop holonomies instead of Wilson lines with open ends, making the construction manifestly gauge invariant. Of course, in actual practice, how improve the detailed construction for better performance of must be subjected to numerical investigations. And just like in the case of nl$\sigma$m, we guess it might be a consistent approximation to just let $|\nu| = 0$ whenever any $m_{p \in \partial c} = -$, and this would largely simplify the implementation. How good this approximation is in capturing the physics is, again, subjected to numerical investigations.

In 3d, we can define the non-abelian CS phase with level $k \in \mathbb{Z}$ on lattice as

$$W_{CS}^k := e^{ik \oint_{3d} \mathcal{C}} := e^{ik \sum_c \mathcal{C}_c} . \tag{75}$$

If the 3d space has boundary, Dirichlet boundary condition is required to avoid gauge dependence on the boundary. Unlike the abelian CS (25) which depends on the traditional link gauge field explicitly, here the non-abelian CS only depends on the link gauge field probabilistically through the aforementioned weights $W_2, W_3$, constituting a CS-Yang-Mills theory. (Even in the continuum CS theory, a Yang-Mills term with tiny coefficient is secretly understood in the regularization of the eta-invariant [75].)

In 4d, on the hypercube, we Villainize $e^{id\mathcal{C}_h}$ by introducing an integer d.o.f. $\iota_h \in \mathbb{Z}$, so that the non-abelian instanton density is defined as

$$\mathcal{I}_h := \frac{d\mathcal{C}_h}{2\pi} + \iota_h \tag{76}$$

(cf. (25)).[58] There is a hypercube weight $W_4(\mathcal{I}_h)$ that is positive and decreasing with $|\mathcal{I}_h|$. The

---

[57]An important requirement of the procedure is that, just like in the previous footnote, the description of the standard interpolation into the cube must be stated in terms of Wilson loop holonomies, so that the description is gauge independent except at the starting point of the loop.

[58]Our definition of instanton density reduces to that in [8], if we consider weak enough field strength, and use the saddle point approximation on the new local weights $W_4, W_3, W_2$, i.e. always choose those new d.o.f. $\iota_h, e^{i\mathcal{C}_c}, m_p(=+)$ that maximize $W_4, W_3, W_2$. We will discuss this comparison to [8] in greater details in a separate work [10]; for now let us briefly explain the idea.

In [8], the gauge field (assumed weak field strength) on the links is interpolated via a standard procedure into the plaquettes and the cubes. This corresponds to choosing $m_p = +$ (which maximizes $W_2$ when the field strength is weak) and choosing $e^{i\mathcal{C}_c} = \nu/|\nu|$ (which maximizes $W_3$). Let us explain the latter point. In [8] there was no explicit mention of CS, but the instanton density has been expressed as a sum of terms on the cubes around the

total instanton number

$$I := \oint_{4d} \mathcal{I} = \sum_h \mathcal{I}_h = \sum_h \iota_h \in \mathbb{Z}. \tag{77}$$

A topological theta term can hence be defined in $d = 4$.[59] In $d \geq 5$, the instanton non-conservation defect $d\mathcal{I} \in \mathbb{Z}$ is the Yang monopole, which can be suppressed by some $W_5$, or forbidden by $W_5^{forbid}$ which contains a $(d-5)$-form $U(1)$ Lagrange multiplier, manifesting a $(d-5)$-form dual $U(1)$ global symmetry.

Piecing up the discussions above, we have our second main result, the $SU(2)$ lattice gauge theory refined to include instanton and topological theta term reads, for $d = 4$,

$$Z_\Theta = \left[\prod_{l'} \int_{SU(2)} dg_{l'}\right]\left[\prod_{p'} \sum_{m_{p'}=\pm} \int \frac{d^2\hat{n}_p}{4\pi}\right]\left[\prod_{c'} \int_{-\pi}^{\pi} \frac{d\mathcal{C}_{c'}}{2\pi}\right]\left[\prod_{h'} \sum_{\iota_{h'}\in\mathbb{Z}}\right]$$

$$e^{i\Theta I} \prod_p W_2(\lambda_p, m_p) \prod_c W_3(e^{i\mathcal{C}_c} \nu^*_{g_{l\in\partial c}, m_{p\in\partial c}, \hat{n}_{p\in\partial c}} + c.c.) \prod_h W_4(\mathcal{I}_h). \tag{78}$$

See [10] for the highly technical details of the $\nu$ function. The d.o.f. together form a mathematical structure that implements the second Chern class of the lattice $SU(2)$ Yang-Mills theory, to be explained with (143). For $d \geq 5$ there is no topological theta term, but there can be a weight $W_5$ or $W_5^{forbid}$ for the Yang monopole $d\mathcal{I}$. For $d = 3$, there is no $\iota_h$ and $W_4$, but we can additionally consider a CS term

$$Z_{kCS} = \left[\prod_{l'} \int_{SU(2)} dg_{l'}\right]\left[\prod_{p'} \sum_{m_{p'}=\pm} \int \frac{d^2\hat{n}_p}{4\pi}\right]\left[\prod_{c'} \int_{-\pi}^{\pi} \frac{d\mathcal{C}_{c'}}{2\pi}\right]$$

$$W_{CS}^k \prod_p W_2(\lambda_p, m_p) \prod_c W_3(e^{i\mathcal{C}_c} \nu^*_{g_{l\in\partial c}, m_{p\in\partial c}, \hat{n}_{p\in\partial c}} + c.c.), \tag{79}$$

for any $k \in \mathbb{Z}$. The generalization from $SU(2)$ to $SU(N)$ is straightforward using the Weyl alcove parameterization introduced at the end of Section 4.1.

An important aspect of $SU(N)$ Yang-Mills theory is the 1-form $Z(SU(N)) \cong \mathbb{Z}_N$ global symmetry $g_l \to g_l e^{i\beta_l}$ for $\beta_l \in (2\pi/N)\mathbb{Z}_N$ such that $e^{id\beta_p} = 1$ (which we gauged in (34) to obtain the Villainized $PSU(N)$ gauge theory). Since this transformation leaves $Dg_p$ invariant, it would not interfere with $y_p$ and $W_2$. Moreover, since in the continuum the CS integral also respects this 1-form global symmetry, the $\nu$ function—which is conceptually defined using the continuum CS integral—respects this symmetry on the lattice, hence so does the lattice CS and the lattice instanton density.

---

hypercube, and these terms are effectively playing the role of $\mathcal{C}_c$. A practical difference is that, in [8], the gauge fields in different hypercubes are under different gauge choices (referred to as the "complete axial gauge" in each hypercube), and hence the CS on a cube $c$ has two gauge choices—one each hypercube on the two sides of that cube (let us thus denote the CS value as $\mathcal{C}_c^{(\text{gauge of } h)}$ where $c \in \partial h$); while here, the CS on a cube uses only one gauge, which is good for defining the CS phase $W_{CS}^k$ in 3d (which is not part of the consideration in [8]).

Note that $e^{i2\pi\mathcal{I}_h} = e^{id\mathcal{C}_h}$ is gauge independent. However, if the logarithm of it is defined as $2\pi\mathcal{I}_h = d\mathcal{C}_h^{(\text{gauge of } h)}$ following [8], then gauge choice on $h$ determines the $\mathbb{Z}$ part of $\mathcal{I}_h$; this allows for a non-zero value of $\oint_{4d} \mathcal{I}$ in [8]. On the other hand, we defined $2\pi\mathcal{I}_h = d\mathcal{C}_h + 2\pi\iota_h$ with some new dynamical integer $\iota_h$. For weak enough field strength, if we take the value of $\iota_h$ that minimizes $\mathcal{I}_h$ (hence maximizes $W_4$), then the value of $\mathcal{I}_h$ will agree with that defined by [8].

[59]Just like in the abelian case (footnote 24), one can also let the theta become local and dynamical, but then for consistency we will need to introduce the Villainization integer field for this theta, that lives on the dual lattice link and couples to the CS density. We then obtain the lattice axion theory.

Upon on introducing a 2-form background for this 1-form global symmetry, there should be a self-anomaly in the presence of a non-trivial CS weight in 3d, a breaking of the $2\pi$ periodicity of the topological $\Theta$ angle in 4d,[60] and a mixed anomaly with the dual $(d-5)$-form $U(1)$ that forbids the Yang monopole in $d \geq 5$. We will leave to future works to investigate how to see these anomalies explicitly on the lattice.[61]

Now that we have explained the topologically refined $SU(N)$ lattice gauge theory, let us look back and discuss some important conceptual issue regarding the relation between the topological refinement of the $|SU(N)|$ nl$\sigma$m and that of the $SU(N)$ gauge theory, which is obviously more involved than the relation between Villainized $S^1$ nl$\sigma$m and Villainized $U(1)$ gauge theory.

Recall the link variable in nl$\sigma$m is geometrically interpreted as sampling some representative path in $SU(N)$; when two links are joined together on the lattice, their associated representative paths also concatenate in the obvious manner. But such kind of concatenation interpretation becomes subtle in gauge theory. In gauge theory, the $SU(N)$ is the space of holonomy around some loop. In our refinement at the plaquette level, we deform the loop with one parameter, so that the loop increases its size to wipe over the plaquette, as in (73), and throughout the process the holonomy indeed traces out a representative path in $SU(N)$. Now consider two plaquettes $p, p'$ joined at a shared vertex. Each plaquette has been associated with a path in $SU(N)$, one path connecting $\mathbf{1}$ and $Dg_p$ and the other connecting $\mathbf{1}$ and $Dg_{p'}$:

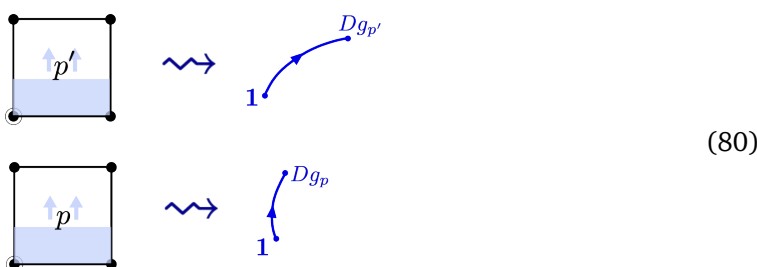

(80)

Figure 21: A path in $SU(N)$ associated with a plaquette.

Now that the two plaquettes $p, p'$ are joined together, do their associated paths somehow get joined together, too? There are two ways to increase the shaded area to fill up the two plaquettes, one filling up $p$ first and then $p'$, the other filling up $p'$ first and then $p$. Suppose we choose the lower left corner of $p$ (see picture below) as the starting point of the loop, then the holonomy around $p'$ (or around any portion of $p'$) needs to be conjugated by suitable Wilson lines. The two ways of filling lead to two different paths in $SU(N)$, though they share the same starting and ending points:

[60]See [101] for a lattice demonstration of this under the traditional notion [8] of lattice instanton.

[61]Let us explain what we should anticipate. To see these anomalies, we need to introduce the 2-form $\mathbb{Z}_N$ background gauge field. What we anticipate (and should further verify) is that there should be no way to define a $\nu$ function whose phase is invariant under the 1-form $\mathbb{Z}_N$ gauge transformation when the associated 2-form background is non-trivial. Suppose this is indeed so, then the remaining is straightforward: In $d = 3$, without a CS weight, this transformation of $\nu/|\nu|$ can be absorbed by a transformation of $e^{iC_c}$, leaving the theory invariant; but when the CS level is non-trivial, the theory will not be left invariant, manifesting the said self-anomaly. Related to this, in $d \geq 5$ with $W_5^{forbid}$, if we introduce a non-trivial Villainized background for the dual $(d-5)$-form $U(1)$ on the dual lattice, its background Dirac string field (which is a $(d-3)$-form integer field on the dual lattice) will couple to $e^{iC_c}$, i.e. a CS weight—which makes the 1-form $\mathbb{Z}_N$ anomalous—is attached on the Dirac string, manifesting the said mixed anomaly.

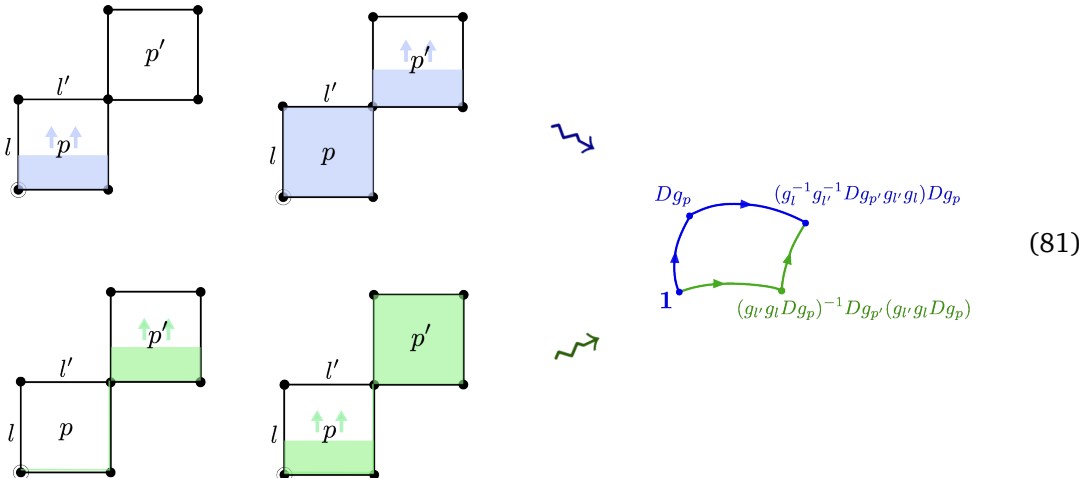

$$\tag{81}$$

Figure 22: Ambiguity in the concatenated paths when joining together two plaquettes.

Unlike joining two links in nl$\sigma$m where there is a natural ordering of which link comes first, when joining two plaquettes there is no natural choice of ordering,[62] so there is no way to determine which of the two ways of composing the representative paths is "the better choice".[63] (Similar issue happens when the two plaquettes are joined together at a shared edge instead of a shared vertex, or even more generally, joined together by a finite length Wilson line. The case of shared vertex pictured above is what will be relevant below.)

Why this issue worths any discussion? Because this is the underlying reason why passing from the topologically refined lattice $|SU(N)|$ nl$\sigma$m to $SU(N)$ gauge theory is not as simple as passing from Villainized $S^1$ nl$\sigma$m to Villainized $U(1)$ gauge theory. And the root of this lies in some important subject in category theory—delooping and Yang-Baxter equation.

Recall in $|SU(N)|$ nl$\sigma$m, the WZW curving d.o.f. on the plaquette is interpreted as sampling some 2d surface in $SU(N)$ (and two surfaces are consider equivalent if they bound a volume over which the WZW integral vanishes—recall the discussion below (54)),[64] and the skyrmion density over the cube is interpreted as (the WZW integral over) some 3d volume in $SU(N)$. However, in gauge theory, it would not be very useful to think of the CS d.o.f. on the cube as some kind of 2d surface in $SU(N)$ and think of the instanton density over a hypercube as some kind of 3d volume in $SU(N)$.

To understand this point, let us suppose we do think in this way and see what difficulties we run into. First consider the six plaquettes around a cube, which are joined together and leave no 1d boundary behind. Moreover, we can choose some ordering of filling up the plaquettes (such as the ordering we used in (36)), and once this ordering is fixed, the paths associated with each plaquette should join together (after conjugations by suitable Wilson lines) unambiguously and form some hexagonal loop in $SU(N)$:[65]

---

[62]We have encountered this when discussing higher form symmetries / degrees of freedom in Section 2.3.

[63]Using Mickelsson product (as in [40] and [96]) instead of geometrical concatenation does not help with this problem.

[64]In particular, some standard choice of surface (for the base of the pyramid in (66)) is used in defining $\mu/|\mu|$, and the deviation of WZW curving d.o.f. $e^{i\mathcal{W}_p}$ away from $\mu/|\mu|$ captures the fluctuation of the surface away from the standard choice (up to the said equivalence relation).

[65]If we have used a simplicial complex as the lattice, then a each tetrahedron will give rise to a quadrangle loop in $SU(N)$.

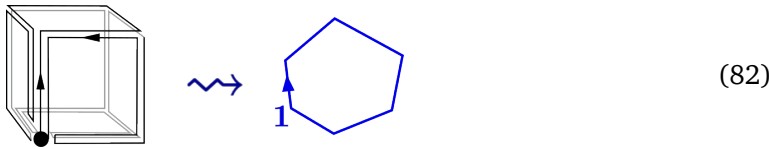

(82)

Figure 23: A hexagonal loop in $SU(N)$ associated with the boundary of a cube.

Suppose we interpret the CS d.o.f. on the cube as sampling some surface bounded by this hexagon (trying to mimic what happens for the WZW curving d.o.f. in nl$\sigma$m). So far there is no problem. Next let us consider the eight cubes around a hypercube, we expect the eight associated hexagonal surfaces to glue up (again after conjugations by suitable Wilson lines) into some closed surface, which will bound some volume whose WZW integral is interpreted as the lattice instanton density. But a careful inspection shows the eight hexagonal surfaces do not glue up to a closed surface, but a truncated octahedron

hypercube 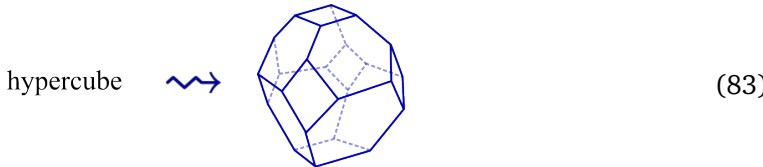

(83)

Figure 24: The hexagonal surfaces do not piece up to a closed surface associated with the boundary of a hypercube—the quadrangle loops are unfilled.

on which the quadrangle loops are left unfilled with surfaces. These quadrangle loops precisely come from (81). While we have fixed the ordering of filling up the plaquettes around each cube, in a hypercube there are still pairs of plaquettes which do not belong to a same cube but nonetheless join at a shared vertex,[66] and for each such pair, both orderings of filling up the two plaquettes will come up when we try to join the cubes around a hypercube, leading to the open quadrangle loops on the truncated octahedron.[67]

Can we fix some standard choice of surfaces to fill up such unfilled quadrangle loops, so that the truncated octahedron that the hypercube associates with becomes a closed surface?[68] We can do so, but a further constraint must be satisfied in the standard choices that we make. Since the WZW integral over the volume bounded by the truncated octahedron is to be interpreted as the lattice instanton density over the hypercube, we need to ensure that $d\mathcal{I}$ as well as $\oint_{4d} \mathcal{I}$ result in an integer. This requirement is equivalent to stating that when ten hypercubes piece up to a 5d-hypercube, we want the ten associated truncated octahedrons in $SU(N)$ to piece up

---

[66]These pairs of plaquettes are those that pair up in defining the cup product in footnote 22.

[67]If we have used a simplicial complex as the lattice, then the five tetrahedra in each 4-dimensional simplex will give rise to five filled faces on a cube in $SU(N)$, with the last face of the cube remaining unfilled due to the issue (81)—and the two plaquettes involved are, indeed, those that pair up in the cup product.

[68]Between two choices of surfaces to fill up the unfilled loop (81), their difference is, again, truncated to a $U(1)$ value, the WZW integral of the volume bounded between the two choice of surfaces.

Importantly, a close inspection shows this $U(1)$ now forms a trivial bundle over the space of choices of the paths around. Briefly speaking, this is because the four paths around are really only determined by two paths, one associated with $p$ and the other associated with $p'$, as shown in (81). In Section 4.1, there is a non-trivial constraint on the space of the link variables (see footnote 49), that gives rise to a non-trivial $\pi_2$ for the space of links variables, which then leads to the non-trivial WZW curving $U(1)$ bundle; by contrast, here, the corresponding constraint will be automatically satisfied, so that the space of the possible choices of paths in (81) has trivial $\pi_2$, and therefore any $U(1)$ bundle on it is necessarily trivial.

to a closed 3d volume without any 2d surface leftover. This is a highly non-trivial constraint imposed on the standard choice of quadrangle surface. In fact, this constraint is a generalized version of Yang-Baxter equation, and specifying a standard choice for the quadrangle surface is an example of braiding data in category theory, as we will discuss in Sections 5.3, in particular (112) and below.

It is in general a very difficult task to find non-trivial solutions to a Yang-Baxter equation. This is why, in our construction here, we do not literally take the geometrical interpretation of those fields in the refined nl$\sigma$m and "put them on lattice cells of one higher dimension" to get the geometrical interpretation of the fields in the refined gauge theory: We do not think of the lattice CS d.o.f. on the cubes as sampling some surface in $SU(N)$ and the instanton density over the hypercube as the WZW phase of some volume in $SU(N)$. Instead, in our construction, the geometrical interpretation is about how to interpolate the gauge fields on the links into the plaquettes and the cubes, much like in the previous work [8], except we consider different possibilities of interpolations in a manner guided by category theory [10]. *But remarkably, although we do not start off associating the lattice CS to some WZW phase, in [10] we show that after some calculations the expression of the CS saddle indeed involves a WZW phase plus some corrections—that is, while the Yang-Baxter equation issue never explicitly came up, it might has been automatically solved.* We will mention this again at the end of Section 6, and hopefully we will understand this more concretely in future works.

# 5  Category theory foundation

In this section we first explain how to cast the previously known examples in Section 2 into the language of category theory, and then we will see how our construction in Section 4 is naturally motivated from there.

The involvement of category theory in the construction of lattice model is, however, more than just being motivational. We have been saying "we want a model that captures the $\pi_3$ topology of the nl$\sigma$m or the Yang-Mills theory on the lattice", but so far we only have some intuitive idea what this "capture" is intended to mean. After some initial build-ups, in Section 5.5, we will turn this intuitive goal into a mathematical statement, i.e. making a proposal for what the mathematical requirements are for a lattice QFT to "capture the $\pi_3$ topology", and why the construction in Section 4 indeed serves this purpose.

We will begin by introducing some basics of category theory so that we can setup our notations and eventually lead the discussion towards the concepts that we will need for our construction. However, this section is not intended as a piece of comprehensive and/or rigorous introductory material to the subject of category theory itself. A gentle introduction to category theory containing some physics oriented perspectives can be found in [44]. For more comprehensive and rigorous introduction, one may consult textbooks and review articles of different levels and with different emphases. The online wiki nLab is a very useful source of knowledge on this subject. And a mathematically rigorous treatment of the particular categories that we need for our construction—built upon but different from what is existing in the current literature (a combination of [37,38,95,96] in particular)—is beyond the scope of this physics paper and will be a task for mathematics oriented subsequent work.

## 5.1  Strict categories, and the known examples

We begin with *strict* higher categories. Being "strict" implies they are straightforward to define and easy to understand, but not as powerful as the more general higher categories in being descriptive. It is not surprising that the previously known examples introduced in Section 2 are all described in terms of strict higher categories.

A 0-category $C$ is just a set $C_0$.[69] The elements in it are often called "objects" in the context of category theory. Often times $C_0$ can be endowed with extra structures, for examples it can be a group, a topological space or smooth manifold, etc.

A 1-category $C$, which is what a "category" usually refers to, has two sets: a set $C_0$ of objects, and a set $C_1$ of all "morphisms", or relations, between objects. Of course the two sets should be somehow related. There are some maps between them:

- Intuitively, $C_1$ should have a "source map" s and a "target map" t to $C_0$, so that $\mathsf{s}(f) = a, \mathsf{t}(f) = b$ means $f$ is a morphism (relation) from object $a$ to object $b$, which we can denote as $b \xleftarrow{f} a$.[70] Because of these two maps, we will often denote a category $C$ as $C_1 \rightrightarrows C_0$. We use $C_1|_{b,a}$ to denote the subset of $C_1$ where the source and the target are restricted to $a$ and $b$ respectively.[71]

- Morphisms (relations) should be able to compose, i.e. there is a map $\circ$ from $C_1 \times_{C_0}^{\mathsf{s},\mathsf{t}} C_1$ to $C_1$ (where the fiber product notation $X \times_Z^{u,v} Y := \{(x,y) \in X \times Y | u(x) = v(y) \in Z\}$, and sometimes we might omit the $u,v$ superscripts), or say from $C_1|_{c,b} \times C_1|_{b,a}$ to $C_1|_{c,a}$ for every $a,c \in C_0$. This specifies how $c \xleftarrow{g} b$ and $b \xleftarrow{f} a$ are composed to some $c \xleftarrow{g\circ f} a$. Moreover, when composing three morphisms, the composition should be associative. (Sometimes we might omit the "$\circ$" in composition.)

- There is a map i from $C_0$ to $C_1$, which for each $a \in C_0$ specifies an "identity morphism" $\mathbf{1}_a := \mathsf{i}(a) \in C_1|_{a,a} \subseteq C_1$, such that under composition, $f \circ \mathbf{1}_a = f$ for any $f$ with $\mathsf{s}(f) = a$, and $\mathbf{1}_a \circ g = g$ for any $g$ with $\mathsf{t}(g) = a$.

If $C_0$ and $C_1$ are endowed with some extra structure, then it is natural to require these maps to respect the extra structure. Particularly, if $C_0$ and $C_1$ are both manifolds, then it is natural to require these maps to be smooth—apparently this will be a key point in our application, and this point will be systematically formulated in terms of *internalization* in Section 5.2.

A morphism $b \xleftarrow{f} a$ is invertible if there exists an $a \xleftarrow{f^{-1}} b$ such that $f^{-1} \circ f = \mathbf{1}_a$, $f \circ f^{-1} = \mathbf{1}_b$.[72] Now, we are ready to see that in category theory, a group can at least be perceived in two ways: either as a 0-category (set) $G$ endowed with the those extra structures that make it a group, or as a 1-category $BG$, where $BG_0$ has only a single object $*$, and every morphism in $BG_1$ is invertible, i.e. $BG := (G \rightrightarrows *)$. Such relation between $G$ and $BG$ is a simple example of *delooping* (from $G$ to $BG$) and *looping* (from $BG$ to $G$), a concept that will be important in our application when elevating the construction for nl$\sigma$m to the construction for gauge field.

One might note that the notation $BG$ is often used to denote the classifying space of $G$. This is not a coincidence. The classifying space, which we will denote as $|BG|$ to avoid confusions,

---

[69]Rigorously speaking, there are large versus small categories, where large categories can involve collections such as proper classes which are logically "larger" than any possible set (e.g. the collection of all sets is a proper class, which is not a set in the sense that it is not allowed to be taken as an element of any set or proper class, hence resolving the Russell paradox). However, such issue does not seem very important in physics. The categories that are directly involved in our detailed construction are all small categories.

[70]Usually the arrow is drawn from left to right. In this paper we will often use the convention from right to left, because when we compose functions or operations this is the conventional order of action.

[71]Usually the notation is $\mathrm{Hom}(a,b)$ or $\mathrm{Hom}_C(a,b)$. Here we emphasize that we prefer to primarily view the morphisms altogether as a whole set $C_1$, rather than to primarily view the morphisms between each given pair of objects, $\mathrm{Hom}(a,b)$, as a set. Of course these two views are equivalent for now, but when we impose more structure on sets, the former view will be more suitable for generalization via *internalization* in Section 5.2, which will be important for our work, while the later view is more suitable for generalization via *enrichment*, which we will not focus on (though also important in general).

[72]It is also possible that only one of these two conditions can be satisfied, or each condition can be satisfied but with two different "$f^{-1}$"s. Therefore in general we should define the notions of left inverse $f_L^{-1}$ and right inverse $f_R^{-1}$ of $f$.

is usually infinite dimensional, and it can be obtained from the category $BG$ via the procedure of *geometric realization* which we will introduce near the end of Section 5.4.

More generally, a 1-category where every morphism in $C_1$ is invertible, but not necessarily with only a single object in $C_0$, is called a *groupoid*. (It is therefore said that the notion of groupoid is the "horizontal categorification" of the notion of group. More generally, a 1-category with a single object—but not necessarily with every morphism invertible—can be viewed as the delooping of a monoid, and thus the notion of 1-category is the "horizontal categorification" of the notion of monoid.) An intuitive example of a groupoid is an *action groupoid* $X \times G \rightrightarrows X$, where there is a set $X$ and a group $G$ acting on $X$, so that a morphism $(x, g) \in X \times G$ is depicted as $gx \xleftarrow{(x,g)} x$ or more simply $gx \xleftarrow{g} x$.

Groupoids are very common in our application. For some examples:

1. Given a continuum manifold $\mathcal{M}$ we can define its free path space $\mathcal{PM}$.[73] Now we consider $\bar{\mathcal{P}}\mathcal{M}$, which is $\mathcal{PM}$ with equivalence up to "thin homotopy"—roughly speaking up to reparametrization and identifications like

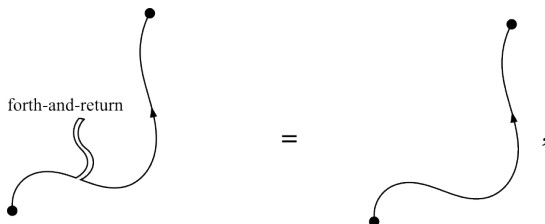

   so that the concatenation of paths is associative, and has the identity path for each point and the inverse path for each path.[74] We thus defined the *path groupoid* $\bar{P}_1\mathcal{M} := (\bar{\mathcal{P}}\mathcal{M} \rightrightarrows \mathcal{M})$.[75]

   A closely related concept is the *fundamental groupoid* $\Pi_1\mathcal{M}$ (we will explain this name in the next subsection), which is like the path groupoid except the identification of two paths that share the same end points is not only made under thin homotopy, but any homotopy (any interpolation). Clearly, if $\mathcal{M}$ is 1d, then $\Pi_1\mathcal{M} = \bar{P}_1\mathcal{M}$, because any homotopy between paths is necessarily thin.

2. A lattice keeping only the vertices and the links but ignoring plaquettes and higher cells gives a groupoid $\bar{\mathcal{L}}_1 \rightrightarrows \bar{\mathcal{L}}_0$, where $\bar{\mathcal{L}}_0$ is the set of all vertices, and $\bar{\mathcal{L}}_1$ is the set of all lattice paths obtained by joining links. Each vertex indeed has the trivial identity path, and each path indeed has the inverse path by reversing the arrow.

3. For $\mathcal{M} = S^1$, the path groupoid is $S^1 \times \mathbb{R} \rightrightarrows S^1$, which (upon introducing a metric on the circle) is also an example of action groupoid. Apparently this structure will be related

---

[73]Compared to the pointed path space $\mathcal{P}_*\mathcal{M}$ we introduced before, the free path space $\mathcal{PM}$ does not fix a starting point for the paths. They are related by the fibre bundle $\mathcal{P}_*\mathcal{M} \to \mathcal{PM} \to \mathcal{M}$.

[74]More exactly, a path is a smooth function $\gamma(\tau) : [0,1] \to \mathcal{M}$. Composition, i.e. concatenation, is defined by $(\gamma' \circ \gamma)(\tau)$ equals $\gamma(2\tau)$ if $0 \leq \tau \leq 1/2$ and $\gamma'(2\tau - 1)$ if $1/2 \leq \tau \leq 1$. To ensure the smoothness around the concatenation point, we need a "sitting instant" condition that $\gamma(\tau)$ stays constant for $|\tau - 0| < \epsilon$, $|\tau - 1| < \epsilon$ for some small $\epsilon$.

The "thin homotopy" identification is that, two paths $\gamma_0$ and $\gamma_1$ that share the same end points are considered identified if they are related by a "thin homotopy", i.e. there is a 2-parameter function $\widetilde{\gamma}(\tau, \lambda)$ interpolating from $\widetilde{\gamma}(\tau, 0) = \gamma_0(\tau)$ to $\widetilde{\gamma}(\tau, 1) = \gamma_1(\tau)$, such that everywhere the differentials $(\partial_\tau \widetilde{\gamma}, \partial_\lambda \widetilde{\gamma})$ fails to be full rank, i.e. spans a less than 2 dimensional vector space tangent to $\mathcal{M}$ (so the image of $\widetilde{\gamma}(\tau, \lambda)$ in $\mathcal{M}$ has zero area, hence "thin"); moreover, $\widetilde{\gamma}$ itself satisfies the sitting instant condition in both $\tau$ and $\lambda$. See e.g. [50].

[75]We would like the spaces $\mathcal{PM}, \bar{\mathcal{P}}\mathcal{M}$ (as well as any maps involved) to be "smooth". But $\mathcal{PM}, \bar{\mathcal{P}}\mathcal{M}$ are in general infinite dimensional, and a suitable generalization of "smooth" to the infinite dimensional cases is known as "diffeological". We will mention this in Section 5.2.

to the d.o.f. used in the Villain model, and (47) is essentially taking the thin homotopy identification in defining this path groupoid.

4. Another example of action groupoid is $S^2 \times SU(2) \rightrightarrows S^2$, which will apparently be related to the spinon decomposition.

Having introduced 0- and 1-category, it is not hard to envision that higher categories involve more layers of higher morphisms equipped with suitable maps in-between. But now there arise definitions of different levels of strictness. The more strict ones are easier to define, but the less strict ones are more flexible and thus have higher descriptive power. Here we first introduce the strict higher categories, which are sufficient to describe the known examples in Section 2; in Section 5.3 we briefly introduce more flexible higher categories which are commonly used in describing topological phases and generalized symmetries; for our construction in Section 4, we will need an even more flexible version of higher categories to be introduced in Section 5.4.

A *strict* 2-category has, first of all, a 1-category $C_1 \rightrightarrows C_0$, but in addition, there is a set $C_2$ of "2-morphisms" between pairs of 1-morphisms which share the same source and target objects. Pictorially a 2-morphism $\varphi$ takes a globular shape

$$b \overset{f}{\underset{g}{\Longleftarrow}} \Downarrow \varphi \quad a \tag{84}$$

.

Thus $C_2$ has a source and a target map to $C_1$, such that when further taking the source or target map to $C_0$, we require $\mathsf{ss}(\varphi) = \mathsf{st}(\varphi)$, $\mathsf{ts}(\varphi) = \mathsf{tt}(\varphi)$. There are maps $\circ_v$ from $C_2 \times_{C_1}^{\mathsf{s,t}} C_2$ to $C_2$ and $\circ_h$ from $C_2 \times_{C_0}^{\mathsf{ss,tt}} C_2$ that define the *vertical and horizontal compositions*

$$
\begin{aligned}
\Downarrow \varphi' \circ_v \varphi \;&=\; \overset{\Downarrow \varphi}{\underset{\Downarrow \varphi'}{\phantom{x}}} \\[2em]
\overset{f' \circ f}{\underset{g' \circ g}{\Downarrow \varphi' \circ_h \varphi}} \;&=\; \overset{f'}{\underset{g'}{\Downarrow \varphi'}} \;\; \overset{f}{\underset{g}{\Downarrow \varphi}}
\end{aligned}
\tag{85}
$$

.

We can also denote them as $\circ_1$ and $\circ_0$ respectively, as the 2-morphisms being composed are joined along a 1- and a 0-morphism respectively. They are required to satisfy vertical and horizontal associativity, as well as interchangeability, i.e. in each kind of these situations,

$$\tag{86}$$

the composition result is unique regardless of the order of composition, similar to the associativity requirement for 1-morphisms. Finally there is a map i from $C_1$ to $C_2$ that specifies, for each 1-morphism $f$, the identity 2-morphism $\mathsf{i}(f) = \mathbf{1}_f$ under vertical composition. On the other hand, in horizontal composition, when $g' \overset{\varphi'}{\Longleftarrow} f'$ is given by $f' \overset{\mathbf{1}_{f'}}{\Longleftarrow} f'$, it is conventional to collapse its globular shape in the pictorial notation:

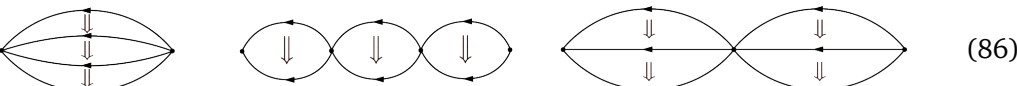

$$\tag{87}$$

and such a horizontal composition is called a left *whiskering*. Similar for right whiskering. A strict 2-groupoid is a strict 2-category in which each 1-morphism is invertible as in a 1-groupoid and moreover each 2-morphism is vertically invertible (hence also horizontally invertible, using whiskering and interchangeability).

The generalization to strict $n$-categories should be obvious. Now we emphasize two very useful and illuminating points of view:

- A strict $n$-category can be naturally seen as a special kind of strict $m$-category for arbitrary $m \geq n$, such that all $k$-morphisms for $n < k \leq m$ are identity morphisms, i.e. $C_k = \{\mathbf{1}_u | u \in C_{k-1}\} \cong C_{k-1}$ for $n < k \leq m$. (We may as well take $m$ towards infinity.) This point of view will be extremely useful for understanding many discussions below.

- While in a 1-category, the 1-morphisms between two given objects forms a set $C_1|_{b,a}$, in a strict $n$-category, the $q$-morphisms between two objects for all $1 \leq q \leq n$ form a strict $(n-1)$-category, sometimes called the *hom-category* between $a$ and $b$, which we will denote as $C|_{b,a}$, with $(C|_{b,a})_0 = C_1|_{b,a}$.[76]

Some examples of strict higher categories—more particularly, strict higher groupoids—that will appear in our lattice theory application include:

1. Given a $d$-dimensional continuum manifold $\mathcal{M}$ we can define a *strict path $d$-groupoid* $\bar{P}_d\mathcal{M} := (\bar{\mathcal{P}}_d\mathcal{M} \rightrightarrows \cdots \rightrightarrows \bar{\mathcal{P}}\mathcal{M} \rightrightarrows \mathcal{M})$, where $\mathcal{P}_k\mathcal{M}$ is the space of "$k$-paths", the interpolation between two elements of $\mathcal{P}_{k-1}\mathcal{M}$ that share the same source and target in $\mathcal{P}_{k-2}\mathcal{M}$, starting with $\mathcal{P}_0\mathcal{M} = \mathcal{M}$ and $\mathcal{P}_1\mathcal{M} = \mathcal{P}\mathcal{M}$, and $\bar{\mathcal{P}}_k$ is $\mathcal{P}_k$ with identification under thin homotopy, in order for this $d$-groupoid to be strict.[77] Geometrically, if $k \leq d$, a generic element in $\bar{\mathcal{P}}_k\mathcal{M}$ wipes over a $k$-dimensional surface (topologically a $k$-disc) in $\mathcal{M}$.

   Being strict makes this category easy to think of, but then not so powerful in capturing the full homotopy information of $\mathcal{M}$.[78] However, it is still sufficient for many physical application purposes. This perspective of continuum manifold is useful for relating lattice QFT to continuum QFT.

2. A $d$-dimensional lattice gives rise to a strict $d$-groupoid $\bar{\mathcal{L}} = (\bar{\mathcal{L}}_d \rightrightarrows \cdots \rightrightarrows \bar{\mathcal{L}}_1 \rightrightarrows \bar{\mathcal{L}}_0)$. The $\bar{\mathcal{L}}_1 \rightrightarrows \bar{\mathcal{L}}_0$ part has been introduced before, while $\bar{\mathcal{L}}_i$ for $i \geq 2$ is roughly speaking $i$-dimensional surfaces (including degenerate ones) on the lattice, but the source and

---

[76] This can be phrased in terms of *enrichment*, which roughly speaking means the hom-set $C_1|_{b,a}$ in a 1-category is replaced by some structure richer than merely a set. Thus, a strict $n$-category is a 1-category enriched by strict $(n-1)$-category. In this paper we will not have much emphasis on the enrichment perspective, though it is generally important in category theory.

[77] Similar to footnote 74, there are the higher dimensional versions of the sitting instant requirement and the thin homotopy (non-full rank interpolation) equivalence. We can take the notion of e.g. "strong 2-track" in [102] and generalize it to higher dimensional paths.

[78] Any approach to construct a strict higher groupoid out of a manifold, regardless of the detailed method, is incapable of capturing the full homotopy information of a generic manifold [42, 43]. (In general, the information of Whitehead product and beyond will be lost. Our particular construction further losses all the homotopy $n$-type information for $n > d$.) We will mention more about this in footnote 97. In order to capture the full homotopy information, suitable notion of weak higher category must be used, and in Section 5.4 we will introduce one such notion, simplicial groupoid, i.e. Kan complex, that is widely used.

If we want to define a path $n$-groupoid that captures the full homotopy $n$-type information for some finite $n$, there are some other particular constructions. For instance, in order to construct a weak path 3-groupoid that captures the full homotopy 3-type information, in [102], identification of 2-paths under a "laminated" condition, which is more stringent than thin homotopy, is taken, so that some 2-paths identified under thin homotopy now become distinct under this laminated condition, and the path 3-groupoid becomes a less strict kind of category—a Gray 3-category, which will be introduced in Section 5.3. Holonomies valued in Gray 3-categories can hence be considered.

target have to be specified. In $\bar{\mathcal{L}}_2$, two elements that wipe over the same plaquette(s) can still be different, but related by whiskering, e.g.

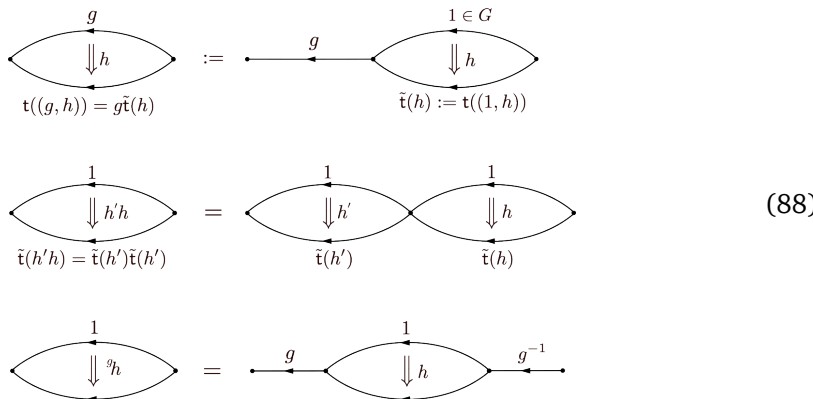

Likewise for $\bar{\mathcal{L}}_i$, $i > 2$. We will denote by $\bar{\mathcal{L}}_{\leq m}$ (where $m \leq d$) the $m$-category obtained from $\bar{\mathcal{L}}$ by keeping up to $\bar{\mathcal{L}}_m$ and ignoring the higher morphisms (or equivalently, keeping only the identity higher morphisms).

Just like the strict path $n$-groupoid for a continuum manifold, the strict $d$-groupoid $\bar{\mathcal{L}}$ does not capture the full homotopy information of the manifold that the lattice is discretizing, but it is sufficient for many physical applications.

3. We can ask whether $BG$ can be delooped once more into a strict 2-groupoid $B^2 G := (G \rightrightarrows * \rightrightarrows *)$. This is only well-defined when $G$ is abelian, due to the requirement of interchangeability between vertical and horizontal compositions (this is known as the Eckmann-Hilton argument). Obviously, when $G$ is abelian, it can be delooped arbitrarily many of times into $B^n G$. And obviously, this will be related to what we discussed in Section 2.3, that higher form gauge fields must be abelian.

4. More generally, a strict 2-groupoid with a single object, but not necessarily with a single 1-morphism, is called a *strict 2-group*. It can be proven that strict 2-groups always take the "crossed module" form $B\mathcal{G} := (G \ltimes H \rightrightarrows G \rightrightarrows *)$ where $G, H$ are groups with a homomorphism $\tilde{\mathsf{t}}$ from $H$ to $G$ [43, 50, 103]:

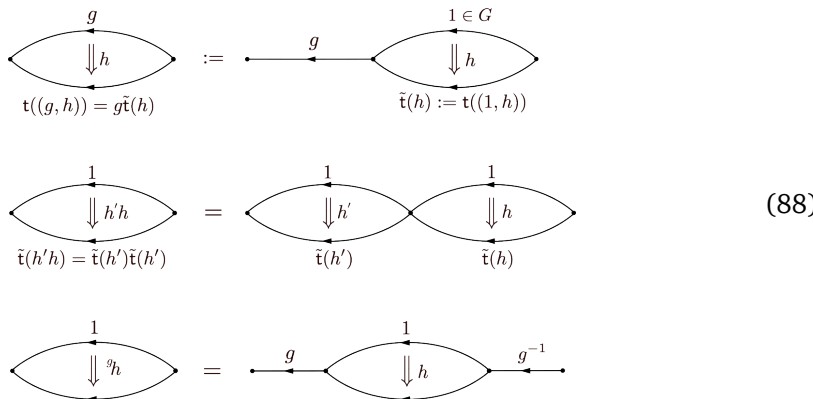

$$\tag{88}$$

(more general compositions can be derived using the associativity and interchangeability conditions, with the fact that $(1,1) \in G \ltimes H$ is the identity for both the vertical and the horizontal composition), and this is the delooping of an action groupoid $\mathcal{G} := (G \times H \rightrightarrows G)$ equipped with some extra structures (that make $G$ a group, to which $H$ has a homomorphism $\tilde{\mathsf{t}}$, along with a $G$ action back on $H$). The interchangeability between vertical and horizontal compositions requires $\ker(\tilde{\mathsf{t}})$ to be a subgroup of the center $Z(H)$. It is apparent that the case of $U(1) \times \mathbb{R} \rightrightarrows U(1) \rightrightarrows *$ will be related to the d.o.f. in Villainized $U(1)$ gauge theory, and it deloops the groupoid $S^1 \times \mathbb{R} \rightrightarrows S^1$ that we have discussed before.

(Even more generally, a strict 2-category with single object can be viewed as the delooping of a 1-category equipped with extra structure, and a 1-category with such extra structure is called a *strict monoidal category*.)

5. The strict 2-groupoid $S^2 \times SU(2) \times \mathbb{R} \rightrightarrows S^2 \times SU(2) \rightrightarrows S^2$, whose elements look like

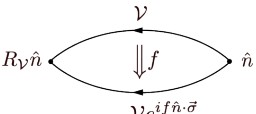

will apparently be related to the spinon decomposition of $S^2$ nl$\sigma$m. The structure (38) is contained in the maps involved in the definition of this strict 2-category. In particular, given source and target objects, $(S^2 \times SU(2))|_{\hat{n}',\hat{n}} \cong U(1)$, and given source and target 1-morphisms, $(S^2 \times SU(2) \times \mathbb{R})|_{(\hat{n},\mathcal{V}),(\hat{n},\mathcal{V}e^{i\theta\hat{n}\cdot\vec{\sigma}})} \cong \mathbb{Z}$. Unwinding more structurally in order to compare with (38), we have

$$
\begin{aligned}
(S^2 \times SU(2) \times \mathbb{R})^{[2]} &\rightrightarrows S^2 \times SU(2) \times \mathbb{R} \\
&\quad\downarrow {}^{(s,t)} \\
(S^2 \times SU(2))^{[2]} &\rightrightarrows S^2 \times SU(2) \\
&\quad\downarrow {}^{(s,t)} \\
(S^2)^2 &\rightrightarrows S^2 \,,
\end{aligned}
\tag{89}
$$

where $(S^2 \times SU(2))^{[2]} := (S^2 \times SU(2)) \times^{(s,t),(s,t)}_{(S^2)^2} (S^2 \times SU(2)) \cong S^2 \times SU(2) \times U(1)$ and $(S^2 \times SU(2) \times \mathbb{R})^{[2]} := (S^2 \times SU(2) \times \mathbb{R}) \times^{(s,t),(s,t)}_{S^2 \times SU(2) \times U(1)} (S^2 \times SU(2) \times \mathbb{R}) \cong S^2 \times SU(2) \times \mathbb{R} \times \mathbb{Z}$.

6. The structure (54) is captured by the strict 2-groupoid $\bar{\mathcal{P}}_2 S^3 \times U(1)/WZW \rightrightarrows \bar{\mathcal{P}} S^3 \rightrightarrows S^3$. (Here we have identified paths related by thin homotopy, while in (54) we did not; this is not a big issue because our purpose is to capture the WZW evaluation, which is indeed unaffected by any thin homotopy. In particular, $\bar{\mathcal{P}} S^3 \neq \mathcal{P} S^3$, but $\mathcal{P}_2 S^3 \times U(1)/WZW = \bar{\mathcal{P}}_2 S^3 \times U(1)/WZW$.) Including the Villainzation layer in (51) above (54), the structure is captured by the strict 3-groupoid $(\bar{\mathcal{P}}_2 S^3 \times U(1)/WZW) \times \mathbb{R} \rightrightarrows \bar{\mathcal{P}}_2 S^3 \times U(1)/WZW \rightrightarrows \bar{\mathcal{P}} S^3 \rightrightarrows S^3$. As mentioned there, the problem of using this structure for a lattice theory is that $\bar{\mathcal{P}} S^3$ is infinite dimensional. Our task is to find a finite dimensional 3-category in Section 5.5 which is equivalent to this infinite dimensional strict 3-category in a suitable sense. Understanding such "equivalence in a suitable sense" is why higher category theory is necessary; otherwise, without category theory, it is hard to move beyond (54).

So far we have described the general structure of strict higher categories. But more interesting is the relation between structures.

Given two 0-categories, i.e. sets, we would think about functions mapping between them, $D \xleftarrow{F} C$. Just from this notation, we realize a deep, interesting point, that all 0-categories together form a 1-category Set, or say 0Cat, where the objects in $\mathsf{Set}_0$ are sets, and the morphisms in $\mathsf{Set}_1$ are functions between sets.[79] This point of view is not only important purely mathematically, but is directly useful for the concept of *internalization* in Section 5.2, which will in turn underlie our construction of lattice d.o.f..

It is then natural to ask what maps between two 1-categories. The notion of *functor* naturally comes up (although for our application we will need a more general notion of functor, i.e. *anafunctor*, which we will explain in Section 5.2): A functor $F$ from 1-category $C$ to 1-category

---

[79]The issue mentioned in footnote 69 appears here. To equate "all 0-categories" to "all sets", we should really mean "all small 0-categories". The same is understood in further discussions below. (Set itself is a large 1-category because the collection of all sets is not a set but a proper class. If we want—though often there is no intrinsic problem to work with large categories—we can always further restrict "all sets" to sets whose cardinalities are not too large, so that the collection of them is still a set, and the collection thus becomes a small 1-category. All these should not matter in physics, because we do not expect sets with indefnitely large cardinalities to be directly involved in physics anyways.)

$D$, again denoted as $D \xleftarrow{F} C$, involves a function $F_0$ from $C_0$ to $D_0$ and a function $F_1$ from $C_1$ to $D_1$, pictorially

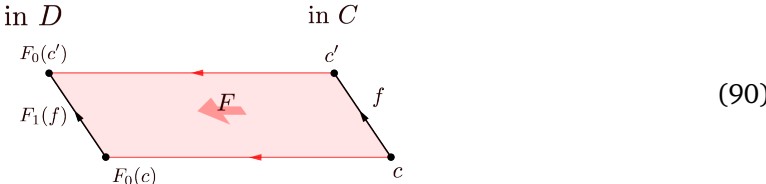

$$(90)$$

such that the source and target maps, the composition, and the identity specifications are all preserved.[80] (In the special case when $C$ and $D$ are $BG$ and $BH$, a functor from $BG$ to $BH$ is apparently a group homomorphism from $G$ to $H$.) Similar to functions between 0-categories, functors between 1-categories can be composed in the obvious manner, $(E \xleftarrow{G \circ F} C) := (E \xleftarrow{G} D \xleftarrow{F} C)$, and the composition is associative.

A fundamental reason that makes the notion of 1-category more powerful than the notion of set (0-category) is, now that we have two layers, there is a new kind of relation from $C$ to $D$ that has no non-trivial counter-part in 0-categories: We can also consider a map from $C_0$ to $D_1$. But then what would $C_1$ map to? Recall we may view a 1-category as a special kind of 2-category which only has identity 2-morphisms, i.e. $D_2 = \{\mathbf{1}_h | h \in D_1\} \cong D_1$, therefore $C_1$ must somehow map to this $D_2$. This leads to the notion of *natural transformation*. We can think of a natural transformation $\Phi$ pictorially as

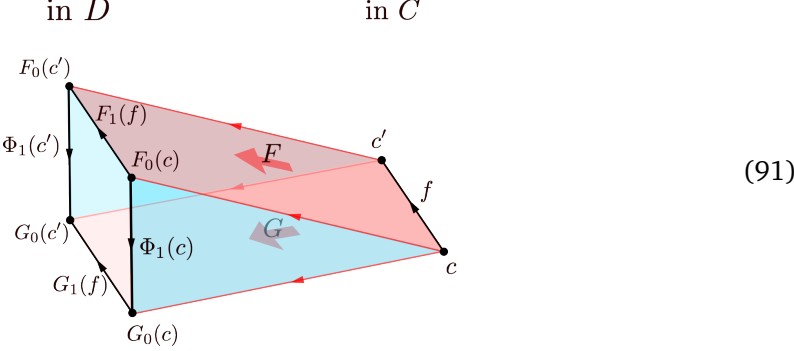

$$(91)$$

where the top and bottom surfaces reduce to two functors $F, G$ mapping from $C$ to $D$, and there is a function $\Phi_1$ mapping from $C_0$ to $D_1$ such that it reduces to $F_0$ and $G_0$ when taking the source and target in $D$. Moreover there is a $\Phi_2$ mapping from $C_1$ to $D_2$, where $D_2$ only contains identity 2-morphisms. More exactly, $f \in C_1$ is mapped to the rectangular shape on the left, which should represent a 2-morphism in $D_2$, and since the only available 2-morphisms in a 1-category are identity 2-morphisms, we conclude the only possibility is $\Phi_2(f) = \mathbf{1}_{\Phi_1(c') \circ F_1(f)} = \mathbf{1}_{G_1(f) \circ \Phi_1(c)} \in D_2$, which in turn implies $\Phi_1(c') \circ F_1(f) = G_1(f) \circ \Phi_1(c) \in D_1$. Therefore, $\Phi_2$ does not contain any new information than what is already contained in $F_1, G_1, \Phi_1$, rather it provides a consistency constraint between these three functions. Such a $\Phi$ is said to be a natural transformation from functor $F$ to functor $G$. Thus, apparently we should denote a natural transformation as

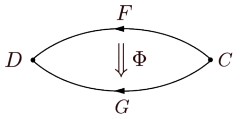

From this picture, we see all 1-categories together should form a strict 2-category $\mathsf{Cat}$, or say $\mathsf{1Cat}$, and they indeed do. The vertical composition of natural transformations is obvious;

---

[80]It is common to abbreviate both $F_0$ and $F_1$ as just $F$, but keeping the subscript in mind is helpful for generalizing towards the crucial notion of *anafunctor* in Section 5.2.

using the interchangeability condition, horizontal composition can be defined, too, known as Godemant product.

A natural transformation $\Phi^{-1}$ from $G$ to $F$ is the inverse (under vertical composition) of $\Phi$ if $(\Phi^{-1})_1 = (\Phi_1)^{-1}$—and this may or may not exist for a given $\Phi$. Just like how the equivalence (equipotence) between two sets is established by the existence of an invertible function between them, we can say two functors are equivalent if there is an invertible natural transformation (also called *natural isomorphism*) between them, though the two functors may not be equal.

With this notion of equivalence between functors, now we can define the notion of "inverse" for a functor at two levels of strictness. Intuitively we can define *the* inverse $C \xleftarrow{F^{-1}} D$ of $D \xleftarrow{F} C$ by strictly requiring $C \xleftarrow{F^{-1} \circ F} C = \mathbf{1}_C$ and $D \xleftarrow{F \circ F^{-1}} D = \mathbf{1}_D$. If two categories are related by invertible functors in such a strict sense, the two categories are strictly isomorphic at each level (we may colloquially say they are the same). However, often a less strict notion is more useful, especially when the strict inverse does not exist. We say a functor $C \xleftarrow{\bar{F}} D$ is *an* inverse of a functor $D \xleftarrow{F} C$, if the composed functor $C \xleftarrow{\bar{F} \circ F} C$ has a natural isomorphism to $\mathbf{1}_C$, and $D \xleftarrow{F \circ \bar{F}} D$ also has a natural isomorphism to $\mathbf{1}_D$. We say the existence of such pair $F, \bar{F}$ establishes a *natural equivalence* between the 1-categories $C$ and $D$.

This is the first scenario where the flexibility of category theory manifests—and we will need more kinds of flexibility later in order to arrive at the lattice construction we desire. It can be seen that the definition of natural equivalence between 1-categories looks remarkably similar to the definition of homotopy equivalence between topological spaces, whose contrast with the strict notion of homeomorphism shows the power of flexibility. Indeed, a homotopy between two manifolds induces a natural equivalence between their fundamental groupoids.

It is easy to prove that an equivalent—but often more useful in practice—way to state natural equivalence between $C$ and $D$ is to say $F$ is "essentially surjective and fully faithful". "Essentially surjective" means while $D_0 \xleftarrow{F_0} C_0$ might not be surjective, any $d \in D_0$ must be related via some invertible morphism in $D_1$ to (in generalization of being strictly equal to) some $F_0(c) \in D_0$. "Fully faithful" means for any $a, b \in C_0$, the restriction of $F_1$ to $C_1|_{b,a}$ is a bijection between $C_1|_{b,a}$ and $D_1|_{F_0(b),F_0(a)}$ (in particular, "full" refers to the surjection condition and "faithful" the injection condition). From these conditions it is not hard to construct an inverse functor $\bar{F}$ that is also essentially surjective and fully faithful.[81] Thus, the map between two naturally equivalent 1-categories is still bijective in the traditional sense at the morphism layer given the source and the target, but becomes more flexible at the object layer.

Let us discuss a simple example of natural equivalence relevant to lattice QFT. Consider two 1d lattice loops, but with different numbers of vertices. We feel they should be equivalent in some suitable sense, since they both discretized a 1d space(time) circle. Indeed, as 1-categories they are naturally equivalent, and both naturally equivalent to a lattice loop with a single vertex, i.e. $B\mathbb{Z}$—and this $\mathbb{Z}$ in the 1-morphism captures the $\pi_1$ of a loop. Readily from here, we can feel that natural equivalence is related to the invariant information under renormalization (coarse graining of lattice), and the notions of "same" versus "naturally equivalent" are, roughly speaking, respectively suitable for discussing UV versus IR. We will see more and more of such intuition in the proceeding.

With this example, we can introduce the concept of *skeletal* category, which means in such a category, if two objects are related by an invertible morphism, then these two objects

---

[81]An important caveat is that one needs a "choice function" to define $\bar{F}_0$ for the essentially surjective $F_0$. The choice function will lead to discontinuity issue when we work with topological spaces, making the use of the more general notion of *anafunctor* necessary in many situations, in generalization of ordinary functor. This is the subject of the next subsection.

must be the same object. Starting with a generic category, we can arrive at a skeletal category naturally equivalent to the original category, by identifying objects that are related by invertible morphisms. We will often use a skeletal category to represent its natural equivalence class, calling it the *skeleton* of the class. In the example above, $B\mathbb{Z}$ is the skeleton.

Now let us try to generalize the notions of functor and natural transformation for 1-categories to 2-categories. It is not hard to see that besides functors, natural transformation between functors, now we can define a new kind of relation called *modification* between natural transformations, which maps $C_0$ to $D_2$ (and $C_1, C_2$ to $D_3, D_4$ which contain only identity 3- and 4-morphisms). We will not delve into modification. But now, even for functors and natural transformations, there arise the possibility of having definitions at different levels of strictness. In the below we will discuss these different levels of strictness and see how they arise in the familiar lattice theories.

A strict 2-functor $F$ is such that it has functions $F_k$ ($k = 0, 1, 2$) that map $C_k$ to $D_k$ and strictly preserve all the source, target, composition and identity maps. A strict 2-natural transformation is basically the same as a natural transformation for 1-category, i.e. $\Phi_2$ still maps $C_1$ to the subset of identity morphisms $\{\mathbf{1}_h | h \in D_1\} \subseteq D_2$.

But even between two strict 2-functors, strict 2-natural transformations are not the only option. We can consider the more general notion of *lax 2-natural transformation*, where $\Phi_2$ can map $C_1$ to $D_2$ in the generic way, i.e. $\Phi_2(f) \in D_2$ (the rectangular surface on the left of (91)) does not have to be any $\mathbf{1}_h \in D_2|_{h,h} \subset D_2$, and there are consistency constraints, whose details we will omit, provided by $\Phi_3$ that maps $C_2$ to $D_3$ which contains only identity 3-morphisms. (It is sometimes desired to require $\Phi_2(f)$ to be an invertible 2-morphism which is not necessarily an identity 2-morphism, and such a "slightly stricter" version of lax 2-natural transformation is called a *pseudo 2-natural transformation*.)

And more general than strict 2-functors, there are *lax 2-functors*, where the composition of 1-morphisms and the assignment of identity 1-morphisms do not have to be preserved strictly, but only up to some 2-morphisms, i.e. we can specify $F_1(g \circ f) \stackrel{\varphi_{g,f}}{\Longleftarrow} F_1(g) \circ F_1(f)$, $\mathbf{1}_{F_0(a)} \stackrel{\psi_a}{\Longleftarrow} F_1(\mathbf{1}_a)$ in generalization of having equalities in the middle, and these 2-morphisms must be chosen to satisfy certain consistency constraints—whose details we will omit, but they finally come from the fact that $D_3$ contains identity 3-morphisms only. (Again, it is sometimes desired to require $\varphi_{g,f}$ and $\psi_a$ to be invertible 2-morphisms, and such a 2-functor is called *pseudo 2-functor*.) The definition of lax 2-natural transformation between two lax 2-functors also requires some changes compared to when the 2-functors are strict, though the spirit is the same.

And thus we can envision that, more generally, for strict $n$-categories, there are $(n, q)$-*transfors* between two $(n, q-1)$-transfors, which map $C_k$ to $D_{k+q}$ (so $q = 0$ are $n$-functors and $q = 1$ are $n$-natural transformations), such that the maps for $k + q \leq n$ contains information that defines the transformation, and the map for $k+q = n+1$ provides consistency constraints. The transfors can be defined at different levels of strictness. The collection of strict $n$-categories along with their strict $(n, q)$-transfors ($0 \leq q \leq n$) form a strict $(n + 1)$-category, but often it is desired to include laxer transfors, which will in general result in a less strict $(n + 1)$-category. Plunging more deeply into this is beyond our scope.

Now it can be readily seen how functors and natural transformations are useful in lattice QFT:

1. In traditional lattice nl$\sigma$m, a field configuration is a function from $\bar{\mathcal{L}}_0$ to $\mathcal{T}$.

2. In traditional lattice gauge theory, a field configuration is a functor from $\bar{\mathcal{L}}_{\leq 1}$ to $BG$. A gauge transformation is a natural transformation, which is automatically invertible

because all morphisms are invertible in $BG$. Hence field configurations that are related by gauge transformations are indeed equivalent as functors.

3. In Villainized $S^1$ nl$\sigma$m, a field configuration is a functor from $\bar{\mathcal{L}}_{\leq 1}$ to the action groupoid $S^1 \times \mathbb{R} \rightrightarrows S^1$.[82] More generally, a field configuration in a Villainized nl$\sigma$m is a functor from $\bar{\mathcal{L}}_{\leq 1}$ to $\widetilde{\mathcal{T}}^2/\Gamma \rightrightarrows \mathcal{T}$, where $\widetilde{\mathcal{T}}$, the universal cover of $\mathcal{T}$, is a $\Gamma$ bundle over $\mathcal{T}$ for some discrete group $\Gamma$, and the mod $\Gamma$ is by a $\Gamma$ action on both $\widetilde{\mathcal{T}}$'s.

With this perspective we can systematically understand what it means for the d.o.f. in a lattice path integral to be local, especially in situations like Villainization (recall the discussion we had at the end of Section 2.1). Locality just means each field configuration sampled in the path integral is a functor from the lattice to some target category (possibly a higher category, which we will discuss later)—in generalization of the usual notion of target space—so that each vertex is mapped to some field valued in $C_0$, each link is mapped to some field valued in $C_1$, and so on. But the path integral is in general not locally factorizable, in the sense that $C_1$ in general cannot be factorized into the form $C_0 \times C_0 \times X$—either not in this form as a set, or not in this form as a manifold though as a set—and likewise for higher morphisms. However, $C_1$ does have the source and target maps to $C_0$, which can be viewed as local constraints (e.g. the $e^{i\gamma_l} = e^{id\theta_l}$ constraint in the Villain model). In practice, when sampling the fields, we parametrize $C_1$ by $C_0' \times C_0' \times X$ using some large enough $C_0'$ and $X$ (e.g. we write $\gamma_l = d\theta_l + 2\pi m_l \in \mathbb{R}$ in the Villain model, with $\theta_v, \theta_{v'} \in (-\pi, \pi]$ and $m_l \in \mathbb{Z}$).

While these descriptions above seem nice and systematic, they are not entirely satisfactory. Let us take the traditional nl$\sigma$m case as example. Looking only at $\bar{\mathcal{L}}_0$ means we ignore which vertices are connected by links and which ones are not, but the path integral weight should be associated with links and cares about the difference of the fields between the two ends of each link. So it is desirable to be able to talk about the lattice $\bar{\mathcal{L}}$ entirely, instead of truncating it to, say, $\bar{\mathcal{L}}_0$. Can we say a field configuration in traditional lattice nl$\sigma$m is a functor from $\bar{\mathcal{L}}$ to $\mathcal{T}$, instead of from $\bar{\mathcal{L}}_0$ to $\mathcal{T}$? It turns out the statement becomes incorrect, because $\bar{\mathcal{L}}_1$ contains all lattice paths, meanwhile, since $\mathcal{T}$ is a 0-category, $\mathcal{T}_1 = \{\mathbf{1}_a | a \in \mathcal{T}\}$ only contains identity 1-morphisms, thus a functor from $\bar{\mathcal{L}}$ to $\mathcal{T}$ must have a constant field over each connected component of the lattice, which is certainly not what we want in general.

The correct statement is:

1. In traditional lattice nl$\sigma$m, a field configuration is a functor from $\bar{\mathcal{L}}$ to the *pair groupoid* $E\mathcal{T} := (\mathcal{T}^2 \rightrightarrows \mathcal{T})$ in which from any object to any other object there is exactly one morphism, $E\mathcal{T}_1|_{b,a} = \{(b,a)\}$, $E\mathcal{T}_1 = \{(b,a) \in \mathcal{T}^2\}$. Physically, this just means an almost trivial fact in traditional lattice nl$\sigma$m: the d.o.f. associated with a link $l = \langle v'v \rangle$ is just specified by the d.o.f. on the two vertices $v$ and $v'$ together, no more and no less.

It is easy to see any pair groupoid $E\mathcal{T}$ is naturally equivalent to the trivial category $*$ for arbitrary $\mathcal{T}$, because any functor between $E\mathcal{T}$ and $*$ is automatically essentially surjective and fully faithful. (If $\mathcal{T}$ is furthermore a group $G$, then similar to the relation between the category $BG$ and the classifying space $|BG|$, the category $EG$ is also related to the universal bundle $|EG|$ via the procedure of geometric realization that we will introduce in Section 5.4. Just like the space $|EG|$ is a $G$ bundle over $|BG|$, in a suitable sense the category $EG$ is also a $G$ bundle over $BG$.[83] And the fact that $EG$ is naturally equivalent to the trivial category is related to the fact

---

[82]What we called a $\mathbb{Z}$ gauge transformation in Section 2.1 when viewing Villainization as gauging a $\mathbb{Z}$ symmetry from an $\mathbb{R}$-valued theory is *not* a natural transformation. In fact it does not act on this description, because $e^{i\theta} \in S^1$ and $\gamma \in \mathbb{R}$ are already physically meaningful variables. The categorical nature of the $\mathbb{Z}$ gauge transformation will be explained below (103) after we introduce anafunctor.

[83]This means we have a functor from the 0-category $G$ to the 1-category $EG$ and then a functor from $EG$ to $BG$, such that any 1-morphism $\mathbf{1}_{\tilde{g}} \in G_1$ is mapped to $(\tilde{g}, \tilde{g}) \in EG_1 = G^2$, which is in turn mapped to

that $|EG|$ is contractible. More generally, the space $|E\mathcal{T}|$ can be constructed in the same way and is also contractible, although there is no $B\mathcal{T}$ when $\mathcal{T}$ does not have a group structure.) Does this natural equivalence between $E\mathcal{T}$ and $*$ mean the traditional lattice nl$\sigma$m is a trivial theory? Certainly not—the theory is non-trivial because of the presence of the path integral weight. This is the crucial distinction between a generic lattice QFT and a topological lattice QFT that worths some detailed explanation.

A topological lattice theory (e.g. [19, 20, 23, 34, 35, 45], which should be viewed as an effective theory already coarse grained to the deep IR limit) has a key feature that the path integral weight can only take value 0 or a complex phase of magnitude 1, and is invariant under natural transformations of field configurations. Thus, for a topological theory, the target category being natural equivalent to the trivial category means the theory is necessarily trivial. By contrast, it is familiar that a dynamical traditional lattice nl$\sigma$m can explore at least two phases by tuning the path integral weight—the trivial (disordered) phase and the spontaneous symmetry breaking (ordered) phase. Consider two topological limits, and the more generic physical situations in-between:

- If the link weight is 1 for any field configuration, then the path integral is sampling the functors from $\bar{\mathcal{L}}$ to $E\mathcal{T}$ freely, such that the weight, being constantly 1, is invariant under any natural transformation. This is the extreme case of the disordered (trivial) phase, as if we have replaced $E\mathcal{T}$ by its ananaturally equivalent skeleton, the trivial category $*$.

- If the link weight is a delta function, then only identity 1-morphisms are kept, in which case the target category becomes $\mathcal{T}$, a non-trivial subcategory of $E\mathcal{T}$. A functor from $\bar{\mathcal{L}}$ to $\mathcal{T}$ is indeed a completely ordered configuration, where each connected component of the lattice has a constant field. This is the extreme case of the ordered phase.

- For a traditional dynamical lattice nl$\sigma$m, a generic link weight lies in-between these two topological limits—the weight does not respect invariance under natural transformation of the field configuration, but it has not gone as far as to reduce the target category from $E\mathcal{T}$ to $\mathcal{T}$ either. The phase transition between the ordered and disordered is not determined at the lattice level but only by the dynamics towards the IR.

Due to this physically very intuitive reason, for a dynamical (as opposed to topological) lattice theory, we *must not* conclude the phase of the theory based on the natural equivalence class of the target category. We will have more thorough discussions of this in Sections 5.2 and 5.5.

After these explanations, we are ready to see how the known examples of lattice theories in Section 2 are described by strict higher categories along with functors and natural transformations (since the strict higher categories involved are strict higher groupoids where all morphisms are strictly invertible, hence all "lax" below are automatically "pseudo"):

1. In traditional lattice nl$\sigma$m, a field configuration is a functor from $\bar{\mathcal{L}}$ to the pair groupoid $E\mathcal{T} := (\mathcal{T}^2 \rightrightarrows \mathcal{T})$, which means the field on a link $l = \langle v'v \rangle$ is just specified by the fields on $v$ and $v'$ together.

   A generic natural transformation is going to change the relative values of the fields across a link (since $\Phi_1(v \in \bar{\mathcal{L}}_0)$ can be any element in $E\mathcal{T}_1$), therefore physically we do not demand the path integral weight to be invariant under natural transformations, otherwise the weight would be a constant.

---

the identity $\tilde{g}\tilde{g}^{-1} = \mathbf{1} \in BG$. Alternatively, the pullback category (which we did not systematically define) $EG \times_{BG} EG := \left( \{((g_1, g_2), (g'_1, g'_2)) | g_2 g_1^{-1} = g'_2 g'^{-1}_1 \} \rightrightarrows \{(g, g')\} \right)$ has a functor to the 0-category $G$, given by $g'^{-1}g = \tilde{g} \in G_0 = G$, and consistently, $g'^{-1}_2 g_2 = g'^{-1}_1 g_1 = \tilde{g}$ for $\mathbf{1}_{\tilde{g}} \in G_1 \cong G$, specifying the $G$ action on $EG$, just like $|EG| \times_{|BG|} |EG|$ has a function to $G$, specifying the $G$ action on $|EG|$. The mathematical idea and physical interpretation behind this will become clearer as we proceed (in particular as discussed below (123)).

2. In traditional lattice gauge theory, a field configuration is a functor from $\bar{\mathcal{L}}$ to $BEG := (G^2 \rightrightarrows G \rightrightarrows *)$ (the deloopling of $EG$), which means the field on a plaquette bounded by two Wilson lines is just specified by the two Wilson lines together, or equivalently, it can be specified by one Wilson line along with the holonomy.

   If $G$ is a discrete group (as is often the case in the effective theory of topological phase, which we will discuss more in Section 5.3), then it is physically possible to forbid the gauge flux, in which case only identity 2-morphisms are left, so that the target category becomes just $BG$. But for Lie group it is not physical to demand so.[84]

   A gauge transformation is a strict 2-natural transformation. The holonomy around a plaquette or a non-contractible loop remains invariant (up to conjugation by Wilson lines) because the image of $\Phi_2$ in a strict 2-natural transformation only contains identity 2-morphisms. On the other hand, a generic lax 2-natural transformation changes the holonomy. Therefore, physically we demand the path integral weight to be invariant under strict 2-natural transformations, but not under generic lax 2-natural transformations, otherwise the weight would be a constant.

3. In Villainized $S^1$ nl$\sigma$m, a field configuration is a functor from $\bar{\mathcal{L}}$ to $S^1 \times \mathbb{R} \times \mathbb{Z} \rightrightarrows S^1 \times \mathbb{R} \rightrightarrows S^1$, where the $\mathbb{R}$ in the space 1-morphisms is the $\gamma$ we saw in Section 2.1, and the $\mathbb{Z}$ in the space of 2-morphisms represents the vorticity; in particular, it comes from $(S^1 \times \mathbb{R})^{[2]} := (S^1 \times \mathbb{R}) \times_{(S^1)^2}^{(s,t),(s,t)} (S^1 \times \mathbb{R}) \cong S^1 \times \mathbb{R} \times \mathbb{Z}$. If vortices are forbidden, i.e. only identity 2-morphisms are allowed, then the target category can be reduced to the action groupoid $S^1 \times \mathbb{R} \rightrightarrows S^1$. More generally, a field configuration in a Villainized nl$\sigma$m is a functor from $\bar{\mathcal{L}}$ to $\widetilde{\mathcal{T}}^2/\Gamma \times \Gamma \rightrightarrows \widetilde{\mathcal{T}}^2/\Gamma \rightrightarrows \mathcal{T}$ (note that $\mathbb{R}^2/\mathbb{Z} \cong S^1 \times \mathbb{R}$); if the $\Gamma$ vortices are forbidden, only the identity 2-morphisms are left.

   Again, in a nl$\sigma$m, physically we do not demand the path integral weight to be invariant under 2-natural transformation, otherwise the weight would be a constant.

4. In Villainized $U(1)$ gauge theory, a field configuration is a functor from $\bar{\mathcal{L}}$ to the deloopping of the target category above, $U(1) \times \mathbb{R} \times \mathbb{Z} \rightrightarrows U(1) \times \mathbb{R} \rightrightarrows U(1) \rightrightarrows *$, where the $\mathbb{Z}$ in the 3-morphism represents the monopole. If monoples are forbidden, i.e. only identity 3-morphisms are allowed, then the target category can be reduced to the 2-group $U(1) \times \mathbb{R} \rightrightarrows U(1) \rightrightarrows *$. More generally, a field configuration in a general Villainized gauge theory appears similar, as long as we replace $U(1) \times \mathbb{R}$ by $G \ltimes H$, and $\mathbb{Z}$ by $H/G = \ker(\widetilde{t})$ which must be abelian as explained before.

   Now it becomes particularly interesting to ask if the path integral weight should be invariant under natural transformations at some certain level of strictness.

   As a Villainized gauge theory, the path integral weight should be invariant under strict 3-natural transformations only, i.e. those where $\Phi_{k+1}$ maps $C_k$ to identity $(k+1)$-

---

[84]In fact, when $G$ is a Lie group, if we impose this unphysical condition of forbidding fluxes, we will run into non-extensive/non-local divergence in the partition function. Suppose there are $N_2$ plaquettes on the lattice. If we forbid fluxes, by locality each plaquette should be treated alike, so we have to impose a delta function on each plaquette, i.e. we have a $\prod_p \delta(Dg_p = \mathbf{1})$ in the path integral. When we integrate over the gauge fields $g_l$ on the links, most of the delta functions will be absorbed by constraining some $g_l$ in the integral; however, on each closed 2d surface, one delta function is redundant, since if $Dg_{p'} = \mathbf{1}$ on all but one plaquette on this surface, $Dg_p = \mathbf{1}$ is automatically true on that remaining plaquette. Thus we are left with $\delta(0)^{N_3+B_2}$ where $N_3$ is the number of cubes/tetrahedra, i.e. the number of contractable closed surfaces, and $B_2$ is the second Betti number, the number of non-contractable closed surfaces. The appearance of the topological number $B_2$ means there is a non-local factor of Dirac delta divergence that cannot be formally dropped. By contrast, when $G$ is a finite group, there is no divergence problem with the Kronecker delta, therefore in the case of finite gauge group we can—and we often do to make connection to continuum gauge theory [19]—forbid fluxes.

morphisms in $D_{k+1}$ for $k > 0$, and to generic 1-morphisms for $k = 0$. These are the usual gauge transformations on the lattice.

What if we impose a stronger requirement that the path integral weight be invariant under 3-natural transformations that are less strict? In particular, let us consider invariance under those laxer 3-natural transformations where $\Phi_{k+1}$ maps $C_k$ to identity $(k + 1)$-morphisms in $D_{k+1}$ for $k > 1$ and generic $(k + 1)$-morphisms for $k = 0, 1$. This is what is called *2-group gauge theory* that has been studied relatively early on as an application of category theory in physics [34–36, 48–50]. In this case the flux is no longer gauge invariant up to conjugation, but the $\ker(\widetilde{t})$-valued monopole in the 3-morphism is still physically well-defined. We will discuss more about this in Section 5.3.

From here, we can see that in general, even for the same target category, we can still demand invariance of the path integral weight under natural transformations of different levels of strictness. As usual, by tuning the path integral weight, we may access different phases of a theory; as we demand the path integral weight to remain invariant under laxer and laxer natural transformations, the accessible phases of a theory become more and more limited. This is why the Villainized $U(1)$ gauge theory can access both the confined and the Coulomb phases (for $d \geq 4$ [67]), while the 2-group gauge theory with the same target category only represents the confined phase [34–36].

5. Obviously, when both $G$ and $H$ in the target category above are abelian, we can deloop the category arbitrarily many times, and obtain Villainized higher form gauge theories.

6. In spinon decomposed $S^2$ nl$\sigma$m, a field configuration is a functor from $\bar{\mathcal{L}}$ to $S^2 \times SU(2) \times \mathbb{R} \times \mathbb{Z} \rightrightarrows S^2 \times SU(2) \times \mathbb{R} \rightrightarrows S^2 \times SU(2) \rightrightarrows S^2$, where the $\mathbb{Z}$ in the 3-morphism represents the hedgehog (see (89)). If hedgehogs are forbidden, i.e. only identity 3-morphisms are allowed, then the target category reduces to $S^2 \times SU(2) \times \mathbb{R} \rightrightarrows S^2 \times SU(2) \rightrightarrows S^2$.

Again, in a nl$\sigma$m, physically we do not demand the path integral weight to be invariant under 3-natural transformation.

7. Consider two smooth functions $f, g$ from manifold $\mathcal{M}$ to manifold $\mathcal{N}$. Function $f$ determines a $d$-functor $F$ from the strict path $d$-groupoid $\bar{P}_d\mathcal{M}$ to the strict path $d$-groupoid $\bar{P}_d\mathcal{N}$ (where $d$ is the max of the dimensions of $\mathcal{M}, \mathcal{N}$), because knowing how every point on $\mathcal{M}$ maps to $\mathcal{N}$ determine how every path, surface and so on on $\mathcal{M}$ maps to that on $\mathcal{N}$. Likewise for $g$. A homotopy from $f$ to $g$ determines a lax $d$-natural transformation from $F$ to $G$. Homotopy equivalence between $\mathcal{M}$ and $\mathcal{N}$ implies natural equivalence between $\bar{P}_d\mathcal{M}$ and $\bar{P}_d\mathcal{N}$.[85]

8. $\bar{\mathcal{L}}$ and $\bar{\mathcal{L}}'$ for two lattices that discretize the same space or two homotopically equivalent spaces are naturally equivalent, where the natural equivalence is again established by lax $d$-natural transformations.

From these discussions we can experience that category theory is a natural language for organizing our thoughts about lattice QFT and potentially their relation to continuum QFT. To describe the known lattice QFTs in Section 2, we only used strict higher categories; so we may indeed anticipate that the generalization problem discussed in Section 3 might find its solution when the more flexible higher categories are taken into consideration.

---

[85]But the converse is not necessarily true, because any strict groupoids cannot capture the full homotopy information of the manifold. See footnote 97. To establish the converse, weak higher groupoids are needed, see Section 5.4 for one construction.

To go towards this direction, next we shall motivate the introduction of *anafunctors* as a necessary (and actually familiar and intuitive, as we shall see) generalization of the ordinary functors, whenever we are concerned with the continuity/smoothness of spaces and functions—which is indeed the very problem our work aims at.

## 5.2   Internalization and anafunctor

Let us begin with a motivating problem. In the above we have seen that the homotopy between two manifolds implies natural equivalence of their strict path $d$-groupoids; the same holds when both manifolds are discretized into lattices. But an obvious question to ask is: Consider the strict $d$-groupoid of a lattice and the strict path $d$-groupoid of the manifold that the lattice is discretizing, are they also naturally equivalent in some suitable sense?

The subtlety here lies in that the manifold is not only a set of points, but has the extra structure of being smooth. So, as mentioned before, it is intuitive to require whatever maps that are involved to be smooth maps. This intuition can be more systematically phrased in terms of *internalization*. Consider all the sets and functions involved in defining a category $C = (C_1 \rightrightarrows C_0)$,

$$
C_1 \times_{C_0}^{\mathsf{s,t}} C_1 \xrightarrow{\quad\circ\quad} C_1 \underset{\mathsf{t}}{\overset{\mathsf{s}}{\underset{\longleftarrow}{\rightrightarrows}}} \; C_0 \;\; , \tag{92}
$$

where the functions satisfy some consistency constraints (such as $\mathsf{si} = \mathsf{ti} = \mathbf{1}_{C_0}$, associativity, etc.). While this diagram represents a category $C$, if we stare at this diagram, we realize it also represents a few objects and a few morphisms within some category—and this ambient category is Set, because $C_0$, $C_1$ and $C_1 \times_{C_0}^{\mathsf{s,t}} C_1$ are indeed sets, and $\mathsf{s,t,i}$ and $\circ$ are indeed functions. So this diagram, along with the diagrams that describe the consistency constraints (which are straightforward to draw, and we omitted here), formed by certain picked objects and morphisms from the 1-category Set of sets, *define* the 1-category $C$. We say $C$ is "a category *internal to* the ambient category Set"—which is what we often mean by default when we say "a category".[86]

With this perspective in mind, it is easy to generalize to 1-categories internal to other ambient 1-categories. For example, let the ambient 1-category be Manifold instead, where the objects are finite dimensional smooth manifolds, and the morphisms are smooth maps between them. Then we can pick some objects and morphisms from Manifold to form the same diagram as above, satisfying the same consistency constraints (which can also be presented as diagrams), and this will define a 1-category internal to Manifold, which means $C_0$, $C_1$ and $C_1 \times_{C_0}^{\mathsf{s,t}} C_1$ are smooth manifolds, and all maps involved are smooth maps.[87] This is the systematic description of the intuition before. Likewise we can define the internalization of higher categories or other structures in Manifold. A familiar example is a group internal to Manifold, which is, apparently, a Lie group; similarly, a groupoid internal to Manifold is a Lie groupoid.[88]

---

[86]More precisely, $C$ defined by such a diagram in Set must be a small category. So the answer to the self-referencing question of whether Set can be defined by such a diagram within Set is "No".

[87]$\mathsf{s,t}$ must be surjective submersions to ensure that, by transversality, $C_1 \times_{C_0}^{\mathsf{s,t}} C_1$ and $C_1 \times_{C_0}^{\mathsf{s,t}} C_1 \times_{C_0}^{\mathsf{s,t}} C_1$ also exist as smooth manifolds (the latter space is for describing the consistency condition of associativity).

[88]In the above we drew the diagram that defines a category. To draw the diagram for a groupoid, we have an additional arrow from $C_1$ to $C_1$ (satisfying suitable constraints) that assigns inverses. Further, to define a group, we require $C_0$ to be the manifold with a single point—if we break away from the set theoretic language, such a "single point manifold" can be described as being a *terminal object* in Manifold, i.e. it is an object such that all objects in Manifold has a unique morphism to it (so it is easy to see that a terminal object is unique up to unique invertible morphisms).

We can further consider more general ambient categories, as long as products of the form $X \times_Z^{u,v} Y$ are defined in the ambient category.[89,90] In particular, when discussing the relation between lattice and continuum, we will often need the spaces of paths, surfaces and so on in a manifold (such as in defining the strict path $d$-groupoid), and these spaces are infinite dimensional. Therefore we will need a notion of "smoothness" for infinite dimensional spaces. A notion suitable for our usage would be "diffeological", whose detailed definition we will not get into (see e.g. [96]) since we are not aiming at a comprehensive and rigorous mathematical exposition in this work. The category Difflg with diffeological spaces as objects and diffeological maps as morphisms will often be used as the ambient category, generalizing Manifold by including the infinite dimensional cases. In the below, we may colloquially use the familiar word "smooth" to mean diffeological when the space involved is infinite dimensional.

The problem we face is, the definitions of functor and natural transformation introduced in Section 5.1 are designed for categories internal to Set, but when applied to categories internal to Manifold or Difflg—as we do—i.e. when requiring the maps involved in the definitions of functor and natural transformation to be smooth, the definitions would become too restrictive to capture many interesting situations. So we must generalize the definitions.

Let us consider the simplest case in our motivating problem: Is $\bar{\mathcal{L}}$ for a 1d lattice loop (in the extreme case where the lattice loop has only one vertex and one link, we get $\bar{\mathcal{L}} = B\mathbb{Z}$, the skeleton) naturally equivalent to the path groupoid $\bar{P}_1 S^1 = (S^1 \times \mathbb{R} \rightrightarrows S^1)$ of the circle that it is discretizing? Indeed, we can have an essentially surjective and fully faithful functor from $\bar{\mathcal{L}}$ to $\bar{P}_1 S^1$, for instance

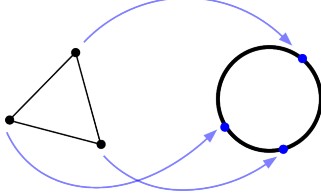

where we indicated how the lattice vertices map to points on the circle, and then the links are mapped to paths on the circle in the obvious way. Conversely, there must be an essentially surjective and fully faithful functor from $\bar{P}_1 S^1$ to $\bar{\mathcal{L}}$ that is an inverse of the functor above. One such inverse functor is

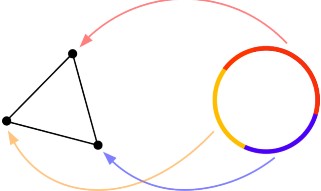

where we indicated how the points on the circle map to the lattice vertices, and then the paths on the circle are mapped to paths on the lattice depending on the starting and ending points and the winding, in the intuitive way. This is all good if the categories are internal to Set, but now that we want them to be internal to Manifold,[91] there is a problem: The inverse functor obviously involves discontinuous functions.

---

[89]This is to require the ambient category to admit finite limits. Limit is a crucial general concept in category theory which we, nevertheless, did not introduce. One may consult relevant texts on this.

[90]The ambient category can also be a higher category. In that case, some equality signs in the consistency constraints can be replaced by invertible 2- or higher morphisms in the ambient category (where "invertible" itself may also be defined in a weak sense). Some of our discussions below can be phrased in this language, for example multiplicative bundle gerbe crucial to our main construction can be described as 2-group internalized in the bicategory of Lie groupoids [37]. But we will not go deeply into the details.

[91]The lattice $\bar{\mathcal{L}}_0$ and $\bar{\mathcal{L}}_1$ are discrete, but discrete topology is a special case of topology.

A familiar treatment allows us to avoid such discontinuity, and will lead us towards the definition of anafunctor soon. Instead of thinking about the circle itself, we cover the circle with some patches (open charts) $U_\alpha$, and map each patch to a lattice vertex:

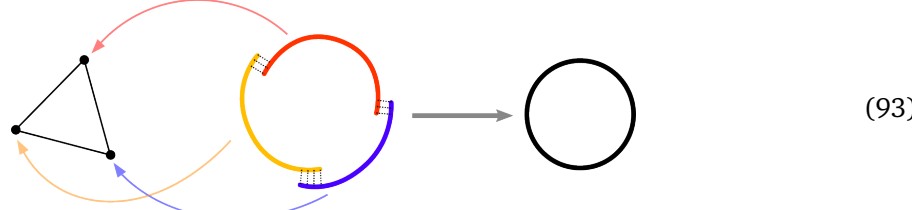

$$(93)$$

More particularly, given the patches we can form a category $F$, where $F_0 = \sqcup_\alpha U_\alpha$, the disjoint union of the patches, and $F_1$ contains two kinds of basic morphisms: one is the paths in $\bar{\mathcal{P}}U_\alpha$ within each patch, and the other is the identification morphisms specifying which points on different patches will be identified when mapped to $\mathcal{M}$ (denoting the map as $\sqcup_\alpha U_\alpha \xrightarrow{\Pi} \mathcal{M}$), i.e. there is a morphism from $x \in U_\alpha$ to $y \in U_\beta$ whenever $\Pi(x) = \Pi(y) \in S^1$; moreover, these two kinds of basic morphisms can be composed, up to the intuitive identification

$$(94)$$

Such a category $F$ has a surjective (rather than just essentially surjective) and fully faithful functor to each of $\bar{P}_1\mathcal{M}$ and $\bar{\mathcal{L}}$, and moreover all maps involved are smooth. In the below, we will extract the essence behind this familiar treatment, to define the notions of *anafunctor*, *ananatural transformation*, and *ananatural equivalence*. The example here will turn out to be an ananatural equivalence between the lattice $\bar{\mathcal{L}}$ and the continuum $\bar{P}_1\mathcal{M}$, established by an invertible anafunctor $F$.

Before giving the precise definitions, it is helpful to look at another motivating example. Let us consider a manifold $\mathcal{M}$ with identity morphisms only, $\mathcal{M} \rightrightarrows \mathcal{M}$. What are the possible functors from $\mathcal{M} \rightrightarrows \mathcal{M}$ to $BG = (G \rightrightarrows *)$? Somehow we feel there should be different possibilities, to do with different principal $G$ bundles over $\mathcal{M}$. However, in fact there is only one possible functor—which maps each point on $\mathcal{M}$ to the single object $*$, and the identity morphism of each point on $\mathcal{M}$ to the identity element of $G$. This is not unexpected—the definition of functor is suitable for categories internal to Set, and if we view $\mathcal{M}$ as merely a set rather than a manifold, indeed there should be no distinction of different bundles—as a set without topology we only have $\mathcal{M} \times G$. In order to define different principal $G$ bundles over $\mathcal{M}$, one familiar treatment is, again, to cover $\mathcal{M}$ by some patches $\sqcup_\alpha U_\alpha \xrightarrow{\Pi} \mathcal{M}$, and then specify the transition functions. In the category theory language, given the patches, along with the aforementioned identification morphisms, we form a category $F = (\mathcal{U} \times_\mathcal{M}^{\Pi,\Pi} \mathcal{U} \rightrightarrows \mathcal{U})$ where $\mathcal{U} := \sqcup_\alpha U_\alpha$. Via the $\Pi$, this category $F$ has a smooth, surjective and fully faithful functor to $\mathcal{M} \rightrightarrows \mathcal{M}$. Moreover, this category $F$ can now have different smooth functors to $BG$, which specify the transition functions. Thus we obtain different principal $G$ bundles.

Here we used patches and transition functions to describe a principal bundle, but there is another familiar way to describe a principal bundle, namely the total space $\mathcal{E} \xrightarrow{\Pi} \mathcal{M}$ of the bundle. It turns out that this corresponds to another choice $F'$ that replaces the $F$ above, given by $F' = (\mathcal{E} \times_\mathcal{M}^{\Pi,\Pi} \mathcal{E} \rightrightarrows \mathcal{E})$, which again has a smooth, surjective and fully faithful map to $\mathcal{M} \rightrightarrows \mathcal{M}$ via $\Pi$. On the other hand, note that $\mathcal{E} \times_\mathcal{M} \mathcal{E} \cong \mathcal{E} \times G$ through the $G$ action on the fibres of $\mathcal{E}$, thus $F'$ has a smooth functor to $BG$ by keeping the $G$ and dropping the $\mathcal{E}$.

Now we have two ways to describe a principal bundle as a functor from some intermediate category $F$ covering $\mathcal{M}$ to $BG$, one where $F_0 = \mathcal{U} = \sqcup_\alpha U_\alpha$ consists of patches, and the other where $F_0' = \mathcal{E}$ is the total space of the bundle. But given a principal bundle, these two ways of description must be equivalent in a suitable sense. Usually how one sees the equivalence is by covering $\mathcal{E}$ by patches as well—pullback from the cover $\mathcal{U}$ of $\mathcal{M}$—and checking the consistency between the transition functions and the $G$ action on the fibres, up to gauge transformations. This will motivate us to define the notion of ananatural transformation in the below.

Gathering the experience from these familiar treatments, it is now clear that we should define an *anafunctor* $D \overset{F}{\leftarrow} C$, in generalization to an ordinary functor, as

$$
\begin{array}{ccccc}
D_1 & \longleftarrow & F_1 & \overset{\text{f.f.}}{\longrightarrow} & C_1 \\
\Downarrow & & \Downarrow & & \Downarrow \\
D_0 & \longleftarrow & F_0 & \overset{\text{surj}}{\longrightarrow} & C_0
\end{array} , \tag{95}
$$

where there is an intermediate category $F$, called the "span", such that it has a surjective (rather than just essentially surjective) and fully faithful ordinary functor to $C$ (so that $F$ is in some sense "equivalent to $C$ but larger in appearance"), and another ordinary functor to $D$.[92] The notation $F_0, F_1$ here seems to be in conflict with the notation we used for ordinary functor before, but in fact this is a generalization rather than a conflict—the function $F_0$ in the ordinary functor can be viewed as a set of ordered pairs $\{(c, F_0(c)) | c \in C_0\}$, which is a set that has a bijective map to $C_0$, and now we are generalizing this to a set with a surjective map to $C_0$. Pictorially,

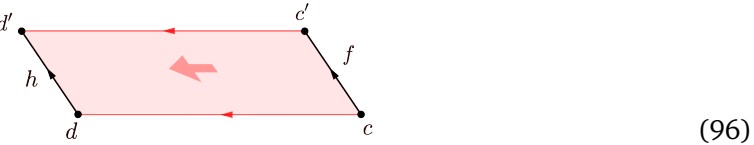

$$\tag{96}$$

in an ordinary functor, there is a unique red arrow emanating from any given $c$, but in an anafunctor, there can be one or more red arrows emanating from a given $c$, and moreover they may end at different $d$'s; the collection of all such red arrows emanating from all $c$ is $F_0$. On the other hand, the collection of all pink surfaces in the middle forms $F_1$; the requirement of "fully faithful" is that, given the two red arrows on the sides and the black arrow $f$ on the right, there is a unique choice of pink surface in the middle, and hence the black arrow $h$ on the left is also uniquely determined.[93]

The composition of anafunctors $D \overset{F}{\leftarrow} C$, $E \overset{G}{\leftarrow} D$ can be defined by such a category $H$ (the arrows here represent ordinary functors)

$$
\begin{array}{ccccccc}
& & & H & & & \\
& & \swarrow & = & \searrow & & \\
E & \leftarrow & G & \rightarrow & D & \leftarrow & F & \rightarrow & C \,,
\end{array} \tag{97}
$$

---

[92]Before the general notion of anafunctor was formulated, it has already had other names in specific contexts. In particular, anafunctor between Lie groupoids has been known as *bibundle* or *Hilsum-Skandalis morphism*. In the general context of higher homotopy theory, which is what we are concerned about, the span $F$ is known as a *resolution*. Some may generalize the usage of the terms *Morita morphism* and *Morita equivalence* (which are originally from ring theory) to refer to anafunctor and ananatural equivalence. In this paper we will just use the general context terminology *anafunctor*.

[93]This description can be casted in the language of *double category*, where a category has the objects from both $C_0$ and $D_0$, and then there are two kinds of morphisms: those black arrows from $C_1$ and from $D_1$, and those red arrows from $F_0$. Although we will not directly use this language in the below, this perspective is helpful for understanding and unifying many concepts.

where $H_0 = F_0 \times_{D_0} G_0$, and $H_1$ is determined by the requirement that the ordinary functor from $H$ to $C$ via $F$ is surjective and fully faithful; the ordinary functor from $H$ to $D$ via $G$ is required to be equal to that via $F$. Then $E \overset{H}{\leftarrow} C$ is the resulting anafunctor. Note that the composition of anafunctors is not strictly associative, but the two results are equivalent up to invertible ananatural transformations, which we now introduce.[94]

Between two anafunctors $D \overset{F}{\leftarrow} C$, $D \overset{F'}{\leftarrow} C$ we can define an *ananatural transformation*. Consider the category $H$ (the arrows here represent ordinary functors)

$$
\begin{array}{ccccc}
 & & F & & \\
 & \swarrow & \uparrow & \searrow & \\
D & \overset{\Phi}{\Downarrow} & H & \| & C\,, \\
 & \searrow & \downarrow & \nearrow & \\
 & & F' & &
\end{array}
\tag{98}
$$

where $H_0 = F_0 \times_{C_0} F'_0$, and $H_1$ is determined by the condition that the ordinary functor from $H$ to $C$ via $F$ is equal to that via $F'$, and is surjective and fully faithful. (Such an $H$ is, intuitively, called the strict pullback of $F \to C \leftarrow F'$, although we skipped the general definition of pullback in category theory.) On the left half of the diagram, the two ordinary functors from $H$ to $D$ via $F$ and via $F'$ are in general distinct. If there is an ordinary natural transformation $\Phi$ between these two ordinary functors, then $H$ together $\Phi$ define an ananatural transformation between the two anafunctors of interest. If $\Phi$ is an invertible ordinary natural transformation, then $H$ and $\Phi$ together is an invertible ananatural transformation (ananatural isomorphism), and in this case the two anafunctors $F$ and $F'$ are considered equivalent.

Two anafunctors $D \overset{F}{\leftarrow} C$, $C \overset{\bar{F}}{\leftarrow} D$ are considered inverse of each other, if their compositions $\bar{F} \circ F$ and $F \circ \bar{F}$ are related to $\mathbf{1}_C$ and $\mathbf{1}_D$ respectively via ananatural isomorphisms; we say they establish an *ananatural equivalence* between $C$ and $D$. It is not hard to see that if $C$ and $D$ are ananaturally equivalent, then there exists some span $F$ such that there are strictly surjective and fully faithful functors from $F$ to both $C$ and $D$.

Before we proceed, let us pause and wonder: What is the fundamental difference between Set and Manifold (or Difflg) that makes the notion of anafunctor, defined by the diagrams above, necessary when internalized in Manifold (or Difflg), but not in Set? Let us denote the anafunctor $D \overset{F}{\leftarrow} C$ by two ordinary functors $D \overset{\mathsf{F}^t}{\leftarrow} F \overset{\mathsf{F}^s}{\rightarrow} C$ where $\mathsf{F}^s$ is surjective and fully faithful. In Set, recall that this means $\mathsf{F}^s$ has an inverse $F \overset{\bar{\mathsf{F}}^s}{\leftarrow} C$. With this inverse, we can see the anafunctor $D \overset{\mathsf{F}^t}{\leftarrow} F \overset{\mathsf{F}^s}{\rightarrow} C$ of interest is equivalent (via invertible ananatural isomorphism) to the ordinary functor $D \overset{\mathsf{F}^t \circ \bar{\mathsf{F}}^s}{\longleftarrow} C$. But crucially, the existence of such $\bar{\mathsf{F}}^s$ requires the axiom of choice; while the axiom of choice can be imposed (as is usually done) in set theory, it is in general violated upon the introduction of topology—simply because in general a projection cannot be lifted back to a continuous section.[95] Therefore, anafunctor is the more useful

---

[94]In the usual construction of "direct product" in set theory, $(X \times Y) \times Z$ and $X \times (Y \times Z)$ are different sets, but there is a bijection between them. Similarly as we involve fibre products now. Thus, the collection of all 1-categories internal to some ambient category, along with their anafunctors and ananatural transformations, form a *bicategory* which we will introduce in Section 5.3.

[95]The axiom of choice is the statement that, if there is a collection of non-empty sets $S_a$ ($a \in A$), then there exists a "choice function" $f$ from $A$ to $\sqcup_a S_a$ such that $f(a)$ is an element of $S_a$ (so the image $\text{im}(f)$ contains exactly one element from each $S_a$). This statement is obviously true (as $f$ can be explicitly constructed) if $A$ is a finite set, but when $A$ is an infinite set, whether such $f$ exists depends on whether we impose the existence as an axiom—and either way is consistent in set theory.

Even if we did impose the axiom of choice in set theory, when we introduce extra structures such as topology to the sets involved, the axiom of choice may become incompatible with the extra structures. For a familiar example, suppose $S_a$ are fibres in a fibre bundle $E$ that project to points $a$ on a base manifold $A$. Then in general there does

notion in generic ambient categories, and only in those ambient categories where the axiom of choice is respected can it be reduced to ordinary functor.

With the definitions (95)(97)(98), we can come back to the examples introduced at the beginning of this subsection:

1. In our lattice loop versus circle example, the lattice loop $\bar{\mathcal{L}}$ (which reduces to $B\mathbb{Z}$ if there is only one vertex) and $\bar{\mathcal{P}}S^1 \rightrightarrows S^1$ are ananaturally equivalent, established by

$$
\begin{array}{ccccc}
\bar{\mathcal{L}}_1 & \xleftarrow{\text{f.f.}} & \widetilde{\mathcal{P}}^{\Pi}\mathcal{U} & \xrightarrow{\text{f.f.}} & \bar{\mathcal{P}}S^1 \\
\Downarrow & & \Downarrow & & \Downarrow \\
\bar{\mathcal{L}}_0 & \xleftarrow{\text{surj}} & \mathcal{U} & \xrightarrow{\text{surj}} & S^1
\end{array}
, \tag{99}
$$

where $\mathcal{U}$ is the disjoint union of patches in (93), and $\widetilde{\mathcal{P}}^{\Pi}\mathcal{U}$ is illustrated by (94)—in general, we define $\widetilde{\mathcal{P}}^{\Pi}X$ for $X \xrightarrow{\Pi} Y$ as the space of piecewise continuous (or Borel) paths on $X$ that can project via $\Pi$ to a continuous path on $Y$, with identification if two such paths sharing the same end points in $X$ project down to the same path in $\bar{\mathcal{P}}Y$ (in order to be fully faithful on the upper right).

$\bar{\mathcal{P}}S^1 \rightrightarrows S^1$ is an example of fundamental groupoid. More generally, the fundamental groupoid of a manifold $\mathcal{M}$ is ananaturally equivalent to a skeletal category $\widetilde{\pi}_1(\mathcal{M}) \rightrightarrows \pi_0(\mathcal{M})$, where the elements of $\pi_0(\mathcal{M})$ represent connected components of $\mathcal{M}$, and the elements of $\widetilde{\pi}_1(\mathcal{M})$ represents classes of non-contractible loops on $\mathcal{M}$, such that $\widetilde{\pi}_1(\mathcal{M})|_{a,a} \cong \pi_1(\mathcal{M}, x)$ is the usual fundamental group based at some point $x$ on a given connected component $a$. Hence the name "fundamental groupoid". For $\mathcal{M} = S^1$, the skeleton of its fundamental groupoid is $B\mathbb{Z}$, a lattice loop with only one vertex.

2. In the principal bundle example, we have two familiar choices of the span $F$ for the anafunctor $BG \xleftarrow{F} \mathcal{M}$: one using the patches $\mathcal{U}$, and the other using the total space $\mathcal{E}$:

$$
\begin{array}{ccccc}
G & \longleftarrow & \mathcal{U} \times_{\mathcal{M}} \mathcal{U} & \xrightarrow{\text{f.f.}} & \mathcal{M} \\
\Downarrow & & \Downarrow & & \Downarrow \\
* & \longleftarrow & \mathcal{U} & \xrightarrow{\text{surj}} & \mathcal{M}
\end{array}
, \quad \text{or replacing } \mathcal{U} \text{ with } \mathcal{E}. \tag{100}
$$

In the first choice, the upper left arrow $G \leftarrow \mathcal{U} \times_{\mathcal{M}} \mathcal{U}$ is transition functions (in a certain gauge). In the second choice, the upper left arrow $G \leftarrow \mathcal{E} \times_{\mathcal{M}} \mathcal{E} \cong \mathcal{E} \times G$ specifies the $G$ action on the fibres of $\mathcal{E}$.

In fact, we can further specify $G$-connections on the principal bundle by considering paths on $\mathcal{M}$ (since path spaces are in general infinite dimensional, we now need to internalize the discussion in Difflg):

$$
\begin{array}{ccccc}
G & \longleftarrow & \widetilde{\mathcal{P}}^{\Pi}\mathcal{U} & \xrightarrow{\text{f.f.}} & \bar{\mathcal{P}}\mathcal{M} \\
\Downarrow & & \Downarrow & & \Downarrow \\
* & \longleftarrow & \mathcal{U} & \xrightarrow{\text{surj}} & \mathcal{M}
\end{array}
, \quad \text{or replacing } \mathcal{U} \text{ with } \mathcal{E}, \tag{101}
$$

where the notation $\widetilde{\mathcal{P}}^{\Pi}$ is introduced below (99). In the first choice, $G \leftarrow \widetilde{\mathcal{P}}^{\Pi}\mathcal{U}$ is specified by the Wilson lines within patches $G \leftarrow \bar{\mathcal{P}}\mathcal{U}$ (in a certain gauge) along with the

---

not exist a continuous lifting function $f$ from $A$ to $E$.

Given an essentially surjective and fully faithful ordinary functor F in some ambient category, the axiom of choice in that ambient category is needed for (and is, in fact, equivalent to) the existence of an essentially surjective and fully faithful inverse ordinary functor $\bar{\text{F}}$, because $\text{F}_0$ is in general non-injective.

transition functions across patches $G \leftarrow \mathcal{U} \times_{\mathcal{M}} \mathcal{U}$ (in a certain gauge); in the second choice, $G \leftarrow \widetilde{\mathcal{P}}^{\Pi}\mathcal{E}$ is specified by the Wilson lines $G \leftarrow \bar{\mathcal{P}}\mathcal{E}$ given by the total connection over $\mathcal{E}$ (the total connection over $\mathcal{E}$ reduce to the connection over $\mathcal{M}$, and the remaining components along the fibres are gauge transformations, familiarly known as the BRST—or Faddeev-Popov—ghosts [104]), along with the $G$ action on the fibres $G \leftarrow \mathcal{E} \times_{\mathcal{M}} \mathcal{E} \cong \mathcal{E} \times G$.

We can describe "a same principal bundle" (whether without or with connection specified) using different choices of gauges on $\mathcal{U}$, different choices of the cover $\mathcal{U}$ itself, and using $\mathcal{E}$ instead of $\mathcal{U}$, and what we mean by "same" is that these anafunctors are related to one another by ananatural isomorphisms.

Thus we see the general notion of anafunctor is already implicitly used in familiar contexts.

We can envision how $n$-anafunctor is to be defined for strict $n$-categories internal to some ambient category.

$$
\begin{array}{ccccc}
D_n & \longleftarrow & F_n & \xrightarrow{\text{f.f.}} & C_n \\
\Downarrow & & \Downarrow & & \Downarrow \\
\cdots & \longleftarrow & \cdots & \xrightarrow{\text{full}} & \cdots \\
\Downarrow & & \Downarrow & & \Downarrow \\
D_1 & \longleftarrow & F_1 & \xrightarrow{\text{full}} & C_1 \\
\Downarrow & & \Downarrow & & \Downarrow \\
D_0 & \longleftarrow & F_0 & \xrightarrow{\text{surj}} & C_0
\end{array}
. \tag{102}
$$

Still $D \xleftarrow{F} C$ takes the form $D \xleftarrow{\mathsf{F}^{\mathsf{t}}} F \xrightarrow{\mathsf{F}^{\mathsf{s}}} C$, where $\mathsf{F}^{\mathsf{t}}$ and $\mathsf{F}^{\mathsf{s}}$ are ordinary $n$-functors (we can choose whether to use strict ones only or also allow non-strict ones, and in the examples below using strict ones is sufficient), and moreover $\mathsf{F}^{\mathsf{s}}$ satisfies the condition that $\mathsf{F}^{\mathsf{s}}_k$ is full for $k < n$ and fully faithful for $k = n$, which means: Given the source and target $(k-1)$-morphisms $g, f \in F_{k-1}$ (it is implied that $g, f$ themselves share the same source and target $(k-2)$-morphisms in $F_{k-2}$), the restriction of $\mathsf{F}^{\mathsf{s}}_k$ from $F_k|_{g,f}$ to $C_k|_{\mathsf{F}^{\mathsf{s}}_{k-1}(g), \mathsf{F}^{\mathsf{s}}_{k-1}(f)}$ is surjective for $k < n$ and bijective for $k = n$.[96] And $F$ establishes a higher ananatural equivalence between $C$ and $D$ if $\mathsf{F}^{\mathsf{t}}$ also has these properties of $\mathsf{F}^{\mathsf{s}}$.

Compositions and ananatural transformations of higher anafunctors are essentially defined by the same diagrams (97), (98) as before. But there is some new ingredient. Consider two strict 2-categories $C$ and $D$. Even between two ordinary 2-functors from $C$ to $D$, we can have 2-ananatural transformations that are beyond the ordinary 2-natural transformations. This is because an ordinary 2-natural transformation involves a functor from $C_1 \rightrightarrows C_0$ to $D_2 \rightrightarrows D_1$ (in a generic 2-category $D_2$ not only contains identity 2-morphisms), and now we can consider the possibility that this functor becomes an anafunctor. This is intuitive if we think pictorially (along the lines of (96)): In (91), even if the $F_0$ red arrow and the $G_0$ red arrow emanating from $c$ are unique for each given $c$ (so that $F, G$ are ordinary 2-functors), the blue surface in between (and hence the vertical black arrow on the left) might still admit multiple choices. Of course, in a more general 2-ananatural transformation between two anafunctors, both the

---

[96]We want to make sure this is a sensible definition. In particular, we shall make sure that, if we view an $n$-category as an $(n+1)$-category with identity $(n+1)$-morphisms only, then "bijection" for $k = n$ can be replaced by "surjection", as long as we have "bijection" for $k = n+1$. This is indeed true. Given $\phi, \psi$ in $F_n$, the restriction $F_{n+1}|_{\phi,\psi}$ is empty if $\phi \neq \psi$ and has a unique element $\mathbf{1}_\psi$ if $\phi = \psi$; likewise for $C_{n+1}$. So, first of all, for $k = n+1$, the map from $F_{n+1}|_{\phi,\psi}$ via $\mathsf{F}^{\mathsf{s}}_{n+1}$ to $C_{n+1}|_{\mathsf{F}^{\mathsf{s}}_n(\phi), \mathsf{F}^{\mathsf{s}}_n(\psi)}$ is automatically injective; then the only non-trivial requirement is that it is also surjective. It being surjective means, whenever $C_{n+1}|_{\mathsf{F}^{\mathsf{s}}_n(\phi), \mathsf{F}^{\mathsf{s}}_n(\psi)} = \{\mathbf{1}_{\mathsf{F}^{\mathsf{s}}_n(\phi)} = \mathbf{1}_{\mathsf{F}^{\mathsf{s}}_n(\psi)}\}$ instead of being empty, we must have $F_{n+1}|_{\phi,\psi} = \{\mathbf{1}_\psi = \mathbf{1}_\phi\}$ instead of being empty, which indeed establishes the injectivity for $k = n$, as desired.

red arrows and the blue surface emanating from any given $c$ may admit non-unique choices. We will see how such new ingredient is relevant in our main construction in Section 5.5, in particular in (134).

After introducing the notion of anafunctor, let us come back to the motivating problem at the beginning of this subsection. We can say if two manifolds are homotopic, then the strict path $d$-groupoids of the manifolds, the strict path $d$-groupoids of patches covering the manifolds (with the identification morphisms between different patches), and the strict $d$-groupoids of the lattices discretizing the manifolds, are all ananaturally equivalent to each other as strict $d$-groupoids.[97]

Clearly, in physics, not only does the homotopy information of the spacetime (as a continuum manifold or a lattice) matter. Besides the topological properties, usually we are also interested in the non-topological correlations of observables at generic energy/length scales—for example, how confinement happens in Yang-Mills theory is an important problem at the intermediate energy scale $\Lambda_{QCD}$. Roughly speaking, the homotopy class (and hence the ananatural equivalence class) of the spacetime becomes most important towards the IR limit, because this information is unchanged under coarse graining; but in general we care about interesting physics problems at generic energy/length scales, and so we also need to care about more details of a category depending on the problem of interest.

The discussions above are about the spacetime, appearing as the source category of a field configuration. Similar situation happens on the side of the target category, i.e. the d.o.f., too. This has already been explained in Section 5.1, except "natural equivalence" should more precisely be "ananatural equivalence": Unlike in topological lattice field theory, in a generic dynamical lattice field theory, the physics is not determined by the ananatural equivalence class of the target category, because the weight of the path integral does not respect invariance under general ananatural transformations.

That said, in our main context of "topologically refining" a lattice QFT,[98] it is still useful to consider the ananatural equivalence class of a target category. It turns out the (higher category analogue of) skeleton of the ananatural equivalence class tells us which topological operators (mathematically, homotopy type) a topological refinement of the lattice theory enables us to explicitly define, regardless of the details of the path integral weight, and in particular regardless of whether these topological operators will play any important role in the IR—since this is usually hard to know *a priori*. Let us go through the known examples:

1. Consider a Villainized nl$\sigma$m. For simplicity let us first assume the vortices are forbidden. As said in the previous subsection, the target category is $\widetilde{\mathcal{T}}^2/\Gamma \rightrightarrows \mathcal{T}$. This is ananaturally equivalent to the skeleton $B\Gamma = (\Gamma \rightrightarrows *)$, established by

$$
\begin{array}{ccccc}
\Gamma & \xleftarrow{\text{f.f.}} & \widetilde{\mathcal{T}}^2 \times \Gamma & \xrightarrow{\text{f.f.}} & \widetilde{\mathcal{T}}^2/\Gamma \\
\downdownarrows & & \downdownarrows & & \downdownarrows \\
* & \xleftarrow{\text{surj}} & \widetilde{\mathcal{T}} & \xrightarrow{\text{surj}} & \mathcal{T}
\end{array} , \tag{103}
$$

---

[97]Using higher category theory to lay the foundation of homotopy theory is an important program in mathematics [41, 42], and weak higher categories must be used. This is because the skeleton (under ananatural equivalence) of any strict $n$-groupoid (may as well take $n \to \infty$) can be expressed in terms of *crossed complex*, which is a generalization of the crossed module introduced before that describes the strict 2-group [105, 106], and a crossed complex does not contain information about the Whitehead product of the $\pi_m$'s. That is why weak higher category is in general needed to capture the full homotopy information. The Kan complex to be introduced in Section 5.4 is one such construction [41].

[98]We have not yet defined "topological refinement" mathematically, but we already have the experience what this means from the previous sections. An important goal of the remaining parts of this paper is to lead towards a suitable notion. One definition will be given below through the known examples, in terms of extending the traditional target category by the homotopy type, and another closely related notion will be seen in Section 5.5, in terms of anafunctor from traditional target category to the category of defects. Further discussions will be made in Section 6.

where the span has a surjective and fully faithful ordinary functor to the right by identifying $(x, y, \gamma) \in \widetilde{\mathcal{T}}^2 \times \Gamma$ with $(\gamma'x, \gamma''y, \gamma'\gamma\gamma''^{-1}) \in \widetilde{\mathcal{T}}^2 \times \Gamma$ for any $\gamma', \gamma'' \in \Gamma$ (and this is the categorical nature of what we called the $\Gamma$ gauge invariance in Section 2.1 where $\mathcal{T} = S^1$ and $\Gamma = \mathbb{Z}$ and in Section 2.3 for more general $\mathcal{T}$ and $\Gamma$), and a surjective and fully faithful ordinary functor to the left by mapping $\widetilde{\mathcal{T}}$ to $*$. The $\Gamma$ at the 1-morphism in $B\Gamma$ originates from the fact that $\pi_1(\mathcal{T}) \cong \Gamma$; in more formal terms, $B\Gamma$ is the *homotopy 1-type* of $\mathcal{T}$. Physically, it means the Villainized nl$\sigma$m allows us to explicitly describe $\Gamma$-valued windings, regardless of whether it is important or not in the dynamics due to the path integral weight.

It is illuminating to think about the deep IR limit, where the lattice is so coarse grained such that, the $\bar{\mathcal{L}}_1 \rightrightarrows \bar{\mathcal{L}}_0$ part becomes the skeleton of the fundamental groupoid, $\widetilde{\pi}_1(\mathcal{M}) \rightrightarrows \pi_0(\mathcal{M})$. If we also reduce the target category to its skeleton $B\Gamma$, then a field configuration is a homomorphism from the non-contractible loops to $\Gamma$, as expected.

When the vortices are not forbidden, the target category is $(\widetilde{\mathcal{T}}^2/\Gamma) \times \Gamma \rightrightarrows \widetilde{\mathcal{T}}^2/\Gamma \rightrightarrows \mathcal{T}$, which is ananaturally equivalent to $BE\Gamma = (\Gamma^2 \rightrightarrows \Gamma \rightrightarrows *)$. The extra $\Gamma$ at the 2-morphism describes the vortices. This category is in turn ananaturally equivalent to the trivial category $*$, which physically suggests that the theory describes a trivial phase if the plaquette weight is sufficiently insensitive (just like in the case of traditional lattice gauge theory that we describe in the previous subsection).

This is one way to mathematically motivate the target category to be used for Villainized nl$\sigma$m: The desired target category, i.e. the right column of (103), has the properties that it covers the traditional target category $E\mathcal{T}$, and is moreover ananaturally equivalent to $B\Gamma$, the homotopy 1-type of $\mathcal{T}$. This can be described as a $B\Gamma$ extension of $E\mathcal{T}$, keeping the objects $\mathcal{T}$ unchanged. In Section 5.5, in particular (127), we will systematically introduce a closely related but alternative way towards the same goal, started out by allowing rather than forbidding vortices.

2. A Villainized gauge theory is similar as long as we replace the 0-category $\mathcal{T}$ by the 1-category $BG$—or we can say, as long as we take $\mathcal{T} = G$ and deloop everything said above. In particular, $\Gamma$ must be abelian. "$\pi_1(\mathcal{T}) \cong \Gamma$" stays "$\pi_1(G) \cong \Gamma$", but "a homomorphism from non-contractible loops in $\mathcal{M}$ to $\Gamma$" becomes "a homomorphism from non-contractible surfaces in $\mathcal{M}$ to $\Gamma$".

3. If $G$ itself abelian, we can further deloop arbitrarily many times.

4. Consider the spinon decomposed $S^2$ nl$\sigma$m. For simplicity we first assume the hedgehogs are forbidden. The target category is $S^2 \times SU(2) \times \mathbb{R} \rightrightarrows S^2 \times SU(2) \rightrightarrows S^2$ (recall (89)), which is ananaturally equivalent to the skeleton $B^2\mathbb{Z} = (\mathbb{Z} \rightrightarrows * \rightrightarrows *)$, established by

$$
\begin{array}{ccccc}
\mathbb{Z} & \xleftarrow{\text{f.f.}} & SU(2)^2 \times \mathbb{R}^2 \times \mathbb{Z} & \xrightarrow{\text{f.f.}} & S^2 \times SU(2) \times \mathbb{R} \\
\Downarrow & & \Downarrow & & \Downarrow \\
* & \xleftarrow{\text{full}} & SU(2)^2 \times \mathbb{R} & \xrightarrow{\text{full}} & S^2 \times SU(2) \\
\Downarrow & & \Downarrow & & \Downarrow \\
* & \xleftarrow{\text{surj}} & SU(2) & \xrightarrow{\text{surj}} & S^2
\end{array}
\quad , \quad (104)
$$

where the span has an ordinary functor to the right, full at the lower layers and fully faithful at the top layer, by identifying $(\mathcal{U}, \mathcal{U}', a, a', s) \in SU(2)^2 \times \mathbb{R}^2 \times \mathbb{Z}$ with $(\mathcal{U}e^{i\alpha\sigma^z}, \mathcal{U}'e^{i\alpha'\sigma^z}, a + \alpha - \alpha' + 2\pi k, a' + \alpha - \alpha' + 2\pi k', s + k - k') \in SU(2)^2 \times \mathbb{R}^2 \times \mathbb{Z}$ for any $k' \in \mathbb{Z}$ and $\alpha, \alpha' \in \mathbb{R}$ (this is the categorical nature of the 1-form $\mathbb{Z}$ gauge invariance and the $\mathbb{R}$ mod $2\pi\mathbb{Z}$ gauge invariance in Section 2.4), and a surjective and

fully faithful ordinary functor to the left by collapsing $SU(2)$ and $\mathbb{R}$ to $*$. Similar to the Villainization case, the $\mathbb{Z}$ in the 2-morphism is related to the fact that $\pi_2(S^2) \cong \mathbb{Z}$; in more formal terms, $B^2\mathbb{Z}$ is the *homotopy 2-type* of $S^2$. Physically it means the spinon decomposition allows us to explicitly describe $\mathbb{Z}$-valued skyrmions.

When the hedgehogs are not forbidden, the target category has the space of 3-morphisms being $S^2 \times SU(2) \times \mathbb{R} \times \mathbb{Z}$, where the extra $\mathbb{Z}$ (compared to the space of 2-morphisms) describes the hedgehogs. The target category is ananaturally equivalent to $B^2 E\mathbb{Z}$, which is in turn ananaturally equivalent to the trivial category, and this physically suggests the theory can describe the trivial phase if the cube weight is sufficiently insensitive.

Again, the desired target category, i.e. the right column of (104), can be viewed as an extension of the traditional target category $ES^2$ by the homotopy 2-type $B^2\mathbb{Z}$ of $S^2$, keeping the objects $S^2$ unchanged. And in Section 5.5, in particular (131), we will systematically introduce a closely related but alternative way towards the same goal, started out by allowing rather than forbidding hedgehogs.

From these discussions, it becomes clear that to tackle the main problems we aim at, for nl$\sigma$m we need a topological refinement for $\mathcal{T} = S^3$ so that the target category is internal to Manifold (which implies finite dimensional) and has a ananatural equivalence—similar to the ones in the examples above—to $B^3\mathbb{Z}$ (when baryon non-conserving hedgehogs are forbidden) or $B^3 E\mathbb{Z}$ (when baryon non-conserving hedgehogs are allowed); then, for Yang-Mills theory we take $\mathcal{T} = SU(N)$ and suitably deloop the refined target category, to obtain one that has a ananaturally equivalence to $B^4\mathbb{Z}$ (when Yang monopoles are forbidden) or $B^4 E\mathbb{Z}$ (when Yang monopoles are allowed). If we do not care about the d.o.f. being finite dimensional—so that we are internalizing in Difflg instead of Manifold—then, as we said in Section 5.1, we can simply turn (54) (along with the Villainizing layer at the top of (51)) into a strict higher category (just like how (48) is related to (89)) to fulfill the goal. However, for an actual lattice model, the d.o.f. being finite dimensional is crucial. To satisfy all these conditions, it turns out we have to work with more flexible higher categories, in generalization to the strict higher categories that we have been working with so far.

## 5.3 Weak categories: Bicategories and tricategories

Let us introduce some weak categories that are more flexible than the strict ones, but not as flexible as what we will finally need. In particular we will focus on the weak 2- and 3-categories called *bicategories* and *tricategories*. They have been extensively used in the study of topological phases and generalized symmetries, which we will briefly mention but not go deeply into. We will mainly emphasize the conceptual aspects which will lead us towards what we actually need in the next subsections; also, we will see the mathematical origin of the Yang-Baxter equation issue mentioned below (83).

In this subsection we will ignore topology, so that all the structures are internalized in Set. In fact, our very reason to go towards even more flexible definitions of categories in the next subsection is to take topology into account.

From the definitions of lax 2-natural transformation and lax 2-functor, we have learned that, when non-trivial 2-morphisms are available, we may replace the equality signs that appear in some consistency conditions between 1-morphisms by more general (i.e. possibly non-identity) 2-morphisms between 1-morphisms. Now, we note that even in the definition of category itself, there are some equality signs describing consistency conditions between 1-morphisms—the associativity condition $(h \circ g) \circ f = h \circ (g \circ f)$, and the unital condition $f \circ \mathbf{1}_a = f = \mathbf{1}_b \circ f$. These equality signs can be understood as identity 2-morphisms, which are the only 2-morphisms available in a 1-category. However, if we have a 2-category with

more general 2-morphisms, it is possible to relax these conditions on 1-morphisms, by replacing the equality signs (identity 2-morphisms) with more general 2-morphisms:

$$h \circ (g \circ f) \overset{\alpha_{h,g,f}}{\Longleftarrow} (h \circ g) \circ f, \qquad f \overset{\lambda_f^L}{\Longleftarrow} \mathbf{1}_b \circ f, \qquad f \overset{\lambda_f^R}{\Longleftarrow} f \circ \mathbf{1}_a, \tag{105}$$

where $\alpha_{h,g,f}$ is called the *associator* for $h, g, f$, and $\lambda_f^L$ and $\lambda_f^R$ are the *left and right unitors* for $f$; we require these 2-morphisms to be invertible under vertical composition. We expect the associators and unitors to satisfy suitable consistency conditions which ultimately follow from the fact that the only available 3-morphisms are identity 3-morphisms. It is helpful to explain the details of these conditions, because over the process we will develop some important perspectives.

To begin, we picture an associator as

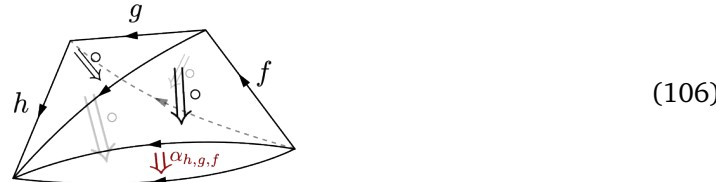

$$\tag{106}$$

which looks like a tetrahedron but with one edge becoming a slit filled with the associator 2-morphism.

- The most crucial point to read-off from this picture is that, the composition ∘ should be thought of as a generalized kind of 2-morphism, or say a 2-cell, which has a triangular shape that takes two source 1-morphisms to one target 1-morphism, rather than the previous globular shape (84) which takes one source 1-morphism to one target 1-morphism. Let us denote by $[g \circ f]$ the triangular shape 2-morphism that takes $g, f$ to $g \circ f$.

  To appreciate the importance of this point of view, let us consider such a situation (which will actually appear in our discussion soon). It is possible that, as 1-morphisms, $h \circ (g \circ f)$ and $(h \circ g) \circ f$ are equal. In that case, however, we can still have an associator $\alpha_{h,g,f}$ which is not the identity 2-morphism. What would the associator mean if the two 1-morphisms are equal already? The point of view above explains it: While $h \circ (g \circ f)$ denotes a 1-morphism, we shall also view the process as a trapezoidal shaped 2-morphism, denoted as $[h \circ (g \circ f)]$, that takes three source 1-morphisms $h, g, f$ to one target 1-morphism which we called $h \circ (g \circ f)$; likewise for $[(h \circ g) \circ f]$. Thus, regardless of whether $h \circ (g \circ f)$ and $(h \circ g) \circ f$ are equal as 1-morphisms, $[h \circ (g \circ f)]$ and $[(h \circ g) \circ f]$ may still be different as trapezoidal shaped 2-morphisms, and the difference is captured by the associator $\alpha_{h,g,f}$. So (106) means

$$[h \circ (g \circ f)] = \alpha_{h,g,f} \circ_\mathrm{v} [(h \circ g) \circ f], \tag{107}$$

  where the equality is made sense of as trapezoidal shaped 2-morphisms taking three sources to one target. This is what the picture (106) really means.

  In particular, the "equality as 2-morphisms" is because there is no non-identity 3-morphisms—in the picture, the bounded 3d volume represents the equality sign in the formula above.[99]

---

[99]Our perspective becomes closer and closer to that of a *simplicial set*, which is indeed what we will get to in the next subsection. Intuitively, it becomes more and more similar to a lattice theory (which is desired), or to a high dimensional tiling game with certain rules, such as what kind of tiles are available and which ones can join together. This is indeed the nature of it.

- Since the only available 3-morphisms are identity 3-morphisms, it is easy to see that the 2-morphisms satisfy strict associativity under consecutive vertical compositions, and strict interchangeability between vertical and horizontal compositions. On the other hand, the associativity under consecutive horizontal compositions is slightly modified. Replacing the arrows for $h, f, g$ in (106) by slits $h' \overset{\varphi}{\Leftarrow} h, g' \overset{\psi}{\Leftarrow} g, f' \overset{\rho}{\Leftarrow} f$ under consecutive horizontal compositions, it is not hard to see in the end we will be left with a 3d volume bounded by four slits, which represents the equality between 2-morphisms

$$(\varphi \circ_{\mathsf{h}} (\psi \circ_{\mathsf{h}} \rho)) \circ_{\mathsf{v}} \alpha_{h,g,f} = \alpha_{h',g',f'} \circ_{\mathsf{v}} ((\varphi \circ_{\mathsf{h}} \psi) \circ_{\mathsf{h}} \rho). \tag{108}$$

- When four 1-morphisms $j, h, g, f$ are consecutively composed, starting with the upper left in the diagram below, by using associators and whiskering, we conclude that the two results at the lower right must be equal as 2-morphisms, as indicated by the blue equal sign. Since their triangular parts are the same, the equality becomes that of the vertical compositions of the (whiskered) associators.

$$
\begin{array}{c}
\text{(diagram)}
\end{array}
\tag{109}
$$

This is often called the "pentagon equation" of the associators (the pentagon refers to the five red equal signs, at which associators are introduced).

The pentagon equation can also be thought of as five tetrahedra (106) piecing up to a 4d simplex, where the slits are taken care of by filling in two extra 3d volumes representing the whiskerings. And the existence of such an equation is simply because the only available 4-morphisms are identites ones.

- In (106) or (107), if $g = \mathbf{1}_b$ is some identity 1-morphism, by applying the unitors and suitable whiskerings, we obtain

$$\mathbf{1}_h \circ_{\mathsf{h}} (\lambda_f^L)^{-1} = \alpha_{h,\mathbf{1}_b,f} \circ_{\mathsf{v}} ((\lambda_h^R)^{-1} \circ_{\mathsf{h}} \mathbf{1}_f), \tag{110}$$

which is often called the "triangle equation", similar to the pentagon equation above.

This explains how a single, simple fact that all available 3- and higher morphisms are identities lead to a set of seemingly complicated consistency conditions satisfied by the associators and the unitors—so these conditions can be thought of as being *derived*, rather than being imposed at will. Such a 2-category is called a *weak 2-category*, or a *bicategory*. Between weak 2-categories, it is generally impossible to define strict 2-functors, so we must use pseudo or lax 2-functors and 2-natural transformations.

There is a coherence theorem for 2-categories stating that every weak 2-category is naturally equivalent to some strict 2-category (and this is not true for higher categories), but practically there are many advantages to work with weak 2-categories [44, 103].

In our main construction we do not directly use bicategories. However, they are widely used in both mathematics and theoretical physics. Here we briefly review some applications in physics related contexts. Most of these applications concentrate on bicategories with a single object. (A bicategory with a single object can be viewed as the delooping $BM$ of a 1-category $M$ equipped with suitable extra structures; such an $M$ is called a *monoidal category*.)

One major application is on the classification of 2-groups [107]. It is proven that every 2-group (recall we ignore topology in this subsection) is naturally equivalent to a skeletal weak 2-group $K \ltimes A \rightrightarrows K \rightrightarrows *$ where $A$ is abelian, and being skeletal at the 1-morphism level means $\mathsf{s}((k,a)) = \mathsf{t}((k,a)) = k$;[100] the unitor is trivial, and the associator $\widetilde{\alpha} : K^3 \to A$ (where $\alpha_{k,k',k''} = (kk'k'', \widetilde{\alpha}_{k,k'k''}) \in K \ltimes A$, so $\mathsf{s}(\alpha_{k,k',k''}) = \mathsf{t}(\alpha_{k,k',k''}) = kk'k''$ and the non-trivial content in $\alpha$ is $\widetilde{\alpha}$) is well-defined as an element of the group cohomology $H^3_{\mathrm{group}}(K;A)$. This classification is important in physics because, as we have seen before, the phase of a system is characterized by the natural equivalence class of the target category of the low energy effective theory (which is in general different from that of the original target category in a UV lattice theory), and thus the phases of those systems described by 2-group symmetries or 2-group gauge theories at low energies are classified using the skeletal weak 2-groups [36, 46, 47]. In particular, given a strict 2-group $G \ltimes H \rightrightarrows G \rightrightarrows *$, the naturally equivalent skeletal weak 2-group has $A = \ker(\widetilde{\mathsf{t}}), K = \mathrm{coker}(\widetilde{\mathsf{t}})$, forming the exact sequence $* \to A \to H \xrightarrow{\widetilde{\mathsf{t}}} G \to K \to *$; the associator arises from the fact that, as groups, in general $H \neq A \times (H/A)$ and $G \neq K \times (H/A)$.[101,102] (Conversely, by the coherence theorem mentioned above, given a weak skeletal 2-group, there always exists a naturally equivalent strict 2-group; more particularly, given $A$ and $K$ in the exact sequence, the "2-extension problem" of finding the possible choices of $H$ and $G$ is indeed classified by $H^3_{\mathrm{group}}(K;A)$, the data encoded by the associator.)

It can also be noted that, when $A = U(1)$, we may use the associator as the Dijkgraaf-Witten phase [19] for a 3d topological order with $K$ lattice gauge field (recall we ignore topology here, so $K$ is discrete, and thus we may forbid its flux, as is assumed in finite group Dijkgraaf-Witten theory), or as the WZW phase for a 2d symmetry protected topological order with $K$ global symmetry [23]. But in these applications, it is better not to view $\alpha$ as an associator 2-morphism, but rather, equivalently, as a non-identity 3-morphism, i.e. (107) becoming

$$[h \circ (g \circ f)] \overset{\alpha_{h,g,f}}{\Longleftarrow} [(h \circ g) \circ f], \tag{111}$$

or more pictorially, (106) becomes a tetrahedron (this is possible since $(h \circ g) \circ f = h \circ (g \circ f) \in K$) with the associator 3-morphism $\alpha_{h,g,f}$ filling the 3d volume—indeed this is how we usually think of the Dijkgraaf-Witten phase. Moreover this will make better connection to the cases of continuous-valued d.o.f. to be discussed in the next two subsections.

---

[100]So this is an example of the situation we explained before, that $k \circ (k' \circ k'') \overset{\alpha_{k,k',k''}}{\Longleftarrow} (k \circ k') \circ k''$ has the same source and target 1-morphisms, but the associator 2-morphism is still meaningful.

[101]While we will not rigorously prove the natural equivalence here, we can explain how the associator arises in more details. Let $g_k \in G$ denote a chosen lift of $k \in K$. We have $g_{kk'} = \widetilde{\mathsf{t}}(\beta_{k,k'})g_k g_{k'}$ where $\beta_{k,k'} \in H$; it is in general impossible to simultaneously make all $\widetilde{\mathsf{t}}(\beta_{k,k'}) = 1$ as $G \neq K \times (H/A)$ in general. When three elements in $K$ are composed, we have $g_{kk'k''} = \widetilde{\mathsf{t}}(\beta_{kk',k''})\widetilde{\mathsf{t}}(\beta_{k,k'})g_k g_{k'} g_{k''} = \widetilde{\mathsf{t}}(\beta_{k,k'k''})(g_k \widetilde{\mathsf{t}}(\beta_{k',k''})g_k^{-1})g_k g_{k'} g_{k''}$. In the language of strict 2-group we write $g_{kk'} \overset{\beta_{k,k'}}{\Longleftarrow} g_k g_{k'}$, and $g_{kk'k''} \overset{\beta_{kk',k''}\beta_{k,k'}}{\Longleftarrow} (g_k g_{k'})g_{k''}$ and $g_{kk'k''} \overset{\beta_{k,k'k''}(^k\beta_{k',k''})}{\Longleftarrow} g_k(g_{k'}g_{k''})$. Thus we find $\widetilde{\alpha}_{k,k',k''} := (\beta_{k,k'k''}\beta_{k',k''})^{-1}(\beta_{kk',k''}(^k\beta_{k,k'}))$ satisfies $\widetilde{\mathsf{t}}(\widetilde{\alpha}_{k,k',k''}) = 1$, i.e. $\widetilde{\alpha}_{k,k',k''} \in A$. The pentagon equation and the facts that in general $\beta \notin A$ but has an ambiguity parametrized by $A$ implies $\widetilde{\alpha} \in H^3_{\mathrm{group}}(K;A)$.

[102]In the Villainized gauge theories we discussed, $K = \mathrm{coker}(\widetilde{\mathsf{t}})$ is trivial. But as we said in Section 5.1, the Villainized gauge theory is not a 2-group gauge theory, because it has non-trivial path integral weight that is only invariant under strict 2-natural transformations (gauge transformations) but not the laxer ones for a 2-group gauge theory. That is, in a dynamical lattice QFT, we not only care about the natural equivalence class of the target category, but also the target category itself, so $K$ being trivial does not mean the theory is trivial. On the other hand, there are physical applications with non-trivial $K$ [34–36, 46, 47], including studies on the possible low energy phases after Yang-Mills confinement and/or Higgsing.

Now let us briefly introduce weak 3-category, or *tricategory*. In our main construction, when we go from $S^3$ nl$\sigma$m to $SU(2)$ gauge theory (which, as we can tell now, is some kind of delooping process) in Section 4.2, recall there is a potential issue (83) involving Yang-Baxter equation that we could have had encountered. Now we can understand the origin of this problem in terms of tricategory.

When non-identity 3-morphisms are available, the consistency conditions between 2-morphisms in a strict or weak 2-category can be relaxed by replacing equalities (identity 3-morphisms) with more general invertible 3-morphisms. These conditions include the unital law for identity 2-morphisms, the interchangeability, the vertical associativity, the modified horizontal associativity (108), the pentagon equation (109), and the triangle equation (110).

Let us mainly look at the case where a 3-category is almost like a strict 3-category, except the interchangeability of 2-morphisms—the last diagram of (86)—is weakened. Such weak 3-categories are called *Gray 3-categories*. There is a coherence theorem for 3-categories stating that every weak 3-category is naturally equivalent to some Gray 3-category, but not to any strict 3-category in general [108].[103]

In a Gray 3-category, the last diagram of (86) is relaxed, i.e. composing vertically first and then horizontally and composing horizontally first and then vertically may result in different 2-morphisms, but we may specify an invertible *interchanger* 3-morphism between them. (Even when the two resulting 2-morphisms are equal, a non-identity interchanger is still meaningful, just like the associator case we discussed above (107).) But practically it is more convenient to do the following. We first define vertical composition and left and right whiskerings, and then use the whiskerings and vertical composition to define two kinds horizontal compositions,

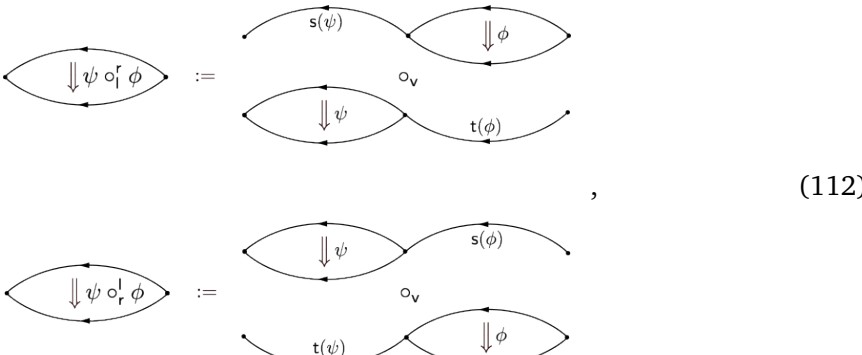

$$\tag{112}$$

and then introduce the "interchanger" invertible 3-morphism $\rho_{\psi,\phi}$ that relates these two kinds of horizontal compositions:

$$\psi \circ_r^l \phi \overset{\rho_{\psi,\phi}}{\Leftarrow} \psi \circ_l^r \phi. \tag{113}$$

It is not hard to see that the 3-morphism $\rho_{\psi,\phi\circ_v\chi}$ can be written as the composition ($\circ_2$ means composition of 3-morphisms by joining along source and target 2-morphisms)

$$\psi \circ_r^l (\phi \circ_v \chi) \overset{\rho_{\psi,\phi\circ_v\chi}}{\Leftarrow} \psi \circ_l^r (\phi \circ_v \chi)$$

$$= \left[ \psi \circ_r^l (\phi \circ_v \chi) \overset{1_{t(\psi)\circ_h\phi}\circ_v\rho_{\psi,\chi}}{\Leftarrow} ((\psi \circ_l^r \chi) \circ_r^l \phi = (\psi \circ_r^l \phi) \circ_l^r \chi) \overset{\rho_{\psi,\phi}\circ_v 1_{s(\psi)\circ_h\chi}}{\Leftarrow} \psi \circ_l^r (\phi \circ_v \chi) \right]$$

$$= \psi \circ_r^l (\phi \circ_v \chi) \overset{\left[1_{t(\psi)\circ_h\phi}\circ_v\rho_{\psi,\chi}\right]\circ_2\left[\rho_{\psi,\phi}\circ_v 1_{s(\psi)\circ_h\chi}\right]}{\Leftarrow} \psi \circ_l^r (\phi \circ_v \chi), \tag{114}$$

---

[103]Such a level of strictness, that lets the *n*-categories under consideration to "appear as strict as possible", meanwhile any generic weak *n*-category is still equivalent to at least one of those *n*-categories under consideration, is called "semi-strict". For $n = 2$, semi-strict is strict. For $n = 3$, semi-strict can be Gray, but there are also other options, such as Simpson, which weakens the unital laws instead of interchangeability.

where we used the assumption that all composition rules other than interchangeability are strict, and the fact that the only available 4-morphisms are identities. From this we can further derive that, when three 2-morphisms are consecutively "horizontally composed", there will be $3! = 6$ different definitions related by six interchangers (with suitable 2-whiskerings), and they satisfy a consistency equation. Such a "hexagon equation"[104] constraint is in fact the Yang-Baxter equation; it is due to the fact that the only available 4-morphisms are identities, just like the pentagon equation (109) for the associators comes from the fact that the only available 3-morphisms are identities.

Recall from (88) that a strict 2-group can be described as a "crossed module", and moreover any 2-group is equivalent to some strict 2-group. Similarly, any Gray 3-group (i.e. Gray 3-category with single object and strictly invertible 1-, 2- and 3-morphisms) can be described as a "2-crossed module", and in this context the interchanger is known as "Peiffer lifting", see e.g. [43]; moreover, any 3-group is equivalent to some Gray 3-group.

If we go beyond Gray 3-category by weakening more composition rules (such as the associativities of 1- and/or 2-morphisms), then the form of (114) and hence the form of the Yang-Baxter equation will change,[105] but the spirit is the same—all constraints come from the fact that the only available 4-morphisms are identities.

One major application of weak 3-category in physics occurs in delooping, which, as we have seen, is important for gauge theory. Recall a (delooped) group $BG$ cannot be further delooped to a 2-category $B^2G$ if $G$ is non-abelian, due to the interchangeability condition. If we really do want to deloop, we may discard elements of $G$ by only keeping its center $Z(G)$ and delooping to $B^2Z(G)$. However, if we have a (delooped) strict 2-group $G \ltimes H \rightrightarrows G \rightrightarrows *$, we may deloop it to a Gray 3-category not by discarding information, but by specifying more information—the interchangers (note that in general there are more than one inequivalent ways to consistently specify interchangers; also, even if $G$ is abelian, specifying the interchangers is still meaningful as explained before). The same is true for more general 2-categories.[106]

In physics, delooping usually occurs in two ways. One is when we take $\mathcal{T} = |G|$ in the target category of a nl$\sigma$m and deloop it to the target category of a gauge theory, as we have seen before. Another is when we start with a gauge theory, but look at the gauge invariant (up to conjugation) fluxes, so that $G$-valued link d.o.f. (as 1-morphisms) are ignored, and we only look at the $G$-valued fluxes on plaquettes (as 2-morphisms). The second way is commonly seen in the study of anyons.[107] Here we will focus on the first way.

---

[104]This is different from what is usually called the "hexagon equation" in topological order. There, the "hexagon equation" is a generalized version of (114)—when the associativity is also non-strict, on the right-hand-side of (114) there will be three associators involved in addition, forming a hexagonal picture.

[105]In particular, (114) will become what is usually called the hexagon equation, see the previous footnote. It appears commonly in the description of topological order, see footnote 107.

[106]Instead of choosing one way of specifying the interchangers, we may also enlarge the 2-category by having many copies of the original one, such that each copy uses one possible consistent set of specifications of interchangers. The detailed construction is called *Drinfeld center*, see e.g. [109]. In a suitable sense, the Drinfeld center is the generalization of the notion of "center" for usual groups.

[107]In the lattice Turaev-Viro-Levin-Wen models of 2+1d topological orders (see e.g. [20, 22, 45, 109]), we start with a (possibly weak) 2-category with single object, i.e. a delooped monoidal category. For instance, for Dijkgraaf-Witten model with discrete group $K$, the 2-category will be the skeletal bicategory $K \ltimes U(1) \rightrightarrows K \rightrightarrows *$ (recall the discussions before (111)). By further specifying the interchangers, this 2-category can be further delooped to a weak 3-category, whose 2-morphisms are supposed to describe anyon types (indeed, anyons are co-dimension 2 in 2+1d); in this context, a monoidal category with the interchanger specified (so that it can be delooped twice into a weak 3-category) is called a *braided monoidal category*, and the interchanger is also called *braiding*. And we consider all possible consistent ways of braiding specification, so the resulting weak 3-category is in fact (the twice delooping of) the Drinfeld center introduced in the previous footnote. For instance, in the case of Dijkgraaf-Witten model, the 2-morphisms of this weak 3-cateogry has a $K$ label which describes an anyon's $K$-valued fluxes around a plaquette (which transforms by conjugation under gauge transformation), and it also has a label that specifies the choice of braiding, which represents the anyons's charge under $K$; thus, in this case, the Drinfeld center just encodes the anyon's flux and charge.

Consider the strict 2-groupoid $\bar{\mathcal{P}}_2 S^3 \times U(1)/WZW \rightrightarrows \bar{\mathcal{P}} S^3 \rightrightarrows S^3$ which, as we said in Section 5.1, re-expresses the structure (54)—which describes WZW curving of $S^3$ nl$\sigma$m in the continuum.[108] Now we view the space $S^3$ as a group $SU(2)$ and try to deloop the category to understand CS in $SU(2)$ gauge theory in the continuum. We note that the 1-category part $\bar{\mathcal{P}} SU(2) \rightrightarrows SU(2)$ cannot be delooped because the interchangeability could not be satisfied: When we attempt to deloop the above to something like "$\bar{\mathcal{P}} SU(2) \rightrightarrows SU(2) \rightrightarrows *$", we can define the vertical composition as the concatenation of paths, and define the left/right whiskering as the group multiplication of an $SU(2)$ element on the left/right of each point on a path in $SU(2)$. Then the two horizontal compositions in (112) indeed yield two different 2-morphisms, i.e. two different paths in $SU(2)$. This is nothing but what we have encountered in (81). Fortunately, originally we also have non-trivial 2-morphisms, $\bar{\mathcal{P}}_2 S^3 \times U(1)/WZW$, which will become 3-morphisms after delooping to $\bar{\mathcal{P}}_2 S^3 \times U(1)/WZW \rightrightarrows \bar{\mathcal{P}} S^3 \rightrightarrows S^3 \rightrightarrows *$, so we can choose suitable elements from $\bar{\mathcal{P}}_2 S^3 \times U(1)/WZW$ as interchangers, which explains what we discussed below (83). (A crucial point emphasized there is that, in the actual construction there—whose mathematical origin will be discussed in Section 5.5—we did not have to really make effort to choose these interchangers and the issue is automatically resolved. Why this the case needs to be better understood in future works.)

## 5.4 Weak categories: Simplicial and cubical ones

The crucial perspective brought to us by (106), that we should think of the composition $\circ$ of 1-morphisms as a 2-morphism of triangular rather than globular shape, opens up an new possibility: Can we consider 2-categories with more general triangular shaped 2-morphisms?

The simplest case is to consider triangular shaped 2-morphisms obtained by vertically composing the composition triangle $\circ$ with a globular shaped 2-morphism:

$$
\begin{array}{ccc}
g \, \Downarrow\psi \, f \\ h
\end{array}
\quad = \quad
\begin{array}{ccc}
g \, \Downarrow\circ \, f \\ \Downarrow\phi \\ h
\end{array}
\quad . \tag{115}
$$

If this covers all the cases, then of course we achieve nothing new. We want to consider scenarios where a triangular shaped $\psi$ cannot be suitably decomposed into an ordinary composition $\circ$ and a globular shaped $\phi$.

The requirement of continuity/smoothness makes such more general scenarios necessary. Suppose the space of all triangular shaped 2-morphisms $\psi$ form a fibre bundle over the space $C_1 \times^{\mathsf{s},\mathsf{t}}_{C_0} C_1 \ni (g, f)$. Being able to define a unique $\circ$ in a continuous manner for every pair $(g, f)$ means this bundle has a continuous global section. But then we can conceive scenarios in which the bundle has no continuous global section—so that there is no good way to define a unique composition "$\circ$" that is continuous in its two source 1-morphisms; rather, we should just think about generic triangular shaped 2-morphisms, interpreted as non-unique compositions, such that the different compositions can be related by globular shaped 2-morphisms:

$$
\begin{array}{ccc}
g \, \Downarrow\psi \, f \\ h
\end{array}
\quad = \quad
\begin{array}{ccc}
g \, \Downarrow\psi' \, f \\ h' \\ \Downarrow\phi \\ h
\end{array}
\quad . \tag{116}
$$

(The composition of a triangular shaped 2-morphism and a globular shaped 2-morphisms appended on one of the source 1-morphisms, rather than appended on the target 1-morphism as shown above, should also be specified.)

---

[108] At the beginning of this subsection we said we will ignore smoothness in this subsection. Although this example involves smoothness, the issue we will describe now is largely independent of smoothness.

This can be casted in terms of anafunctor—which, as we have seen in Section 5.2, also becomes necessary when continuity/smoothness is required. Recall the hom-category $C|_{b,a}$ introduced in Section 5.1, with $(C|_{b,a})_0 = C_1|_{\text{t}=b,\text{s}=a}$ being the space of objects, and $(C|_{b,a})_1 = C_2|_{\text{tt}=b,\text{ss}=a}$ (the globular shaped 2-morphisms between 1-morphisms in $C_1|_{\text{t}=b,\text{s}=a}$) composing under $\circ_\text{v}$ being the space of 1-morphisms. The composition $\circ$ along with $\circ_\text{h}$ forms an ordinary functor from $C|_{c,b} \times C|_{b,a}$ to $C|_{c,a}$. However, once continuity/smoothness is taken into consideration, we should also consider anafunctor to capture more interesting possibilities, and "the triangular shaped 2-morphisms" are nothing but the objects of the span $F$ in this anafunctor.

A problem familiar in topological order exemplifies the necessity to introduce such generality. When we introduced (106), continuity/smoothness was not part of the consideration. Once continuity/smoothness is taken into account, we may want the associator $\alpha_{h,g,f}$ to be continuous/smooth in $h, g, f$. But this requirement is too restrictive to capture many interesting cases. For instance, recall the symmetry protected nl$\sigma$m [23] mentioned in the previous subsection, where the associator $\widetilde{\alpha}_{h,g,f}$ specifies an element of the group cohomology $H^3_{\text{group}}(K; U(1))$ and serves as 2d WZW phase. When $K$ is discrete, everything is fine. When $K$ is a Lie group, it is well-known that to suitably define $H^3_{\text{group}}(K; U(1))$, we should not only consider those $\widetilde{\alpha}$ that are continuous from $K^3$ to $U(1)$, but also those that are only piecewise continuous (Borel), in order to capture many interesting cases.[109] While such definition of group cohomology is mathematically consistent, the discontinuity makes the lattice models constructed out of it unphysical. Part of the problem here is indeed due to that we wanted to define a unique notion of composition. What we will explain in the next subsection, in particular (140) that leads to the construction in Section 4.1 for the 2d WZW phase of lattice $S^3$ nl$\sigma$m, is the solution to this kind of problem. (If we go to 3d and gauge the global symmetry $K$ by introducing dynamical gauge field, we obtain a Dijkgraaf-Witten theory [19]. But $K$ is a Lie group now, there is yet another key distinction with the cases of finite groups: It becomes unphysical to demand the $K$ gauge field to be flat.[110] Then the problem indeed becomes that of defining $K$ CS [19] on the lattice, and the solution will be (143), that leads to the construction in Section 4.2.)

With these motivations in mind, we are now ready to introduce the simplicial model of weak categories, which is sufficiently weak so that it is powerful enough to fulfill our goal.

We first define a *simplicial set*—which is not a single set, but a collection of sets related in a "simplicial fashion". To begin, we still have a set $C_0$ of objects, pictured as points, and a set $C_1$ of 1-morphisms (or 1-cells), pictured as arrows between objects. But now $C_2$ is a set of 2-morphisms (or 2-cells) not of globular shape between two 1-morphisms, but of triangular shape between three 1-morphisms. Likewise, $C_3$ is a set of tetrahedral shape 3-morphisms (or 3-cells) between four 2-morphisms, and so on. Thus, just like the source and target maps in Section 5.1:

- We have $k + 1$ maps $\partial_i$ ($i = 0, 1, \cdots, k$) from $C_k$ to $C_{k-1}$, called the *face maps*. We may think of $\partial_i$ as removing the $i$th vertex from a $k$-simplex to obtain a $(k-1)$-simplex. For example for $k = 2$:

$$
\begin{array}{c}
\partial_0\psi \quad \psi \quad \partial_2\psi \\
\partial_1\psi
\end{array}
\tag{117}
$$

.

---

[109]More mathematically, $H^3_{\text{group}}(K; U(1))$ defined under this piecewise continuous (Borel) condition, rather than the strictly continuous condition, is isomorphic to $H^4(|BK|; \mathbb{Z})$.

[110]See footnote 84 for a formal problem associated with this unphysical flatness requirement.

Moreover, we would like to be able to view a $(k-1)$-cell as some special kind of $k$-cell, much like the identity maps in Section 5.1:

- We have $k$ maps $\delta_i (i = 0, 1, \cdots, k-1)$ from $C_{k-1}$ to $C_k$, called the *degeneracy maps*. There are $k$ of them, because we may think of $\delta_i$ as repeating the $i$th vertex of a $(k-1)$-simplex to obtain a $k$-simplex. For example for $k = 2$:

$$
\tag{118}
$$

The face maps and degeneracy maps satisfy some pictorially obvious constraints (sometimes called simplicial identities), such as $\partial_1 \partial_0 \psi = \partial_0 \partial_2 \psi$ and $\partial_0 \delta_1 f = \delta_0 \partial_0 f (= \delta_0 b)$ in the diagrams above. Just like (92), a simplicial set can thus be viewed as a diagram

$$
\cdots \; C_2 \; \rightrightarrows \; C_1 \; \leftleftarrows \; C_0,
\tag{119}
$$

internalized in the ambient category Set. Now, if we internalize the same diagram in the ambient category Manifold instead, we get a simplicial set whose $C_k$ are manifolds, and whose maps in between are smooth maps. Such a simplicial set is called a *simplicial manifold*—which must not be confused with a manifold discretized into a simplicial complex.

Note that we did not need to separately include globular shaped higher cells. This is because the role played by globular shaped $k$-cells can be effectively covered by simplicial shaped $k$-cells. For instance for $k = 2$:

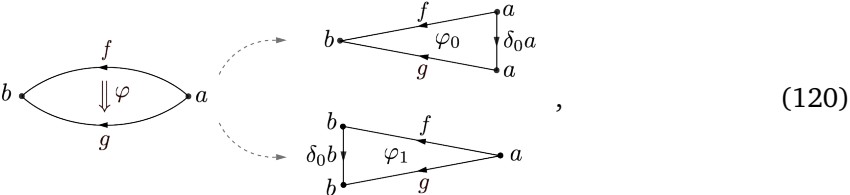

$$
\tag{120}
$$

and likewise for higher $k$'s.

Sometimes we may want to impose some additional conditions on a simplicial set so that it becomes more like a usual category. Consider such a "horn",

$$
\tag{121}
$$

which means two 1-cells are about to compose. In a usual category, there is a unique composition $[g \circ f]$. We motivated by saying we want more possibilities. But the definition of simplicial set also allows the scenario where we end up with less possibility—as there might just exist no 2-cell $\psi$ that satisfies $\partial_0 \psi = g, \partial_2 \psi = f$. Sometimes we may want to avoid such scenario. For many of our applications, we will impose the *Kan condition* that,

- For any *horn* formed by $k$-many $(k-1)$-cells that looks like $k$ out of the $k+1$ faces of a $k$-cell, there indeed exists at least one $k$-cell that takes them as $k$ out of its $k+1$ faces. The $k$-cell, possibly non-unique, can be viewed as one way of composing these $(k-1)$-cells, and the result of this particular composition is the remaining face.

Such a simplicial set is called a *Kan complex* [41]. It is a simplicial version of higher groupoid where all cells are "invertible", because the Kan condition does not distinguish between "source" and "target"—given a $k$-cell, we can view any $k$ out of its $k+1$ faces as sources, and the remaining one face as target.[111] In our application we will also consider Kan complexes internalized in Manifold, sometimes called *Kan simplicial manifolds*.

We may as well work with $n$-simplicial sets, which means we only have $C_k$ up to $k \leq n$. Given an $n$-simplicial set, there are two natural ways to form an $m$-simplicial set for $m > n$ (or a simplicial set, as $m \to \infty$):

1. We may form $C_k$ for $n < k \leq m$ such that its elements are all degenerate $k$-cells. Such an $m$-simplicial set is said to be *n-skeletal*. (Note: It is very unfortunate that the term "skeletal" has two unrelated meanings, one is introduced in Section 5.1 that a category is "skeletal" if any two objects related by an invertible morphism must be the same object, and the other is the notion of "$n$-skeletal" introduced here. Which meaning is being used needs to be distinguished through the context. Fortunately, in this paper we will not really use the notion of "$n$-skeletal".)

2. Alternatively, we may form $C_k$ for $n < k \leq m$ such that, for any set of $(k+1)$-many $(k-1)$-cells that share $(k-2)$-faces in such a manner as if they are the faces of some $k$-cell, there will be one and exactly one $k$-cell that takes them as its faces. Such an $m$-simplicial set is said to be *n-coskeletal*.

We will immediately see that the notion of coskeletal is very useful.

We said simplicial sets provide a very powerful model of weak categories, so we should be able to encompass the previously introduced categories from Section 5.1 to 5.3 by the notion of simplicial set. This procedure is known as taking the *nerve* of a category. The nerve of a category $C$ is commonly denoted as $\mathcal{N}(C)$, but in this paper, by a slight abuse of notation, we will often use just $C$ to denote the nerve. Given a strict $n$-category, to form the nerve,

– the 0- and 1-cells of the nerve are the objects and 1-morphisms of the category;

– the 2-cells of the nerve are those of the form (115), and more generally, for $2 \leq k \leq n+1$, the $k$-cells of the nerve are essentially higher versions of (115), made of the composition rules of the $(k-1)$-morphisms along with the $k$-morphisms;

– finally, for $k > n+1$, the $k$-cells are introduced by demanding the nerve to be $(n+1)$-coskeletal.

(As a consistency check, if we view an $n$-category as an $m$-category for $m > n$ with only identity $q$-morphisms for $n < q \leq m$, the nerve formed is the same.) Let us take 1-category as example: the 2-cells specify the composition rules of the 1-morphisms; the 3- and higher cells express the associativity of the composition of multiple 1-morphisms—the 2-coskeletal condition means that, the composition of three or more 1-morphisms is uniquely defined once the composition of two is defined. Note the nerve of a set $X$ is 0-skeletal and 1-coskeletal, the nerve of a pair groupoid $EX$ (a 1-category) is 0-coskeletal instead of merely 2-coskeletal, and the nerve of $BEG$ (a 2-category) is 1-coskeletal instead of merely 3-coskeletal. The nerve of globular weak (as opposed to strict) higher categories such as bicategories and tricategories

---

[111]If we impose other less stringent conditions in replacement of the Kan condition, we get more general kinds of simplicially modeled weak categories. One example is *quasi-category*: In the Kan condition, we consider any horn that appears as $k$ out of the $k+1$ faces of an anticipated $k$-cell (so there are $(k+1)$ possible horns), and require at least one such anticipated $k$-cell exists. For quasi-category, we distinguish outer and inner horns, where an outer horn has either the $\partial_0$ or the $\partial_k$ face removed, and an inner horn has any one $\partial_i$ ($i \neq 0, k$) face removed; we then only require the anticipated $k$-cell to exist for inner horns, but not necessarily for outer horns. The intuition is that the 1-cell labeled by $(0k)$ must be part of the target.

can also be defined with extra technical details [110–112] along the lines of (116). Note that the nerve of a (possibly weak) $n$-groupoid is a Kan complex, as expected.

We also need to introduce maps between simplicial sets. The analogue of an ordinary functor between simplicial sets is obviously a *simplicial map* such that each $C_k$ is mapped to $D_k$ with the face maps and degeneracy maps preserved. On the other hand, when internalized in Manifold or Difflg, we should use the simplicial version of anafunctor to map between simplicial manifolds—which is the same as (102) except the columns become simplicial sets of the form $(\cdots \rightrightarrows\!\!\rightarrow C_2 \rightrightarrows\!\!\rightarrow C_1 \rightrightarrows C_0)$, the horizontal arrows become simplicial maps, and the condition of "given the source and target" in the notions of "full" and "faithful" becomes "given the faces". We will see examples in our main construction in the next subsection.

Reviewing the categorical notions we have introduced so far, now it appears that the simplicially modeled weak categories are weakened to such an extent that they become conceptually simple to understand again. The strict categories and strict functors in Section 5.1 were easy to understand because the rules are all dictated by strict equality signs. As we began to involve lax functors, anafunctors, bicategories and tricategories, the rules became a little harder to follow, because there are some weakened rules as well as some strict equalities. But now, as we further weaken the rules and arrive at simplicial set, all the rules are essentially of the form "given the $(k-1)$-cells around, which $k$-cells are we allowed to fill-in", much like some kind of tiling board game, so the concept becomes simple to comprehend again. Different simplicial sets just have different details in the "rules of the game".

And it is not hard to sense that being "simplicial" is not of crucial importance here. We can as well consider "cubical sets/manifolds", whose definition is obvious from the name. A simplicial set and a cubical set can be equivalent in a suitable sense. (In fact, the notion of Kan complex was first defined in the cubical rather than simplicial setting.)

Let us now review the previously discussed categories in the purview of the more general notion of simplicial/cubical sets/manifolds.

1. All the target categories of those previously known lattice models, discussed in Section 5.1 to 5.3, can be described by Kan complexes by taking the nerves of those categories.

2. For a manifold $\mathcal{M}$, the collection $\Delta_m\mathcal{M}$ of singular simplicial $m$-cells (those which we use as the basis for singular $m$-chain in singular homology) for all $m$ form a Kan complex $S\mathcal{M}$ internal to Difflg, with $S\mathcal{M}_m = \Delta_m\mathcal{M}$. This is more powerful than the strict path groupoid $\bar{P}_{n\to\infty}\mathcal{M}$, in that $S\mathcal{M}$ captures all the homotopy information of $\mathcal{M}$; in fact, $S\mathcal{M}$ is one way to realize the notion of the (fully fledged rather than strict) fundamental groupoid $\Pi_{n\to\infty}\mathcal{M}$ [41] (see brief discussion at the end of the subsection). Likewise we can consider singular cubical $m$-cells.

3. A lattice that takes the form of a simplicial complex with a given branching structure[112] is naturally a simplicial set, with $C_m$ the set of all $m$-dimensional simplices, including degenerate ones in the sense of (118). Likewise for a cubical lattice as a cubical set. We will denote this simplicial/cubical set as $\mathcal{L}$. Note that $\mathcal{L}$ is not the nerve of the strict $d$-category $\bar{\mathcal{L}}$ that we introduced before. In particular, $\mathcal{L}$ captures the full homotopy $d$-type information of the manifold (what this means will be briefly mentioned at the end of this subsection) that the lattice is discretizing, while the strict $\bar{\mathcal{L}}$ does not.

   An interesting subtlety is noteworthy. $\mathcal{L}$ as defined above is *not* a Kan complex, because two consecutive links might not be two edges of any plaquette. As a consequence, $\mathcal{L}$ does not fully contain the nerve of $\bar{\mathcal{L}}$, as the later is a Kan complex—any two consecutive links can be composed to a lattice path in $\bar{\mathcal{L}}$ made of two links. Likewise for higher dimensional cells. Of course, we can enlarge $\mathcal{L}$ into a Kan complex that contains $\bar{\mathcal{L}}$, by

---

[112]I.e. an ordering of vertices, which is needed when defining cup and higher cup product.

including into $\mathcal{L}_1$ the lattice paths with more than one link, and into $\mathcal{L}_2$ those degenerate triangles representing the composition of paths, and so on.[113] But it turns out that we do not want such enlargement when we describe a lattice field configuration, i.e. a simplicial map from $\mathcal{L}$ to some target Kan simplicial manifold to be introduced below. Indeed, recall the explicit description in Section 4, the d.o.f. in the path integral only live on the actual lattice cells in $\mathcal{L}$; no extra d.o.f. lives on the concatenation of several cells. We want to understand the mathematical root of this subtlety in the future.

Before we move on, we can finally introduce the procedure of *geometric realization*, which has been mention before. Given a simplicial set $C$ (or a category $C$, and then take its nerve), we can construct such a simplicial complex that each $m$-dimensional simplex is labeled by a non-degenerate element of $C_m$, and these simplices are geometrically glued together according to the face maps of $C$ to form a simplicial complex and hence a topological space $|C|$, the geometric realization of $C$, which is in general infinite dimensional. If $C$ is a simplicial manifold to begin with, then the manifold structure on $C_m$ is also inherited onto the set of $m$-dimensional simplices, on top of the manifold structure of each simplex— that is, the set $m$-dimensional simplices, before any gluing, forms a space with topology $(C_m\backslash\{\text{degenerate elements}\}) \times (\text{one } m\text{-dimensional simplex}).$[114]

We should remark that most often, the space constructed out of geometric realization is understood up to homotopy equivalence. A familiar example is $|B\mathbb{Z}|$, whose construction through this procedure is infinite dimensional, but it is homotopic to a circle, so we may as well take $|B\mathbb{Z}| = S^1$. In most cases there are only infinite dimensional realizations.

One may note that the procedure of taking the geometric realization of a simplicial set/manifold and the procedure of taking the singular simplicial set of a topological space seem to be some kind of inverse of each other. The former is a functor from the category of simplicial sets to the category of topological spaces, while the later is a functor from the category of topological spaces to the category of simplicial sets. In fact, these two functors are not inverses of each other; rather, the latter/former functor is a *right/left adjoint* functor to the former/latter, which is an important generalization of the notion of inverse. This means given any simplicial set $\Sigma$ and any topological space $X$, the hom-set of continuous functions from $|\Sigma|$ to $X$ is in one-to-one correspondence with the hom-set of simplicial maps from $\Sigma$ to $SX$—a fact that is intuitive to see. (It is also easy to see that, however, a continuous function from $X$ to $|\Sigma|$ in general has no corresponding simplicial map from $SX$ to $\Sigma$, hence taking geometrical realization and taking singular simplicial set are indeed not inverses of each other, and right vs left adjoints must be distinguished.) This pair of adjoint functors has an important additional property that they establish a *Quillen equivalence*, which gives a precise definition of what it means that "the homotopy theory of topological spaces is fully captured by studying simplicial sets"; likewise for cubical sets. In this paper we will have this idea in mind but we will not further delve into it.

## 5.5 Topological refinement as anafunctor

Now we are ready to tackle our main problem. Everything discussed below will be internalized in Manifold, or Difflg when necessary. Let us think more closely what we really want when we say we want to define the skyrmion that counts $\pi_3 \cong \mathbb{Z}$. The continuum expression (55) of the skyrmion density in fact represents the generator of $H^3(\mathcal{T};\mathbb{Z})$.[115] In the case of $\mathcal{T} = S^3$, this is

---

[113]This is to enlarge $\mathcal{L}$ by pushing out $\mathcal{L} \hookleftarrow \mathcal{L} \cap \bar{\mathcal{L}} \hookrightarrow \bar{\mathcal{L}}$ in the category of simplicial/cubical sets.

[114]One may also drop the "non-degenerate" condition in $C_m$. We are including this condition because we want $|\mathcal{M}| = \mathcal{M}$. If we drop this condition, then the $|\mathcal{M}|$ constructed will be infinite dimensional, though homotopic to $\mathcal{M}$ (see next paragraph for related discussions), for instance $|*|$ will be constructed as $S^\infty$.

[115]More precisely, as mentioned in footnote 42, the continuum WZW curving and the transition functions introduced from (55) to (58) form the generator of the *Deligne-Beilinson double cohomology* $H^2_{\mathrm{DB}}(\mathcal{T};U(1))$, where the

related to $\pi_3$ via the universal coefficient theorem $H^3(S^3;\mathbb{Z}) \xrightarrow{\sim} \mathrm{Hom}(H_3(S^3;\mathbb{Z});\mathbb{Z})$ along with the Hurewicz theorem $H_3(S^3;\mathbb{Z}) \xleftarrow{\sim} \pi_3(S^3)$. Now let us understand the topological classes from the perspective of anafunctor.

Along the way, we will also make a remarkable observation: While our "topological refinement" is studied primarily for taking care of the continuity/smoothness of continuous-valued d.o.f., we will see that when applied to finite groups, our approach will recover the celebrated group cohomology models familiar in topological orders [19, 23]. This strongly suggests that there should be some unifying theme to be uncovered.

We begin with the simple case $H^1(S^1;\mathbb{Z}) \cong \mathbb{Z}$. From (100) in Section 5.2 we have seen that it is natural to represent the basic $\mathbb{Z}$ bundle over $S^1$ as an anafunctor from $S^1$ to $B\mathbb{Z}$:

$$
\begin{array}{ccccc}
\mathbb{Z} & \longleftarrow & \mathbb{R} \times_{S^1} \mathbb{R} & \xrightarrow{\text{f.f.}} & S^1 \\
\Downarrow & & \Downarrow & & \Downarrow \\
* & \longleftarrow & \mathbb{R} & \xrightarrow{\text{surj}} & S^1
\end{array}
\;,
\tag{122}
$$

where $\mathbb{R}$ maps to $S^1$ by modding out $2\pi\mathbb{Z}$, and $\mathbb{R} \times_{S^1} \mathbb{R} \cong \mathbb{R} \times \mathbb{Z}$ with the $\mathbb{Z}$ here mapping to that in the left column identically. Anafunctors from a manifold $\mathcal{T}$ to $B\mathbb{Z}$ are classified by $H^1(\mathcal{T};\mathbb{Z})$—here "classify" means up to ananatural isomorphism between anafunctors—which agrees with the familiar classification of $\mathbb{Z}$ bundles.[116] And the particular anafunctor (122) realizes the generator of the classification $H^1(S^1;\mathbb{Z}) \cong \mathbb{Z}$.

This looks almost like the Villainzation process. From the Villain model we can observe that the Villainization process is described by the following 2-anafunctor (we may often omit the "2-" or "higher" in the below) from $ES^1$ to $B^2\mathbb{Z}$:

$$
\begin{array}{ccccc}
\mathbb{Z} & \longleftarrow & S^1 \times (\mathbb{R} \times_{S^1} \mathbb{R}) & \xrightarrow{\text{f.f.}} & S^1 \times S^1 \\
\Downarrow & & \Downarrow & & \Downarrow \\
* & \longleftarrow & S^1 \times \mathbb{R} & \xrightarrow{\text{full}} & S^1 \times S^1 \\
\Downarrow & & \Downarrow & & \Downarrow \\
* & \longleftarrow & S^1 & \xrightarrow{=} & S^1
\end{array}
\;.
\tag{123}
$$

The right column $ES^1$, as we have seen in Section 5.1, is nothing but the target category used in traditional lattice $S^1$ nl$\sigma$m. The left column $B^2\mathbb{Z}$ is the category characterizing the topological defects that we want to describe—the vortices on the plaquettes. The span, i.e. the middle column, as we have seen in Section 5.1, is the desired target category to be used for the Villain model. (Also, as discussed below (103), this target category, i.e. the middle column in (123), has an ananatural equivalence to $BE\mathbb{Z}$. This $BE\mathbb{Z}$ then maps to the left column $B^2\mathbb{Z}$ by picking up the holonomies in the 2-morphisms.)

How is the Villainization anafunctor (123) related to the anafunctor (122) that realizes the generator of the classification $H^1(S^1;\mathbb{Z}) \cong \mathbb{Z}$? In this example, roughly speaking, we can spot that (122) is "part of" (123) if in (123) we ignore an $S^1$ in the right and middle columns and ignore the bottom row. In the lattice model this corresponds to fixing the $S^1$ d.o.f. on one vertex and looking at the (traditional or Villainized) d.o.f. on the lattice paths and surfaces attached to this vertex—it is familiar to do so if the model has an exact $U(1)$ global symmetry over $S^1$ (because we can indeed use the symmetry to fix the $S^1$ d.o.f. on some given vertex without changing the weight), although we are focusing on topological configuration here

---

double cochain has a de Rham exterior derivative coboundary operator and a Čech transition function coboundary operator. $H^2_{\mathrm{DB}}(\mathcal{T};U(1))$ contains the topological classification information in $H^3(\mathcal{T};\mathbb{Z})$ as well as the flat 2-holonomy information [38]. (The case of $H^1_{\mathrm{DB}}(X;U(1))$ is presented in details in e.g. [65].)

[116]Roughly speaking, the 1-cocycle condition corresponds to the fact that $B\mathbb{Z}$ only has identity 2-morphisms, and 1-coboundaries corresponds to ananatural isomorphisms.

which does not really rely on having this symmetry. Let us now develop an understanding that formalizes this idea, although this understanding will need some important modification when applied to more general Villainizations later.

The idea above is that we deloop (122), which means each column is being delooped, and the functors between columns are consistently carried through. We get an anafunctor from $BU(1)$ to $B^2\mathbb{Z}$:

$$
\begin{array}{ccccc}
\mathbb{Z} & \longleftarrow & \mathbb{R} \times_{S^1} \mathbb{R} & \xrightarrow{\text{f.f.}} & U(1) \\
\Downarrow & & \Downarrow & & \Downarrow \\
* & \longleftarrow & \mathbb{R} & \xrightarrow{\text{full}} & U(1) \\
\Downarrow & & \Downarrow & & \Downarrow \\
* & \longleftarrow & * & \xrightarrow{=} & *
\end{array}
\quad \cdot
\tag{124}
$$

The classification of anafunctors from $BU(1)$ to $B^2\mathbb{Z}$ is denoted, as a definition, as the cohomology $H^0(BU(1); B^2\mathbb{Z})$ or $H^2(BU(1); \mathbb{Z})$,[117] which in turn is given by the familiar cohomology of the classifying space, $H^0(BU(1); B^2\mathbb{Z}) = H^2(BU(1); \mathbb{Z}) \cong H^2(|BU(1)|; \mathbb{Z}) \cong \mathbb{Z}$. This is isomorphic to $H^1(S^1; \mathbb{Z}) \cong \mathbb{Z}$ (recall the discussion below (122)) via the *transgression map*, $H^2(|BU(1)|; \mathbb{Z}) \xrightarrow{\sim} H^1(S^1; \mathbb{Z})$.[118] Then, given (124), we take product with $S^1$ on the right and middle columns, which maps trivially to the left, to arrive at (123), where $S^1 \times S^1$ is identified with $S^1 \times U(1)$ by mapping $(e^{i\theta}, e^{i\theta'})$ to $(e^{i\theta}, e^{i(\theta'-\theta)})$.

There are, however, some aspects of this idea that are not entirely satisfactory:

- This idea only applies when $\mathcal{T} = |G|$ for some group $G$ (whose $\pi_1$ topological configurations we want to look at), otherwise the first delooping step does not make sense. Associated to this, we knew from the $\mathbb{R}P^2$ example in Section 2.3 that 1-morphisms in the Villain model form $\widetilde{\mathcal{T}}^2/\Gamma$ (see also (103); recall $\Gamma = \pi_1(\mathcal{T})$ and $\widetilde{\mathcal{T}}$ is the universal cover of $\mathcal{T}$) which in general cannot be expressed as $\mathcal{T} \times \widetilde{\mathcal{T}}$, unlike $\mathbb{R}^2/\mathbb{Z} \cong S^1 \times \mathbb{R}$ in our example above.

- $H^1(S^1; \mathbb{Z}) \cong \mathbb{Z}$ does *not* classify anafunctors from $ES^1$ to $B^2\mathbb{Z}$ (to which (123) belongs), as such a classification would have been trivial because any pair groupoid $EX$ is naturally equivalent to the trivial category $*$ (recall from Section 5.1). From the discussions around (124), we can roughly see the key is that the $S^1$ that we finally multiply to (124) to get (123) is irrelevant to the appearance of topological configuration, so at least in the example there, we looked at the isomorphism $H^2(|BU(1)|; \mathbb{Z}) \xrightarrow{\sim} H^1(S^1; \mathbb{Z})$, for which (124) and (122) are the respective generators; there seems no obvious involvement of

---

[117]The general notion of cohomology, first developed in [113], can be sketched as follows. Consider an ambient infinite category (suitably weak in general) in which all 2- and higher morphisms are invertible in a suitable sense (here the ambient category is that of smooth higher groupoids, with anafunctors as 1-morphisms and ananatural transformations as 2-morphisms and so on). Then the cohomology $H^0(X; Y)$ from an object $X$ to another object $Y$ (here $BU(1)$ and $B\mathbb{Z}^2$) is indeed defined as the hom-groupoid of 1-morphisms from $X$ to $Y$, with identification if related by 2-morphisms. When $Y = B^n A$ for some abelian group $A$, we will usually denote $H^0(X; B^n A)$ as $H^n(X; A)$; the $H^0(X; Y)$ notation allows $Y$ to be general.

[118]The transgression map is constructed in the following intuitive way. A singular $n$-cochain $\phi$ on $|BG|$ can be pulled-back to an $n$-cochain $\varphi$ in $|EG|$. Since $|EG|$ is contractible, $H^n(|EG|; A)$ must be trivial, so $\varphi = d\rho$ for some $(n-1)$-cochain $\rho$ in $|EG|$. Restricting $\rho$ to some fibre $F \cong |G|$, we have some $(n-1)$-cochain $\varrho$ in $|G|$ that satisfies $d\varrho = 0$, since $d\rho = \varphi$ becomes a trivial $n$-cochain when restricted to a fibre $F$. Thus $\varrho$ defines a class in $H^{n-1}(|G|; A)$.

The transgression map with coefficient $A = \mathbb{Z}$ can be refined to a map in the Deligne-Beilinson double cohomology from $H^{n-1}_{\text{DB}}(|BG|; U(1))$ to $H^{n-2}_{\text{DB}}(|G|; U(1))$ [38]. The interpretation of this map is familiar in physics: For instance, when $G = U(1), n = 2$, this map is an isomorphism, and it says that gauge covariance requires the ends of a $U(1)$ Wilson line of charge $w \in \mathbb{Z}$ to be particles of charges $\pm w$; for $G = SU(N), n = 4$, this maps is also an isomorphism, that says gauge covariance requires the boundary of a level-$k$ CS term to be a level-$k$ WZW. This later point is related to the discussion below (58).

$ES^1$. But how should we formalize this idea more generally so that it is applicable when $\mathcal{T} \neq |G|$?

Thus we are coming back to the important conceptual problem discussed in Section 5.1: Why we in general do not just treat $E\mathcal{T}$ as trivial. We gave the physical explanation there. We now discuss the more formal aspect of it, which will in turn allow us to generalize the relation between (122) and (123) to generic Villainization procedures. This will lead us to the understanding of topology configurations of continuous-valued fields in terms of *relative cohomology*.

Let us think of not just $E\mathcal{T}$, but the inclusion functor $\mathcal{T} \hookrightarrow E\mathcal{T}$, where the objects map identically, and the identity morphisms in $\mathcal{T}$ map to the identity morphisms in $E\mathcal{T}$. Physically, this means we have a well-defined notion of when the two d.o.f. across a lattice link "take the same value". (And the physical intuition for the lattice nl$\sigma$m is that the link weight will be maximized when this happens.) In this purview, let us look back at the ananatural equivalence between $E\mathcal{T}$ and $*$, equipping the latter with the trivial functor $\mathcal{T} \rightarrow *$. While any pair of functors between $*$ and $E\mathcal{T}$ establish their (ana)natural equivalence, if we impose the additional requirement that the equipped $\mathcal{T} \rightarrow *$ and $\mathcal{T} \hookrightarrow E\mathcal{T}$ must be preserved, then we can see there is no functor from $*$ to $E\mathcal{T}$ that respects this requirement (unless $\mathcal{T}$ itself is a single point). Related to this, while any functor from $E\mathcal{T}$ to $*$ preserves the equipped functor from $\mathcal{T}$ and is surjective and fully faithful, if we carefully inspect the bijection in the definition of "fully faithful", we find the map from $*_1|_{*,*} = \{\mathbf{1}_*\}$ to $E\mathcal{T}_1|_{b,a} = \{(b,a)\}$ for $a \neq b \in \mathcal{T}$ does not preserve the image of any identity 1-morphism in $\mathcal{T}$. Therefore, we conclude that *while $E\mathcal{T}$ and $*$ are equivalent as categories, they are inequivalent as categories equipped with the specified functors from $\mathcal{T}$*. We call categories with such specified functors from $\mathcal{T}$ "categories under $\mathcal{T}$".[119]

Now with this perspective in mind, we shall think of (123) as an "anafunctor under $\mathcal{T} = S^1$", which means:

– Each column of (123) is implicitly equipped with an obvious functor from $\mathcal{T} = S^1$, which maps identically to the objects of the right and the middle columns, and maps trivially to the left column.

Crucially, the left column describes the topological defect of interest, and equipping a trivial map from $\mathcal{T}$ to the left column intuitively means, in the lattice nl$\sigma$m, when all the vertex d.o.f. on the same connect component of the lattice take the same value, i.e. when the field configuration is completely ordered, there obviously should not be any topological defect.

– The equipped functors from $\mathcal{T} = S^1$ are being preserved along the horizontal functors in (123); moreover, any notion of "surjection" and "bijection" in the definition of anafunctor are also $\mathcal{T}$-preserving.

Anafunctors as such are classified, up to ananatural isomorphisms under similar $\mathcal{T}$-preserving conditions, by the *relative cohomology* $H^2(ES^1, S^1; \mathbb{Z}) \cong H^2(|ES^1|, S^1; \mathbb{Z})$. Since $ES^1$ is trivial (or say $|ES^1|$ is contractible), by the long exact sequence of relative cohomology

$$\cdots \rightarrow H^n(X,Y;A) \rightarrow H^n(X;A) \rightarrow H^n(Y;A) \rightarrow H^{n+1}(X,Y;A) \rightarrow H^{n+1}(X;A) \rightarrow \cdots, \quad (125)$$

we have an isomorphism $H^2(|ES^1|, S^1; \mathbb{Z}) \xleftarrow{\sim} H^1(S^1; Z) \cong \mathbb{Z}$, hence explaining the relation of (123) to (122), and in particular (123) realizes the generator of this classification, which is

---

[119]In general, given a category $C$, the "under category" $c/C$ (for some specified $c \in C_0$) is made of objects $(c/C)_0 := \{f \in C_1 | \mathsf{s}(f) = c\}$ and morphisms $(c/C)_1|_{f',f} := \{g \in C_1 | g \circ f = f'\}$ ("over category" $C/c$ is if we replace $\mathsf{s}$ with $\mathsf{t}$ and replace $g \circ f = f'$ with $f' \circ g = f$). This can be generalized to higher categories. Here, $\mathcal{T} \rightarrow *$ and $\mathcal{T} \hookrightarrow E\mathcal{T}$ ("categories under $\mathcal{T}$", or more precisely "Lie groupoids under $\mathcal{T}$") are objects in the 2-category $\mathcal{T}/\mathsf{LieGroupoids}$, and there is no 1-morphism ($\mathcal{T}$-preserving anafunctor) from the former to the latter.

the canonical class—it captures the intuition that "going around the $S^1$ once will indeed give an increment by 1 in $\mathbb{Z}$".

Thus, we extracted from the Villain model a main proposal of this paper: *The target category of a "topologically refined lattice theory" is the span of an anafunctor, that maps from the target category of the traditional lattice model, to the category that characterizes the topological defects that we want to describe. Anafunctors as such are classified by a suitable relative cohomology, and the desired anafunctor realizes a canonical class of this relative cohomology.*

Let us see how this applies to general Villain models. Recall $\widetilde{\mathcal{T}}$ denotes the universal cover of $\mathcal{T} = \widetilde{\mathcal{T}}/\Gamma$, with $\Gamma$ discrete. This can be described by an anafunctor from $\mathcal{T}$ to $B\Gamma$:

$$
\begin{array}{ccccc}
\Gamma & \longleftarrow & \widetilde{\mathcal{T}} \times_{\mathcal{T}} \widetilde{\mathcal{T}} & \overset{\text{f. f.}}{\longrightarrow} & \mathcal{T} \\
\big\Downarrow & & \big\Downarrow & & \big\Downarrow \\
* & \longleftarrow & \widetilde{\mathcal{T}} & \overset{\text{surj}}{\longrightarrow} & \mathcal{T}
\end{array} \quad , \tag{126}
$$

where, recall, $\widetilde{\mathcal{T}} \times_{\mathcal{T}} \widetilde{\mathcal{T}} \cong \widetilde{\mathcal{T}} \times \Gamma$. If $\Gamma$ is discrete and abelian, the Villainization procedure is described by such an anafunctor under $\mathcal{T}$

$$
\begin{array}{ccccc}
\Gamma & \longleftarrow & (\widetilde{\mathcal{T}}^2/\Gamma) \times \Gamma & \overset{\text{f. f.}}{\longrightarrow} & \mathcal{T} \times \mathcal{T} \\
\big\Downarrow & & \big\Downarrow & & \big\Downarrow \\
* & \longleftarrow & \widetilde{\mathcal{T}}^2/\Gamma & \overset{\text{full}}{\longrightarrow} & \mathcal{T} \times \mathcal{T} \\
\big\Downarrow & & \big\Downarrow & & \big\Downarrow \\
* & \longleftarrow & \mathcal{T} & \overset{=}{\longrightarrow} & \mathcal{T}
\end{array} \tag{127}
$$

(in general $\widetilde{\mathcal{T}}^2/\Gamma$ cannot be expressed as $\mathcal{T} \times \widetilde{\mathcal{T}}$, even though $\mathcal{T}^2/\Gamma|_{\text{fixing s or t}} \cong \widetilde{\mathcal{T}}$) that realizes the canonical class of $H^2(E\mathcal{T}, \mathcal{T}; \Gamma) \overset{\sim}{\leftarrow} H^1(\mathcal{T}; \Gamma)$. For discrete non-abelian $\Gamma$, the left column that describes the topological defects shall be replaced by the 2-group $\text{InAut}\Gamma \ltimes \Gamma \rightrightarrows \text{InAut}\Gamma \rightrightarrows *$, where the inner automorphism InAut arises because non-abelian holonomies are only well-defined up to conjugation. The cohomology classification will take value in this 2-group instead.[120] These more complicated non-abelian cases have little to do with what we plan to introduce below.

When $\mathcal{T}$ in the above is itself a Lie group $G$, and $\Gamma$ is abelian, we can deloop (127) and obtain the Villainized gauge theory. The classification becomes $H^3(BEG, BG; \Gamma) \overset{\sim}{\leftarrow} H^2(BG; \Gamma) \cong H^2(|BG|; \Gamma)$. If $G$ itself is abelian, we can further deloop the above arbitrarily many times. This completes the discussion of general Villainization.

Now we move on beyond Villainization. Consider $H^2(\mathcal{T}; \mathbb{Z})$; for this case we may keep $\mathcal{T} = S^2$ in mind as the basic example. $H^2(\mathcal{T}; \mathbb{Z})$ classifies the $U(1)$ bundles on $\mathcal{T}$. It is common to represent a $U(1)$ bundle as $U(1) \to \mathcal{E} \to \mathcal{T}$ (including our Sections 2 and 3) where $\mathcal{E}$ is the total space, but this has two disadvantages: the map $U(1) \to \mathcal{E}$ is non-canonical, and moreover it obscures the similarity between a $U(1)$ bundle and a $U(1)$ function. It is more natural to represent a $U(1)$ bundle, again, by the anafunctor (100)

$$
\begin{array}{ccccc}
U(1) & \longleftarrow & \mathcal{E} \times_{\mathcal{T}} \mathcal{E} & \overset{\text{f. f.}}{\longrightarrow} & \mathcal{T} \\
\big\Downarrow & & \big\Downarrow & & \big\Downarrow \\
* & \longleftarrow & \mathcal{E} & \overset{\text{surj}}{\longrightarrow} & \mathcal{T}
\end{array} \quad , \tag{128}
$$

where the map $U(1) \leftarrow \mathcal{E} \times_{\mathcal{T}} \mathcal{E} \cong \mathcal{E} \times U(1)$ represents the $U(1)$ action on the fibres of $\mathcal{E}$ and is therefore canonical, and moreover this is obviously a "higher version" of a $U(1)$ function $U(1) \leftarrow \mathcal{T}$. We can say a $U(1)$ function is a $U(1)$ 0-bundle, while a $U(1)$ bundle is a $U(1)$

---

[120]See footnote 117.

1-bundle. Two anafunctors that appear to be different can describe a same $U(1)$ bundle, with the equivalence established by an invertible ananatural transformation (98), that involves a function $\Phi_1$ (recall (91)) in the natural transformation on the left half of (98):

$$
\begin{array}{c}
BU(1)_1 = U(1) \\
\searrow \\
\qquad\qquad H_0 = \mathcal{E} \times_{\mathcal{T}} \mathcal{E}',
\end{array}
\tag{129}
$$

meaning that two equivalent $U(1)$ bundles only differ by a $U(1)$ function; and $\Phi$ in (98) being a natural transformation means (recall (91)) on $\mathcal{E} \times_{\mathcal{T}} \mathcal{E}' \times_{\mathcal{T}} \mathcal{E} \times_{\mathcal{T}} \mathcal{E}'$, the pullbacks of the four $U(1)$ functions on $\mathcal{E} \times_{\mathcal{T}} \mathcal{E}$, $\mathcal{E}' \times_{\mathcal{T}} \mathcal{E}'$ and two copies of $\mathcal{E} \times_{\mathcal{T}} \mathcal{E}'$ multiply to a trivial (identity) $U(1)$ function. If $\mathcal{E}'$ is also the total space of the bundle, this $\Phi_1$ just familiarly describes a gauge transformation on the total space; but $\mathcal{E}'$ could as well be other choices, such as the patches $\mathcal{U}$ in (100).

To manifest the relation of (128) to $H^2(\mathcal{T}; \mathbb{Z})$, we deloop (122) into (124), and compose it on the left of (128) (according to (97)) to obtain an anafunctor from $\mathcal{T}$ to $B^2\mathbb{Z}$:

$$
\begin{array}{ccccc}
\mathbb{Z} & \longleftarrow & \mathcal{E} \times \mathbb{R} \times \mathbb{Z} & \xrightarrow{\text{f.f.}} & \mathcal{T} \\
\Downarrow & & \Downarrow & & \Downarrow \\
* & \longleftarrow & \mathcal{E} \times \mathbb{R} & \xrightarrow{\text{full}} & \mathcal{T} \\
\Downarrow & & \Downarrow & & \Downarrow \\
* & \longleftarrow & \mathcal{E} & \xrightarrow{\text{surj}} & \mathcal{T}
\end{array}
\tag{130}
$$

Obviously we should call such an anafunctor a $\mathbb{Z}$ 2-bundle over $\mathcal{T}$.

Analogous to the "$\mathcal{T} \hookrightarrow E\mathcal{T}$ step" from (126) to (127), the target category to use for topological refinement would be the span of a suitable anafunctor under $\mathcal{T}$ from $E\mathcal{T}$ to $B^3\mathbb{Z}$:

$$
\begin{array}{ccccc}
\mathbb{Z} & \longleftarrow & (\mathcal{E}^2/U(1)) \times \mathbb{R} \times \mathbb{Z} & \xrightarrow{\text{f.f.}} & \mathcal{T} \times \mathcal{T} \\
\Downarrow & & \Downarrow & & \Downarrow \\
* & \longleftarrow & (\mathcal{E}^2/U(1)) \times \mathbb{R} & \xrightarrow{\text{full}} & \mathcal{T} \times \mathcal{T} \\
\Downarrow & & \Downarrow & & \Downarrow \\
* & \longleftarrow & \mathcal{E}^2/U(1) & \xrightarrow{\text{full}} & \mathcal{T} \times \mathcal{T} \\
\Downarrow & & \Downarrow & & \Downarrow \\
* & \longleftarrow & \mathcal{T} & \xrightarrow{=} & \mathcal{T}
\end{array}
\tag{131}
$$

Such anafunctors are classified by $H^3(E\mathcal{T}, \mathcal{T}; \mathbb{Z}) \xleftarrow{\sim} H^2(\mathcal{T}; \mathbb{Z})$. For our familiar example $\mathcal{T} = S^2$, the $B^3\mathbb{Z}$ on the left represents the hedgehog defects; we have $\mathcal{E} = SU(2)$ and $SU(2)^2/U(1) \cong S^2 \times SU(2)$,[121] and the anafunctor realizes the generator of $H^3(ES^2, S^2; \mathbb{Z}) \xleftarrow{\sim} H^2(S^2; \mathbb{Z}) \cong \mathbb{Z}$. (Also, as discussed below (104), the target category of the refined nl$\sigma$m, i.e. the middle column above, has an ananatural equivalence to $B^2 E\mathbb{Z}$, which in turn maps to the left column $B^3\mathbb{Z}$ by picking up the holonomies in the 3-morphisms.)

We can also arrive at (131) from (128) via an interchanged order of steps. From (128) we can first perform the "$\mathcal{T} \hookrightarrow E\mathcal{T}$ step" from (126) to (127) but with the discrete $\Gamma$ replaced

---

[121]Where $(\mathcal{U}', \mathcal{U}) \in SU(2) \times SU(2)$ is mapped to $(R_{\mathcal{U}}\hat{z}, \mathcal{U}'\mathcal{U}^{-1}) \in S^2 \times SU(2)$, which is invariant under $\mathcal{U}'e^{i\psi\sigma^z}, \mathcal{U}e^{i\psi\sigma^z}$ for $e^{i\psi} \in U(1)$.

with $U(1)$, arriving at

$$
\begin{array}{ccccc}
U(1) & \longleftarrow & (\mathcal{E}^2/U(1)) \times U(1) & \xrightarrow{\text{f.f.}} & \mathcal{T} \times \mathcal{T} \\
\Downarrow & & \Downarrow & & \Downarrow \\
* & \longleftarrow & \mathcal{E}^2/U(1) & \xrightarrow{\text{full}} & \mathcal{T} \times \mathcal{T} \\
\Downarrow & & \Downarrow & & \Downarrow \\
* & \longleftarrow & \mathcal{T} & \xrightarrow{=} & \mathcal{T}
\end{array}
\tag{132}
$$

which corresponds to the first step of spinon decomposition in Section 2.4, and the $B^2 U(1)$ describes the $U(1)$ Berry curvature on plaquettes. Then we compose on its left the twice delooping of (122), which corresponds to the second step that Villainizes the Berry curvature in Section 2.4, to arrive at (131). This order of steps, i.e. going from (128) via (132) to arrive at (131), is closer to how we think about the physical lattice model, compared to the other order of going from (128) via (130) to arrive at (131).

After these discussions, it becomes obvious that if we have a nl$\sigma$m with $\mathcal{T} = S^3$ or $\mathcal{T} = |SU(N)|$ and want to capture the physics due to $H^3(\mathcal{T}; \mathbb{Z})$, we shall begin with a $U(1)$ 2-bundle on $\mathcal{T}$, i.e. an anafunctor from $\mathcal{T}$ to $B^2 U(1)$, classified up to ananatural isomorphism (see (134) below) by $H^3(\mathcal{T}; \mathbb{Z})$:

$$
\begin{array}{ccccc}
U(1) & \longleftarrow & L \times_{Y \times_{\mathcal{T}} Y} L & \xrightarrow{\text{f.f.}} & \mathcal{T} \\
\Downarrow & & \Downarrow & & \Downarrow \\
* & \longleftarrow & L & \xrightarrow{\text{full}} & \mathcal{T} \\
\Downarrow & & \Downarrow & & \Downarrow \\
* & \longleftarrow & Y & \xrightarrow{\text{surj}} & \mathcal{T}
\end{array}
\tag{133}
$$

In particular, we will let $Y$ be a surjective submersion covering $\mathcal{T}$, so that $Y \times_{\mathcal{T}} Y$ exists as a manifold, and $L$ is the total space of a $U(1)$ bundle over $Y \times_{\mathcal{T}} Y$, and thus $U(1) \leftarrow L \times_{Y \times_{\mathcal{T}} Y} L \cong L \times U(1)$ is the $U(1)$ action on the fibres of $L$. Such a construction of $U(1)$ 2-bundle over $\mathcal{T}$ is called a *bundle gerbe* over $\mathcal{T}$ [39] (with basic idea from [114]); the Lie groupoid part $L \rightrightarrows Y$ in the span is the analogue of the total space $\mathcal{E}$ in a $U(1)$ 1-bundle (128), except it is no longer a single space. (Just like a $U(1)$ bundle can be presented as $U(1) \to \mathcal{E} \to \mathcal{T}$ but with the map from $U(1)$ to $\mathcal{E}$ non-canonical, a $U(1)$ 2-bundle can also be presented in a similar way, except each entry becomes a Lie groupoid, i.e. there is a non-canonical anafunctor from $BU(1)$ to $L \rightrightarrows Y$ and then a projection from $L \rightrightarrows Y$ to $\mathcal{T}$.)

Just like the equivalence of two 1-bundles (128) is established by an ananatural isomorphism (129), the same is true for 2-bundles: To establish the equivalence of two 2-bundles (133), we use an 2-ananatural isomorphism that involves an anafunctor (instead of functor, in general)

$$
\begin{array}{ccccc}
B^2 U(1)_2 = U(1) & & & & \\
\Downarrow & \searrow & & & \\
B^2 U(1)_1 = * & \searrow \quad (\Sigma \times_{\mathcal{T}} \Sigma) \times_{Y \times_{\mathcal{T}} Y' \times_{\mathcal{T}} Y \times_{\mathcal{T}} Y'} (L \times_{\mathcal{T}} L') & \xrightarrow{\text{f.f.}} & H_1 = L \times_{\mathcal{T}} L' \\
& \searrow \qquad \Downarrow & & \Downarrow \\
& \Sigma & \xrightarrow{\text{surj}} & H_0 = Y \times_{\mathcal{T}} Y' ,
\end{array}
\tag{134}
$$

which means two equivalent $U(1)$ 2-bundles only differ by a $U(1)$ 1-bundle $\Sigma$ over $Y \times_{\mathcal{T}} Y'$, and moreover two $\Sigma$'s along with $L$ and $L'$ can together "piece up" to a trivialized $U(1)$ bundle over $Y \times_{\mathcal{T}} Y' \times_{\mathcal{T}} Y \times_{\mathcal{T}} Y'$. (By "piece up" we mean the two $\Sigma$'s and $L$ and $L'$ each pulls-back to a $U(1)$ bundle over $Y \times_{\mathcal{T}} Y' \times_{\mathcal{T}} Y \times_{\mathcal{T}} Y'$, and then we take the tensor product of these four

$U(1)$ bundles into one $U(1)$ bundle. The same when we say "piece up" of $U(1)$ bundles in the below.) We can illustrate this as

$$
\begin{array}{ccc}
y_1' \in Y' \bullet & \xleftarrow{\;\sigma_1 \in \Sigma\;} & \bullet\, y_1 \in Y \\
\ell' \in L' \downarrow & & \downarrow\, \ell \in L \\
y_2' \in Y' \bullet & \xleftarrow{\;\sigma_2 \in \Sigma\;} & \bullet\, y_2 \in Y
\end{array}
\tag{135}
$$

where $y_1, y_2, y_1', y_2'$ all project to a same element in $\mathcal{T}$, and a $U(1)$ value specifying the trivialization (the upper left of (134)) is assigned to the quadrangle as a function of $\sigma_1, \sigma_2, \ell, \ell'$. This anafunctor (134) in the 2-ananatural transformation is often called a *stable isomorphism* between the two bundle gerbes [115]. While this general notion of stable isomorphism might look a little complicated, in certain cases it can be greatly simplified: When $Y$ has an embedding $Y \hookrightarrow Y'$ that preserves the projection to $\mathcal{T}$, and moreover $L$ is (or is equivalent as a $U(1)$ bundle to) the pullback of $L'$ along $Y \times_{\mathcal{T}} Y \hookrightarrow Y' \times_{\mathcal{T}} Y'$, then these two bundle gerbes are automatically equivalent, with $\Sigma$ given by the pullback of $L'$ along $Y \times_{\mathcal{T}} Y' \hookrightarrow Y' \times_{\mathcal{T}} Y'$.[122] We will use this in our main construction below, (142) that constructs the $\Lambda$ in (140).

We should make some comments about the cover $Y$. Originally, in [39] $Y$ was required to be a fibre bundle over $\mathcal{T}$, and with such restriction it can be proven [39] that, in order for the principal 2-bundle to be non-trivial in $H^3(\mathcal{T}; \mathbb{Z})$,[123] $Y$ must be infinite dimensional (hence internalized in Difflg rather than Manifold). One such example is the *tautological bundle gerbe* [39], with $Y_{taut} = \bar{\mathcal{P}}_* \mathcal{T}$, $Y_{taut} \times_{\mathcal{T}} Y_{taut} \cong \bar{\mathcal{P}}_* \mathcal{T} \times \bar{\Omega}_* \mathcal{T}$, and $L_{taut} \cong \bar{\mathcal{P}}_* \mathcal{T} \times \left( \bar{\mathcal{P}}_* \bar{\Omega}_* \mathcal{T} \times U(1) / WZW \right)$ in the sense of (54), which, when $\mathcal{T} = |SU(N)|$, realizes the generator of $H^3(\mathcal{T}; \mathbb{Z})$.[124] (We

---

[122]Let us demonstrate this idea in the simpler context of ananatural isomorphism (129) for $U(1)$ bundles, instead of the more involved stable isomorphism (134) for $U(1)$ 2-bundles (bundle gerbes).

Consider the familiar canonical $U(1)$ bundle over $S^2$. We can present the same $U(1)$ bundle using three different but ananaturally isomorphic anafunctors (128): with the space of objects of the span being the total space $\mathcal{E} = SU(2)$ (with $SU(2) \times_{S^2} SU(2) \cong SU(2) \times U(1)$ specifying the $U(1)$ action on the fibre), or being the patches $\mathcal{U} = (S^2 \backslash \{-\hat{z}\}) \sqcup (S^2 \backslash \{+\hat{z}\})$ (with a $U(1)$ transition function on $\mathcal{U} \times_{S^2} \mathcal{U}$ with winding number 1), or being the pointed path space $\bar{\mathcal{P}}_* S^2$ (with $\bar{\mathcal{P}}_* S^2 \times_{S^2} \bar{\mathcal{P}}_* S^2 \cong \bar{\mathcal{P}}_* S^2 \times \bar{\Omega}_* S^2$ mapping to $U(1)$ by the Berry phase bounded by the loop in $\Omega_* S^2$). The last presentation is the "tautological" one, and for concreteness we can choose the fixed starting point of $\mathcal{P}_*$ to be $* = \hat{z}$.

Let us see how embedding the patches $\mathcal{U} = (S^2 \backslash \{-\hat{z}\}) \sqcup (S^2 \backslash \{+\hat{z}\})$ into $\bar{\mathcal{P}}_{\hat{z}} S^2$ will give the desired ananatural isomorphism between the different anafunctors—the patches description and the path space description—for the same $U(1)$ bundle. For an element in the $(S^2 \backslash \{-\hat{z}\})$ patch, we associate it with the shortest geodesic from $+\hat{z}$ to the target point in $(S^2 \backslash \{-\hat{z}\})$; while for an element in the $(S^2 \backslash \{+\hat{z}\})$ patch, we associate it with a curve that first go from $+\hat{z}$ to $-\hat{z}$ along, say, the $x$-direction (i.e. the $\phi = 0$ longitude), and then along the shortest geodesic from $-\hat{z}$ to the target point in $(S^2 \backslash \{+\hat{z}\})$. The function $\mathcal{U} \times_{S^2} \bar{\mathcal{P}}_{\hat{z}} S^2 \to U(1)$ in the natural isomorphism (129) is then given by the Berry phase bounded by such an embedded curve and an arbitrary curve sharing the same end points. Moreover, the $U(1)$ transition function between the two patches can also be inherited from the Berry phase bounded by two embedded curves, one from an element of each patch—this will make transition function given by the longitude $e^{i\phi}$ of the target point. If we do not want the $x$-direction to be special, then we may replace the $(S^2 \backslash \{+\hat{z}\})$ patch by $(S^2 \backslash \{+\hat{z}\} \times S^1)$, with the $S^1$ specifying the direction (seen from $\hat{z}$) along which the longitude is set to zero—obviously, our interpretation of $Y$ in (62) is a generalization of this to one dimension higher.

For comparison, the ananatural isomorphism between the total space description and the path space description is not established via an embedding of $SU(2)$ into $\bar{\mathcal{P}}_{\hat{z}} S^2$. Note that $SU(2) \times_{S^2} \bar{\mathcal{P}}_{\hat{z}} S^2 \cong \widetilde{\mathcal{P}}_1^\Pi SU(2)$ (where $\Pi : SU(2) \to S^2$ is given by the action on $\hat{z}$, and we fixed the starting point of $\widetilde{\mathcal{P}}^\Pi$ at $\mathbf{1} \in SU(2)$), which maps to $U(1)$ by the integration with the Berry connection—the end point living in $SU(2)$ instead of $S^2$ manifests the gauge dependence at the end point, and the equivalence in the notion of $\widetilde{\mathcal{P}}$ is due to independence of the gauge in the middle of the path. So in the ananatural isomorphism between this pair of anafunctors there is no involvement of any embedding of $SU(2)$ into $\bar{\mathcal{P}}_{\hat{z}} S^2$.

[123]Unwinding the definition of ananatural transformation, non-trivial means the bundle gerbe $(L \rightrightarrows Y)$ is "non-exact", which means it is not stably isomorphic to any $(L' \rightrightarrows Y')$ such that the $U(1)$ bundle $L'$ over $Y' \times_{\mathcal{T}} Y'$ is formed by piecing up (i.e pulling-back and then taking tensor product) two copies of some $U(1)$ bundle $E'$ over $Y'$.

[124]It does not matter whether we take identification under thin homotopy or not. We can as well take $Y_{taut} = \mathcal{P}_* \mathcal{T}$, since any thin homotopy will not affect the WZW evaluation. Or we can as well use $\bar{\mathcal{P}}_* \mathcal{T}$ in (54).

also need to specify the identity map from $Y_{taut}$ to $L_{taut}$. It is the naturally trivial element of $\left(\bar{\mathcal{P}}_*\bar{\Omega}_*\mathcal{T} \times U(1)/WZW\right)$, i.e. given a path in $Y_{taut}$, we take the trivial surface in $\bar{\mathcal{P}}_*\bar{\Omega}_*\mathcal{T}$ sweeping from this path to itself, and at the same time take 1 in the $U(1)$ factor.) It is not hard to see the tautological bundle is closely related to (54); we will discuss their relation after we introduce (139).

Later, $Y$ that are more general surjective submersions covering $\mathcal{T}$ (as opposed to having to be fibre bundles over $\mathcal{T}$) have been considered, and such $Y$ can be finite dimensional even for non-trivial 2-bundle in $H^3(\mathcal{T};\mathbb{Z})$. This explains Section 3. A particularly nice choice of finite dimensional $Y$ for $\mathcal{T} = S^3 \cong |SU(2)|$ is $Y = (SU(2)\backslash\{-\mathbf{1}\}) \sqcup (SU(2)\backslash\{+\mathbf{1}\})$ [95], where each patch is invariant under $SU(2)$ conjugation—and such a bundle gerbe over $|G|$ is said to be "$G$-equivariant". Then $Y \times_{\mathcal{T}} Y$ has four patches, given by $Y \times_{\mathcal{T}} Y = (SU(2)\backslash\{-\mathbf{1}\}) \sqcup (SU(2)\backslash\{\pm\mathbf{1}\}) \sqcup (SU(2)\backslash\{\pm\mathbf{1}\}) \sqcup (SU(2)\backslash\{+\mathbf{1}\})$, and $L$ is the $U(1)$ bundle over it such that, over the $(SU(2)\backslash\{-\mathbf{1}\})$ and $(SU(2)\backslash\{+\mathbf{1}\})$ patches which are topologically trivial, the $U(1)$ bundle is necessarily trivial, while over the $(SU(2)\backslash\{\pm\mathbf{1}\}) \cong S^2 \times [0,1]$ patches, the $U(1)$ fibres form the Hopf fibration over $S^2$.[125,126] (We also need to specify the identity map from $Y$ to $L$, which is choosing a trivialization section over the $(SU(2)\backslash\{-\mathbf{1}\}) \sqcup (SU(2)\backslash\{+\mathbf{1}\})$ part of $Y \times_{\mathcal{T}} Y$.) Clearly this will be related to the $Y$ in the lattice model in Section 4.1. We will see later how the model arises from these mathematical considerations.

This completes our introduction of the notion of $U(1)$ bundle gerbe, i.e. $U(1)$ principal 2-bundle, (133), as the higher analogue of $U(1)$ principal bundle (128). Based on the previous experience from (128) to (132), clearly there are two follow-up steps in order to arrive at the desired topological refinement: the "$\mathcal{T} \hookrightarrow E\mathcal{T}$ step", and the "Villainization step", and the order of these two steps is interchangeable. As we said below (132), first taking the "$\mathcal{T} \hookrightarrow E\mathcal{T}$ step" and then the "Villainization step" is closer to how we think about the physical lattice model. This order of steps also makes closer connection to the existing mathematical literature. So this is the order in which we will proceed now.

It turns out performing the "$\mathcal{T} \hookrightarrow E\mathcal{T}$ step" to the $U(1)$ 2-bundle (133) is crucially more non-trivial than doing the same to the $U(1)$ 1-bundle (128). Doing it to (128) will lead to (132) which is still an anafunctor of strict groupoids, while doing it to (133) will in general necessarily lead to an anafunctor of Kan simplicial manifolds, resulting in (140). This is what we shall explain now.

For simplicity, from here on, we will focus on $\mathcal{T} = |G|$ for connected and simply connected semi-simple Lie group $G$, primarily with $|SU(N)|$ or more specifically $S^3 \cong |SU(2)|$ in mind, unless otherwise specified. This is sufficient for our primary purpose of this paper. The convenience this assumption brings in is that we can then use the idea similar to (124), which is not applicable when $\mathcal{T} \neq |G|$. That is, we can first study the delooping of (133).

The delooping of (133) with $\mathcal{T} = |G|$ turns out to have been well-studied in the literature, known as *multiplicative bundle gerbe* [38]. It is an anafunctor from $BG$ to $B^3U(1)$ but with a

---

[125]The non-trivial bundle part of $L$ can be interpreted as follows [95]. When $g \neq \pm\mathbf{1} \in SU(2)$, the diagonalization $g = \mathcal{U}e^{i\lambda\sigma^z}\mathcal{U}^{-1}$ is non-degenerate, so $\mathcal{U} \in SU(2)$ is well-defined up to a $e^{i\kappa\sigma^z} \in U(1)$ action on the right, parametrizing an $S^2$. The $SU(2) \ni \mathcal{U}$ is the desired $U(1)$ bundle over the $S^2$. (Note the $SU(2) \ni \mathcal{U}$ is not the $SU(2) \ni g$ that we started with.) However, such group theoretic interpretation is no longer applicable when we talk about multiplicative bundle gerbe (136) later, as we move closer to what we need.

[126]The generalization to $\mathcal{T} = |SU(N)|$ for $N > 2$ in terms of the Weyl alcove [95] is briefly explained at the end of Section 4.1.

simplicial manifold as span:

$$
\begin{array}{ccccc}
U(1) & \longleftarrow & \Lambda^{(4)} & \xrightarrow{\text{f.f.}} & G^3 \\
\Downarrow & & \Downarrow\Downarrow\Downarrow & & \Downarrow\Downarrow\Downarrow \\
* & \longleftarrow & \Lambda & \xrightarrow{\text{full}} & G^2 \\
\Downarrow & & \Downarrow\Downarrow & & \Downarrow\Downarrow \\
* & \longleftarrow & Y & \xrightarrow{\text{full}} & G \\
\Downarrow & & \Downarrow & & \Downarrow \\
* & \longleftarrow & * & \xrightarrow{=} & *
\end{array}
\tag{136}
$$

Here the right column is just the category $BG$ presented as a simplicial manifold by taking the nerve (with triangular 2-cells being the group composition $\circ$, which forms $(G \times G) \times_G^{\circ,\text{id}} G \cong G^2$, and so on). In the middle column, $\Lambda$ is the manifold of all triangular shaped 2-cells, and is a $U(1)$ bundle over the triangular loop $(Y \times Y) \times_G^{\circ \Pi^2, \Pi} Y$ (where we have denoted the covering $Y \to G$ as $\Pi$, and $(Y \times Y) \times_G^{\circ \Pi^2, \Pi} Y$ is the submanifold of $Y^3$ that satisfies the $G$ composition rule after the $\Pi$ projection), representing the non-unique multiplicative structure on $Y$. The construction of $\Lambda$ from a given $L$ [96] is essentially (141) that we will introduce later,[127] for now let us just accept that a suitable $\Lambda$ is already constructed from a given $L$. $\Lambda^{(4)}$ is formed by demanding the simplicial manifold to be 2-coskeletal, i.e. $\Lambda^{(4)}$ is the manifold of all tetrahedral shaped 3-cells formed by gluing four triangular shaped 2-cells along shared edges:

$$
\begin{aligned}
\Lambda^{(4)} &:= ((\Lambda \times_Y^{\partial_2, \partial_2} \Lambda) \times_{Y \times Y}^{(\partial_1, \partial_1),(\partial_2, \partial_1)} \Lambda) \times_{Y \times Y \times Y}^{(\partial_0, \partial_0, \partial_0),(\partial_2, \partial_1, \partial_0)} \Lambda \\
&\cong ((\Lambda \times_Y^{\partial_2, \partial_2} \Lambda) \times_{Y \times Y}^{(\partial_1, \partial_1),(\partial_2, \partial_1)} \Lambda) \times U(1).
\end{aligned}
\tag{137}
$$

The second line expresses the condition that the four $U(1)$ bundles $\Lambda$ can together piece up to a trivialized $U(1)$ bundle over the horn formed by three $\Lambda$'s, and the trivialization is specified by the map from this $U(1)$ factor in $\Lambda^{(4)}$ to $U(1)$,

$$
\text{lattice WZW curvature} \longleftarrow \quad \text{[diagram]} \quad , \tag{138}
$$

which will later be interpreted as assigning the lattice WZW curvature—the gauge invariance of which is the manifestation of that (137) has trivialized $U(1)$ fibre.

We make a few remarks about multiplicative bundle gerbes:

- Why does it become necessary here for the span to be a simplicial manifold in general? Since we are delooping (133), we want to introduce some notion of composition on $Y$, as well as some notion of horizontal composition on $L$ (the original composition of $L$ becomes the vertical composition). This corresponds to an ordinary functor $(\circ_h \rightrightarrows \circ)$ from $(L \rightrightarrows Y)^2$ to $(L \rightrightarrows Y)$. But to capture the general interesting cases, we should consider anafunctors, and this leads to the use of simplicial manifold [37], as explained in Section 5.4. The globular shaped 2-cells in $L$ can be viewed as special kind of triangular shaped 2-cell in $\Lambda$, recall (120).

- Multiplicative bundle gerbes (136) are classified by $H^4(BG; \mathbb{Z}) \cong H^4(|BG|; \mathbb{Z})$ [38], where the appearance of $\mathbb{Z}$ can be manifested by a Villainization step, i.e. composing on the left of (136) the trice delooping of (122). This $H^4(BG; \mathbb{Z})$ classification of

---

[127]Except there we will apply the idea (141) to (140), which is what we really need, rather than to (136). Some key technical difference with [96] will be explained in footnote 133.

multiplicative bundle gerbes maps to the $H^3(\mathcal{T} = |G|; \mathbb{Z})$ classification bundle gerbes (133) by forgetting about the multiplicative structure (looping); correspondingly, at the level of the ordinary singular cohomology of the classifying space, $H^4(|BG|; \mathbb{Z})$ maps to $H^3(\mathcal{T} = |G|; \mathbb{Z})$ by *transgression* [38] (recall footnote 118). For $G = SU(N)$, the transgression is an isomorphism, $H^4(|BG|; \mathbb{Z}) \xrightarrow{\sim} H^3(\mathcal{T} = |G|; \mathbb{Z}) \cong \mathbb{Z}$.

The relation of $H^4(BG; \mathbb{Z})$ to usual group cohomology $H^3_{\text{group}}(G; U(1)) \cong H^4_{\text{group}}(G; \mathbb{Z})$ will be discussed later, as we make connection between our model and the familiar group cohomology lattice models with finite groups.

- Just like (124) can be familiarly rephrased in terms of a Lie group extension of $U(1)$ by $\mathbb{Z}$, or say of $BU(1)$ by $B\mathbb{Z}$, here we can rephrase (136) as a Lie 2-group extension of $G$ by $BU(1)$, or say of $BG$ by $B^2U(1)$ [37]. The extended Lie 2-group thus obtained is the span of (136) with the constraint of trivial WZW phase;[128] there is an anafunctor from $B^2U(1)$ (which demands the 3-morphisms to be identities, in compatible with the restriction to trivial WZW phase) into this extended Lie 2-group, and then this extended Lie 2-group can project to $BG$, forming a short exact sequence in a suitable sense.fThe classification of possible extensions, denoted as $\text{Ext}(BG; B^2U(1))$, is indeed given by $H^0(BG; B^3U(1)) \cong H^0(BG; B^4\mathbb{Z}) \cong H^4(BG; \mathbb{Z})$.

What we really want for the topologically refined lattice model is slightly different from the multiplicative bundle gerbe (136)—just like how (123) differs from (124). So we cannot directly use the well-studied multiplicative bundle gerbe in the mathematical literature, and some technical modifications must be made, guided by our goal in physics. Instead of delooping (133), we want to perform the "$\mathcal{T} \hookrightarrow E\mathcal{T}$ step", which should lead to an anafunctor from $E\mathcal{T}$ (equipped with the inclusion functor from $\mathcal{T}$) to $B^3U(1)$—with which we want to realize the generator of $H^4(E\mathcal{T}, \mathcal{T}; \mathbb{Z})$. For $\mathcal{T} = |G|$ it satisfies $H^4(E\mathcal{T}, \mathcal{T}; \mathbb{Z}) \xleftarrow{\sim} H^3(\mathcal{T}; \mathbb{Z}) \xleftarrow{\sim} H^4(BG; \mathbb{Z})$ (recall the $H^4(BG; \mathbb{Z})$ classifies the multiplicative bundle gerbe (136) above).

If we do not mind using infinite dimensional d.o.f. and had used the tautological bundle gerbe for (133), then for the "$\mathcal{T} \hookrightarrow E\mathcal{T}$ step" we should naturally consider the following anafunctor of strict categories (or we can take the nerve of each column):

$$
\begin{array}{ccccc}
U(1) & \longleftarrow & (\bar{\mathcal{P}}_2\mathcal{T} \times U(1)/WZW)^{(2)} & \xrightarrow{\text{f.f.}} & \mathcal{T} \times \mathcal{T} \\
\Downarrow & & \Downarrow & & \Downarrow \\
* & \longleftarrow & \bar{\mathcal{P}}_2\mathcal{T} \times U(1)/WZW & \xrightarrow{\text{full}} & \mathcal{T} \times \mathcal{T} \\
\Downarrow & & \Downarrow & & \Downarrow \\
* & \longleftarrow & \bar{\mathcal{P}}\mathcal{T} & \xrightarrow{\text{full}} & \mathcal{T} \times \mathcal{T} \\
\Downarrow & & \Downarrow & & \Downarrow \\
* & \longleftarrow & \mathcal{T} & \xrightarrow{=} & \mathcal{T}
\end{array}
\tag{139}
$$

whose relation to the tautological bundle gerbe is that, fixing a source (or target) object gives

$$\bar{\mathcal{P}}\mathcal{T}|_{\text{fixing } \mathsf{s}} = \bar{\mathcal{P}}_*\mathcal{T} = Y_{taut} \qquad \text{and}$$

$$(\bar{\mathcal{P}}_2\mathcal{T} \times U(1)/WZW)|_{\text{fixing } \mathsf{ss}} \cong \bar{\mathcal{P}}_*\mathcal{T} \times (\bar{\mathcal{P}}\bar{\Omega}_*\mathcal{T} \times U(1)/WZW) = L_{taut}.$$

The left column $B^3U(1)$ represents the WZW phase over a 3d region. The span of (139) is intuitive if we have a continuum nl$\sigma$m and, as motivated around (54), think of the lattice as being embedded in the continuum:

---

[128]The reason that the WZW phase is required to be trivial in describing the 2-group is much like that in $BG$ the flux is trivial, so to specify the group composition rule.

- along a lattice path, the continuum field will trace out a path in $\mathcal{T}$, giving an element in $\bar{\mathcal{P}}\mathcal{T}$;

- over a lattice surface, the continuum field will swipe out a disk-like surface, giving an element in $\bar{\mathcal{P}}_2\mathcal{T}$, and since we only care about the difference between two surfaces according to the WZW phase bounded between them (in the sense of (54)), the space reduces $\bar{\mathcal{P}}_2\mathcal{T} \times U(1)/WZW$;

- over a lattice volume, the continuum field will swipe over a ball-like volume, giving an element in $\bar{\mathcal{P}}_3\mathcal{T}$, but again we only care about the WZW phase over this volume, which is already uniquely specified given the source and target surfaces of this volume, leading to

$$(\bar{\mathcal{P}}_2\mathcal{T} \times U(1)/WZW)^{(2)} := (\bar{\mathcal{P}}_2\mathcal{T} \times U(1)/WZW) \times^{(s,t),(s,t)}_{\bar{\mathcal{P}}\mathcal{T} \times^{(s,t),(s,t)}_{\mathcal{T}\times\mathcal{T}} \bar{\mathcal{P}}\mathcal{T}} (\bar{\mathcal{P}}_2\mathcal{T} \times U(1)/WZW)$$

$$= (\bar{\mathcal{P}}_2\mathcal{T} \times U(1)/WZW) \times U(1),$$

and the last $U(1)$ factor is the WZW phase over the volume. (Since we only care up to the WZW phase, it is intuitive to see that the nerve of the span of (139) is 2-coskeletal.)

Therefore, (139) is formulating the idea introduced at (54) in a more natural categorical language.[129,130]

However, for an actual lattice model, we want the d.o.f. to be finite dimensional, i.e. we want to use a bundle gerbe (133) with a finite dimensional $Y$. Then, just like in the delooping problem (136), in general for "$\mathcal{T} \hookrightarrow E\mathcal{T}$ step" we will need an anafunctor of simplicial manifolds (instead of strict categories) of the form

$$
\begin{array}{ccccc}
U(1) & \longleftarrow & \mathcal{T} \times \Lambda^{(4)} & \xrightarrow{\text{f.f.}} & \mathcal{T} \times G^3 \\
\Downarrow & & \Downarrow\Downarrow\Downarrow\Downarrow & & \Downarrow\Downarrow\Downarrow\Downarrow \\
* & \longleftarrow & \mathcal{T} \times \Lambda & \xrightarrow{\text{full}} & \mathcal{T} \times G^2 \\
\Downarrow & & \Downarrow\Downarrow\Downarrow & & \Downarrow\Downarrow\Downarrow \\
* & \longleftarrow & \mathcal{T} \times Y & \xrightarrow{\text{full}} & \mathcal{T} \times G \\
\Downarrow & & \Downarrow\Downarrow & & \Downarrow\Downarrow \\
* & \longleftarrow & \mathcal{T} & \xrightarrow{=} & \mathcal{T}
\end{array}
\tag{140}
$$

(Recall we made the simplifying assumption that the target space $\mathcal{T} \cong |G|$ for some connected and simply connected semi-simple Lie group $G$, which is needed for passing from (133) to (140) via (136), just like how $\mathcal{T} \cong |G|$ allows passing from (122) to (123) via (124).) The right column is the target category $E\mathcal{T} \cong EG$ used in traditional lattice nl$\sigma$m, presented as a simplicial manifold by taking the nerve, and as in Section 4.1, we represented $(g_1, g_2) \in \mathcal{T}^2$ as $(g_1, g_2 g_1^{-1}) \in \mathcal{T} \times G$, $(g_1, g_2, g_3) \in \mathcal{T}^3$ as $(g_1, g_2 g_1^{-1}, g_3 g_2^{-1}) \in \mathcal{T} \times G^2$, and so on. The left column $B^3 U(1)$ is again nothing but the WZW phase over a 3d region. The middle column, again 2-coskeletal, is almost what we want for the target category of the topologically refined lattice nl$\sigma$m—except we will usually perform the last simple step of Villainization, i.e. composing the trice delooping of (122) on the left of (140), which will make it 3-coskeletal.

Now we come to the crucial technical point of how to construct the multiplicative structure $\Lambda$ in (140), given a finite dimensional bundle gerbe $(L \rightrightarrows Y)$ from (133). We will first present

---

[129]As we said in footnote 124, in (54) we did not take identification under thin homotopy but here we do. This distinction is unimportant since it does not affect the WZW evaluation. We could have taken identification under thin homotopy in (54) too.

[130]In the previous examples, (123) is related to the continuum loop space since $S^1 \times \mathbb{R} \cong \bar{\mathcal{P}}S^1$, i.e. (47), and (131) is related to the continuum loop space since $S^2 \times SU(2) \cong \bar{\mathcal{P}}S^2 \times U(1)/Berry$, i.e. (45) and (48).

the general procedure, illustrated as (141), and then we will discuss the special case we need in practice, (142), which, due to the simplification discussed below (135), is much simpler than the general cases.

It is, again, helpful to begin with what we expect from the continuum QFT, (139), which, although involving infinite dimensional d.o.f., gives us the crucial intuition of what we really want. We first present (139) as an anafunctor of diffeological simpilicial sets by taking the nerve; the space of 2-cells form $\bar{\Delta}_2\mathcal{T} \times U(1)/WZW$, where elements of $\Delta_2\mathcal{T}$ are singular 2-chains in $\mathcal{T}$.[131] Then we can induce the multiplicative structure $\Lambda$ in the finite dimensional (140) using the stable isomorphism between $(L \rightrightarrows Y)$ and the tautological $(L_{taut} \rightrightarrows Y_{taut})$. The general idea is illustrated as

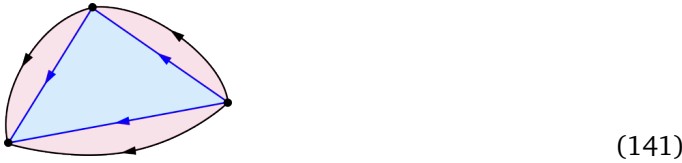

(141)

Figure 25: Illustration of the general idea of the construction.

where each 0-cell is an element from $\mathcal{T} = |G|$, each black 1-cell is from $\mathcal{T} \times Y$, each blue 1-cell is from $\bar{\mathcal{P}}\mathcal{T}$ (which is a $Y_{taut} = \bar{\mathcal{P}}_*\mathcal{T}$ bundle over $\mathcal{T}$), whose "multiplicative structure" is represented by the blue 2-cell from $\bar{\Delta}_2\mathcal{T} \times U(1)/WZW$, and each pink 2-cell between the black and blue 1-cell is a pullback of the stable isomorphism $\Sigma$ between $Y$ and $Y_{taut}$ onto one between $\mathcal{T} \times Y$ and $\bar{\mathcal{P}}\mathcal{T}$. Since the space of each of the four 2-cells in the above is a $U(1)$ bundle over the space of the 1-cells on around it, the spaces of the four 2-cells together can piece up to a $U(1)$ bundle over the space of the three black 1-cells, and this will be identified as the desired $\mathcal{T} \times \Lambda$.

In practice, the construction is much simpler, thanks to the simplification discussed below (135). When the groupoid $(L \rightrightarrows Y)$ can be embedded in $(L_{taut} \rightrightarrows Y_{taut})$ by an ordinary functor while preserving the projections from $Y$ and $Y_{taut}$ to $\mathcal{T}$, the stable isomorphism $\Sigma$ is just the pullback of $L_{taut}$ along the embedding of $Y \times_\mathcal{T} Y_{taut} \hookrightarrow Y_{taut} \times_\mathcal{T} Y_{taut}$.[132] As a result, in (141) the pullback of the $U(1)$ bundle $\bar{\Delta}_2\mathcal{T} \times U(1)/WZW$ over the space of the blue triangular loop $(\bar{\mathcal{P}}\mathcal{T} \times_\mathcal{T}^{\mathsf{s},\mathsf{t}} \bar{\mathcal{P}}\mathcal{T}) \times_{\mathcal{T}^2}^{(\mathsf{s},\mathsf{t}),(\mathsf{s},\mathsf{t})} \bar{\mathcal{P}}\mathcal{T}$ along the embedding of each $\mathcal{T} \times Y \hookrightarrow \bar{\mathcal{P}}\mathcal{T}$ directly gives rise to the desired $U(1)$ bundle $\mathcal{T} \times \Lambda$ over the space of the black triangular loop $\mathcal{T} \times ((Y \times Y) \times_G^{\circ\Pi^2,\Pi} Y)$. In more explicit terms, given three elements of $\mathcal{T} \times Y$, denoted as $(g_0, y_{10})$, $(g_1, y_{21})$ and $(g_0, y_{20})$ which satisfy $\Pi(y_{ij})g_j = g_i$, we can embed each into $\bar{\mathcal{P}}\mathcal{T}$, obtaining three paths $\gamma_{10}, \gamma_{21}, \gamma_{20}$ that form a loop:

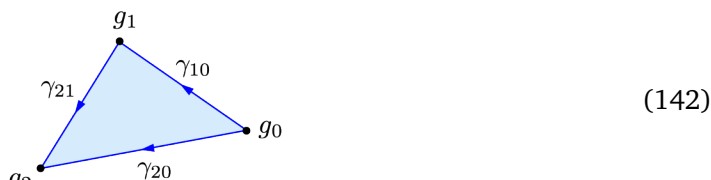

(142)

Figure 26: Illustration of the actual construction.

---

[131]$\bar{\Delta}_2\mathcal{T}$ means $\Delta_2$ taking identification under thin homotopy. $\bar{\Delta}_2\mathcal{T} \times U(1)/WZW = \Delta_2\mathcal{T} \times U(1)/WZW$ because a thin homotopy between two $\Delta_2\mathcal{T}$ must have zero WZW phase.

[132]It is helpful to recall the simpler case of $U(1)$ bundle in footnote 122.

and the $U(1)$ fibre in $\mathcal{T} \times \Lambda$ on $(g_0, y_{10}, y_{21}, y_{20}) \in \mathcal{T} \times ((Y \times Y) \times_G^{\circ\Pi^2,\Pi} Y)$ is the $U(1)$ fibre in $\bar{\Delta}_2 \mathcal{T} \times U(1)/WZW$ on $(\gamma_{10}, \gamma_{21}, \gamma_{20}) \in (\bar{\mathcal{P}}\mathcal{T} \times_{\mathcal{T}}^{\mathsf{s,t}} \bar{\mathcal{P}}\mathcal{T}) \times_{\mathcal{T}^2}^{(\mathsf{s,t}),(\mathsf{s,t})} \bar{\mathcal{P}}\mathcal{T}$, whose elements are understood as surfaces in $\mathcal{T}$ bounded by the loop $(\gamma_{10}, \gamma_{21}, \gamma_{20})$, with identification if two surfaces bound zero WZW phase.

We remark that our construction (142) of $\Lambda$ in (140) has some key technical differences with the construction [96] of $\Lambda$ in multiplicative bundle gerbe (136).[133] But they are still topologically equivalent, in the sense that we want (140) to realize the generator of $H^4(E\mathcal{T}, \mathcal{T}; \mathbb{Z}) \cong \mathbb{Z}$, while (136) is to realize the generator of $H^4(BG; \mathbb{Z}) \cong \mathbb{Z}$. These generator classes map to each other via $H^4(E\mathcal{T}, \mathcal{T}; \mathbb{Z}) \xleftarrow{\sim} H^3(\mathcal{T}; \mathbb{Z}) \xleftarrow{\sim} H^4(BG; \mathbb{Z})$.

Now we are ready to make connection to the lattice model construction (70) in Section 4.1. A field configuration in the path integral is a functor[134] from $\mathcal{L}$ (introduced near the end of Section 5.4) to the refined target category, i.e. the span of (140). Each such functor is weighted by the path integral weight, and the locality of the weight means the total weight is a product of local weights at each lattice cell, where the local weight $W_i$ is a smooth map from the $i$-cells of the target cateogry to $\mathbb{R}_+$.[135] A technical difference is that in (140) we used simplicial manifold, which is more suitable when the lattice is a simplicial complex (so that $\mathcal{L}$ is also a simplicial set); in practice the lattice is usually a cubic lattice (so that $\mathcal{L}$ is a cubical set), so the simplicial target category will be replaced by a cubical one. Let us look at the d.o.f. and the weights more closely.

- The vertex d.o.f. in (70) is the traditional one that takes value in $\mathcal{T}$, which is $S^3 \cong |SU(2)|$ there. This is indeed the space of objects in the span of (140).

- The link d.o.f. along with the vertex d.o.f. on the two ends of the link in (70) take value in $\mathcal{T} \times Y$, the space of 1-morphisms in the span of (140).

  Regarding the choice of $Y$, we do not follow [95] which uses

  $$Y = (SU(2)\backslash\{-\mathbf{1}\}) \sqcup (SU(2)\backslash\{+\mathbf{1}\});$$

  rather, we use $Y = (SU(2)\backslash\{-\mathbf{1}\}) \sqcup (SU(2)\backslash\{+\mathbf{1}\} \times S^2)$, (60). The embedding $\mathcal{T} \times Y \hookrightarrow \bar{\mathcal{P}}\mathcal{T}$ is given by (62), establishing the desired stable isomorphism needed for constructing $\Lambda$ in the below. This embedding explains why we have the extra $S^2$ factor: With this extra $S^2$, the embedding becomes rotationally covariant, as explained below (62);[136] without this $S^2$, the embedding can be chosen as if $\hat{n}$ is fixed to a certain

---

[133]In our construction of $\Lambda$, we started with (139) (whose relation to the tautological bundle gerbe is explained there), in which the 1-morphisms in $\bar{\mathcal{P}}\mathcal{T}$ simply compose by concatenation. By contrast, if we try to deloop the tautological bundle gerbe, elements in $Y_{taut} = \bar{\mathcal{P}}_*\mathcal{T}$ cannot compose by concatenation, because the starting points are fixed at the identity, while the ending points are arbitrary. Therefore, in [96], to facilitate the delooping, $Y'_{taut} = \mathcal{P}_*G$ (recall $\mathcal{T} = |G|$ in our discussion) with the starting point fixed at $\mathbf{1} \in G$ and without identification under thin homotopy is used, so that, having specified the parametrization of the path, the composition can be defined by pointwise group mutiplication. To define the 2-morphisms of the delooping, in [96] the Mickelsson product is used for horizontal composition and geometrical concatenation is used for vertical composition before modding out WZW (while in [40] the Mickelsson product is used for both horizontal and vertical compositions), hence defining a 2-group $\mathcal{P}_*G \ltimes (\mathcal{P}_*\Omega_*G \times U(1)/WZW) \rightrightarrows \mathcal{P}_*G \rightrightarrows *$ in comparison to the 2-groupoid part $\bar{\mathcal{P}}_2\mathcal{T} \times U(1)/WZW \rightrightarrows \bar{\mathcal{P}}\mathcal{T} \rightrightarrows \mathcal{T}$ in the span of (139). This non-topological, detailed difference is finally carried over to the definitions of $\Lambda$ via the process (141).

[134]Anafunctor is not needed, because $\mathcal{L}$ itself is discrete and there is no discontinuity issue here.

[135]This only covers (70) which does not contain topological terms which are complex phases (such as the 2d WZW term). At this point we have not fully understood how to formulate a general local path integral weight in the categorical language. We will mention this in Section 7.

[136]It seems this extra $S^2$ may be some reminiscent (though not the same as) of the extra data in the more general bundle gerbe construction introduced in [116, 117]. But we are not sure about this yet and this should be further investigated.

direction,[137] hence not rotationally covariant.[138]

The link weight $W_1$ (70) maps each element of $\mathcal{T} \times Y$ to a positive number in a rotationally invariant manner, hence depends only on the $\lambda$ and $m$. Crucially, we require the weight to approach 0 in suitable ways as explained below (61).

- The plaquette d.o.f. in (70) takes value on a $U(1)$ fibre, parametrized by $e^{i\mathcal{W}}$, over the base space $\mathcal{T} \times ((Y \times Y) \times_G^{\circ\Pi^2, \circ\Pi^2} (Y \times Y))$, i.e. (63), of the vertex and link d.o.f. on the square-shaped loop around the plaquette. The $U(1)$ bundle is still denoted as $\mathcal{T} \times \Lambda$, except the 2-cells in it are now square-shaped rather than triangular-shaped, i.e. each 2-cell has four boundary maps to 1-cells; other than this difference, the construction of this $U(1)$ bundle is the same as (142). The plaquette weight $W_2$ in (70), introduced below (65), is constructed based on (the quadrangle version of) the understanding (142) of the $U(1)$ bundle, together with the desired physical properties, as we now explain.

  What we did below (65) is essentially defining two different sections in $\mathcal{T} \times \Lambda$. Since this is a non-trivial $U(1)$ bundle, there can be no global section, so each choice of section must involve some singularity that needs to be suitably handled. The two different sections are:

  – To describe the $U(1)$ bundle $\mathcal{T} \times \Lambda$, we want to parametrize it by the base space $\mathcal{T} \times ((Y \times Y) \times_G^{\circ\Pi^2, \circ\Pi^2} (Y \times Y))$ (whose element is specified by the link and vertex d.o.f. around the plaquette, see (63)) and the fibre $U(1)$ (parametrized by $e^{i\mathcal{W}}$). Since this is a non-trivial bundle, the parametrization must develop singularity somewhere. More exactly, the point on each $U(1)$ fibre at which we set $e^{i\mathcal{W}} = 1$ is a choice of section, and there must be some codimension-2 loci on the base manifold, at which the choice of the $e^{i\mathcal{W}} = 1$ point becomes singular. (In the simpler example of $SU(2)$ as a $U(1)$ bundle over $S^2$, this singularity is the familiar Dirac string.) But the weight and the observables should not depend on our choice of parametrization, and hence such singularity in the parametrization should not appear in any physical effects.

  – With the plaquette weight $W_2$, we are assigning a positive value to each element of $\mathcal{T} \times \Lambda$, therefore on each $U(1)$ fibre we can ask which point has the maximum weight within this fibre. This will pick out a section, and this is physical. However, since there can be no global section, it is impossible that each $U(1)$ fibre has a unique point that has a maximum weight within this fibre. On some loci on the base space $\mathcal{T} \times ((Y \times Y) \times_G^{\circ\Pi^2, \circ\Pi^2} (Y \times Y))$, we will let the weight to become uniform over the entire $U(1)$ fibre.

  According to the construction of (the quadrangle verion of) (142), we use the aforementioned embedding $\mathcal{T} \times Y \hookrightarrow \bar{\mathcal{P}}\mathcal{T}$, (62), to think of the link and vertex d.o.f. around the plaquette as specifying a quadrangle-shaped loop in $\mathcal{T}$. Then the $U(1)$ fibre consists of the possible interpolating surfaces bounded by such a loop, with two different surfaces identified if they bound trivial WZW phase. Physically it is intuitive to demand the dynamical property that, the maximum weight within each $U(1)$ fibre occurs at the point that represents, in the sense of (142), the minimal

---

[137]It may be helpful to understand this point in the simpler case of $U(1)$ bundle over $S^2$, see footnote 122.

[138]The reason why this $S^2$ was absent in [95] is that, the multiplicative structure and the stable isomorphism to the tautological bundle gerbe were not under consideration in [95]. While in [96] the multiplicative structure constructed using the stable isomorphism to the tautological bundle gerbe was indeed the main consideration, being rotationally covariant was unimportant there. But we would like our construction to be physically natural, in particular to be covariant under global symmetry, hence we introduced this extra $S^2$. This will become even more important when we deloop to gauge theory later.

surface bounded by the loop. (This is the "base" surface of the pyramid in (66). In the discussions below (66), we introduced two possible procedures to make standardized choice of interpolating surfaces on which the weight is maximized; they are not exactly the minimal surfaces, but at least close to minimal surfaces when the loops are small.) When the loop becomes large such that the minimal surface (or in practice, our standardized choice of interpolating surface) becomes ambiguous, we demand the weight to become uniform over the $U(1)$ fibre, which is when we demand $|\mu| \to 0$.

In (65), our choice of the parametrization of $\mathcal{T} \times \Lambda$ is so that, the $e^{i\mathcal{W}} = 1$ section from our choice of parametrization is *not* the section where $W_2$ is maximized on the fibres. Rather, the $W_2$-maximizing section occurs at $e^{i\mathcal{W}} = \mu/|\mu|$, where the phase $\mu/|\mu|$ is constructed according to (66). Since we said above that the $W_2$-maximizing section is associated with the "base" surface of the pyramid in (66), the $e^{i\mathcal{W}} = 1$ section is therefore associated with the surface made of the four "sides" of the pyramid. (As we said there, this is motivated by (46) in the $S^2$ spinon decomposition.)

- The WZW phase over a cube is the last $U(1)$ factor in (the cubical analog of) (137), which will be mapped identically to the top left of (140). The advantage of our choice parametrization (66) of $e^{i\mathcal{W}}$ is that, the WZW phase over a cube will then be $e^{id\mathcal{W}}$, in retrospect rendering $\mathcal{W}$ the meaning of WZW curving. (Alternatively, if we had redefined $e^{i\mathcal{W}'} := e^{i\mathcal{W}}\mu^*/|\mu|$, then the $e^{i\mathcal{W}'} = 1$ section from this new parametrization would be the $W_2$-maximizing section, which is physical and therefore gauge invariant, but then the WZW phase $e^{id\mathcal{W}}$ is not given by $e^{id\mathcal{W}'}$.)

  In (70) we further Villainized the WZW phase, which means we composed on the left of (the cubic analogue of) (140) the trice delooping of (122), as mentioned below (140). The WZW phase $U(1) \ni e^{id\mathcal{W}}$ factor in the space of 3-cells is then replaced by $\mathbb{R} \ni \mathcal{S}$, interpreted as the skrymion density over a cube, subjected to the constraint $e^{i2\pi\mathcal{S}} = e^{id\mathcal{W}}$, and the cube weight $W_3$ maps it to $\mathbb{R}_+$ in a way that prefers small $|\mathcal{S}|$. Finallyx, the space of 4-cells has a factor of $\mathbb{Z} \ni d\mathcal{S}$, interpreted as the hedgehog defect number in a hypercube, and its fugacity $W_4$ maps it to $\mathbb{R}_+$; or we can use $W_4^{forbid}$ that enforces $d\mathcal{S}$ to vanish via a $U(1)$ Lagrange multiplier, which corresponds to restricting the 4-cells to be degenerate 4-cells (the cubic counterpart of identity 4-morphisms).

This completes the explanation how our model construction of Section 4.1 is guided by (140).

Before we move on to Yang-Mills theory, we would like to point out a remarkable observation: Our anafunctor perspective of "topological refinement", which primarily aims to handle the smoothness of Lie groups, is also useful for finite groups—in which cases our construction will reduce to the familiar group cohomology lattice models [19, 23].

It is familiar that when applying the usual group cohomology to Lie groups, we need to include cochains $G^n \to U(1)$ that are only piecewise continuous (more precisely, Borel), hence making the lattice model thus built manifestly discontinuous [23]. There is a generalization of the usual group cohomology, known as Segal's double cohomology [118] or as Brylinski's differentiable group cohomology [119], that aims to avoid such discontinuous cochains. (Indeed, very recently, differentiable group cohomology has been used to carefully study anomalies of Lie group symmetries [120].) And the gerbe and anafunctor perspective we employed above is a further generalization of this double cohomology—roughly speaking, the double cohomology corresponds to special cases where $Y$ is chosen to be a "good cover" of $\mathcal{T}$ with sufficiently fine patches; but such $Y$ cannot be equivariant, hence not suitable for our purpose.[139] This is

---

[139]Let us go into more details [37]. Our choice of $Y$ is nice for being equivariant, i.e. the patches are invariant under conjugation, which is important for manifesting the global symmetry in the context of Section 4.1. However,

why for our problem we indeed need to generalize our perspective to the level of (140).

Indeed because (140) is such a natural generalization of the familiar group cohomology, we can in fact manifestly see that (140) encompasses the familiar group cohomology lattice models [23] when $G$ becomes a finite group and still $\mathcal{T} = |G|$. We can simply use $Y = G$ and $\Lambda = G^2 \times U(1)$ (of the form (115)), and $\Lambda^{(4)} \cong G^3 \times U(1)^4$; then we map $G^3 \times U(1)^4$ to $U(1)$ (assigning the WZW curvature) by multiplying the four $U(1)$ phases along with an extra $U(1)$ phase that depends on $G^3$—the associator from $H^n_{\text{group}}(G; U(1))$ seen in Section 5.3. In practice, since the extra $U(1)$ in $\Lambda$ (and hence the extra $U(1)^4$ in $\Lambda^{(4)}$) does not contain any topological information of $G$, it is usually discarded,[140] leaving only the associator. This explains why in (111) we said it is more natural to view the associator as a non-identity 3-morphism.

Related to this, the Dijkgraaf-Witten theory [19] with finite gauge group—so that the flat connection condition can be (and is indeed) imposed—is encompassed by (136) (instead of (140) in the previous paragraph), with $Y, \Lambda$ and the relevant discussions the same as in the previous paragraph. This is quite different from the case of continuous gauge group, where the flat connection condition becomes unphysical[141] —so that we cannot start with $BG$ but must start with $BEG$—and we need more involved treatment to be introduced in (143) below.

Now we see these familiar group cohomology lattice models can be derived in two ways: First as special cases of the powerful tensor category formalism that has been well-developed for TQFT (see e.g. [109]), and second, as we now see, as special cases of our anafunctor formalism. This hints that there is hope for a future unification of the usage of category theory in UV dynamical QFT and in IR TQFT, as we will briefly discuss in Section 7.

Finally we discuss the construction for lattice Yang-Mills theory. Recall the target category of Villainized $U(1)$ lattice gauge theory is the delooping of the target category of Villainized $S^1$ lattice nl$\sigma$m, so, clearly, what we now expect is a suitable notion of delooping the target category (140) of $\mathcal{T} = |G| = |SU(N)|$ nl$\sigma$m, which should result in an anafunctor from the nerve of $BEG$ (the target category of traditional lattice gauge theory), equipped with the inclusion

---

if we do not care about the global symmetry but only the topology, then we can choose $Y$ to be a "good cover" made of finely cut patches, such that each of $Y$, $Y^{[2]} := Y \times_G Y$, $Y^{[3]} := Y \times_G Y \times_G Y$ and so on is a disjoint union of contractible spaces. In this case, the $U(1)$ bundle $L$ over $Y \times_G Y$ is automatically trivial. While such choice of $Y$ has the disadvantage of not being equivariant (hence not suitable for our construction of lattice QFT), it has the advantage that there is no non-trivial $U(1)$ bundle on any $Y^{[m]}$. Let us define the space $K_{n,m}$ as the pullback of $G^n$ (viewed as the space of $n$-cells of the nerve of $BG$) along the covering $Y^{[m]}$ over each $G$ factor; the elements of $K_{n,m}$ form the basis of a double chain complex $C_{\text{Segal } n,m}(G)$. Then we can consider smooth mappings from $K_{n,m}$ to $U(1)$, which leads to a double cochain complex $C^{n,m}_{\text{Segal}}(G; U(1))$. From this we can define the *Segal double cohomology* $H^k_{\text{Segal}}(G; U(1))$, for which a representative element involves one element from each of $C^{n,m}_{\text{Segal}}(G; U(1))$ that satisfies $n + m = k$. It turns out that $H^k_{\text{Segal}}(G; U(1))$ is isomorphic to $H^{k+1}(BG; \mathbb{Z}) \cong H^{k+1}(|BG|; \mathbb{Z})$ [37, 119]. Now, with such choice of $Y$, it is not hard to check that a multiplicative bundle gerbe is described by a representative element of a class in $H^3_{\text{Segal}}(G; U(1))$ [37,119]. Segal double cohomology is a generalization of the usual group cohomology, with the advantage that we only consider those mappings from $K_{n,m}$ to $U(1)$ that are smooth. When $G$ is discrete, we can let $Y = G$ and Segal double cohomology reduces to usual group cohomology.

[140]This discarded $U(1)$ can however be recognized as the complex phase of the $\mathbb{C}$-linear enrichment in the definition of a tensor category—a standard language used to describe of topological theories with discrete d.o.f. (e.g. [109]). This will be mentioned again in Section 7.

[141]See footnote 84 for a formal problem associated with this unphysical flatness requirement.

$BG \hookrightarrow BEG$ that represents flat connections, to $B^4 U(1)$:

$$
\begin{array}{ccccc}
U(1) & \longleftarrow & G^4 \times \widetilde{\Lambda}^{(5)} & \xrightarrow{\text{f.f.}} & G^{10} \\
\Downarrow & & \Downarrow\Downarrow\Downarrow\Downarrow & & \Downarrow\Downarrow\Downarrow\Downarrow \\
* & \longleftarrow & G^3 \times \widetilde{\Lambda} & \xrightarrow{\text{full}} & G^6 \\
\Downarrow & & \Downarrow\Downarrow\Downarrow\Downarrow & & \Downarrow\Downarrow\Downarrow\Downarrow \\
* & \longleftarrow & G^2 \times Y & \xrightarrow{\text{full}} & G^3 \\
\Downarrow & & \Downarrow\Downarrow\Downarrow & & \Downarrow\Downarrow\Downarrow \\
* & \longleftarrow & G & \xrightarrow{=} & G \\
\Downarrow & & \Downarrow & & \Downarrow \\
* & \longleftarrow & * & \xrightarrow{=} & *
\end{array}
\tag{143}
$$

Here $G^3$ is the three 1-cells around a triangular 2-cell, and we can equivalently say $G^3 \cong G^2 \times G$ where the last $G \ni Dg$ is the holonomy around the 2-cell in the lattice theory, and $Y$ is a surjective submersion covering of this holonomy $G$. Likewise, $G^6$ is the six 1-cells around a tetrahedral 3-cell, and we can say $G^6 \cong G^3 \times (G^3 \times_G G)$ where the last $G^3 \times_G G$ are the holonomies around four 2-cells around the tetrahedral 3-cell subjected to $DDg = \mathbf{1}$,[142] and $\widetilde{\Lambda}$ is a suitable $U(1)$ bundle (discussed below) over the four triangular 2-cells $Y^3 \times_G Y$ covering the $G^3 \times_G G$. Five such tetrahedral 3-cells can glue along their faces into a 4-cell, so that the span is by definition 3-coskeletal, and we denote the space of all such glues as $G^4 \times \widetilde{\Lambda}^{(5)}$[143] which forms a trivial $U(1)$ bundle over the horn formed by four tetrahedral 3-cells (an analogue of (137) in one dimension higher), and the trivialization map from the trivial $U(1)$ fibre to the $U(1)$ at the upper left represents the CS phase around a 4d lattice cell $e^{idC} = e^{i2\pi\mathcal{I}}$ (the gauge invariance of this quantity is the manifestation that the $U(1)$ fibre in $\widetilde{\Lambda}^{(5)}$ is trivial).

To complete the construction, we perform the last Villainization step on this trivialized $U(1)$ fibre in the 4-cells, i.e. we compose on the left of (143) the quarce (i.e. four times) delooping of (122), such that the span now becomes 4-coskeletal. The 5-morphisms valued in $\mathbb{Z}$ represent Yang monopoles. The desired anafunctor realizes the generator of $H^5(BEG, BG; \mathbb{Z})$, which, through the isomorphism $H^5(BEG, BG; \mathbb{Z}) \xleftarrow{\sim} H^4(BG; \mathbb{Z}) \cong \mathbb{Z}$, corresponds to the second Chern class as expected. If we forbid Yang monopoles, the d.o.f. in the span becomes a 5-coskeletal Kan complex with single object, that can be interpreted as a weak 4-group which extends $BEG$ by $B^4\mathbb{Z}$ such that the extension is trivial on the image of $BG \hookrightarrow BEG$. This essentially recovers the perspective introduced at the end of Section 5.2. This 4-group perspective is how the insightful interpretation of Villainized gauge theory in terms of 2-group [34–36] crucially motivated this entire program.

Section 4.2, and the follow-up paper [10] in greater details, describe how the intuitions from continuum QFT, partially involving ideas from the previous efforts [8, 100], help us construct such a structure (143)—in particular the $\widetilde{\Lambda}$—except there we used cubical cells rather than simplicial cells. In the below we will explain our rationale behind the construction. As far as we are aware of, the construction of (143) or something closed related has not been formally discussed in the mathematical literature. (This is in contrast to the case of nl$\sigma$m, where the step from (139) to (140) via (142) is a generalization of—although not exactly the same as—the construction of finite dimensional multiplicative bundle gerbe (136) in [96]; see footnote 133. We believe using the elements from [96], it should be straightforward to make the construction (140) mathematically rigorous.) We hope our rationale behind the construction (143), as we shall introduced below, to be formalized into rigorous mathematical treatment in the near future.

---

[142]If a tetrahedron has vertices labeled by $0, 1, 2, 3$, then in our parametrization an element of $G^3 \times (G^3 \times_G G)$ is $(g_{32}, g_{21}, g_{10}, Dg_{210}, Dg_{310}, Dg_{320}, g_{10}^{-1} Dg_{321} g_{10})$.

[143]If a 5-simplex has $0, 1, 2, 3, 4$, the first $G^4$ factor consists of $g_{43}, g_{32}, g_{21}, g_{10}$.

We want to construct the $\widetilde{\Lambda}$ in (143) similar in idea to how we constructed the $\Lambda$ in (140) via (142). So we want to think of the lattice as being embedded in the continuum, and then, as a first step, to each element of $G^2 \times Y$ find a representative continuum gauge field configuration over an embedded triangular plaquette. This is what (73) is doing, expect it is the cubical version instead of simplicial.

To render more geometrical intuition (the previous (142) is indeed very geometrical), recall it is customary to think of a $G$ gauge theory as a $|BG|$ nl$\sigma$m [19]: The Wilson line along a path in the spacetime is given by the embedding of the path in $|BG|$ integrated with the universal $G$ connection on $|BG|$; in terms of anafunctors, the universal bundle and universal connection are captured by (100) and (101) with $\mathcal{M}$ substituted by $|BG|$ and $\mathcal{E}$ by $|EG|$.[144] Of course, in the end, a $G$ gauge theory is not exactly a $|BG|$ nl$\sigma$m, because which particular point in $|BG|$ a point in the spacetime maps to is not a physical observable; but up to this issue, this "$|BG|$ nl$\sigma$m" perspective is very useful. With this perspective, given what is done in the previous paragraph, we may formally specify a representative functor from $G^2 \times Y \rightrightarrows G \rightrightarrows *$ to the singular 2-simplicial set $\Delta_2|BG| \overset{\rightarrow}{\rightrightarrows} \Delta_1|BG| \rightrightarrows |BG|$ (or the cubical version, in practice), in analogy to the embedding of $\mathcal{T} \times Y \ni (g, y)$ into $\bar{\mathcal{P}}\mathcal{T} \ni \gamma$ in (142). Let us denote the space of closed 2d surfaces in $|BG|$ given by piecing up four singular 2-cells as $(\Delta_2|BG|)^{(4)}$, then we have an important $U(1)$ bundle over $(\Delta_2|BG|)^{(4)}$, i.e. $\Delta_3|BG| \times U(1)/CS$, where two elements $(\sigma, e^{i\theta})$ and $(\sigma', e^{i\theta'})$ from $\Delta_3|BG| \times U(1)$ are considered equivalent if the two singular tetrahedra $\sigma, \sigma'$ from $\Delta_3|BG|$ share the same boundary in $(\Delta_2|BG|)^{(4)}$ and moreover the closed 3d volume formed by $\sigma \cup \bar{\sigma}'$ has a CS phase (making use of the universal CS 3-form over $|BG|$) that is equal to $e^{i\theta'}e^{-i\theta}$. Pulling back this $U(1)$ bundle onto the four $G^2 \times Y$'s we obtain the desired $G^3 \times \widetilde{\Lambda}$ over $G^3 \times (Y^3 \times_G Y)$. We then parametrize the $U(1)$ fibre by $e^{i\mathcal{C}}$, and the remaining discussions about in what sense (74) is related to $\widetilde{\Lambda}$ is then parallel to our previous discussions of how (65) is related to $\Lambda$ in nl$\sigma$m.

While this is the geometric rationale of how $G^3 \times \widetilde{\Lambda}$ is constructed, this is hardly useful in practice because $|BG|$ is infinite dimensional. Fortunately, since the universal CS 3-form on $|BG|$ only depends on the universal gauge connection—familiarly, $(AdA + 2A^3/3)/4\pi$—but not on the particular placement of points in $|BG|$, the constructed $U(1)$ bundle $G^3 \times \widetilde{\Lambda}$ must be describable in terms of the Wilson lines in a 3-cell in $\Delta_3|BG|$ without referring the points in $|BG|$. We claim that the construction of the CS saddle in terms of "interpolation into the interior of cube", sketched in Section 4.2 and detailed in [10] based on the previous works [8, 100], describes (the cubical version of) the desired $U(1)$ bundle (recall that the topology of a non-trivial bundle can, again, be encoded in how the saddle, as a section, develops singularities, just like the Berry connection saddle in the spinon decomposed $S^2$ nl$\sigma$m and the WZW curving saddle in our refined $S^3$ nl$\sigma$m). *What remains to be rigorously proven (and carefully defined at the mathematical level, in the first place) is that the claim indeed holds, i.e. that the $U(1)$ bundle thus described in practice indeed agrees with the $U(1)$ bundle that we constructed in the previous paragraph using $|BG|$ conceptually.* Our confidence in its validity relies on the evidences that: 1) there certainly exists 3d volumes in $|BG|$ in which the gauge holonomies conincide with our specified interpolations, 2) given a specified interpolation, the relation between the lattice formula that we use to evaluate the interpolation's associated CS phase (to be used as the lattice CS saddle) and the continuum CS 3-form integral has been studied in [10, 100], although not in the formal mathematical literature.

---

[144]In [38], the notion of *CS bundle 2-gerbe* is introduced. It is a $U(1)$ 3-bundle over $|BG|$, given by the composition of the multiplicative bundle gerbe (136) on the left of (100); the universal CS bundle 2-gerbe is when (100) describes the universal bundle, i.e. with $\mathcal{M}$ substituted by $|BG|$ and $\mathcal{E}$ by $|EG|$. Despite its name, we are not directly using it in our construction of (143).

# 6 Sketching a relation between continuum QFT and lattice QFT

From the Villain model to our constructions, and from the explicit descriptions in physical terms to the systematic derivations in categorical terms, we have seen the geometrical intuition from the continuum played a crucial role. This is natural, because the very purpose of our work is to realize in lattice QFT those topological operators that are present in continuum QFT.

In Section 5.5 we explained how our constructions in Section 4 is guided by mathematical principles, and gave of mathematical notion of what "topological refinement" means. We started from the desired algebraic information $H^3(\mathcal{T};\mathbb{Z})$ or $H^4(|BG|;\mathbb{Z})$, and the geometrical intuition from continuum is to facilitate the realization of the desired algebraic information. In this section, we want to reverse the emphasis. We want to begin with the geometrical picture from continuum QFT and come up with a corresponding lattice QFT, such that the algebraic information is to facilitate a suitable truncation of the geometrical details. This should lead to a systematic relation between continuum QFT and lattice QFT.

To motivate in another way, traditionally, we are familiar with the idea that a lattice QFT in the UV leads to a continuum QFT when renormalized towards the IR. However, there are also many situations—such as lattice QCD—in which we want to do the reverse, i.e. we want to find a lattice QFT that suitably describes some given continuum QFT. For TQFT, such a connection has been well-developed [19, 22, 23], simply because the UV and the IR really are not different if the QFT is topological. Now we want to explore whether such a connection can be drawn for more general QFTs with dynamical d.o.f..

At this stage, such a broader picture, extending beyond our primary goal of arriving at the constructions in Section 4, is only a sketched one. We however do believe this is a good starting point for more systematic exploration of the relation between continuum QFT and lattice QFT.

We begin with nl$\sigma$m. When we say "a field configuration" in a continuum nl$\sigma$m, we simply mean a smooth function from the spacetime manifold to the target manifold,

$$\mathcal{M} \to \mathcal{T}. \tag{144}$$

The path integral is intended to integrate over the space of all such functions. But this space is infinite dimensional and the path integral is not well-defined.

Let us ask what we intend to mean when we say "a field configuration" in a lattice nl$\sigma$m. Of course, in traditional lattice nl$\sigma$m, it is just a function from the lattice vertices, $\mathcal{L}_0$, to the target manifold $\mathcal{T}$. As we have seen in the previous sections, it is helpful to think of the lattice as being embedded in the continuum, then we can say, traditionally, a field configuration on the lattice is just a sampling of the continuum field at some discrete points on $\mathcal{M}$,

$$\mathcal{L}_0 \hookrightarrow \mathcal{M} \to \mathcal{T}. \tag{145}$$

Obviously a lot of information in the continuum field configuration is lost after the sampling.

To solve this problem, it turns out useful to not only think of $\mathcal{M}$ the manifold itself, but also its higher path spaces, which together form a higher (or infinite) groupoid. The realization is non-unique. We can use the singular simplicial complex $(\cdots \Delta_2 \mathcal{M} \overset{\to}{\underset{\to}{\to}} \Delta_1 \mathcal{M} \rightrightarrows \mathcal{M})$, or the cubical analogue $(\cdots \mathcal{P}^2 \mathcal{M} \overset{\to}{\underset{\to}{\to}} \mathcal{P}\mathcal{M} \rightrightarrows \mathcal{M})$ where $\mathcal{P}$ again means taking the path space,[145] or some notion of weak globular higher category $(\cdots \mathcal{P}_2 \mathcal{M} \rightrightarrows \mathcal{P}\mathcal{M} \rightrightarrows \mathcal{M})$ where $\mathcal{P}_n \mathcal{M} \subset \mathcal{P}^n \mathcal{M}$ is the space of interpolation of two elements of $\mathcal{P}_{n-1}\mathcal{M}$ that share boundaries in $\mathcal{P}_{n-2}\mathcal{M}$.[146]

---

[145]$\mathcal{P}^2 \mathcal{M}$ has four rather than two arrows to $\mathcal{P}\mathcal{M}$, because a path between two paths in general swipes out a square shape rather than a globular shape (which would be the case if we require the end points to be fixed—and that would become what we denoted as $\mathcal{P}_2\mathcal{M}$).

[146]This globular higher category is weak because without identification under thin homotopy, there is no identity path in the strict sense, and composition of paths is not strictly associative, and likewise for higher paths [42].

These realizations can capture the full homotopy information of $\mathcal{M}$; while for many physical applications, such as those considered in the present work which only concern the lowest non-trivial $\pi_n$, using the strict higher path groupoid $(\cdots \bar{\mathcal{P}}_2 \mathcal{M} \rightrightarrows \bar{\mathcal{P}} \mathcal{M} \rightrightarrows \mathcal{M})$ would also be sufficient.[147] Similarly for the target manifold $\mathcal{T}$. We then have the simplicial map (assuming we used the simplicial realization, but we can also use the other realizations mentioned before; same below)

$$
\begin{array}{ccc}
\cdots & \rightarrow & \cdots \\
\Downarrow\Downarrow\Downarrow\Downarrow & & \Downarrow\Downarrow\Downarrow\Downarrow \\
\Delta_2 \mathcal{M} & \rightarrow & \Delta_2 \mathcal{T} \\
\Downarrow\Downarrow\Downarrow & & \Downarrow\Downarrow\Downarrow \\
\Delta_1 \mathcal{M} & \rightarrow & \Delta_1 \mathcal{T} \\
\Downarrow\Downarrow & & \Downarrow\Downarrow \\
\mathcal{M} & \rightarrow & \mathcal{T}
\end{array}
\quad , \tag{146}
$$

induced from $\mathcal{M} \rightarrow \mathcal{T}$. This simplicial map contains exactly the same amount of information as the original function $\mathcal{M} \rightarrow \mathcal{T}$, simply because the paths, surfaces and so on are all made of points.

The reason why we make things seemingly more complicated by including the higher path spaces is so that we can make better connection to the lattice. While $\mathcal{L}_0 \hookrightarrow \mathcal{M}$ is sampling some points in the continuum and lost the interpolation information, $(\mathcal{L}_1 \rightrightarrows \mathcal{L}_0) \hookrightarrow (\Delta_1 \mathcal{M} \rightrightarrows \mathcal{M})$ is sampling some paths, hence retrieving more information about how the field interpolates from point to point. We can repeat this for higher dimensional cells (assuming the lattice is also a simplicial complex), until the $d$-dimensional cells completely fills up the continuum manifold. We obtain

$$
\begin{array}{ccccc}
\mathcal{L}_d & \hookrightarrow & \bar{\Delta}_d \mathcal{M} & \rightarrow & \bar{\Delta}_d \mathcal{T} \\
\downarrow\cdots\downarrow & & \downarrow\cdots\downarrow & & \downarrow\cdots\downarrow \\
\cdots & \hookrightarrow & \cdots & \rightarrow & \cdots \\
\Downarrow\Downarrow\Downarrow\Downarrow & & \Downarrow\Downarrow\Downarrow\Downarrow & & \Downarrow\Downarrow\Downarrow\Downarrow \\
\mathcal{L}_2 & \hookrightarrow & \Delta_2 \mathcal{M} & \rightarrow & \Delta_2 \mathcal{T} \\
\Downarrow\Downarrow\Downarrow & & \Downarrow\Downarrow\Downarrow & & \Downarrow\Downarrow\Downarrow \\
\mathcal{L}_1 & \hookrightarrow & \Delta_1 \mathcal{M} & \rightarrow & \Delta_1 \mathcal{T} \\
\Downarrow\Downarrow & & \Downarrow\Downarrow & & \Downarrow\Downarrow \\
\mathcal{L}_0 & \hookrightarrow & \mathcal{M} & \rightarrow & \mathcal{T}
\end{array}
\quad , \tag{147}
$$

where we have truncated the $\mathcal{M}$ column and the $\mathcal{T}$ column to the $d$th layer by taking identification of $d$-cells up to thin $(d+1)$-homotopy. After the truncation, the $\mathcal{L}$ column and the $\mathcal{M}$ column become ananaturally equivalent,[148] which roughly speaking means the $d$-dimensional lattice captures all the essential information of this truncated path $d$-groupoid of $\mathcal{M}$. We indeed do not expect the lattice theory to be able to capture those information that we truncated away—which, we believe, are physically unimportant anyways, as those truncated information are either unimportant UV details within each lattice cell (geometrically a tiny region), or higher homotopy information in $\mathcal{T}$ that seem not to be accessible by a $d$-dimensional QFT even in the continuum.

The above describes a functor from the lattice $\mathcal{L}$ to a target category, the $\mathcal{T}$ column, so it is almost interpretable as a lattice field configuration. Except there is one problem—the configuration is still essentially a continuum configuration, in the sense that, in general, the

---

[147] See footnote 97.

[148] Established by taking as the span the pullback of a Čech nerve over $\mathcal{M}$ (such that each patch is labeled by a lattice vertex) wtih the $\mathcal{M}$ column itself.

higher layers $\Delta_1\mathcal{T}, \Delta_2\mathcal{T}, \cdots, \bar{\Delta}_d\mathcal{T}$ in the target category are infinite dimensional spaces that came from the continuum picture (146),[149] which is undesired for a lattice theory.

What we gained is that now it becomes clear how the vague physical problem of defining a desired "topologically refined" lattice QFT should be turned into a well-posed mathematical problem:

$$
\begin{array}{ccccccc}
\mathcal{L}_d & \hookrightarrow & \bar{\Delta}_d\mathcal{M} & \to & \bar{\Delta}_d\mathcal{T} & & \mathbf{ET}_d \\
\downarrow\cdots\downarrow & & \downarrow\cdots\downarrow & & \downarrow\cdots\downarrow & & \downarrow\cdots\downarrow \\
\cdots & \hookrightarrow & \cdots & \to & \cdots & & \cdots \\
\downarrow\downarrow\downarrow\downarrow & & \downarrow\downarrow\downarrow\downarrow & & \downarrow\downarrow\downarrow\downarrow & & \downarrow\downarrow\downarrow\downarrow \\
\mathcal{L}_2 & \hookrightarrow & \Delta_2\mathcal{M} & \to & \Delta_2\mathcal{T} & \overset{\text{equiv up to}}{\underset{\text{what we care}}{\longrightarrow}} & \mathbf{ET}_2 \\
\downarrow\downarrow\downarrow & & \downarrow\downarrow\downarrow & & \downarrow\downarrow\downarrow & & \downarrow\downarrow\downarrow \\
\mathcal{L}_1 & \hookrightarrow & \Delta_1\mathcal{M} & \to & \Delta_1\mathcal{T} & & \mathbf{ET}_1 \\
\downarrow\downarrow & & \downarrow\downarrow & & \downarrow\downarrow & & \downarrow\downarrow \\
\mathcal{L}_0 & \hookrightarrow & \mathcal{M} & \to & \mathcal{T} & & \mathcal{T}
\end{array}
\qquad (148)
$$

*We want an anafunctor that reduces the third column, the simplicial path d-groupoid of $\mathcal{T}$, which in general involves infinite dimensional spaces, to an ananaturally equivalent (up to whatever topological information we care about) but finite dimensional Kan simplicial manifold* **ET***, with the objects $\mathcal{T}$ (and accordingly the identity morphisms) in the third column mapping identically to* $\mathbf{ET}_0 = \mathcal{T}$*. A topologically refined lattice nlσm field configuration is a functor (a simplicial map) from the lattice $\mathcal{L}$ to the target category* **ET***, which covers E$\mathcal{T}$, the target category of traditional lattice nlσm. Moreover, if $\mathcal{T}$ admits a global symmetry action $G \times \mathcal{T} \to \mathcal{T}$, then the action extends to an automorphism of simplicial manifold $G \times \mathbf{ET} \to \mathbf{ET}$.*[150] We make three crucial remarks:

- By "up to what we care about", we mean, if $d > n$ but we only care about up to the homotopy $n$-type of $\mathcal{T}$, then we can first further reduce the third column to a fundamental $n$-groupoid by taking identification in $\Delta_n\mathcal{T}$ under any $(n+1)$-homotopy, and then demand **ET** to only be ananaturally equivalent to this fundamental $n$-groupoid. In practice, in (48), we realize this for $n = 2$ by integrating the continuum Berry curvature over a 2d surface in $\Delta_2\mathcal{T}$. Similarly, in (54) for $n = 3$, we first further reduce the third column to a fundamental 3-groupoid by integrating the WZW curvature over a 3d surface in $\Delta_3\mathcal{T}$.

- We demand the continuum target space, i.e. the objects $\mathcal{T}$ of the third column, to map identically to $\mathbf{ET}_0 = \mathcal{T}$ because we still want to keep the ordinary vertex observables that take value in $\mathcal{T}$, acted on by the global symmetry in the ordinary way. If we do not demand this, then we will lose the dynamical information. For instance, suppose $d = 1$ and $\mathcal{T} = S^1$, we have $(\bar{\Delta}_1\mathcal{T} \rightrightarrows \mathcal{T}) = (S^1 \times \mathbb{R} \rightrightarrows S^1)$ (which is already finite dimensional and can be readily used as **ET**), but recall in (103) we said this is ananaturally equivalent to $B\mathbb{Z} = (\mathbb{Z} \rightrightarrows *)$. In the Villain model, we use $S^1 \times \mathbb{R} \rightrightarrows S^1$ as the target category, rather than $B\mathbb{Z}$ which only keeps the homotopy type, because we want to also keep the dynamics of the $S^1$ d.o.f..

- The lattice nlσm field configurations constructed according to (148) forbid topological defects, simply because the construction started from smooth field configurations in the continuum, which do not contain defects. In many situations. this is desired, if we want the lattice nlσm to represent a continuum nlσm which does not contain defect up to any accessible energy scale; by comparison, in a traditional lattice nlσm, the effects

---

[149]Except for when $\mathcal{T} = S^1$, in which case we are basically done by now.

[150]Allthough the last arrow in (148) is an anafunctor, the resulting functor from $\mathcal{L}$ to to **ET** can nonetheless be an ordinary functor, because $\mathcal{L}$ is discrete.

from defect fluctuation cannot be forbidden because the defects are not well-defined on the lattice.[151]

In other situations, we might want to include the effects of defects on the lattice (meanwhile still being able to explicitly define the defects; otherwise we can just use the traditional lattice nl$\sigma$m). To do so, we need a minimal enlargement **ET'** of **ET**, such that **ET'** contains the **ET** in (148) as a subcategory, and **ET'** is ananaturally equivalent to a trivial category; moreover, **ET'** is the smallest category that satisfies these two properties. The rationale behind these properties is the same as that explained below (103) and (104) through examples.

(Interestingly, the algebraic perspective in Section 5.5 constructs target categories that allow defects by default, and an extra step is needed if we want to forbid defects. While the geometrical perspective in this section constructs target categories that forbid defects by default, and an extra step is needed if we want to allow defects.)

This explains our basic idea of how higher category theory leads to a more systematic understanding of what it means to "discretize a continuum QFT". At this stage, the connection is only built at the level of field configurations in the path integral. In future works, it is important to also cast the path integral weight into this language.

Now we attempt to suggest a reasonable systematic relation between continuum gauge theory and lattice gauge theory. Further work is needed to complete the understanding.

In the continuum, there are two ways to think about a gauge field configuration,

$$
\begin{array}{ccc}
\bar{\mathcal{P}}\mathcal{M} & & G \\
\Downarrow & \rightarrow & \Downarrow \\
\mathcal{M} & & *
\end{array}
\qquad \text{versus} \qquad \mathcal{M} \rightarrow |BG|,
\tag{149}
$$

where the first way, shown on the left, is (101) (the arrow now represents an anafunctor), while the second way, shown on the right, makes use of the universal gauge connection on $|BG|$ [19]. The advantage of the first way is that the target category is finite dimensional and the anafunctor is readily the Wilson lines, and thus a field configuration in traditional lattice gauge theory is just a sampling[152]

$$
\begin{array}{ccccc}
\mathcal{L}_1 & \hookrightarrow & \bar{\mathcal{P}}\mathcal{M} & & G \\
\Downarrow & & \Downarrow & \longrightarrow & \Downarrow \\
\mathcal{L}_0 & \hookrightarrow & \mathcal{M} & & *
\end{array}.
\tag{150}
$$

The advantage of the second way is that a gauge theory can now be seen as a nl$\sigma$m valued in $|BG|$, so that we can connect the problem to what we already know for nl$\sigma$m, albeit there is a difference that $|BG|$ is in general infinite dimensional and the points on it are not physically observable.

If we view a continuum gauge field in the second way—which is indeed what we did at

---

[151]A recent work [97] also considered forbidding defects in a lattice nl$\sigma$m. The construction in [97] is by discretizing the target space $\mathcal{T}$ into a simplicial complex, so the target category is also a simplicial set. This way, while the homotopy type information of $\mathcal{T}$ is kept, $\mathbf{ET}_0$ is no longer $\mathcal{T}$ but only some discrete points in $\mathcal{T}$, so the local dynamics of the continuous-valued d.o.f. is lost, and moreover the original continuous global symmetry on $\mathcal{T}$ cannot act on **ET** anymore. By comparison, the target category we constructed in (148) has the ordinary $\mathcal{T}$ d.o.f., with ordinary global symmetry action.

[152]Although the functor from the path groupoid $\bar{P}\mathcal{M}$ to $BG$ is an anafunctor, the functor from the lattice to $BG$ can be an ordinary functor, because the lattice is discrete, similar to footnote 150.

the end of Section 5.5—then we are almost done. Following the reasoning of (148), we have

$$
\begin{array}{ccccccc}
\mathcal{L}_d & \hookrightarrow & \bar{\Delta}_d\mathcal{M} & \to & \bar{\Delta}_d|BG| & & \mathbf{BEG}_d \\
\downarrow\cdots\downarrow & & \downarrow\cdots\downarrow & & \downarrow\cdots\downarrow & & \downarrow\cdots\downarrow \\
\cdots & \hookrightarrow & \cdots & \to & \cdots & & \cdots \\
\downarrow\downarrow\downarrow\downarrow & & \downarrow\downarrow\downarrow\downarrow & & \downarrow\downarrow\downarrow\downarrow & & \downarrow\downarrow\downarrow\downarrow \\
\mathcal{L}_2 & \hookrightarrow & \Delta_2\mathcal{M} & \to & \Delta_2|BG| & \overset{\text{equiv up to}}{\underset{\text{what we care}}{\longrightarrow}} & \mathbf{BEG}_2 \\
\downarrow\downarrow\downarrow & & \downarrow\downarrow\downarrow & & \downarrow\downarrow\downarrow & & \downarrow\downarrow\downarrow \\
\mathcal{L}_1 & \hookrightarrow & \Delta_1\mathcal{M} & \to & \Delta_1|BG| & & G \\
\downarrow\downarrow & & \downarrow\downarrow & & \downarrow\downarrow & & \downarrow\downarrow \\
\mathcal{L}_0 & \hookrightarrow & \mathcal{M} & \to & |BG| & & *
\end{array}
\tag{151}
$$

where the desired topologically refined target category on lattice is a finite dimensional Kan simplicial manifold **BEG** that is ananaturally equivalent (up to whatever topological information we care about) to the simplicial path $d$-groupoid of $|BG|$. But instead of the $\mathbf{ET}_0 = \mathcal{T}$ condition in (148), here we require $(\mathbf{BEG}_1 \rightrightarrows \mathbf{BEG}_0) = (G \rightrightarrows *)$, which is obtained from $(\Delta_1|BG| \rightrightarrows |BG|)$ using the connection on the universal bundle. This is because, unlike $\mathcal{T}$ in actual nl$\sigma$m, $|BG|$ is already infinite dimensional in general, and moreover the points in $|BG|$ are not physical observables, only the Wilson lines are, so we want to only keep the finite dimensional Wilson line information instead of $|BG|$ itself; indeed, the target category of traditional lattice gauge theory is $BEG$, whose two lowest layers are $(G \rightrightarrows *)$, and we expect **BEG** to cover the traditional $BEG$ by refining the holonomies. Again, the target category constructed by (151) forbids topological defects; the way to include topological defects is the same as that discussed below (148).

On the other hand, it is currently unclear to us how to think about the problem of topological refinement if we directly begin with the first way in (149) of viewing a continuum gauge field. It should be a suitable notion of delooping of the target categroy of nl$\sigma$m, the rightmost column of (148). As we discussed at both the ends of Sections 4.2 and 5.3, a Yang-Baxter equation should arise in the delooping process. This implies the CS phase 3-morphisms after the delooping should come from the WZW phase 2-morphisms before the delooping, with some extra correction terms in the composition rules (interchangers) that entails a solution to the Yang-Baxter equation. In our follow-up paper [10] which construct a models with more technical details, we indeed observe that the CS phase saddle around a hypercube is automatically expressed in terms of the WZW phase of a certain 2-parameter family of Wilson loops, plus some correction terms. How to concretely interpret this observation as a delooping from WZW to CS via a solution to a Yang-Baxter equation will be an interesting and important problem for the future.

# 7 Further thoughts

This final section is for our further, scattered thoughts. We will begin with some near term problems. Then we will discuss some long term prospects.

**Numerical implementation.** Actual numerical implementation in the near future is definitely the primary aim of this paper. Our constructions in Section 4 for $S^3$ nl$\sigma$m and $SU(N)$ gauge theory on lattice serve to introduce the key concepts that allow the topological operators to become well-defined. For actual numerical implementation, a more explicit proposal is presented in the subsequent work [10] (focusing on gauge theory). We emphasize that, given the principles stated in the present paper, the detailed implementation is not unique,

and there may be better ways to practically construct the suitable (and, desirably, numerically optimized) path integral weights, especially the $W_2(e^{i\mathcal{W}}\mu^* + c.c.)$ in nl$\sigma$m and $W_3(e^{i\mathcal{C}}\nu^* + c.c.)$ in gauge theory, either through some clever analytical method, or some automated optimization program such as some form of machine learning or so. While the actual implementation takes some extra efforts, the traditional fundamental obstacle to defining topological operators on the lattice have been lifted by now with the key concepts we introduced.

Even aside of the purpose of explicitly defining the topological operators, it is still interesting to compare our construction to the traditional lattice QFT. In traditional lattice QFT, in order to better converge to the continuum limit, *Symanizk improvement* has been introduced [9, 57–61]. Roughly speaking, the Symanzik improvement introduced extra tuning parameters by going beyond nearest neighbor coupling; for gauge theory, this means to consider the gauge holonomy around more than one plaquette. By constrast, even without going beyond nearest neighbor coupling, our topological refinement introduces extra tuning parameters by weighing the higher morphisms in the target category, which roughly represent the interpolations of fields if we think of the lattice as being embedded in the continuum. It seems the extra weights introduced in the latter way are physically better interpretable. For the simplest example, consider the vortex fugacity weight introduced in the Villainized $S^1$ nl$\sigma$m (13), which obviously controls the likelihood of vortices; this is important for setting up the renormalization analysis for the BKT transition [14,56] (we will discuss more about renormalization later). Moreover, summing over the Villain integer variable $m_l$ with non-trivial vortex fugacity weight will indeed generate beyond-nearest neighbor couplings between the traditional $S^1$ variables $e^{i\theta_v}$ (compared to (7) when the vortex fugacity weight is trivial), although the result cannot be expressed analytically. Similarly, integrating out the Berry connection field (along with its Dirac string field) with non-trivial Maxwell weight in the spinon-decomposed $S^2$ nl$\sigma$m (39) will generate beyond-nearest neighbor coupling between the traditional $S^2$ variables. Based on this, we expect that, in general, the higher morphism weights from the topological refinement will (at least partly) play the role of Symanzik improvement, in a physically more interpretable manner; and since the topological operators are explicitly controlled, it is interesting to understand whether there is a relation to the numerical problem of topological freezing. These problems are in their own right worthwhile to be studied numerically.

**Mathematical establishment.** The mathematical context of our models is explained in Section 5.5 and Section 6. As we said there, we expect it to be straightforward to make the "refined target category" of lattice nl$\sigma$m a mathematically established concept, given the previous mathematical literature on constructing multiplicative bundle gerbes [96] which is closely related to what we need—see footnote 133. On the other hand, the counterpart for gauge theory (as well as its relation to nl$\sigma$m via delooping and Yang-Baxter equation) may take a little more efforts, partly because the relevant technical details we need are so far only found in the physics literature [8, 10, 100] (as opposed to the mathematical, and more precisely category theoretical, literature) as far as we are aware of; but we expect there to be no intrinsic difficulty.

**Generalizations.** There are some directions of generalization that worth working out.

1. Throughout this paper we have only been interested in those topological operators that are captured by the lowest non-trivial $\pi_n$, for $n \leq 3$. We should also consider cases with multiple types of topological operators of interest, captured by several non-trivial $\pi_n$'s, since they might have non-trivial interplay. Physically relevant examples include $S^2$ nl$\sigma$m with both $\pi_2$ and $\pi_3$ in consideration [121,122] (rather than just $\pi_2$ in Section 2.4), and $\mathbb{R}P^2$ nl$\sigma$m with $\pi_1, \pi_2$ and $\pi_3$ in consideration (rather than just $\pi_1$ in Section 2.3). We will study these examples in subsequent works. For gauge theories, it is also im-

portant to consider non-abelian gauge groups such as $O(N)$ that have non-trivial $\pi_0, \pi_1$ before $\pi_3$ [19], and for these cases the general multiplicative bundle gerbes constructed in [116, 117] will be useful.

2. Throughout this paper our examples are either pure nl$\sigma$m or pure gauge theory. We should also consider the topological operators when we couple lattice nl$\sigma$m to lattice gauge field (background or dynamical), especially for those constructed in Section 4. As mentioned there, this is in particular important for manifesting the anomalies on lattice. (In Section 2, the anomalies in the known lattices models have been manifested, with details presented in the footnotes 12 and 33.)

3. Constructions for $\pi_n$ topological operators for $n > 3$ seem to require some further efforts on the mathematical side. $\pi_5$ is particularly physically relevant for the 4d WZW term in the low energy nl$\sigma$m of QCD [98, 99] (and also $\pi_4$ if the nl$\sigma$m is the pion $S^3$). And there are other examples in strongly coupled theories in both high energy physics and condensed matter physics.

**More general observables and representations of weak higher groups.** Consider our topologically refined $SU(N)$ lattice gauge theory for example. At the end of Section 5.5, we explained that the target category is a weak Lie 4-group, realized as a Kan simplicial manifold with single object. Mathematically, there should exist a suitable notion of "representations of the weak Lie 4-group", which should be worked out explicitly.

Physically, this corresponds to answering the following question. Suppose the Yang-Mills theory lives on a spacetime of dimension $d \geq 4$. We know there is a class of observables living on 1d submanifolds, the Wilson lines, characterized by representations of $G$, where $G$ is the 1-morphisms of the Lie 4-group. There is a class of observables living on (oriented) 3d submanifolds, the CS terms, characterized by the integer CS levels, which are representations of $U(1)$, where $U(1)$ is the new d.o.f. in the 3-morphisms of the Lie 4-group. There is a class of observables living on (oriented) 4d submanifolds, the topological theta terms, characterized by the theta angles, which are representations of $\mathbb{Z}$, where $\mathbb{Z}$ is the new d.o.f. in the 4-morphisms of the 4-group. But can we also characterize some observables living on 2d submanifolds? The new d.o.f. in the 2-morphisms of the weak 4-group do not form a group in the ordinary sense, so they do not have representation in the ordinary sense, but since the whole structure forms a weak 4-group, it is reasonable to anticipate that we can organize observables living on submanifolds from 1d to 4d into some notion of representation of the weak 4-group.

Similarly, we should also ask, for a nl$\sigma$m that lives in $d \geq 3$, on 0d submainifolds there are the order parameters, on 2d submanifolds there are WZW levels, on 3d submanifolds there are topological theta terms, then how shall we characterize some observables living on 1d submanifolds, so that all these observables together form a coherent categorical structure?

**Hamiltonian formalism.** It is natural to ask if the topologically refined lattice constructions we introduced on the Euclidean spacetime lattice have corresponding versions on the spatial lattice in the Hamiltonian formalism. While we expect there to be, it takes further efforts to work out the details. In particular, for ordinary group valued operators, their canonical operators are characterized by the representations, so now the weak higher group representation problem described above might become particularly relevant. That is, mathematically, we want a suitable notion of "representation" that allows us to do harmonic analysis on weak higher group, so that we can turn Lagrangian into Hamiltonian.

There is an extra issue to be noted as the d.o.f. of interest are continuous-valued—even when the d.o.f. are ordinary groups, such as in Villainization. We emphasized that the d.o.f. in the target category in general do not factorize; on the other hand, a lot of times in the Hamiltonian formalism it is desired that the physical Hilbert space factorizes locally on the spatial

lattice. If we indeed demand so, there is a familiar treatment when the d.o.f. are discrete-valued [21, 22]: We can let the physical Hilbert space be an enlarged, locally factorized one, and then have energy penalty terms in the Hamiltonian, such that a low energy subspace is exactly the desired, non-factorized Hilbert space, and moreover all higher energy states have a finite gap above this low energy subspace. However, when the d.o.f. are continuous-valued, under the same treatment there is no such gap because the energies of the states vary continuously (unless we use a Hamiltonian with discontinuous matrix elements, but such unphysical treatment will lead to other problems). Suitably modified treatment has been developed in the case of Villainized $U(1)$ gauge theory [79, 80], in order to ensure the emergence of a low energy subspace with the desired non-factorized properties, meanwhile having a finite gap separated from the higher energy states. We expect similar issue occurs for more general target categories, if we want a locally factorized physical Hilbert space.

The above are more or less well-defined problems that we believed can be solved in the forseeable future. In the below, we sketch some directions that we believe worth explorations for the long term. The discussions below are highly speculative at this point.

**Renormalization.** As we have seen, a field configuration in a lattice QFT is a functor from the lattice to a target category, where the latter is constructed based on the target space of the desired continuum QFT, either from the more algebraic perspective described in Section 5.5, or the more geometric perspective described in Section 6. The path integral is to integrate over the space of all such functors; at least in the examples that we have seen, the measure to use for the integral is obvious, due to the global symmetry or gauge group. On the other hand, the integrand, i.e. the path integral weight, still awaits to be casted in a categorical language. Of course, the weight is a suitably constructed map from the space of field configuration (which are functors) to the non-zero complex numbers $\mathbb{C}_*$. But what is meant by "suitable" needs to be clarified.

Locality is a crucial requirement. In the constructions we presented, the weight is a product of factors contributed by individual vertices, links, plaquettes, and so on, therefore a map from the space of $n$-morphisms of the target category, for each $n$, to $\mathbb{C}_*$ is involved. But more general weights are also legitimate—those short ranged but beyond nearest neighbor couplings (we have mentioned this when discussing numerical implementation at the beginning of this section). So we need to find a concise way to convey the requirement of locality in the weight assignment.

Another layer of the problem is that there are two kinds of weight contributions: the "non-topological" ones which contributes a positive magnitude, and the "topological" ones which contributes a $U(1)$ phase, such as the topological theta terms, Berry phase, WZW phase, and CS phase. As required by reflection positivity [123, 124]—the Euclidean version of unitarity—under orientation reversal of the spacetime,[153] the positive magnitude contributions must be invariant, while the phase contributions must become complex conjugation. But there are further distinctions between the two kinds of contributions: The "non-topological" weights seem to be locally well-defined "outright", whilst the "topological" weights (such as the Berry phase, WZW phase, CS phase) may not be well-defined on individual lattice cells or, more generally, regions with boundaries, in the sense that there will be dependence on some notion of "gauge" on the boundary conditions, and related to this the $U(1)$ phase contribution from such a region in general takes value from a non-trivial $U(1)$ bundle over the space of boundary conditions. (For $U(1)$ CS-Maxwell, see [71] for details.) We need a concise way to capture these aspects into a complete categorical definition of lattice QFT with generic dynamics.

---

[153]It is understood that, if there are extra background structures on the spacetime, such as background gauge field, branching structure, etc., involved in defining the theory, these background structures are also transformed under the orientation reversal.

Suppose the above can be achieved in the foreseeable future. Then we can try to formulate renormalization in the categorical language. It can envisioned that there should be a category of lattice QFTs, whose objects contain information about the (topologically refined) target category and the path integral weight assignment. The coarse graining of lattice can certainly be realized in terms of inclusion functors between lattices (with the IR limit being some notion of skeletal lattice). And we want the coarse graining inclusion functor, as morphisms in the category of lattices (discrete Kan complexes), to induce certain "renormalization morphisms" in the category of lattice QFTs.

Perhaps a better way to realize the coarse graining of lattice is by general anafunctor, rather than ordinary inclusion functor, despite that according to Section 5.2 there seem to be no necessity to use anafunctor when dealing with discrete spaces. The reason is, over the past two decades, it has become increasingly clear that a good way to think about renormalization is to think about an $\mathrm{AdS}_{d+1}$ spacetime, with the extra "radial direction" representing the renormalization scale, and such ideas should apply to lattice as well [125]. Then, by (96), naturally the lattice links, plaquettes and so on connecting two consecutive radial slices constitute the span of the anafunctor for one step of coarse graining. (It is furthermore illuminating to think of the lattice $\mathrm{AdS}_{d+1}$ as a double category—recall footnote 93.) Then, the problem becomes how this perspective of coarse graining a lattice is lifted to the level of renormalizing a lattice QFT.

**Relation to categories involved in topological quantum field theory.**  In the long term, if we have a good categorical understanding of what renormalization is, we can then discuss what a renormalization fixed point means. Hopefully, we can see how the familiar categorical description of the IR fixed point emerges from a description of generic QFT after renormalization.

Nonetheless, even at the present stage, it may be a good idea to begin pondering the difference between the categories familiar in the TQFT context versus the categories involved in the present work for QFT with generic dynamics. The categories that familiarly describe TQFT are equipped with a long list of extra structures and requirements (see e.g. [109]), in order to reproduce all the desired nice physical properties that an IR fixed point should have [22]; moreover, the systematically well-studied ones involve discrete-valued d.o.f. only, while continuous-valued d.o.f. still pose a crucial challenge. In comparison, the categories we used in the present work are much "simpler" in certain aspects of the definition, with less structures and requirements; moreover, they can, and are primarily designed to, describe continuous-valued d.o.f. with homotopy properties that are of interest.

We would like to particularly remark on the difference that, the categories used in TQFT are $\mathbb{C}$-linear enriched, which means the morphisms between any two objects form a $\mathbb{C}$ linear space, so that the objects carry Hilbert spaces and allow superpositions; more generally, $\mathbb{C}$ linear space structure of the $n$-morphisms (usually interpreted as evolution in $n$-dimensional spacetime) allow the lower morphisms (usually describe particle excitations, line excitations, domains walls, etc.) to carry Hilbert space and allow superpositions [109]. Meanwhile, for the categories we used in this paper, we did not introduce a built-in linear structure, but we know our constructions do have the quantum mechanical linearity, simply because, in the end, we are constructing well-defined path integrals. Let us try to ponder about this important distinction.

Having a build-in linear structure in the former case is both more imminent needed and more convenient. Because for TQFT there is no actual distinction between the UV and the IR, a same categorical structure is to be used to describe both the UV d.o.f. (say, in constructing a lattice model) and the IR states, and we usually want to be imminently able to talk about the superposition of the IR states. And since the theory is topological, there are only a few actions we can perform on the states (such as the fusion and braiding of anyons) and they will

only generate superimposed states with a *fixed* set of coefficients (such as the Clebsch-Gordon coeffcients or the components of the F-, R-symbols), so having a built-in linear structure is convenient for incorporating all these structures all at once.

On the other hand, in the latter case, we are introducing the UV d.o.f. to a generic QFT, and they *a priori* do not have to have any direct correspondence with the IR states, especially in the strongly interacting situations of interest. We believe it should be viable to carefully construct the UV Hilbert space by looking at the square integrable complex functions in the continuous-valued variables on each spatial lattice cell, so to talk about UV states, and this should probably be done in the near future, as part of the Hamiltonian formalism program mentioned before. But most often this does not directly help with whatever physics that we want to understand. And in a dynamical QFT, superpositions will happen on UV states with arbitrary coefficients depending on the details of the UV dynamics, so unlike the aforementioned TQFT case, there does not seem to be a nice fixed structure awaiting to be incorporated all at once just by building-in a linear structure in the categorical structure that describe the UV d.o.f..

That said, we can still observe some important roles, played by the built-in linear structure in the categories that describe TQFT, being fulfilled in other ways in our present constructions. In particular, let us consider the notion of fusion in a fusion category. There, the linear structure allows one to have the notion of semi-simplicity: there is a set of "simple objects" (interpreted as the states of simple anyons in 3d TQFT, which are more naturally 2-morphisms as we said in Section 5.3, and to call them "objects" we have essentially looped twice) and we can take their direct sums, such that every object is some finite direct sum of simple objects; when we fuse two simple objects, the result is in general a non-simple object, a key feature of non-abelian topological order. How do we reproduce this if we do not build-in linear structure? The answer is to simply phrase the above is a plainer language—what we usually say is, when two simple anyons fuse, there can be multiple possible fusion channels, giving rise to multiple possible results of simple anyon; there is no mention of non-simple anyon. But this is literally what a simplicial set does. That is, the role of the linear space in the fusion process can be played by the non-unique composition in simplicial sets.

Interestingly, if we really want to, we can still catch some reminiscence of the linear enrichment. Recall we said in Section 5.5 (see footnote 140 in particular) that when applying (136) to discrete $BG$ for Dijkgraaf-Witten theory, the $\Lambda$ in the span has an independent $U(1)$ d.o.f.—which is often ignored in the lattice models. We can recognize this $U(1)$ in the fusion category language as the phase of the $\mathbb{C}$ in the linear enrichment.

What we have discussed here is just one aspect of the full problem. We chose to discuss this aspect because, at least at the technical level, a difficulty of generalizing the notion of fusion category to include cases with continuous d.o.f. is the loss of semi-simplicity. There are many more problems to be explored. Through all these discussion that we had, we should anticipate that there will likely be a route towards a future unification of the use of categories in QFT, regardless of whether the QFT is IR TQFT or generic dynamical QFT, and whether the d.o.f. are discrete or continuous-valued.

**Constructive quantum field theory.**   An ultimate question about a lattice QFT is whether some suitable notion of continuum limit exists. Numerically there are good evidences for the convergence in lattice QCD, but one may wonder whether this can be shown analytically. In fact, this is one possible route towards the program of *constructive QFT*, i.e. towards constructively defining what a continuum QFT is. This route has some crucial advantages compared to other possible routes, aside from being more intuitive: Most importantly, reflection positivity and locality (see our discussion about renormalization above) are built-in as long as the lattice QFT itself is a legitimate path integral; moreover, if dynamical gauge field is involved, the gauge redundancy (for compact Lie group) requires literally no treatment at the fundamental

level [1] (though if one wants to one can still fix the gauge).

Remarkable partial results have been achieved by Balaban in this regard. Through highly technical analyses, Balaban showed that, in 3d [126] and 4d [127, 128], given a finite size three/four-torus Euclidean spacetime, as the lattice spacing decreases towards zero, Wilson's lattice Yang-Mills theory [1] is renormalized such that the value of the partition function remains stable within a finite bound. This program is, unfortunately, almost not being carried on since then, perhaps due to its highly involved technicality.

It is natural to ask how our topological refinement of Wilson's traditional lattice gauge theory affects the analyses in this program. This is a technically very difficulty yet important question in the long term.

Since the topological refinement introduces new higher morphisms d.o.f. on the lattice and new weight factors for them, the renormalization flow is affected. The optimistic hope is, now that the topologically refined lattice QFT has a more systematic relation to the desired continuum QFT (Section 6), and moreover the non-perturbative topological operators such as instantons have become well-defined and explicitly controllable in the path integral, the renormalization towards the continuum may also be under better control.

Another optimistic hope is, now that we have a categorical understanding (at a preliminary level for now) of what a lattice QFT is in relation to a desired continuum QFT, and in the future such an understanding may hopefully be extended to cover renormalization, eventually we may hope for a reorganization of the currently highly involved analyses towards the continuum limit. Even though the essential technicality most likely will not be eliminated, a more systematic reorganization, if possible, may help with the progress on the analyses and the physical understanding of it.

## Acknowledgments

The lattice QCD instanton problem was first brought to the author by Dam Than Son and Mikhail Stephanov, and by David Kaplan in a separate occasion. The author is grateful to Qing-Rui Wang for pointing towards the studies on string groups at an early stage of this research, and thanks Zhen Huan, Shi-Yun Liu and Ze-An Xu for regular discussions about category theory and its applications in physics. The maintainers of the online wiki nLab on higher category theory are greatly appreciated.

**Funding information** This work is supported by NSFC under Grants No. 12174213, No. 12342501 and No. 12447104.

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
