# Peer review of "Instanton Density Operator in Lattice QCD from Higher Category Theory"

_SciPost Physics, doi:SciPost Phys. 19, 158 (2025)_

## Round 1 · Referee Report · Anonymous (Referee 1) · 2024-8-28

Strengths

1- intriguing motivation 2- detailed motivation and review of background 3- potential high impact, potential high novelty 4- potentially relevant sophisticated mathematical tools identified correctly

Weaknesses

1- key definitions remain hard to parse 2-consistency checks of key definitions seem lacking 3-relation between key sections 4 and 5 remains vague

Report

+++ General statement +++

The author's motivation and broad proposal is in intriguing, and success of this ambitious approach would have high impact. From the introductory sections, I was highly motivated to see the proposed solution.

However, the main section 4 still is largely motivational in style and the key fine-print of the proposed definitions remains unclear to me after spending some time mulling over it.

I would urge the author to restate the key definitions (64) and (71) in something closer to math style, where every ingredient is explicitly declared. This concerns particularly the arguments of the key term $W_2(..)$ and $W_3(...)$, respectively. Define them, concretely.

Morever, consistency checks are missing: After introducing all the new fields, there should be an argument that the resulting lattice models are indeed locally still suitably equivalent to the ordinary ones which they are meant to enhance. This seems far from obvious and needs an argument. (Note that a broad appeal to Elitzur's theorem advertized in the introduction has little bearing on this, as it does not concern the higher gauge fields nor the various constraints introduced by the authors)

Section 5 is a review of sophisticated mathematics that the authors plausibly argues to be necessary for coming up with these definitions, and it may serve a purpose as exposing some of this math to a math-remote pure physics community -- but its actual relation to the proposal in section 4 is left quite unclear.

It may be worth going for a much shorter article which just defines the proposed lattice models concretely. If the category/homotopy theory helps with establishing their intended behaviour then add that proof explicitly, but if the category/homotopy theory is just a vague motivation for the author, then leave it away and instead focus on tangible results.

+++ List of comments +++

Here is a list of random comments going linearly through the document.

general comment:
The text speaks throughout about "higher category theory", but what it really uses is just "higher groupoid theory" also known as (simplicial-)homotopy theory. This conflation is common in the literature, but it may be worthwhile to beware of it.In simplicial homotopy theory, the notion of "anafunctors" is much further developed, known as maps out of cofibration resolutions.

p. 4 "theoretical appeal[s]"

p. 4: appeal to Elitzur's theorem:
However, Elitzur's theorem applies to ordinary gauge transformation, while the lattice models that the author is about to introduce have (also) higher gauge symmetries. It is maybe not a priori clear that the analog of Elitzur's theorem will still apply to these.

p. 9 "only [a] sketched"

p. 9 "that we care [about]"

p. 12 "in the below"
either: "below" or "in the following"

p. 12 "we do not say ... is constant"
the mathematical term is: it is *locally constant*

p. 13: 
Figure below (7): It is hard to see from the figure on the right what is being illustrated.

p. 21 equation (18):
probably "f_c" and "s_c" should be "f_p" and "s_p"

p. 39: "at the 2d boundary between two patches"
What must be meant instead is "the is 2d overlap of two patches"

p. 39, 40: "Stoke's " must be "Stokes' "

p. 41 Section 4.1

Just to note that only now, over 40 pages into the text, does the first promised definition begin to slowly emerge. This lack of conciseness not only puts a burden on the reader at this point but may also not be helpful for the author's own thought development.

At some point the motivational commentarey must be set aside and an actual, concrete detailed definition must be written down and checked to make sense.

p. 41 "To understand why..."
To understand this one may simply and immediately observe that the fiber over +1 or -1 is a singleton but the fiber over any other point is a set of two elements.

p. 42 "What this extra S^2 does... will be explained later."
Why not just say where it will be explained.

p. 42: "Note that while m_l is a two-valued label it by no means for a Z_2 group"
The technical term for this is that m_l is an element of a *torsor* over Z_2.
And the words "by no means" are misleading: A simple means makes any torsor a group: namely the choice of any one element.

p. 42: "each patch of Y now has an open boundary"
Strictly speaking, since the components of Y are open balls (as opposed to closed balls), they do not have any boundary in the technical sense, much less an open boundary.

p. 43 "the contributions from all other continuum paths"
This does not seem to be true: There are paths that stay neither entirely in the patch SU(2)\{+1} not entirely in the patch SU(2)\{-1}.

This is the beginning of me feeling increasingly unsure about the definition that is incrementally being sketched here.

p. 43 "pick a representative path"
It is unclear why to pick any representative paths at all.

p. 46 "it is easy to picture the following desired properties for mu"

I find the logic now hard to follow. Easier than incrementally motivating a definition would be to just state it and then discuss that it satisfies desired properties.

It looks like a definition of the argument of W_2 in (59) is being indicated. And this is indeed needed to make of the key claim (64) to follow. But what exacty the definition of (59) is I am not sure from reading the text. This ought to be clarified.

p. 50 equation (64)
This seems to be the statement of the first main new proposal of the article -- it might want to be highlighted as such.
I am left wondering to know that definition (64) is consistent. Apart from it reproducing the intended topological charges -- which the author has motivated but maybe not proven -- one also needs to check backwards-compatibility, namely that all the new degrees of freedom added (notably the hat-n_l) do not locally change the intended dynamics of the sigma model.
This may be clear to the author, but it is far from clear to this referee at this point.

p. 54 equation (71)
This seems to be the statement of the second main new proposal of the article -- it might want to be highlighted as such.
The same comments apply as to the analogous statement (64) on p. 50 above, only more urgently so, since the actual definition of the term W_3(...) now is even less clear to me than the previous W_2(term).
Also, reading ahead I get the impression that discussion around (126) on p. 106 is meant to be relevant here in giving this definition of (64). If so, this ought to be said. If not, it needs to be said what else (126) is about.

p. 52: "Recall in the case... patches were chosen to be invariant under conjugation..."
Better to give an explicit equation number from which the reader can specifically recall this, since it is not easy to find.

p. 53: 
At this point I'd like to see a concise definition of the construction. The discussion of the ingredients has been spread out over many pages -- which may be good for motivation --- but it makes it hard to know at a glance what all the symbols mean.
Maybe one could point ahead for such a definition, to around (126)

p. 53, equation (71): 
The first integral sign seems to be lacking its "dg"

p. 71, item 2:This is maybe the first point that the notation "BEG" appears, which is used again many pages later (pp. 106) without (re-definition)
Clarification is needed for what is meant, and also for this choice of notation.
Note that usually, "EG" denotes the universal G-bundle, or else the simplicial complex 
 (WG)_n = G_n x G_{n-1} x ... x G_0
This happens to be a simplicial group (arXiv:1204.4886) hence has a further delooping via a bi-simplicial construction, which would deserve to be called B E G.But this does not appear to be what the author is after here.
For one, EG has contractible homotopy type, and hence so does its delooping, which would make it unsuitable for the author's purpose.

p. 106, equation (126): 
Best to (re-)state the definition of "Y" 
an open cover of G?
a more general surjective submersion of G?
subject to which conditions?

This is crucially important now to make sense of the discussion, and leaving it unclear casts doubt on the whole edifice.

Requested changes

1- state precisely the actual definitions of the lattice models 2-add consistency checks that these definitions are indeed backwards compatible with the ordinary ones. 3-clarify the claimed relation of the lattice models to homotopy theory and gerbes

Recommendation

Ask for major revision

  • validity: low
  • significance: high
  • originality: high
  • clarity: low
  • formatting: good
  • grammar: acceptable

Author:  Jing-Yuan Chen  on 2025-02-18  [id 5228]

(in reply to Report 1 on 2024-08-28)

I would like to thank Referee 2 for their recognition of the originality and the potential impact of this work. Regarding their major concerns of weakness:

1 & 2: In the present paper, the construction of the new models, especially that for the refined Yang-Mills theory in Section 4.2, was indeed only a sketch. In the original version, I said more details would be contained in a follow-up paper. The follow-up paper is now completed as arXiv:2411.07195, which will be cited in the updated version of the current paper. The follow-up paper is self-contained for an audience who only care about the model and its physical intuition but not the overall big picture and mathematics, in complimentary to the present paper.

The reason why I decided to present the details in a separate paper is that the detailed construction itself is highly technical and takes tens of pages to explain. On the other hand, this is just one particular construction---there is no single "canonical construction", only the principles are crucial. My prospect is that in the future, numerics will help us find better optimized constructions, different from the one being described in 2411.07195, but still following the same principles. So I decided that the present paper should focus on the principles, while the highly technical details of one particular construction is left to a separate paper 2411.07195.

3 & 2: I will update the version so that at the beginning of Section 5, the relation to Section 4 is more clearly stated: The category theory description is not only motivational; more crucially, we kept saying our motivation is to ``capture the $\pi_3$ physics into the refined model", and what that really means can be given a precise mathematical meaning. Having such a precise mathematical principle for ``refining a lattice model to capture the topology", I believe, is very important for the development of the field. In the updated version, I will split the original Section 5.4 into 5.4 and 5.5, and in the new Section 5.5 state a more exact mathematical principle for what this ``refinement that captures topology onto lattice" means.

Before further replying to the detailed comments, I would also like to respond to the general comment that the writing of this paper is in a largely motivational style. This is indeed true and intentional. A relavant reality in the community of theoretical physics is, while some groups of people are excited about category theory, some other groups are skeptical about its usefulness in the so-called "real physics" (in the sense of describing the real physical world). Since this work is to solve a physics problem using category theory in a necessary manner, I believe it is important to convince a broad audience that the appearance of category theory here is natural, not being anything fancy or artificial. More particularly, there are ideas that used to be familiar to different groups: lattice QCD, condensed matter, TQFT and homotopy theory, and the motivational style serves the purpose of illustrating that if one digs deeper into these existing ideas, they really do come to a confluence in terms of category theory, leading to a natural and useful picture. I hope this motivational style can help with bringing in more consensuses to our community. And, finally, again, a detailed technical description is separately presented in the follow-up 2411.07195, serving the complimentary purpose.

Now I reply to the more detailed comments:

-- Typos will be corrected and various unclear or inaccurate explanations will be improved per the referee's suggestions. I would like to thank the referee for pointing them out.

-- Indeed, the higher categories involved are higher (simplicial) groupoids that appear in homotopy theory, since homotopy is indeed what we want to capture in physics. Should we use the general term "anafunctor", or more field-specific names such as bibundle, H-S morphism, (and as the referee bring up) cofibration resolution etc? My inclination is to use the more general name, because I suppose category theory might be new to a certain proportion of the audience, and I want to show the natural progression of function-->functor-->anafunctor as generally useful concepts without bringing up too much specific knowledge which might bring barriers to readers. In the end, unifying seemingly different concepts in different branches of mathematics is what category theory is good for. That said, in a footnote, I did mention that anafunctor in specific fields has specific names; in the update I will add "cofibration resolution" brought up by the referee, since it is particularly relevant.

-- Those (generalized) higher gauge redundancies in this paper are actually dealt with in a gauge fixed manner. In the Villain model, we discussed why we can fix e.g. $\theta_v\in (-\pi, \pi]$ instead of $\mathbb{R}$, in the spinon decompostion we lifted $S^2$ to $SU(2)$ in certain gauge convention based on the north pole, and likewise for our new constructions. In principle one could also use arbitrary gauge, and the redundancy will be manifestly local. But in this paper we just fixed the (generalized) higher gauge, and each time the independence of the choice of gauge has been explained in the text.

-- The m_l label in the refined non-linear sigma model is not only not a $Z_2$ group, nor is it naturally a $Z_2$ torsor, because the $W_1$ weight for the two choices of $m_l$ are not symmetric under exchange, i.e. there is no natural $Z_2$ symmetry acting on these two labels. Also, topologically, one could have in principle used more patches, which obviously then does not admit a $Z_2$ action. I am not aware of mathematical literature referring to the choice of patches that we described as naturally forming a torsor.

-- When considering "the contributions from all other continuum paths", it is unnecessary for the entirety of the path to stay within one patch. This is just a schematic way of understanding what the $m_l$ label represents, just like in Kadanoff's block spin procedure there is some choice for what a renormalized spin represents.

-- I thank the referee for stressing on this important problem: Do those new d.o.f. added along with the new weights back-react on the dynamics and ruins the ``original theory''?
The same question could have been asked to the Villain model (after the vortex fugacity weight is introduced), so in the update I will add a paragraph elaborating on this in Section 2.1, and re-emphasize this point in the later sections. In the Villain model, which has been well-studied for decades, it is well-known that adding a suitable vortex fugacity term weighing the new d.o.f. actually facilities the renormalization convergence of the theory, rather than ruins the theory (see e.g. the cited Kogut's review paper). The physical idea is that, even if one does not include those higher d.o.f. and higher weights to begin with, along the coarse-graining procedure, those effects will be effectively generated anyways (since those higher d.o.f. are to capture how the fields interpolate, which is like the reminiscence of the d.o.f. on a finer lattice), so having those higher d.o.f. and weights in the action actually helps keeping track such effects along the renormalization flow. This has been analyzed in the Villain model. Since the new constructions are based on the same physical idea, it is physically intuitive to expect the same advantage (although the reasonable range of choices for the new weights should be determined numerically). Discussions along this line appeared in the final section in the original version, but in the updated version I will have the discussion throughout the text.

-- I thank the referee for bringing up the important issue that, given that EX and (when X=|G|) its delooping BEG are naturally equivalent to the trivial category, why the theories can still have non-trivial topology.
In the original version (mainly Section 5.2), I have explained the physical reason for this: The path integral weight in a dynamical QFT does not respect natural transformation, and this is a crucial difference with TQFT. So one cannot conclude the triviality of the theory just based on the triviality of EX or BEG.
While this explanation is correct, I myself also found this not satisfactory enough, both in terms of the way it was written and in terms of the sharpness of the statement. So in the updated version I will greatly improve the explanation on this point:
First, I will move certain discussions in the original Section 5.2 to Section 5.1 where EX and BEG were first introduced, emphasizing the physical reason mentioned above, and emphasizing the crucial difference between dynamical QFT vs TQFT (where the category theory foundation of the latter is much more familiar to certain groups of readers).
Second, in the new Section 5.5 (which used to be the second half of the original Section 5.4), I will make it clear that while EX is trivial, what we are looking at is the inclusion of X into EX, because the path integral weight is able to distinguish a locally constant configuration from a non-constant one; likewise, while BEG is trivial, what we are looking at is the inclusion of BG into BEG, because the path integral weight is able to discern a flat connection from a non-flat one. This will lead to relative cohomology classification. This is a substantial improvement in the scientific content of the updated version.

---

## Round 1 · Referee Report · Anonymous (Referee 2) · 2024-10-4

Strengths

  1. Well-written motivation.
  2. Detailed review of the status of the problem.
  3. Novel math tools.

Weaknesses

  1. Lack of clear statement of results.
  2. The validity of the proposal is not tested.

Report

A well-known problem is that the most popular lattice discretization of gauge theories and sigma-models do not respect the topology of these models. In the continuum the space of classical configurations has several topologically distinct components, on the lattice the space of configurations is connected. This disagreement is supposed to be resolved only after the continuum limit is taken. It would be desirable to have a lattice discretization which correctly reflects the expected topological classification before taking such a limit. For some models such discretizations exist, but the most interesting case (non-abelian Yang-Mills in 4d) remains open. The paper under review proposes a lattice discretization of 4d Yang-Mills with a microscopically defined instanton number, as well as other related models (sigma-model with target S^3). The proposal is quite complicated and is not clear to the reviewer whether it "works". I am not even sure what this would mean. The only way to test the proposal, as far as I can see, is to perform numerical simulations of the model and verify that it behaves qualitatively the same as the usual discretization of 4d Yang-Mills. Section 5 provides a sort of mathematical motivation for the proposal, but it is really hard for me to see the connection between these abstract considerations and the concrete proposal in Section 4. I would also add that the proposal itself is spread over many pages. It would be a good idea to formulate it more concisely.

Requested changes

  1. Formulate the proposed discretization of 4d Yang-Mills clearly and concisely.
  2. If possible, provide some tests of the proposal.
  3. The mathematical part of the paper does not really help to understand the proposal. While it informs the author's thinking, it does not make the paper more convincing and just adds to its length. I suggest dropping it altogether.

Recommendation

Ask for major revision

  • validity: ok
  • significance: good
  • originality: high
  • clarity: ok
  • formatting: excellent
  • grammar: good

Author:  Jing-Yuan Chen  on 2025-02-18  [id 5230]

(in reply to Report 2 on 2024-10-04)

The main purpose of this work, as the referee points out, is indeed that the lattice discretization should correctly reflect the expected topological classification before taking a continuum limit---this serves both the fundamental purpose (since we want a lattice QFT with good enough properties) and the practical purpose (since in practice there is always a finite lattice length) of lattice QFT.

The first main concern of the referee is that the model construction is not explicit enough in this paper. Indeed, the purpose of this paper is to introduce the big picture, the principles and the mathematics required to solve this problem. A follow-up work, which was mentioned in the original version, has now appeared as arXiv:2411.07195, in which a more technical explicit construction is given. The follow-up paper is self-contained for an audience who only care about the model and its physical intuition but not the overall big picture and mathematics, in complimentary to the present paper. Perhaps the follow-up paper 2411.07195 is closer to what the referee has in mind, according to the report.

I will then take this opportunity to explain why I decided to have the present paper and the follow-up paper 2411.07195 serving such complimentary roles. First of all, the detailed model construction itself is highly technical and takes tens of pages to explain. On the other hand, this is just one particular construction---there is no single "canonical construction", only the principles (see below) are crucial. My prospect is that in the future, numerics will help us find better optimized constructions, different from the one being described in 2411.07195, but still following the same principles. So I decided that it is important to have a first paper (i.e. the present paper) stating the principles (see below) which are correct in a model-independent manner, and sketch the model construction, and then have a follow-up paper (i.e. the 2411.07195) presenting the highly technical details of one particular model construction.

A second key question brought up by the referee is, what does it mean for our proposal "to work"? There are two layers of requirements for our proposal "to work". Along the way we will respond to the referee's third question, on why the mathematical Sections 5&6 must not be dropped.

The first layer is to serve our motivation of (in the referee's terms) "having a lattice discretization which correctly reflects the expected topological classification before taking the continuum limit". Thus, we need to make it precise what it means for a lattice theory to "correctly reflect" the topology.

In the past, this used to be only a matter of intuition. There has been no precise principle for this (except for TQFT, but here we are working with dynamical QFT). One major contribution of the present paper is to uncover a precise mathematical principle for what this really means---this is indeed why we wrote the mathematical Sections 5&6, and why these sections must not be dropped. (On the other hand, in the complimentary paper 2411.07195 whose goal is to introduce a detailed construction, there is no discussion of these mathematics but only the physical intuition behind. So this is along the lines of what the referee suggests.)

In the updated version of the present paper, I will state it clearly at the beginning of Section 5 that the mathematics to be introduced is not only motivational; they serve the purpose of uncovering the precise meaning of ``having the lattice theory to correct reflect the topology", hence indeed making the paper convincing. In particular, in the new Section 5.5 of the updated version (which used to be the second half of the original Section 5.4), I will give a sharpened mathematical statement for what it means to ``correct capture the topology", and explain why the construction in Section 4 (and in 2411.07195 in more details) indeed fulfills this principle by design. This explains the first layer of "the theory works"---that the construction indeed correctly captures the topology.

The second layer is that the dynamics of the lattice theory, in particular including the dynamics of the topological configurations, should be right as we approach the continuum limit. Surely this ultimately can only be tested numerically, which we have not yet done, but we have some good reasons to argue that this is highly likely the case:

We introduced some new degrees of freedom and some new weights. If the new weights are chosen in certain limits, then integrating out the new d.o.f. will just reduce the theory back to the traditional Wilson's definition. So first of all, our refinement at least would not make things worse.

On the contrary, it is likely that suitably chosen new weights will make the renormalization convergence of the theory better. In the Villain model, which has been well-studied for decades, it is well-known that adding a suitable vortex fugacity term weighing the new d.o.f. actually facilities the renormalization convergence of the theory, rather than ruins the theory (see e.g. the cited Kogut's review paper). The physical idea is that, even if one does not include those higher d.o.f. and higher weights to begin with, along the coarse-graining procedure, those effects will be effectively generated anyways (since those higher d.o.f. are to capture how the fields interpolate, which is like the reminiscence of the d.o.f. on a finer lattice), so having those higher d.o.f. and weights in the action actually helps keeping track such effects along the renormalization flow. This has been analyzed in the Villain model. Since the new constructions are based on the same physical idea, it is physically intuitive to expect the same advantage (although the reasonable range of choices for the new weights should be determined numerically). Discussions along this line appeared in the final section in the original version, but in the updated version I will make the discussions more clearly throughout the text.

Therefore, this second layer of the meaning of "the theory works" is an educated expectation, that awaits for numerical checks as a next step (following our follow-up explicit construction paper 2411.07195). I believe, nonetheless, it is good to write theoretical papers introducing the important new concepts before the numerical implementation is carried out. As far as I am aware of, Wilson's original 1974 lattice gauge theory paper was motivated by the formal purpose of making good sense of what a QFT really means; only a few years later did Creutz implement it numerically and found it to be numerically useful. Also, Luscher's 1982 geometrical construction of instanton was a theoretical proposal which is only implemented a few years later. I hope this can justify that the present theoretical paper, which introduced the mathematical principles of "capturing topology onto lattice" (which are novel and physically intuitive), is of important values in its own right.

---

## Round 2 · Referee Report · Anonymous (Referee 1) · 2025-3-20

Strengths
Weaknesses
Report
This is unusual procedure. Since I am asked to review the first article [Chen-2406.06673] and not the second [ZhangChen-2411.07195], and since the requested information is claimed to be relegated to the second which I am not asked to review, I think that already on formal grounds I have no choice but to reject the submission at this point, for not providing the requested major revision.
Still, I did read the new article [ZhangChen-2411.07195], which I am not asked to review but which I will comment on now, nevertheless.
On its p. 3, the new article [ZhangChen-2411.07195] claims that
"In this paper we will only describe the intuitive explicit construction, while directing any mathematical formality to [1]"
where [1] of course is the article actually under review here.
But this claim contradicts the claim in revision made by the author, that: "The complimentary follow-up paper arXiv:2411.07195, [...] contains a detailed technical construction."
In conclusion, both articles point to each other for more details.
It seems the only "technical details" which the first article (the one under review) provides on top of the second is its section on higher category theory. However, my complaint from the first report still stands, that this section had (and has) no tangible relation to the actual construction presented. Therefore there is no sense in which the provided discussion of higher category theory (which is largely sketchy anyways) provides details for either of the pair of articles we are looking at.
In fact, the new, second, articles claims on its p. 2 that
"The degrees of freedom altogether form a higher category structure (a suitable weak 4-group) [...] the language of higher category theory really is necessary here"
This sounds superficially like a plausible research plan, and it would be interesting to see this carried out, but the claim is at odds with the material presented, since it is nothing like this.
A potentially acceptable form of the author's claim based on the above quoted advertisement would look like this:
An actual definition of the 4-lattice as a 4-category, then an actual definition of a 4-groupoid coefficient resolving the Yang-Mills monodromies and an actual 4-functor from there to the 4-fold delooping 4-groupoid of Z projecting out the instanton number. Finally an actual definition of 4-functors from the lattice to these coefficients with an actual definition of the path integral over such 4-functors.
This is the kind of construction that the introduction of both articles lead the reader to expect to be presented. But the actual construction offered is not remotely close to this.
Instead, the actual definition, now (16) in the new article, is a decidedly non-categorical expression using a lengthy list of explanations of its symbols which I cannot vouch to be a precise definition, and which in any case seems too roundabout to base any further deductions on.
I am not saying that it might not work. Maybe the authors figured it out and the problem is just in the presentation. But since it is only the presentation that I can base judgement on, and since this presentation is far from living up to its claims, I do not recommend publication (neither of the article that I am asked to review nor, for what it's worth, of the one that was offered in place of its requested major revision).
In closing, I'll to point out references on actual higher-categorical/homotopical constructions of characteristic 4-classes from gauge data:
J.-L. Brylinski and D. A. McLaughlin:
"The geometry of degree-four characteristic classes and of line bundles on loop spaces I."
Duke Math. J., 75(3) (1994) 603–638
J.-L. Brylinski and D. A. McLaughlin:T
"The geometry of degree-4 characteristic classes and of line bundles on loop spaces. II"
Duke Math. J., vol 83 no 1 (1996) 105–139
J.-L. Brylinski and D. A. McLaughlin:
"Cech cocycles for characteristic classes",
Comm. Math. Phys., vol 178 no 1 (1996) 225–236
D. Fiorenza et al:
"Cech Cocycles for Differential Characteristic Classes"
Adv. Theor. Math. Physics vol 16 no 1 (2012) 149-250
with exposition in
D. Fiorenza et al:
"A higher stacky perspective on Chern-Simons theory"
in: "Mathematical Aspects of Quantum Field Theories"
Springer (2014) 153-211
Recommendation
Reject
Warnings issued while processing user-supplied markup:
- Inconsistency: plain/Markdown and reStructuredText syntaxes are mixed. Markdown will be used.
Add "#coerce:reST" or "#coerce:plain" as the first line of your text to force reStructuredText or no markup.
You may also contact the helpdesk if the formatting is incorrect and you are unable to edit your text.
I sincerely thank the referee for seriously reading and thinking about the contents of both papers, and returning their opinions sharply. I do hope the referee can however re-evaluate the manuscript after reading this reply.
I have to first clarify the misinterpretation that the follow-up paper 2411.07195 was written “in reaction to” the previous reports by both referees. It was NOT. It was planned to be written as a complimentary paper, independent of the submission process of this first paper. (One can check that in the original version of this first paper, it was already stated that this follow-up paper was in preparation.) Papers appearing in sequential orders and serving complimentary roles, together tackling one problem, is completely normal; perhaps my coordination of timing was not that optimal here, making it as if the follow-up was written as an “unusual reaction" rather than what it really is---a completely normal, planned complimentary paper.
I feel I particularly need to clarify this, because in the second report the referee seemed to have developed a particularly negative tone/attitude, in relation to us writing a second paper. Let me state clearly: I whole-heartedly respect the referees' first reports and took them seriously; I implemented many suggested revisions thanks to the second referee, and particularly important revision/improvement was made in Section 5.5, changing the over-category perspective to the more generally applicable relative cohomology perspective, thanks to the referee's question on $BEG$; in my response I was bringing up the follow-up paper because I do think it contains (as planned) some of the key technical information that both referees were asking for---i.e. the expression of the partition function in traditional notations.
It seems the referee does consider this overall program to be compelling, and I am sincerely glad about it, despite the referee has judged the current paper's proposal as "hard to discern" and recommended a rejection. To be honest, regardless of the judgements of my current paper(s), my overall goal is that in the long run this program will work out throughout the relevant communities. If other future works can flesh out this program much better than these current papers do, I will only be more than happy.
That said, I will firmly defend my current papers and I of course want the one under review to be accepted. My response to the referee's objections are:
-
I totally disagree with the referee's judgement that the presentation of the complimentary paper 2411.07195 is "too roundabout to base any further deductions on".
-
I totally disagree with---and actually find it hard to understand---the referee's assertions that "the advertised higher categorical construction is nowhere to be seen", "there is nothing like this", and that the Section 5 on categories "is not remotely close to"/"has no tangible relation to" the actual construction presented. I am willing to further improve my wordings in Section 5.5 and Section 4, so to further improve their connection, but that is only going to be incremental. It really is hard to understand the assertion that this connection is in the current version "nowhere to be seen".
-
The referee said "both articles point to each other for more details". This is of course true because they are supposed to serve complimentary roles, and the "more details" that they point to each other for are clearly DIFFERENT aspects of the same research. I don't see how this can in any way be problematic, and how this "contradicted the author's claim in revision".
I will elaborate on points 1 and 2. I believe 3 is self-evident.
1.
It is the best to compare with Seiberg's Phys. Lett. B 148 (1984) 456, in order to respond to the referee's judgement on the presentation of our complimentary paper 2411.07195.
Seiberg's 1984 paper partly refined Luscher's 1982 geometrical construction by introducing a dynamical lattice 3-form U(1) gauge field $\tilde{k}$, which is nothing but the dynamical CS 3-form in our paper. Seiberg indeed also introduced the CS saddle function, $k(U)$, which is also a "symbol defined using a lengthy list of explanations" (in the referee's words).
What our paper 2411.07195 crucially achieved compared to Seiberg's 1984 paper is, we introduced different possible plaquette interpolations with suitable weights (as opposed to fixed plaquette interpolations), and introduced a CS sensitivity function $|\nu|$ with suitable properties (as opposed to the fixed constant $\alpha$ in Seiberg's paper), thereby explicitly resolving the singularities in Seiberg's definition (or more precisely, removing the unphysical and discontinuous "admissibility" condition that used to be imposed in order to artificially avoid those singularities). This comparison is mentioned at footnote 10 in our 2411.07195.
Regarding the presentation, one can see that both Luscher's and Seiberg's work contained construction of functions via lengthy technical steps. Our 2411.07195 is technical along the same line and to the same level. In fact, we have elucidated and removed some unnecessary technicality already (by using Wilson loops instead of Wilson lines for interpolation). We also made additional efforts to improve the readability/transparency by including many pictorial illustrations and intuitive explanations of the technical steps that we are doing, instead of asking the readers to decipher those long formulae themselves.
And over the decades there had been many variants of Luscher's construction, and they are all technical and lengthy at a comparable level. As far as I can see, this technicality is hard to avoid. From Luscher, to Seiberg, through other works over the decades, to us, the technicality has stayed similar, and I believe all these authors, ourselves included, have been trying the best of what they (we) could to present explicit definitions. Nobody tried to "roundabout" stuff. Of course, if someday someone resolved the technicality in one shot by coming up with some simple formulae doing all the same job, I will be extremely happy. But there is no hint of this (we tried very hard and only obtained our limited elucidation mentioned above), so before this happens, I don't see how there can be any problem working with the state-of-art technicality which is lengthy but well-defined.
In summary: In the previous report, the referee asked to see a detailed description of the path integral, so I said such a description, as planned, was presented in our complimentary work 2411.07195; we constructed it using a technical but well-defined state-of-art method. Compared to the published and well-established works of Luscher's and Seiberg's (needless to say, both are physicists of top calibre, and in particular Luscher's work has been very influential and useful), our work made the crucial advance of explicitly resolving the singularities (or more precisely, removing the unphysical "admissibility" condition)---the key problem in these pioneering works---and our presentation is at a level of technicality comparable to theirs, but elucidated and removed some parts of the technicality, and included many pictorial illustrations and intuitive explanations, which could only be helpful.
Given this comparison, I sincerely hope the referee can re-evaluate our complimentary paper 2411.07195, recognizing its presentation of the lengthy (but well-defined) construction as being what one can reasonably do at best rather than dismissing it as something "too roundabout to base any further deductions on". If, on the contrary, the referee still rather disregards our complimentary paper as being apparently inferior to the established papers of Luscher's and Seibergs, to the extent that it is an unacceptable "roundabout" not worthy for publication or even any further deductions/discussions, then I would have to solicit a concrete explanation from the referee of why.
2.
The connection between higher category theory (if any) and our non-categorical description (as a usual lattice path integral) in Section 4 and in the complimentary paper 2411.07195 is, of course, a fair question to ask.
If there was a single inspiration that made this entire program possible, it was 1307.4793 by Gukov and Kapustin and the closely related 1308.2926 by Kapustin and Thorngren. This is acknowledged on page 29 of my manuscript under review. Section V.B of 1307.4793 and Section 5 of 1308.2926 brought to me the key insight that the d.o.f. in the Villain $PSU(N)$ gauge theory organize into a strict 2-group. And these two papers are considered important by the community as of today.
In these two seminal papers, when introducing the relation between the Villainized $PSU(N)$ gauge theory and the strict 2-group, there was no discussion to the level of mathematical rigor required by the referee (in order for my manuscript to be "potentially acceptable"). The discussions there were mostly in physicists' familiar terms. As physics papers, I don't think the connection thus made between Villainized $PSU(N)$ gauge theory and strict 2-group was therefore "sketchy" or "nothing tangible". To me, and perhaps to many other physicists as well, the connection thereby made was sufficiently evident, and truly inspiring.
And this is the kind of communication I want to provide to the physicist readers, albeit the problem now is intrinsically much more technically involved than the Villain case.
To better understand the referee's negative judgement, I would like to ask: For the physically well-established Villain $S^1$ NLsM, Villain $U(1)$ or $PSU(N)$ gauge theory, and spinon-decomposed $S^2$ NLsM, does the referee consider their relations to strict (higher) group(oid)s mathematically well-established in the mathematical physics community? If yes, does the referee think the presentation in Section 5 of the current manuscript helps a broader theoretical physics audience make that connection for these well-known models (given that categories are not something most theoretical physicists will think about at least as of now)? Does the referee agree that the discussion in Section 5 (especially the first few pages of Section 5.5) on these known models, though not pursuing mathematical rigor, is more comprehensive than the brief (but already inspiring enough) discussions in 1307.4793 Section V.B and 1308.2926 Section 5?
Now let's turn to the new models. Let me first summarize the mathematical ideas here, and then discuss my presentation.
If the referee wants, we can begin with the referee's suggested "actual higher-categorical/homotopical constructions" papers. [While I appreciate suggestions, I don't understand why the referee put the word "actual" there. The mathematical literatures that I based my work on and properly cited---which will be mentioned in the below---are, by all measures, no less "actual".] Among the referee's suggested papers, the one on $\mathbb{Z}$-valued Cech cocycles is not so relevant because it only captures the topological class but loses the continuous geometry of the original Lie group---the latter has been emphasized in my manuscript, because the refined d.o.f. must at least cover those in Wilson's traditional theory. The other four papers (among which, the last two by Fiorenza et al were indeed unknown to me before) are somewhat more relevant but still far from what we need.
Let's, say, look at the referee's suggested exposition 1301.2580 by Fiorenza et al, in particular its Section 3. This section is built around the stack map $BG\rightarrow B^3 U(1)$, applying it on different continuum manifolds. But does this directly achieve what we want? Clearly NOT. Stacks are defined on the Grothendieck topology of continuum manifolds; what we ultimately want is a lattice path integral with finite-dimensional d.o.f.. There is no further discussion in 1301.2580 that is directly helpful for moving from the stack description of $BG\rightarrow B^3 U(1)$ towards our actual goal.
If all that is needed is a precise categorical statement of the solution, then I have already stated it in Section 5.5: The proposed 4-group of degrees of freedom for the refined $G=SU(N)$ lattice Yang-Mills is the span of an anafunctor that realizes the generator of $\pi_0 \mathbf{H}(BEG, BG; B^5\mathbb{Z})$ (counting Yang monopole) in the ambient category of finite dimensional smooth Kan complexes (or its cubical counter-part instead of simplicial); and for pion NLsM it would be $\pi_0 \mathbf{H}(B|G|, |G|; B^4\mathbb{Z})$ instead (counting baryon non-conservation defect). (Except that in Section 5.5 $\pi_0 H(X; B^n\mathbb{Z})$ was expressed as the more familiar $H^n(X; \mathbb{Z})$.) And in fact the homotopy of the stack map $BG\rightarrow B^3 U(1)$ is naturally isomorphic to my claimed $\pi_0 \mathbf{H}(BEG, BG; B^5\mathbb{Z}) \cong H^5(BEG, BG; \mathbb{Z})$. So already here, I don't know what the referee might consider is the essential content in 1301.2580 that was missed in my manuscript. (Also, the Deligne-Cech evaluation of this stack map in Section 3.3 of 1301.2580 was already discussed in physicists' familiar notations at the beginning of my Section 4.) But my manuscript had to achieve more, i.e. an actual finite dimensional realization of this generator to be used as lattice d.o.f., which is why the "fancy" category language is at all useful to this traditional long standing problem. So we have to do the following unwinding, using the no-less-actual mathematical literature that I properly cited.
1) While stacks and stack maps are defined on the Grothendieck topology of continuum manifold, they have many other homotopically equivalent realizations, and I particularly looked into the anafunctor (bibundle) between (higher) groupoids, a perspective that I learned from 0911.2483 by Schommer-Pries. Why did I particularly focused on this realization? Because when applied to the simpler case of $BG\rightarrow B U(1) \rightarrow B^2\mathbb{Z}$ discussed in Section 2 of 1301.2580, this anafunctor realization indeed reduces to the strict 2-group Villain structure that I learned from 1307.4793 and 1308.2926. So this must be the right route to take.
There is another route to motivate the same idea (which was my actual original motivation when I started this research three years ago). If we want a direct generalization of Villainization, then, as said in my Section 3, the 3-connected cover on the Whitehead tower will be needed. But the 3-connected cover is an infinite dimensional space, so we need a categorical realization of it (idea learned from Baez and Huerta 1003.4485) that is finite dimensional. This again leads to the same higher groupoids and anafunctors, thanks to the same paper 0911.2483 by Schommer-Pries.
2) The realization of anafunctor between $BG\rightarrow B^3 U(1)$ (now seen as groupoids instead of stacks) has been studied as "multiplicative bundle gerbe" in math/0410013 by Carey et al (which was also cited in the referee's suggested 1301.2580). But they only gave an existential statement about finite dimensional realization, not an actual description. (Similarly in 0911.2483 by Schommer-Pries.)
3) To find an actual finite dimensional description of a multiplicative bundle gerbe, we can begin with that of a bundle gerbe, realizing $|G|\rightarrow B^2 U(1)$. For $G=SU(N)$ an explicitly conjugation-equivariant construction of bundle gerbe has fortunately been given in hep-th/0205233 by Gawedzki and Reis, and the description is easily understandable to physicists.
4) Now the key task is to deloop this known description of a bundle gerbe into that of a multiplicative bundle gerbe. Most mathematical literatures only gave an existential statement about this. The paper that aimed to constructively introduce a delooping protocal was Waldorf's 1201.5052, in particular Section 7, method 2. It makes use of path spaces, and a rigorous treatment requires the path space's diffeological structure. And a key map is need, denoted as $\mathcal{A}$ there, which was however not explicitly defined.
The actual study of this key map $\mathcal{A}$ was done in Waldorf's previous paper 1004.0031, which was, alas, not manifestly expressed either. It was described as a stable isomorphism between the bundle gerbe given in hep-th/0205233 by Gawedzki and Reis and the tautological WZW bundle gerbe constructed on the path space. Then I deciphered that the stable isomorphism can be described by manifest choices of representative paths.
5) The multiplicative bundle gerbe is not yet what we finally need, unless we are considering flat gauge configurations only. Rather, a multiplicative bundle gerbe over $G$ appears as a sub-category of actual the target category of a refined $|G|$ NLsM, obtained by fixing the source (or target) object. This point is most easily seen by comparing to the more well-known case of Villain model, as has been particularly emphasized in Section 5.5 in relation to relative cohomology. And for refined $G$ gauge theory, we need to further deloop the target category of the $G$ NLsM.
Unwinding this line of logic, I hope the referee can appreciate the facts that: -- The multiplicative bundle gerbe on which my construction is based is well-defined in the "actual" mathematical literature. -- On the other hand, the procedure to explicitly construct a finite dimensional multiplicative bundle gerbe has not been written explicitly in any single mathematical paper, but rather stretched over several papers over the time span of a decade.
I think it would be great if someone writes up a pure mathematics paper that explicitly combines the contents from the several papers mentioned above into one rigorous explicit description of a finite dimensional multiplicative bundle gerbe over $SU(N)$. However, I don't think I should be obligated to do this job in this physics paper in order for it to be just "potentially acceptable", especially considering that this will take tens of pages of rigorous mathematical notions (diffeological structure, Deligne cohomology etc) that are unfamiliar to the even the formal community of theoretical physicists. Over the past, physics papers which introduced important new mathematical concepts or ideas into physics were mostly not subjected to such kind of obligation, including the aforementioned 1307.4793 and 1308.2926 which directly inspired me. I don't understand why this is asked for for my paper to be "potentially acceptable". I believe properly citing (as I did) the mathematical literatures should be appropriate.
Now let's see how I presented the material.
-- From Eq(110) to Eq(114), I started by explaining how the well-known Villainization process is naturally understood from the anafunctor point of view. This serves two purposes: first, it makes precise the connection between Villainization and groupoid or 2-group that was only intuitively seen but not formalized in the aforementioned seminal works 1307.4793 and 1308.2926; second, it sets the stage for how we want to generalize towards our main problem of interest. Of particular importance is the relation between the principle bundle Eq(110) and the actual Villainization Eq(111) (or more generally, between Eq(113) and Eq(114)), which we explained at length---and this is nothing but point 5) in the above. For here we arrived at the key idea that lattice topological configurations for continuous d.o.f. (rather than the well-studied discrete d.o.f.) are characterized by *relative* cohomology, which as far as I know is novel. Then, from Eq(115) to Eq(118) we generalized the formalized ideas to the also well-known spinon decomposition.
-- Based upon the previous step, in Eq(119) I introduced the notion of bundle gerbe from the anafunctor perspective, and it is clear that it is a generalization of Eq(115). In particular, the stable isomorphism (ananatural isomorphism) between the infinite dimensional tautological bundle gerbe (which will make good connection to the continuum field theory) and the finite dimensional realization of the bundle gerbe is discussed. The mathematical literatures that contain rigorous discussions are properly cited.
-- Then I introduced the notion of multiplicative bundle gerbe in Eq(123). This serves two purposes: First, while it is not what we finally need, it is closely related to what we need, and it has been well-studied---the mathematical literatures that contain rigorous discussions are properly cited. Second, we want to see later in the text that the Dijkgraaf-Witten twist for discrete gauge group can be understood in the same manner, establishing the connection between our construction for continuous d.o.f. and the previously well-established formal literatures for discrete d.o.f..
-- Then we introduced Eq(127) and Eq(128), which corresponds to what we have in continuum NLsM and what we want for the lattice NLsM, respectively. The relation between Eq(127) and the tautological bundle gerbe, and the relation between Eq(128) and the finite dimensional bundle gerbe, are parallel to the relation between the aforementioned Eq(118) and Eq(115), and that between the aforementioned Eq(114) and Eq(110). So the physical motivation for these structures is clear.
-- The final task is to construct the desired finite dimensional span, in particular the $\Lambda$, of Eq(128), which is supposed to be ananaturally isomorphic to that in Eq(127). This is explained by Eq(129), i.e. by finding representative paths for each element of $Y$. This procedure is parallel to the idea in point 4) above in the context of multiplicative bundle gerbe. The representative paths are nothing but those introduced in Section 4.1. (Footnote 136 provides a discussion for a simpler and more familiar situation.) With these representative paths, the $\Lambda$ in Eq(128) is already well-defined via the discussion below Eq(129). The $W_2$ weight and in particular its $\mu$ function in Section 4.1 serve the purpose of assigning weight of elements in $\Lambda$. While this is straightforward to see, I can revise the manuscript and explain more about this point. That said, from the discussion in the current manuscript, the correspondence is already plausible, and can be verified straightforwardly--especially given the straightforward comparison to the more familiar Eq(41) whose geometric meaning has been explained at length in Section 2.4.
-- Eq(130) and below discusses how to deloop the structure needed for NLsM to that needed for Yang-Mills. I can explain the connection to Section 4.2 in better details in the revision.
Summarizing the above, I really do find the material (given that it is intrinsically quite non-trivial) has been presented in a well-motivated and well-organized manner, and the relevant mathematical literatures that contain more rigorous discussions have been suitably cited whenever needed. Major revision has already been made after the first report, that led to the current Section 5.5, and on top of that I don't see what must be further substantially revised, were it not for making the mathematical discussions rigorous---but as I said above, this should not be the job of a physics paper, and such requirement was indeed not seen in similar physics papers in the past, not even in most of those seminal ones. As I said, I am happy to further improve my wordings, but any further explanation will just be an increment of an extra page or so, because in the current version the connection between Section 5.5 and Section 4 can be readily seen. So, I really find it hard to understand the referee's assertions that "the advertised higher categorical construction is nowhere to be seen", "there is nothing like this", and that the Section 5 on categories "is not remotely close to"/"has no tangible relation to" the actual construction presented. And I find it equally hard to understand why the referee implied those suggested further references are the "actual" mathematical papers needed while they are actually not directly helpful (though relevant), meanwhile overlooking the many mathematical papers that I cited which are actually directly helpful.

Author: Jing-Yuan Chen on 2025-07-25 [id 5677]
(in reply to Report 2 by Mithat Ünsal on 2025-07-24)I would like to thank Referee 3 for their recognition of the value of this work and their recommendation of publication, based on the currently submitted version (version 2). I would also like to thank Referee 3 for suggesting future directions of research.
It turns out that per Referee 1's previous report, I have already been making major revisions in Sections 2.4, 4.1 and 5.1-5.5, as promised in my previous reply to Referee 1's report. I believe these revisions will improve the quality of the paper, therefore I still plan to complete these revisions and upload a version 3. These revisions are clarifications of ideas (per Referee 1's request), and there will be no major change in the actual scientific content; so I expect these revisions will not interfere with Referee 3's positive remarks.

---

## Round 2 · Referee Report · Mithat Ünsal (Referee 3) · 2025-7-24

Report
the definition of the winding number in the XY model and the monopole charge operator in U(1) lattice gauge theory become unambiguous. The paper argues that to achieve what Villain formulation does for the above mentioned theories
can be performed in lattice Yang-Mills theory with the machinery of the
Higher Category Theory.
If we unpack the contents of what is done in this paper, what is actually done turns out to be the extension of Luscher's formulation of the lattice topological charge. However, Luscher's formulation requires something called admissibility condition from gauge field configurations, i.e. lattice field configurations are sufficiently smooth. This is usually achieved with some sort of gradient flow or cooling method. However, for example, to define winding number or monopole charge in the Villain form of the XY model and U(1) lattice gauge theory, we can work directly with lattice fields at the cut-off. In essense, Higher Category Theory buys this step for lattice Yang-Mills theory. The chart on page 7 is a good summary of what is essentially achieved by the formalism.
To summarize, I find this work well motivated and quite reasonable.
Perhaps, one possible weakness of the paper is the fact that it is a bit abstract. However, a subsequent work (https://arxiv.org/abs/2411.07195) makes it more concrete. So, I do not see this as a problem. Hence, I recommend the paper for publication.
I think the paper opens interesting and important prospects of lattice gauge theory into investigation. In certain sense, Villain brings lattice and continuum description of gauge theories closer. Having a useful definition on the lattice, if I may, suggest the author an examination of the moduli space of instantons, a computation of beta function in instanton background (similar to NSVZ computation) and perhaps, more ambitiously, generalize the ADHM construction into this context.
Recommendation
Publish (easily meets expectations and criteria for this Journal; among top 50%)

---

## Round 2 · Author Response

List of changes
Main: 1) The complimentary follow-up paper arXiv:2411.07195, which contains a detailed technical construction for the refined Yang-Mills theory, is cited wherever needed, especially in Section 4.2. 2) Splitted the original Section 5.4 into 5.4 and 5.5. Moreover, the contents being moved to Section 5.5 are significantly improved. 3) Discussions about the triviality of ET versus the non-triviality of the lattice theory is moved from Section 5.2 to Section 5.1. 4) In Section 2.1, after introducing the vortex fugacity, we added a paragraph emphasizing its role in improving the renormalization behavior (so that the "back-reaction" from the perspective of the "original model" is in fact not only not a problem, but an advantage). The idea has been reiterated in more general models later in the paper.
Besides these main changes, minor changes are made, fixing the typos and/or inaccuracies brought up by the 2nd referee.
Relevant references missed in the original version are added. They are: [2], [7], [10] (the main follow-up), [15], [47], [66, 67] (both appeared after this paper was first posted), [76], [78], [79], [96], [97], [105],

---

## Round 2 · List of Changes

Main: 1) The complimentary follow-up paper arXiv:2411.07195, which contains a detailed technical construction for the refined Yang-Mills theory, is cited wherever needed, especially in Section 4.2. 2) Splitted the original Section 5.4 into 5.4 and 5.5. Moreover, the contents being moved to Section 5.5 are significantly improved. 3) Discussions about the triviality of ET versus the non-triviality of the lattice theory is moved from Section 5.2 to Section 5.1. 4) In Section 2.1, after introducing the vortex fugacity, we added a paragraph emphasizing its role in improving the renormalization behavior (so that the "back-reaction" from the perspective of the "original model" is in fact not only not a problem, but an advantage). The idea has been reiterated in more general models later in the paper.
Besides these main changes, minor changes are made, fixing the typos and/or inaccuracies brought up by the 2nd referee.
Relevant references missed in the original version are added. They are: [2], [7], [10] (the main follow-up), [15], [47], [66, 67] (both appeared after this paper was first posted), [76], [78], [79], [96], [97], [105],

---

## Editorial Decision

published